# Fairness-aware Bayes optimal functional classification

**Xiaoyu Hu** [*]

School of Mathematics and Statistics

Xi'an Jiaotong University

huxiaoyu2019@gmail.com

**Gengyu Xue** [*]

Department of Statistics

University of Warwick

gengyu.xue.1@warwick.ac.uk

**Zhenhua Lin**

Department of Statistics and Data Science

National University of Singapore

linz@nus.edu.sg

**Yi Yu**

Department of Statistics

University of Warwick

yi.yu.2@warwick.ac.uk

## Abstract

Algorithmic fairness has become a central topic in machine learning, and mitigating disparities across different subpopulations has emerged as a rapidly growing research area. In this paper, we systematically study the classification of functional data under fairness constraints, ensuring the disparity level of the classifier is controlled below a pre-specified threshold. We propose a unified framework for fairness-aware functional classification, tackling an infinite-dimensional functional space, addressing key challenges from the absence of density ratios and intractability of posterior probabilities, and discussing unique phenomena in functional classification. We further design a post-processing algorithm Fair Functional Linear Discriminant Analysis classifier (Fair-FLDA), which targets at homoscedastic Gaussian processes and achieves fairness via group-wise thresholding. Under weak structural assumptions on eigenspace, theoretical guarantees on fairness and excess risk controls are established. As a byproduct, our results cover the excess risk control of the standard FLDA as a special case, which, to the best of our knowledge, is first time seen. Our theoretical findings are complemented by extensive numerical experiments on synthetic and real datasets, highlighting the practicality of our designed algorithm.

## 1 Introduction

Driven by technological advancements that enable high-resolution data collection and analysis, functional data analysis (FDA) has gained increasing attention over the past two decades. A wide range of statistical research has been carried out in this area, and we refer readers to Wang et al. (2016) for a comprehensive review of recent developments.

Across a variety of statistical tasks, functional classification has emerged as one of the central focuses, with applications in many areas including neuroscience (e.g. Heinrichs et al., 2023; Lila et al., 2024), genetics (e.g. Coffey et al., 2014), handwriting recognition (e.g. Hubert et al., 2017) and others. Various classifiers have been proposed and thoroughly studied in the literature, among which are classifiers based on projection (e.g. Delaigle and Hall, 2012; Kraus and Stefanucci, 2019), Radon–Nikodym derivatives (e.g. Berrendero et al., 2018; Torrecilla et al., 2020), principal component score

---

[*]Equal contribution

39th Conference on Neural Information Processing Systems (NeurIPS 2025).

densities (e.g. Dai et al., 2017) and partial least squares (e.g. Preda et al., 2007), to name but a few. While these classifiers often perform well in terms of accuracy, our real data analysis on NHANES dataset in Section 4 demonstrates that applying standard functional classification algorithms can lead to substantial unfairness. Despite the growing recognition of fairness as a critical issue in machine learning and statistics, to the best of our knowledge, there is no existing work addressing unfairness in functional data classification. We are to bridge this gap by providing a principled framework for fairness-aware functional classification.

To date, fair classification for multivariate data has been extensively studied, including Yang et al. (2020); Jiang et al. (2020); Wei et al. (2021); Zeng et al. (2024a,b); Hou and Zhang (2024). These algorithms can be classified into pre-, in- and post-processing procedures. Pre-processing methods aim to modify training data prior to model training, allowing models to learn from debiased inputs (e.g. Calmon et al., 2017; Johndrow and Lum, 2019). In-processing ones handle fairness constraints during the training step. Common strategies include fairness-constrained optimisation (e.g. Narasimhan, 2018; Celis et al., 2019) and fairness penalised objective functions (e.g. Cho et al., 2020). Post-processing one, by contrast, seek to reduce disparities by modifying predictions after training is completed (e.g., Kim et al., 2019; Li et al., 2022).

However, the infinite-dimensional nature of functional data poses unique challenges, rendering the aforementioned fairness methods developed for multivariate data ineffective, see numerical results in Appendix A.6. To overcome this, we fully respect the functional nature of the data and derive the Bayes optimal fair classifier. Furthermore, we establish that the excess risk converges to zero and provide an explicit convergence rate. The most related work to this paper is Zeng et al. (2024a), in which they develop a framework for Bayes-optimal fair classifiers under finite-dimensional feature spaces with a strong reliance on posterior probabilities, which are unfortunately intractable in functional spaces. This handicaps the direct application of Zeng et al. (2024a) to functional data. In this paper, we resort to the Radon–Nikodym derivative, which is used naturally as a substitute and plays a central role in characterising optimal decision rules under fairness constraints in the functional setting. Additional challenges, further comparisons and practical considerations are provided in Remark 1.

In fact, even without fairness constraints, the characterisation of excess risk for functional classification is unresolved in general settings with unknown eigenfunctions of the covariance operator. Addressing fairness in this setting is inherently more challenging, and quantifying the trade-off between fairness and accuracy is theoretically more demanding and remains largely unexplored.

## 1.1 List of contributions

In this paper, we study the problem of optimal binary classification for functional data under various fairness constraints. Specifically, we focus on the case when the sensitive attribute is binary and the probability measures of two classes of standard features within each sensitive group are mutually absolutely continuous. The main contributions of this paper are summarised as follows.

Firstly, to the best of our knowledge, this is the first study to explore fair classification for functional data. We propose a unified framework for constructing the fair Bayes-optimal classifier for functional data, providing a functional data-tailored treatment of fairness-aware classification problems. Our framework is sufficiently general to accommodate a wide range of extensions, including settings where the sensitive feature is unavailable at test time and multi-class classification problems. These extensions (detailed in Appendix B) highlight the flexibility of our approach and its potential to serve as a foundation for future advances in fairness-aware functional data analysis..

Secondly, when the non-sensitive features are assumed to be Gaussian processes, we introduce a post-processing algorithm, the Fair Functional Linear Discriminant Analysis classifier (Fair-FLDA) in Algorithm 1, which effectively enforces fairness by group-wise thresholding. Our algorithm accounts for the most general setting where we assume all model parameters, including group-wise covariance functions and their eigenvalues and eigenfunctions, to be unknown.

Thirdly, we further establish the finite-sample theoretical guarantee for the proposed algorithm in terms of both fairness and excess risk control, ensuring our algorithm Fair-FLDA not only adheres to the specified fairness constraint with high probability, but also achieves a satisfactory classification performance, with the cost of fairness explicitly quantified. As a byproduct, our results cover the

special case of functional classification without fairness, which serves as a complement of Wang et al. (2021) under a more general setting when eigenfunctions are assumed to be unknown.

Finally, the proposed algorithm is validated through extensive numerical experiments on both simulated and real datasets in Section 5 and Appendix A. The comparisons with existing multivariate fairness methods to functional data further highlight the superiority and practical necessity of our method for fair functional classification from a numerical perspective.

further supporting our theoretical findings and highlighting their practicality.

**Notation.** In this paper, for $a \in \mathbb{N}_+$, denote $[a] = \{1, \ldots, a\}$. For $a, b \in \mathbb{R}$, let $a \vee b = \max\{a, b\}$. For two sequences of positive numbers $\{a_n\}$ and $\{b_n\}$, denote $a_n \lesssim b_n$ (or $a_n = O(b_n)$) and $a_n \asymp b_n$, if there exists some constants $c, C > 0$ such that $a_n/b_n \leq C$ and $c \leq a_n/b_n \leq C$. Write $a_n \ll b_n$, if $a_n/b_n \to 0$ as $n \to \infty$. For a sequence of random variables $\{X_n\}$ and positive numbers $\{a_n\}$, denote $X_n = O_p(a_n)$ if $\lim_{M \to \infty} \limsup_n \mathbb{P}(|X_n| \geq Ma_n) = 0$. For any two $\sigma$-finite measures $\mu$ and $\nu$, denote $\mu \ll \nu$ if $\mu$ is absolutely continuous with respect to $\nu$ and write $\mathrm{d}\mu/\mathrm{d}\nu$ the Radon–Nikodym derivative; write $\mu \sim \nu$ if they are equivalent. Let $L^2([0,1])$ be the space of square-integrable functions on $[0,1]$. For $f \in L^2([0,1])$, denote $\|f\|_{L^2}^2 = \int_0^1 f^2(s)\,\mathrm{d}s$. For $f, g \in L^2([0,1])$, denote the inner product by $\langle f, g \rangle_{L^2} = \int_0^1 f(s)g(s)\,\mathrm{d}s$. For any bivariate kernel function $K : [0,1]^2 \to \mathbb{R}_+$, let $\mathcal{H}(K)$ denote the reproducing kernel Hilbert spaces (RKHS) generated by $K$. For $f \in \mathcal{H}(K)$, denote $\|f\|_K^2 = \sum_{j=1}^{\infty} \langle f, \phi_j \rangle_{L^2}^2 / \lambda_j$ its RKHS norm, where $\{\phi_j\}_{j=1}^{\infty}$ and $\{\lambda_j\}_{j=1}^{\infty}$ are obtained by Mercer's decomposition of $K$: $K(s,t) = \sum_{j=1}^{\infty} \lambda_j \phi_j(s) \phi_j(t)$, $s, t \in [0,1]$.

## 2 Fair Bayes optimal classifier

### 2.1 Problem setup

Suppose that we have $n$ independent and identically distributed samples $\mathcal{D} = \{(X_i, A_i, Y_i), i \in [n]\}$, where $X_i \in L^2([0,1])$ is the standard (non-sensitive) functional feature, $A_i \in \{0,1\}$ is the sensitive feature (e.g. gender or race) and $Y_i \in \{0,1\}$ is the binary label. Let $\mathcal{F}$ be the class of measurable functions $f : L^2([0,1]) \times \{0,1\} \to [0,1]$ and our goal is to identify a randomised classifier $f^\star \in \mathcal{F}$, as defined in Definition 1, such that the misclassification error $R(f^\star) = \mathbb{P}(\widehat{Y}_{f^\star}(X, A) \neq Y)$ is minimised subject to a specified fairness constraint.

**Definition 1** (Randomised classifier). *For any $x \in L^2([0,1])$ and $a \in \{0,1\}$, a randomised classifier $f \in \mathcal{F}$ is a measurable function such that $f(x, a) = \mathbb{P}(\widehat{Y}_f = 1 | X = x, A = a)$, where $\widehat{Y}_f = \widehat{Y}_f(x, a)$ is the predicted label, i.e. $\widehat{Y}_f | \{X = x, A = a\} \sim \mathrm{Bernoulli}\,(f(x, a))$.*

For $a, y \in \{0,1\}$, let $P_{a,y}$ be the distribution of the random process $X$ given $(A, Y) = (a, y)$. Assume that $P_{a,1} \sim P_{a,0}$, thus the Radon–Nikodym derivative $\eta_a(X) = \mathrm{d}P_{a,1}(X)/\mathrm{d}P_{a,0}$ is well defined. Let $\pi_{a,y} = \mathbb{P}(A = a, Y = y)$ and $\pi_a = \mathbb{P}(A = a)$, then for each sensitive group, the Bayes rule that minimises the misclassification error is given by

$$f^*(x, a) = \mathbb{1}\{\eta_a(X) \geq \pi_{a,0}/\pi_{a,1}\}, \tag{1}$$

where $\mathbb{1}\{\cdot\}$ denotes the indicator function (e.g. Berrendero et al., 2018). However, the above classifier does not take fairness into account. To address this, in order to mitigate bias across groups, existing literature proposed various notions of parity, some of which are listed below.

**Definition 2.** *A classifier $f$ is said to satisfy (i) equality of opportunity (e.g. Hardt et al., 2016) if the true positive rates are the same among protected groups, i.e. $\mathbb{P}(\widehat{Y}_f = 1 | A = 0, Y = 1) = \mathbb{P}(\widehat{Y}_f = 1 | A = 1, Y = 1)$, (ii) predictive equality (e.g. Corbett-Davies et al., 2017) if the false positive rates are the same among protected groups. i.e. $\mathbb{P}(\widehat{Y}_f = 1 | A = 0, Y = 0) = \mathbb{P}(\widehat{Y}_f = 1 | A = 1, Y = 0)$, and (iii) demographic parity (e.g. Cho et al., 2020) if its prediction $\widehat{Y}_f$ is independent of the sensitive attribute $A$, i.e. $\mathbb{P}(\widehat{Y}_f = 1 | A = a) = \mathbb{P}(\widehat{Y}_f = 1)$, for $a \in \{0, 1\}$.*

Enforcing exact parity may lead to a substantial loss in accuracy. In the literature, one popular approach is to instead control the disparity measure, $\mathrm{D} : \mathcal{F} \to [-1, 1]$, i.e. upper bounding the difference in quantities of interest between the sensitive groups. The disparity measures corresponding to the notions of parity in Definition 2 are presented in Definition 3 for completeness.

**Definition 3.** *For a given classifier $f \in \mathcal{F}$, the disparity of opportunity (DO), predictive disparity (PD) and demographic disparity (DD) are defined as* $\mathrm{DO}(f) = \mathbb{P}\{\widehat{Y}_f(X,1) = 1|A = 1, Y = 1\} - \mathbb{P}\{\widehat{Y}_f(X,0) = 1|A = 0, Y = 1\}$, $\mathrm{PD}(f) = \mathbb{P}\{\widehat{Y}_f(X,1) = 1|A = 1, Y = 0\} - \mathbb{P}\{\widehat{Y}_f(X,0) = 1|A = 0, Y = 0\}$ *and* $\mathrm{DD}(f) = \mathbb{P}\{\widehat{Y}_f(X,1) = 1|A = 1\} - \mathbb{P}\{\widehat{Y}_f(X,0) = 1|A = 0\}$.

For any disparity measure D, tolerance level $\delta \geq 0$, we seek the $\delta$-fair Bayes optimal classifier $f^{\star}_{\mathrm{D},\delta}$, such that the misclassification error is minimised over all classifiers that satisfy the $\delta$-disparity, i.e.

$$f^{\star}_{\mathrm{D},\delta} \in \underset{f \in \mathcal{F}}{\arg\min} \left\{ R(f) : |\mathrm{D}(f)| \leq \delta \right\}. \tag{2}$$

### 2.2 Unified framework for fairness-aware functional classification

To characterise the $\delta$-fair Bayes optimal classifier defined in (2), in this section, we apply the generalised Neyman–Pearson lemma (see Lemma 43 in Appendix H) and exploit Radon–Nikodym derivatives, providing a unified framework for fairness-aware functional classification.

Applying the generalised Neyman–Pearson lemma is possible when both objective function and constraint in (2) are linear in the classifier. For the objective function, it follows from Lemma 8 in Appendix C.4 that $R(f) = \sum_{a \in \{0,1\}} \int_{\mathcal{X}} f(x,a)\{\pi_{a,0} - \pi_{a,1}\frac{\mathrm{d}P_{a,1}}{\mathrm{d}P_{a,0}}(x)\}\mathrm{d}P_{a,0}(x) + \mathbb{P}(Y = 1)$ is linear in $f$. For the constraint, we consider the class of bilinear disparity measures defined below.

**Definition 4.** *For all probability measures $P$ and $f \in \mathcal{F}$, a disparity measure $\mathrm{D} : \mathcal{F} \to [-1, 1]$ is bilinear in the classifiers $f$ and $\mathrm{d}P_{a,1}/\mathrm{d}P_{a,0}$ if there exist $s_{\mathrm{D},a}, b_{\mathrm{D},a} \in \mathbb{R}$ such that $\mathrm{D}(f) = \sum_{a \in \{0,1\}} \int_{\mathcal{X}} f(x,a)\{s_{\mathrm{D},a}\frac{\mathrm{d}P_{a,1}}{\mathrm{d}P_{a,0}}(x) + b_{\mathrm{D},a}\}\mathrm{d}P_{a,0}(x).$*

Definition 4 is also considered in Zeng et al. (2024a) and holds for many commonly used disparity measures in the existing literature, including those defined in Definition 3.

**Proposition 1.** *The disparity measures* $\mathrm{DO}, \mathrm{PD}$ *and* $\mathrm{DD}$ *defined in Definition 3 are bilinear with* $s_{\mathrm{DO},a} = 2a - 1$, $b_{\mathrm{DO},a} = 0$; $s_{\mathrm{PD},a} = 0$, $b_{\mathrm{PD},a} = 2a - 1$; *and* $s_{\mathrm{DD},a} = (2a - 1)\pi_{a,1}/\pi_a$, $b_{\mathrm{DD},a} = (2a - 1)\pi_{a,0}/\pi_a$, *for* $a \in \{0, 1\}$.

With both misclassification error and disparity measures being linear in the classifier, the generalised Neyman–Pearson lemma unlocks $f^{\star}_{\mathrm{D},\delta}$, a closed-form solution to the $\delta$-fair Bayes optimal classifier.

**Theorem 2.** *Assume that $\mathrm{d}P_{a,1}/\mathrm{d}P_{a,0}$ is a continuous random variable for $a \in \{0, 1\}$. For any $\tau \in \mathbb{R}$ and a given bilinear disparity measure $\mathrm{D}$ in Definition 4, denote the classifier $g_{\mathrm{D},\tau}(x,a) = \mathbb{1}\{(\pi_{a,1} - \tau s_{\mathrm{D},a})\frac{\mathrm{d}P_{a,1}}{\mathrm{d}P_{a,0}}(x) \geq \pi_{a,0} + \tau b_{\mathrm{D},a}\}$ and $\mathrm{D}(\tau) = \mathrm{D}(g_{\mathrm{D},\tau})$. Then, for $\delta \geq 0$, the $\delta$-fair Bayes optimal classifier is $f^{\star}_{\mathrm{D},\delta} = g_{\mathrm{D},\tau^{\star}_{\mathrm{D},\delta}}$, where*

$$\tau^{\star}_{\mathrm{D},\delta} = \underset{\tau \in \mathbb{R}}{\arg\min} \left\{ |\tau| : |\mathrm{D}(\tau)| \leq \delta \right\}. \tag{3}$$

The proof of Theorem 2 is in Appendix C.2. We remark that our analysis can be easily extended to the case when $\mathrm{d}P_{a,1}/\mathrm{d}P_{a,0}$ is discontinuous, by including a randomised decision rule on the set where $(\pi_{a,1} - \tau s_{\mathrm{D},a})\mathrm{d}P_{a,1}(x)/\mathrm{d}P_{a,0} = \pi_{a,0} + \tau b_{\mathrm{D},a}$.

At a high level, Theorem 2 states that the $\delta$-fair Bayes optimal classifier is shifted from the Bayes classifier (1) by linear factors $s_{\mathrm{D},a}$ and $b_{\mathrm{D},a}$. The linear shift is ensured from the bilinearity of the disparity measure D and the shift level $\tau$ is further optimised in (3). The optimisation in (3), at the core is to minimise the misclassification error among all that satisfy the fairness constraints. The detailed form is a consequence of behaviours of the misclassification error $R(g_{\mathrm{D},\tau})$ and disparity measure $\mathrm{D}(g_{\mathrm{D},\tau})$ as functions of $\tau$. We show in Proposition 7 in Appendix C.4 that the disparity is continuous and non-increasing, while the misclassification error is non-decreasing in $|\tau|$. As a result, finding an optimal threshold $\tau$ that minimises the misclassification error reduces to minimising $|\tau|$.

For $\mathrm{D} \in \{\mathrm{DO}, \mathrm{PD}, \mathrm{DD}\}$ in Definition 3, it holds that $\pi_{a,1} - \tau^{\star}_{\mathrm{D},\delta}s_{\mathrm{D},a} > 0$ and $\pi_{a,0} + \tau^{\star}_{\mathrm{D},\delta}b_{\mathrm{D},a} > 0$ (see Lemma 9 in Appendix C.4). Consequently, we rewrite the fair Bayes-optimal classifier $f^{\star}_{\mathrm{D},\delta}$ as

$$f^{\star}_{\mathrm{D},\delta}(x,a) = \mathbb{1}\left\{ \frac{\mathrm{d}P_{a,1}}{\mathrm{d}P_{a,0}}(x) \geq \frac{\pi_{a,0} + \tau^{\star}_{\mathrm{D},\delta}b_{\mathrm{D},a}}{\pi_{a,1} - \tau^{\star}_{\mathrm{D},\delta}s_{\mathrm{D},a}} \right\}. \tag{4}$$

Compared to the Bayes classifier without fairness constraints in (1), mitigating disparity is achieved by adjusting the classification thresholds, from $\pi_{a,0}/\pi_{a,1}$ to $(\pi_{a,0} + \tau_{D,\delta}^{\star} b_{D,a})/(\pi_{a,1} - \tau_{D,\delta}^{\star} s_{D,a})$, with the shift determined by the chosen disparity measure and the underlying population distributions. When $\tau_{D,\delta}^{\star} = 0$, i.e. $|D(0)| \leq \delta$, we recover the Bayes classifier $f^{\star}(x, a)$ in (1), which is, in this case, automatically fair. We write the classifier in (1) as $f_{D,\infty}^{\star}$ in the rest of the paper.

*Remark* 1. Our framework is inspired by the one in Zeng et al. (2024a), which is also an application of the generalised Neyman–Pearson lemma. The key difference between Zeng et al. (2024a) and Theorem 2 lies in the generalisation of the classifier to a functional feature space. When moving to infinite-dimensional spaces, the absence of a default base measure leads to the use of the Radon–Nikodym derivative $dP_{a,1}(x)/dP_{a,0}$. It is a more natural functional used for functional classification, rather than posterior probabilities $\mathbb{P}(Y = 1|A = a, X = x)$ considered in Zeng et al. (2024a). In particular, in important cases such as when functions are Gaussian processes, $dP_{a,1}(x)/dP_{a,0}$ is analytically tractable but not the posterior. In the cases when $dP_{a,1}(x)/dP_{a,0}$ is not tractable, there are ample tools for its approximation (e.g. Bongiorno and Goia, 2016; Dai et al., 2017).

## 2.3 Fair functional linear discriminant analysis classifier for Gaussian processes

As a concrete and important example, we focus on a specific setting when the functional features are Gaussian processes. We propose the Fair Functional Linear Discriminant Analysis classifier in Algorithm 1, featuring a plug-in estimator built on $f_{D,\delta}^{\star}$ in Theorem 2 and, specifically, in (4).

Additionally to the problem setup in Section 2.1, for any collection of $(X, A, Y) \in \mathcal{D}$, we assume that the functional feature $X$ is a Gaussian process, i.e. $\{X|A = a, Y = y\} \sim \mathcal{GP}(\mu_{a,y}, K_{a,y})$, where $\mu_{a,y}(t) = \mathbb{E}\{X(t)|A = a, Y = y\}$ and $K_{a,y}(s,t) = \mathbb{E}[\{X(s) - \mu_{a,y}(s)\}\{X(t) - \mu_{a,y}(t)\}|A = a, Y = y]$ are mean and covariance functions, $s, t \in [0, 1]$ and $a, y \in \{0, 1\}$. For simplicity, we consider a homoscedastic setting within each group, i.e. $K_{a,0} = K_{a,1} = K_a$. The covariance function $K_a$, consequently, admits the spectral expansion $K_a(s,t) = \sum_{j=1}^{\infty} \lambda_{a,j} \phi_{a,j}(s) \phi_{a,j}(t)$, where $\lambda_{a,1} \geq \lambda_{a,2} \geq \cdots \geq 0$ are eigenvalues and $\{\phi_{a,j}\}_{j \in \mathbb{N}_+}$ are eigenfunctions. Without fairness constraints, the functional linear discriminant analysis (FLDA) classifier in (1) is known to be optimal to minimise the misclassification error (e.g. Berrendero et al., 2018).

To account for fairness and construct a plug-in type classifier for $f_{D,\delta}^{\star}$, it is essential to evaluate the Radon–Nikodym derivative $dP_{a,1}/dP_{a,0}$, which plays a central role in the decision rule and analytically available under Gaussian settings. By standard results of Gaussian measures (e.g. Theorem 1 in Berrendero et al., 2018), the distributions $P_{a,0}$ and $P_{a,1}$ are mutually continuous if and only if the mean difference $\mu_{a,1} - \mu_{a,0}$ belongs to the RKHS space $\mathcal{H}(K_a)$. It then holds that $\frac{dP_{a,1}}{dP_{a,0}}(X) = \exp\{\sum_{j=1}^{\infty} \frac{(\zeta_{a,j} - \theta_{a,0,j})(\theta_{a,1,j} - \theta_{a,0,j})}{\lambda_{a,j}} - \frac{1}{2}\sum_{j=1}^{\infty} \frac{(\theta_{a,1,j} - \theta_{a,0,j})^2}{\lambda_{a,j}}\}$, where $\zeta_{a,j} = \langle X, \phi_{a,j}\rangle_{L^2}$ is the principal component scores of $X$ and $\theta_{a,y,j} = \langle \mu_{a,y}, \phi_{a,j}\rangle_{L^2}$ are the coefficients of the mean functions projected onto the eigenfunctions of $K_a$.

To estimate $dP_{a,1}/dP_{a,0}$ in practice, we assume the availability of an additional training dataset, $\widetilde{\mathcal{D}} = \{(\widetilde{X}_i, \widetilde{A}_i, \widetilde{Y}_i)\}$, which is drawn independently from the same distribution as $\mathcal{D}$ and used to estimate $\widehat{\eta}_a$ in the initial classifier. We refer $\mathcal{D}$ as the calibration data, subsequently used to post-process the initial classifier by selecting the adjusted threshold $\widehat{\tau}_{D,\delta}$.

We decompose the calibration data as $\mathcal{D} = \mathcal{D}_{0,1} \cup \mathcal{D}_{0,0} \cup \mathcal{D}_{1,1} \cup \mathcal{D}_{1,0}$, where for $a, y \in \{0, 1\}$, let $\mathcal{D}_{a,y} = \{(X_i, A_i = a, Y_i = y)\}$, $n_{a,y} = |\mathcal{D}_{a,y}|$ and $n = \sum_{a,y} n_{a,y}$. For notational clarity, we denote the $i$-th feature in $\mathcal{D}_{a,y}$ as $X_{a,y}^i$ for $i \in [n_{a,y}]$. The notation for $\widetilde{\mathcal{D}}$ follows similarly. The resulting classifier is detailed in Algorithm 1, with its theoretical guarantees discussed in Section 3.

*Remark* 2 (Perfect classification). For functional classification without fairness, the intrinsic infinite-dimensional nature of functional data gives rise to vanishing misclassification errors under certain scenarios. This is first discussed in Delaigle and Hall (2012) and known as perfect classification in the existing literature. As discussed in Berrendero et al. (2018), for homogeneous Gaussian processes, perfect classification arises when the class distributions $P_{a,1}$ and $P_{a,0}$ are mutually singular, i.e. $\mu_{a,1} - \mu_{a,0} \notin \mathcal{H}(K_a)$, in which case the Radon–Nikodym derivative $dP_{a,1}/dP_{a,0}$ does not exist. In this paper, we restrict our theoretical analysis to the more challenging regime of imperfect classification with non-vanishing classification error. Notably, in the perfect classification regime, the optimal FLDA classifier is automatically fair when the disparity measure $D \in \{DO, PD\}$. When

---

**Algorithm 1** Fair Functional Linear Discriminant Analysis classifier.

---

**INPUT:** Training data $\widetilde{\mathcal{D}}_{0,1} \cup \widetilde{\mathcal{D}}_{0,0} \cup \widetilde{\mathcal{D}}_{1,1} \cup \widetilde{\mathcal{D}}_{1,0}$, calibration data $\mathcal{D}_{0,0} \cup \mathcal{D}_{0,1} \cup \mathcal{D}_{1,0} \cup \mathcal{D}_{1,1}$, disparity level $\delta$, level of truncation $J$.

    **S1.** Estimating Radon–Nikodym derivatives $\eta_a$ and class probabilities $\pi_{a,y}$ using training data.

1:    For $a, y \in \{0, 1\}$, calculate $\widehat{\pi}_{a,y} = \frac{\widetilde{n}_{a,y}}{\widetilde{n}}$, $\widehat{\mu}_{a,y}(t) = \frac{1}{\widetilde{n}_{a,y}} \sum_{i=1}^{\widetilde{n}_{a,y}} \widetilde{X}_{a,y}^i(t)$ and $\widehat{K}_a(s,t) = \sum_{y \in \{0,1\}} \frac{\widetilde{n}_{a,y}}{\widetilde{n}_{a,0} + \widetilde{n}_{a,1}} \frac{1}{\widetilde{n}_{a,y}-1} \sum_{i=1}^{\widetilde{n}_{a,y}} \{\widetilde{X}_{a,y}^i(s) - \widehat{\mu}_{a,y}(s)\}\{\widetilde{X}_{a,y}^i(t) - \widehat{\mu}_{a,y}(t)\}$

2:    Estimate eigenvalues $\{\widehat{\lambda}_{a,j}\}_{j=1}^J$, eigenfunctions $\{\widehat{\phi}_{a,j}\}_{j=1}^J$ of $\widehat{K}_a$ by spectral expansion.

3:    Estimate projection coefficients $\widehat{\theta}_{a,y,j} = \int_0^1 \widehat{\mu}_{a,y}(t)\widehat{\phi}_{a,j}(t)\,\mathrm{d}t$. For any function $X$, denote $\widehat{\eta}_a(X) = \exp\{\sum_{j=1}^J \frac{(\widehat{\theta}_{a,1,j}-\widehat{\theta}_{a,0,j})(\int_0^1 X(t)\widehat{\phi}_{a,j}(t)\,\mathrm{d}t - \widehat{\theta}_{a,0,j})}{\widehat{\lambda}_{a,j}} - \frac{1}{2}\sum_{j=1}^J \frac{(\widehat{\theta}_{a,1,j}-\widehat{\theta}_{a,0,j})^2}{\widehat{\lambda}_{a,j}}\}$.

    **S2.** Estimating the optimal threshold using calibration data.

4:    Let $\widehat{g}_{\mathrm{D},\tau}(x,a) = \mathbb{1}\big\{(\widehat{\pi}_{a,1} - \tau s_{\mathrm{D},a})\widehat{\eta}_a(x) > \widehat{\pi}_{a,0} + \tau b_{\mathrm{D},a}\big\}$.

5:    Calculate $\widehat{\mathrm{D}}(\tau) = \sum_{a \in \{0,1\}}\{\int_{\mathcal{X}} \widehat{g}_{\mathrm{D},\tau}(x,a)s_{\mathrm{D},a}\,\mathrm{d}\widehat{P}_{a,1}(x) + \int_{\mathcal{X}} \widehat{g}_{\mathrm{D},\tau}(x,a)b_{\mathrm{D},a}\,\mathrm{d}\widehat{P}_{a,0}(x)\}$, where $\int_{\mathcal{X}} f(x,a)\,\mathrm{d}\widehat{P}_{a,y} = n_{a,y}^{-1} \sum_{i=1}^{n_{a,y}} f(X_{a,y}^i, a)$. Set $\widehat{\tau}_{\mathrm{D},\delta} = \arg\min_{\tau \in \mathbb{R}}\{|\tau| : |\widehat{\mathrm{D}}(\tau)| \leq \delta\}$.

**OUTPUT:** $\widehat{f}_{\mathrm{D},\delta}(x,a)$, with $\widehat{f}_{\mathrm{D},\delta}(x,a) = \mathbb{1}\{(\widehat{\pi}_{a,1} - \widehat{\tau}_{\mathrm{D},\delta}s_{\mathrm{D},a})\widehat{\eta}_a(x) > \widehat{\pi}_{a,0} + \widehat{\tau}_{\mathrm{D},\delta}b_{\mathrm{D},a}\}$.

---

$\mathrm{D} = \mathrm{DD}$, (1) is automatically fair if $|\mathbb{P}(Y = 1|A = 1) - \mathbb{P}(Y = 1|A = 0)| \leq \delta$. Further insights into the phenomenon of automatic fair are supported by numerical experiments in Appendix A.4.

## 3 Theoretical Properties

For a general bilinear disparity measure (Definition 4), we provide the theoretical guarantees on the fairness and excess risk control of the Fair-FLDA algorithm (Algorithm 1) in Theorems 3 and 5, with a special case regarding the disparity of opportunity in Corollary 6. We start with assumptions.

**Assumption 1** (Class probabilities). *Assume that there exist absolute constants $0 < C_p \leq C_p' < 1$, such that the class probabilities satisfy $0 < C_p \leq \pi_{a,y} \leq C_p' < 1$, $a, y \in \{0, 1\}$.*

**Assumption 2** (Gaussian processes). *Assume that the standard features $\{\widetilde{X}_{a,y}^i\}_{i \in [\widetilde{n}_{a,y}]} \cup \{X_{a,y}^i\}_{i \in [n_{a,y}]}$ are collections of Gaussian processes $\mathcal{GP}(\mu_{a,y}, K_a)$ with continuous trajectories, $a, y \in \{0, 1\}$. In addition, assume the following holds for any $a \in \{0, 1\}$: **a.** (Covariance function) The covariance function $K_a$ is continuous and there exist absolute constants $C_\lambda, C_\lambda' > 0$ such that the eigenvalues of the covariance operator are decreasing with $j^{-\alpha} \geq C_\lambda \lambda_{a,j} \geq C_\lambda' \lambda_{a,j+1} + j^{-\alpha-1}$ for $\alpha > 1$ and $j \in \mathbb{N}_+$; **b.** (Signal-to-noise ratio) There is an absolute constant $C_K > 0$ such that $\|\mu_{a,1} - \mu_{a,0}\|_{K_a}^2 \geq C_K$; and **c.** (Mean difference) There exists a constant $C_\mu > 0$ such that for any $j \in \mathbb{N}_+$, it holds that $|\langle \mu_{a,1} - \mu_{a,0}, \phi_{a,j}\rangle_{L^2}| \leq C_\mu j^{-\beta}$ with $\beta > (\alpha + 1)/2$.*

In Assumption 1, we assume that the class probabilities are bounded away from 0 and 1, essential to ensure that a sufficient number of samples for each group can be observed. Assumption 2 characterises the properties of the functional features. Assumption 2a specifies the decaying rate of eigenvalues and quantifies the spacings between two consecutive eigenvalues. This is commonly seen in the FDA literature (e.g. Hall and Horowitz, 2007; Dou et al., 2012) involving eigenfunction estimations. Assumption 2b imposes a lower bound on the magnitude of the signal-to-noise ratio $\|\mu_{a,1} - \mu_{a,0}\|_{K_a}^2$. We thus exclude the trivial classification regime and preclude the classifier from degenerating into random guessing. Assumption 2c enforces the alignment between the mean difference and eigenspace, with larger values of $\beta$ indicating better alignment. Assumptions 2a, 2b and 2c jointly imply that $\|\mu_{a,1} - \mu_{a,0}\|_{K_a}^2 \asymp 1$ with the tail sum satisfying $\sum_{j=J+1}^\infty (\theta_{a,1,j} - \theta_{a,0,j})^2/\lambda_j \lesssim J^{\alpha-2\beta+1}$, $J \in \mathbb{N}_+$. Note that decay rates $\alpha$ and $\beta$ are assumed to be invariant for $a \in \{0, 1\}$, but this can be easily generalised to different decaying rates between groups.

**Theorem 3** (Fairness guarantee). *For any $\delta > 0$ and bilinear disparity measure $\mathrm{D}$ in Definition 4, let $\widehat{f}_{\mathrm{D},\delta}$ denote the classifier output by Algorithm 1, we have, for any $\eta \in (0, 1/2)$, that $\mathbb{P}\{|\mathrm{D}(\widehat{f}_{\mathrm{D},\delta})| \leq \delta + C\sqrt{\log(1/\eta)/n}\} \geq 1 - \eta$, where $C > 0$ is an absolute constant.*

Theorem 3 is proved in Appendix D and shows that with probability at least $1 - \eta$, after calibration step, the disparity level of $\widehat{f}_{\mathrm{D},\delta}$ does not exceed the pre-specified level $\delta$ by a small offset term up to $O(\sqrt{\log(1/\eta)/n})$. The magnitude is quantified by the high probability upper bound on $|\widehat{\mathrm{D}}(\widehat{\tau}_{\mathrm{D},\delta}) - \mathrm{D}(\widehat{\tau}_{\mathrm{D},\delta})|$, measuring the deviation of the empirical distribution from its population counterpart.

If insisting on controlling the population unfairness $\mathrm{D}(\widehat{f}_{\mathrm{D},\delta})$ below $\delta$, then provided that $\sqrt{\log(1/\eta)/n} \lesssim \delta$, it suffices to adjust the input $\delta$ to $\delta - C\sqrt{\log(1/\eta)/n}$, i.e. set $\widehat{\tau}_{\mathrm{D},\delta} = \arg\min_{\tau \in \mathbb{R}}\{|\tau| : |\widehat{\mathrm{D}}(\tau)| \leq \delta - C\sqrt{\log(1/\eta)/n}\}$ in **S2** of Algorithm 1. We refer to the resulting method as the Fair Functional Linear Discriminant Analysis classifier with Calibration (Fair-FLDA$_c$). Numerical results comparing the performances are presented in Section 4.

To present the excess risk control of our method, as a preliminary step, we establish the misclassification error of the oracle classifier in Proposition 4.

**Proposition 4** (Misclassification error). *Under the model setup in Section 2.3, for any $\delta \geq 0$ and bilinear disparity measure $\mathrm{D}$ such that $\pi_{a,1} - \tau_{\mathrm{D},\delta}^{\star}s_{\mathrm{D},a} > 0$ and $\pi_{a,0} + \tau_{\mathrm{D},\delta}^{\star}b_{\mathrm{D},a} > 0$, the corresponding misclassification error for $f_{\mathrm{D},\delta}^{\star}$ defined in Theorem 2 is given by $R(f_{\mathrm{D},\delta}^{\star}) = \sum_{a \in \{0,1\}} \pi_{a,0}\Phi[-\frac{\|\mu_{a,1}-\mu_{a,0}\|_{K_a}}{2} - \frac{\log\{(\pi_{a,0}+\tau_{\mathrm{D},\delta}^{\star}b_{\mathrm{D},a})/(\pi_{a,1}-\tau_{\mathrm{D},\delta}^{\star}s_{\mathrm{D},a})\}}{\|\mu_{a,1}-\mu_{a,0}\|_{K_a}}] + \sum_{a \in \{0,1\}} \pi_{a,1}\Phi[-\frac{\|\mu_{a,1}-\mu_{a,0}\|_{K_a}}{2} + \frac{\log\{(\pi_{a,0}+\tau_{\mathrm{D},\delta}^{\star}b_{\mathrm{D},a})/(\pi_{a,1}-\tau_{\mathrm{D},\delta}^{\star}s_{\mathrm{D},a})\}}{\|\mu_{a,1}-\mu_{a,0}\|_{K_a}}]$, where $\Phi$ is the cumulative distribution function of the standard normal distribution.*

See Appendix C.3 for the proof of Proposition 4. Compared to the standard excess risk without fairness constraints, $R(f_{\mathrm{D},\infty}^{\star})$ (e.g. Theorem 2 in Berrendero et al., 2018), the cost of fairness constraints is quantified by the logarithmic term involving $\tau_{\mathrm{D},\delta}^{\star}$, resulted from the adjusted group-wise thresholds in (4). As shown in Proposition 7 in Appendix C.4, a decrease in $\delta$, i.e. stronger fairness constraints, leads to an increase in $|\tau_{\mathrm{D},\delta}^{\star}|$, hence a larger misclassification error $R(f_{\mathrm{D},\delta}^{\star})$. When $f_{\mathrm{D},\infty}^{*}$ is automatically fair, i.e. $\tau_{\mathrm{D},\delta}^{*} = 0$, Proposition 4 recovers Theorem 2 in Berrendero et al. (2018).

Moving towards excess risk controls, as discussed in Zeng et al. (2024b), for any $f \in \mathcal{F}$, the traditional excess risk $R(f) - R(f_{\mathrm{D},\delta}^{\star})$ may be negative as $f_{\mathrm{D},\delta}^{\star}$ does not necessarily minimise the excess risk. To make a meaningful control of the misclassification error, we resort to the quantity $|R(f) - R(f_{\mathrm{D},\delta}^{\star})|$, which can be further decomposed as the non-negative fairness-aware excess risk $d_E(f, f_{\mathrm{D},\delta}^{\star})$ (defined in Definition 5) and a disparity cost $\tau_{\mathrm{D},\delta}^{\star}\{\mathrm{D}(f_{\mathrm{D},\delta}^{\star}) - \mathrm{D}(f)\}$, that $|R(f) - R(f_{\mathrm{D},\delta}^{\star})| = |d_E(f, f_{\mathrm{D},\delta}^{\star}) + \tau_{\mathrm{D},\delta}^{\star}\{\mathrm{D}(f_{\mathrm{D},\delta}^{\star}) - \mathrm{D}(f)\}| \leq d_E(f, f_{\mathrm{D},\delta}^{\star}) + |\tau_{\mathrm{D},\delta}^{\star}||\mathrm{D}(f_{\mathrm{D},\delta}^{\star}) - \mathrm{D}(f)|$. The derivation above is directly via the definition below.

**Definition 5** (Fairness-aware excess risk). *For $\delta \geq 0$, let $f_{\mathrm{D},\delta}^{\star}$ be a $\delta$-fair Bayes optimal classifier defined in (2), recalling $\tau_{\mathrm{D},\delta}^{\star}$ in (3) and $s_{\mathrm{D},a}$, $b_{\mathrm{D},a}$ in Definition 4. For any classifier $f : \mathcal{X} \times \{0,1\} \to [0,1]$, define the fairness-aware excess risk as $d_E(f, f_{\mathrm{D},\delta}^{\star}) = \sum_{a \in \{0,1\}} \int_{\mathcal{X}}\{f(x,a) - f_{\mathrm{D},\delta}^{\star}(x,a)\}[(\pi_{a,0} + \tau_{\mathrm{D},\delta}^{\star}b_{\mathrm{D},a}) - (\pi_{a,1} - \tau_{\mathrm{D},\delta}^{\star}s_{\mathrm{D},a})\frac{\mathrm{d}P_{a,1}}{\mathrm{d}P_{a,0}}(x)] \, \mathrm{d}P_{a,0}(x).$*

We are now ready to present the excess risk control for Algorithm 1 in Theorem 5.

**Theorem 5.** *Denote $\epsilon_{\pi}$, $\epsilon_{\eta}$ and $\epsilon_{\mathrm{D}}$ the estimation error related to $\widehat{\pi}_{a,y}$, $\widehat{\eta}_a$ and $\widehat{\mathrm{D}}$, i.e. for any small $\eta \in (0, 1/2)$, $a, y \in \{0,1\}$ and $X \sim \mathcal{GP}(\mu_{a,y}, K_a)$, it holds with probability at least $1 - \eta/3$ that $|\widehat{\pi}_{a,y} - \pi_{a,y}| \leq \epsilon_{\pi}$, $|\log\{\widehat{\eta}_a(X)\} - \log\{\eta_a(X)\}| \leq \epsilon_{\eta}$ and $\sup_{\tau \in \mathbb{R}}|\widehat{\mathrm{D}}(\tau) - \mathrm{D}(\tau)| \leq \epsilon_{\mathrm{D}}$.*

*Suppose that Assumptions 1 and 2 hold; and for $\delta \geq 0$, bilinear disparity measure $\mathrm{D}$ (Definition 4), satisfying that (i) $\mathrm{D}(0) \notin (\delta - \epsilon_{\mathrm{D}}, \delta] \cup [-\delta, -\delta + \epsilon_{\mathrm{D}})$, (ii) $\pi_{a,1} - \tau_{\mathrm{D},\delta}^{\star}s_{\mathrm{D},a} \geq c_1$, $\pi_{a,0} + \tau_{\mathrm{D},\delta}^{\star}b_{\mathrm{D},a} \geq c_1$, with $\max\{|s_{\mathrm{D},a}|, |b_{\mathrm{D},a}|\} \leq c_2$, and (iii) $|\mathrm{D}(\tau_{\mathrm{D},\delta}^{\star}) - \mathrm{D}(\tau_{\mathrm{D},\delta}^{\star} + \xi)| \geq C_{\mathrm{D}}|\xi|^{\frac{1}{\gamma}}$ for any $\xi$ in the small neighbourhood of $0$ and some $\gamma > 0$, where $c_1, c_2, C_{\mathrm{D}} > 0$ are absolute constants.*

*Then, it holds with probability at least $1 - \eta$, $\eta \in (0, \epsilon_{\pi} + \epsilon_{\eta} + \epsilon_{\tau})$, that the classifier $\widehat{f}_{\mathrm{D},\delta}$ output by Algorithm 1 satisfies that*

$$|R(\widehat{f}_{\mathrm{D},\delta}) - R(f_{\mathrm{D},\delta}^{\star})| \lesssim d_E(f, f_{\mathrm{D},\delta}^{\star}) + |\tau_{\mathrm{D},\delta}^{\star}|\sqrt{\log(1/\eta)/n}, \tag{5}$$

*where $d_E(\widehat{f}_{\mathrm{D},\delta}, f_{\mathrm{D},\delta}^{\star}) \lesssim (\epsilon_{\pi} + \epsilon_{\eta} + \epsilon_{\tau})^2$ with $\epsilon_{\tau} = |\widehat{\tau}_{\mathrm{D},\delta} - \tau_{\mathrm{D},\delta}^{\star}| \lesssim \epsilon_{\mathrm{D}}^{\gamma}\mathbb{1}\{\tau_{\mathrm{D},\delta}^{\star} \neq 0\}$.*

Theorem 5 provides a general characterisation of fairness-aware excess risk and traditional excess risk when $\mathrm{D}(0)$ is not too close to $\delta$, i.e. when $f_{\mathrm{D},\infty}^\star$ is either sufficiently fair or unfair. We impose two extra assumptions on D, with the first one controlling the behaviours of $\tau_{\mathrm{D},\delta}^\star$ near the boundaries. This condition is used to ensure that $\widehat{\tau}_{\mathrm{D},\delta}$ lies within the range of its estimated counterparts, i.e. $\widehat{\pi}_{a,1} - \widehat{\tau}_{\mathrm{D},\delta} s_{\mathrm{D},a} > 0$ and $\widehat{\pi}_{a,0} + \widehat{\tau}_{\mathrm{D},\delta} b_{\mathrm{D},a} > 0$. This guarantees that the misclassification error $R(\widehat{f}_{\mathrm{D},\delta})$ retains a well-structured form, analogous to that in Proposition 4. The second assumption controls the steepness of D in a small neighbourhood of $\tau_{\mathrm{D},\delta}^\star$, with larger values of $\gamma$ corresponding to greater steepness. In the fair-impacted case, i.e. $\tau_{\mathrm{D},\delta}^\star \neq 0$, if D is potentially very flat near $\tau_{\mathrm{D},\delta}^\star$, i.e. $\gamma$ is small, the estimation problem becomes more challenging, resulting in a larger estimation error $\epsilon_\tau$ (as illustrated in Figure 1). We remark that when D is explicitly given, both of the above assumptions can be verified in most of the cases, see Corollary 6 as an example.

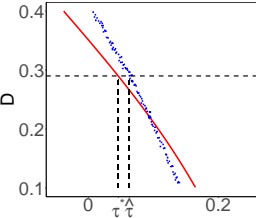 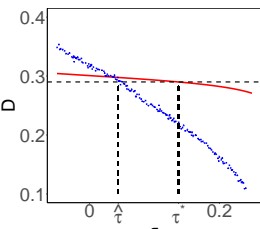

Figure 1: Effects of steepness of disparity levels on the estimation error of $\tau_{\mathrm{D},\delta}^\star$. Left and right panels illustrate steep and flat $\mathrm{D}(\cdot)$. Red solid line: $\mathrm{D}(\cdot)$. Blue dotted line: $\widehat{\mathrm{D}}(\cdot)$.

The disparity cost in the right-hand side of (5) is a direct consequence of the fairness guarantees in Theorem 3. The fairness-aware excess risk, $d_E(\widehat{f}_{\mathrm{D},\delta}, f_{\mathrm{D},\delta}^\star)$, consists of two components, with $(\epsilon_\pi + \epsilon_\eta)^2$ capturing the intrinsic excess risk and $\epsilon_{\mathrm{D}}^{2\gamma}$ reflecting the cost of fairness. This is the first time seen in the FDA literature, echoing the same spirit in the finite-dimensional classification literature (Zeng et al., 2024b; Hou and Zhang, 2024). When D is chosen as one of the disparity measures in Definition 3, $\epsilon_{\mathrm{D}}$ can be explicitly controlled as $\epsilon_{\mathrm{D}} \asymp \epsilon_\pi + \epsilon_\eta + \left(\frac{\log(1/\eta)}{n}\right)^{\frac{1}{2}}$.

To further illustrate Theorem 5, we apply the framework when $\mathrm{D} = \mathrm{DO}$, the disparity of opportunity. The corresponding results up to poly-logarithmic factors are summarised in Corollary 6.

**Corollary 6** (Excess risk under DO). *Under Assumptions 1 and 2, for any $\delta \geq 0$, $\beta \geq \frac{3\alpha+2}{2}$, it holds for any $J$ satisfying $J^{2\alpha+2} \lesssim \widetilde{n}$ that $d_E(\widehat{f}_{\mathrm{DO},\delta}, f_{\mathrm{DO},\delta}^\star) = O_{\mathrm{p}}(\frac{J}{\widetilde{n}} + J^{\alpha-2\beta+1} + \frac{\mathbb{1}\{\tau_{D,\delta}^\star \neq 0\}}{n})$, if we additionally assume that $\mathrm{DO}(0) \notin (\delta - (\frac{J}{\widetilde{n}} \vee J^{\alpha-2\beta+1} \vee \frac{1}{n})^{\frac{1}{2}}, \delta] \cup [-\delta, -\delta + (\frac{J}{\widetilde{n}} \vee J^{\alpha-2\beta+1} \vee \frac{1}{n})^{\frac{1}{2}})$. Additionally, if we further assume $n \asymp \widetilde{n} \asymp N$ and pick $J \asymp N^{\frac{1}{2\beta-\alpha}}$, it holds $d_E(\widehat{f}_{\mathrm{DO},\delta}, f_{\mathrm{DO},\delta}^\star) = O_{\mathrm{p}}(N^{\frac{\alpha-2\beta+1}{2\beta-\alpha}})$ and $|R(\widehat{f}_{\mathrm{DO},\delta}) - R(f_{\mathrm{DO},\delta}^\star)| \leq d_E(\widehat{f}_{\mathrm{DO},\delta}, f_{\mathrm{DO},\delta}^\star) + |\tau_{\mathrm{DO},\delta}^\star| O_{\mathrm{p}}(N^{-\frac{1}{2}})$.*

Corollary 6 is a shorter version of Corollary 11 in Appendix E.2, presenting only the case when $\beta \geq \frac{3\alpha+2}{2}$. Detailed results for the case when $\frac{\alpha+1}{2} < \beta \leq \frac{3\alpha+2}{2}$ are deferred to Corollary 11.

At a high level, Corollary 6 shows the upper bound is of the form : $d_E(\widehat{f}_{\mathrm{DO},\delta}, f_{\mathrm{DO},\delta}^\star) \leq$ variance to estimate $\eta_a$ + squared bias due to truncation + cost of fairness, which highlights the role of truncation parameter $J$ in determining the final convergence rate through the underlying bias-variance trade-off. Moreover, shown in Corollary 11, as the mean difference aligns more strongly with the eigenspace, i.e. as $\beta$ increases, a smaller estimation variance is observed. Note that the cost of fairness is masked when $n \asymp \widetilde{n} \asymp N$, which shares the same finding as Hou and Zhang (2024).

Corollary 6 is the first time providing finite-sample guarantees for functional classification under fairness constraints. There is no existing work in FDA with fairness constraints, and even for the FDA work without fairness, such detailed characterisation is novel in the literature. Without any predecessors regarding the former, we list some relevant works with the latter point.

Excess risk control for functional classification has previously been considered in Meister (2016) and Wang et al. (2021), with the former relying on smoothness assumptions of functional densities and decay rates of the metric entropy of functional space, and the latter assuming that eigenfunctions

of the covariance operator are explicitly known. Instead, our result in Equation (5) is established requiring only weak structural assumptions on the eigenspace. When $\tau^\star_{D,\delta} = 0$, our result in Corollary 6 recovers the excess risk rate established in Wang et al. (2021), provided that $J^{2\alpha+2} \lesssim \widetilde{n}$. We would like to remark that the upper bound on $J$ arises from estimating eigenfunctions; similar condition is also in Hall and Horowitz (2007) and Dou et al. (2012). Another relevant work is Cai and Zhang (2019), studying the linear discriminant analysis (LDA) classifier for $J$-dimensional Gaussian random variables. Our key variance term $\frac{J}{\widetilde{n}}$, resulting from estimating the first $J$ leading terms in $\eta_a$, aligns with the minimax optimal result derived in Cai and Zhang (2019). At a high level, $f^\star_{D,\delta}$ can be viewed as applying LDA to an infinite number of principal component scores. By focusing only on the first $J$ scores in Fair-FLDA, we recover the variance term in $J$-dimensional LDA.

# 4 Numerical experiments

In this section, we demonstrate some key numerical evidence via simulated and real data analysis.

**Simulation.** Generate $(Y, A) \in \{0, 1\}^{\otimes 2}$ according to the distributions $\mathbb{P}(A = 1) = 0.7, \mathbb{P}(Y = 1 | A = 0) = 0.4$ and $\mathbb{P}(Y = 1 | A = 1) = 0.7$. Given $Y = y$ and $A = a$, generate the functional covariate $X_{a,y}(t)$ as $X_{a,y}(t) = \mu_{a,y}(t) + \sum_{k=1}^{50} \zeta_{a,k} \phi_k(t)$, where $\phi_k(t) = \sqrt{2}\cos(k\pi t)$, $\zeta_{a,k} \sim N(0, \lambda_{a,k})$, $\lambda_{0,k} = k^{-2}$, $\lambda_{1,k} = 2k^{-2}$, and the mean functions are specified as follows,

$$\mu_{0,0} = \mu_{1,0} = 0, \ \ \mu_{0,1}(t) = \sum_{k=1}^{50} 0.8(-1)^k k^{-\beta} \phi_k(t), \ \ \mu_{1,1}(t) = \sum_{k=1}^{50} \sqrt{2}(-1)^k k^{-\beta} \phi_k(t).$$

Let $\beta = 1.5$ and $n = 1000$. To conserve space, we defer detailed settings, implementation details, more numerical results on effects of sample sizes ($n$), alignment of mean difference ($\beta$), model misspecification (non-Gaussianity) and perfect classification, and comparisons with multivariate baselines to Appendix A.

Figure 2 reports the medians across 500 times of the classification error and disparity measures of our proposed methods Fair-FLDA and its calibrated version Fair-FLDA$_c$ (in Section 3), with the classical functional linear discriminant classifier (FLDA) and oracle Bayes classifiers as competitors.

The classification errors of Fair-LDA and Fair-FLDA$_c$ exhibit a non-increasing trend as $\delta$ grows, mirroring the behaviour of the oracle Bayes classifier. The error of FLDA remains constant across different values of $\delta$, as it does not incorporate any fairness constraint. When $\delta$ is small, corresponding to fair-impacted regimes, the classification errors of our methods decrease with increasing $\delta$, due to the less stringent fairness constraints. Once $\delta$ exceeds a certain threshold, the fairness constraint becomes inactive, and the classification errors of the fairness-aware classifiers converge to that of the unconstrained FLDA. As for the disparity control, the FLDA consistently fails to meet fairness requirements, but the Fair-FLDA maintains the desired median disparity level and the Fair-FLDA$_c$ typically achieves median disparity levels strictly below $\delta$. As established in Theorem 3, the empirical 95% quantile of disparity of Fair-FLDA may slightly exceed $\delta$, whereas the Fair-FLDA$_c$ effectively corrects for this offset, achieving probabilistic control of the disparity below $\delta$ with probability at least 95%. As $\delta$ increases beyond a critical threshold, both fairness-aware classifiers gradually reduce to the unconstrained classifier FLDA, and their corresponding disparity levels stabilise accordingly. Both Fair-FLDA and Fair-FLDA$_c$ satisfy their respective fairness criteria without significant compromise in classification accuracy.

**Real data.** For the real dataset, we use the 2005-2006 National Health and Nutrition Examination Survey data (CDC, 2006), where the sensitive attribute is race and the classification task is to determine if an individual is under 20 or over 50 years old based on the quantile function of intensity values. Results are plotted in the 4-6th columns in Figure 2, with details and additional experiments presented in Appendices A.1 and A.5. It shows that the FLDA exhibits substantial unfairness, whereas the Fair-FLDA effectively controls the median disparity. In terms of probabilistic disparity control, the 95% empirical quantile of the disparity for Fair-FLDA$_c$ slightly exceeds $\delta$ with the default choice of the calibration constant under DO. We recommend tuning the calibration parameter to achieve more reliable probabilistic disparity control (Appendix A.5). The classification errors of Fair-FLDA and Fair-FLDA$_c$ remain comparable to that of the FLDA, demonstrating that

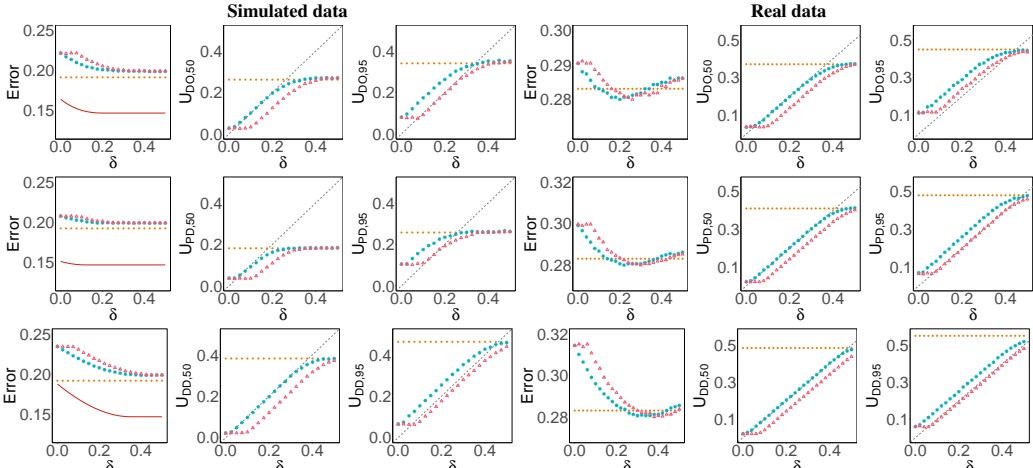

Figure 2: From left to right: medians of classification errors, medians and 95% quantiles of the disparity measures, in the simulated (1st-3rd columns) and real data (4-6th columns). Orange dots: FLDA; blue stars: Fair-FLDA; pink triangles: Fair-FLDA$_c$; red solid line: oracle Bayes classifier; grey dashed line: $y = x$.

our fairness-aware classifiers effectively mitigate unfairness while maintaining competitive classification accuracy.

## 5 Discussions

In this paper, we use the publicly available NHANES dataset as a representative data example to demonstrate the effectiveness of our method. It is worth noting that our approach is broadly applicable to a wide range of functional data classification problems where fairness is a concern. For example, the Siena Scalp EEG dataset in the PhysioNet database contains EEG recordings from male and female subjects and can serve as a natural application of our fair functional classifier, where the response variable can be defined as the occurrence of a seizure. Overall, our method offers a principled and broadly applicable tool for fair classification in functional data analysis.

We envisage several potential extensions. Firstly, our framework and Fair-FLDA algorithm depend on the availability of the sensitive features, which may be restricted in certain practical settings due to privacy concerns. When sensitive features $A$ are available during training but not available during prediction, we provide necessary steps to extend our algorithm in Appendix B. However, when $A$ is not available even during the training process, more refined methods for inferring the sensitive features would be necessary, see Appendix B for additional discussion. Secondly, in many scenarios, the Radon–Nikodym derivatives $dP_{a,1}/dP_{a,0}$ are not explicitly known and easy to work with. To address this, a natural strategy during implementation is to approximate it using the density ratios of projection scores (e.g. Bongiorno and Goia, 2016; Dai et al., 2017). Thirdly, in reality, functions can only be discretely observed over sampling grids. Investigating the effect of sparsity on the excess risk under fairness constraints remains an intriguing area.

## Acknowledgments and Disclosure of Funding

Yu's research is partially supported by the EPSRC (EP/Z531327/1) and the Leverhulme Trust (Philip Leverhulme Prize). Lin's research is partially supported by the Singapore MOE AcRF Tier 1 grant.

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

## Technical Appendices and Supplementary Material

All technical details and additional numerical results are collected in the Appendices. Detailed simulation settings and additional experimental results are collected in Appendix A. We present proofs and properties related to the Bayes optimal classifier $f_{D,\delta}^\star$ in Appendix C. The proof of Theorem 3 is collected in Appendix D, with the proofs of Theorems 5 and 6 presented in Appendix E. All results related to class probability and eigenspace estimation can be found in Appendices F and G. For completeness, necessary technical lemmas are included in Appendix H.

Throughout the appendix, with a slight abuse of notation, unless specifically stated otherwise, let $c_1, C_1, c_2, C_2, \ldots > 0$ denote absolute constants whose values may vary from place to place. For $a, b \in \mathbb{R}$, write $a \wedge b = \min\{a, b\}$. For an $\mathbb{R}$-valued random variable $X$ and $k \in [2]$, let $\|X\|_{\psi_k}$ denote the Orlicz-$\psi_k$ norm, i.e. $\|X\|_{\psi_k} = \inf\{t > 0 : \mathbb{E}[\exp(\{|X|/t\}^k)] \le 2\}$.

## A    Further details of numerical experiments

The figure labels are consistent with those used in Figure 2 in the main text, unless otherwise specified. We implemented all methods in R (version 4.3.1). Experiments were conducted on a server equipped with an Intel(R) Xeon(R) Platinum 8280 CPU @ 2.70GHz (28 cores) and 503GB of RAM. The source code is provided in the Supplementary Material.

### A.1    Detailed setup of Section 4

**Simulation.**    For the simulation results, we generate $(Y, A) \in \{0, 1\}^{\otimes 2}$ according to the distributions $\mathbb{P}(A = 1) = 0.7, \mathbb{P}(Y = 1|A = 0) = 0.4$ and $\mathbb{P}(Y = 1|A = 1) = 0.7$. Given $Y = y$ and $A = a$, generate the functional covariate $X_{a,y}(t)$ as $X_{a,y}(t) = \mu_{a,y}(t) + \sum_{k=1}^{50} \zeta_{a,k}\phi_k(t)$, where $\phi_k(t) = \sqrt{2}\cos(k\pi t)$, $\zeta_{a,k} \sim N(0, \lambda_{a,k})$, $\lambda_{0,k} = k^{-2}$, $\lambda_{1,k} = 2k^{-2}$, and the mean functions are specified as follows,

$$\mu_{0,0} = \mu_{1,0} = 0, \;\; \mu_{0,1}(t) = \sum_{k=1}^{50} 0.8(-1)^k k^{-\beta}\phi_k(t), \;\; \mu_{1,1}(t) = \sum_{k=1}^{50} \sqrt{2}(-1)^k k^{-\beta}\phi_k(t).$$

Let $n$ denote the size of the training sample. We implement the proposed fairness-aware classifier under two calibration settings,

- Fair-FLDA: calibration constant set to 0;
- Fair-FLDA$_c$: calibration constant set to $\min\{\sqrt{2\log(1/\rho)/n}, \delta\}$, with $\rho = 0.05$.

Truncation levels are selected via 5-fold cross-validation, specifically by minimising the average classification error associated with the unconstrained classifier.

During implementation, the training set $\mathcal{D}$ is randomly split into two equal-sized subsets, $\mathcal{D}_1 \cup \mathcal{D}_2$. One subset is used to estimate $\widehat{\eta}_a$ and $\widehat{\pi}_{a,y}$, while the other is used to estimate the threshold $\widehat{\tau}$. Let $\widehat{f}_1$ denote the classifier estimated using $\mathcal{D}_1$ for model estimation and $\mathcal{D}_2$ for threshold calibration, and let $\widehat{f}_2$ denote the classifier constructed with the roles of $\mathcal{D}_1$ and $\mathcal{D}_2$ reversed. To mitigate the randomness caused by random splitting, we adopt a cross-fitting approach and define the final probabilistic classifier as the average $\widehat{f} = (\widehat{f}_1 + \widehat{f}_2)/2$.

**Real data.**    For the real data analysis, we apply the proposed method to data from the 2005-2006 National Health and Nutrition Examination Survey (CDC, 2006). Further details about this dataset can be found in Lin et al. (2023). Following the preprocessing steps in Lin et al. (2023), we exclude observations with questionable data reliability according to NHANES protocol, remove observations with intensity values higher than 1000 or equal to 0, and retain subjects with at least 100 remaining observations. The response variable is whether an individual is under 20 or over 50 years old, with the quantile function of intensity values as the functional covariate. The sensitive attribute refers to race, categorised as non-Hispanic white and non-Hispanic black. The final dataset consists of 3252 instances, which we randomly split into equal-sized training and test subsets. The methods are implemented following the same procedures described for simulation experiments.

## A.2 Additional simulation results under Gaussian models

**Results under varying sample sizes.** We evaluate the model from Appendix A.1 under varying sample sizes. As shown in Figures 3-5, the excess risk of both Fair-FLDA and Fair-FLDA$_c$ decreases with increasing $n$. Moreover, with larger $n$, the difference between Fair-FLDA and Fair-FLDA$_c$ in disparity control becomes less significant.

**Error-unfairness trade-off.** We illustrate the error-unfairness trade-off in Figures 6-8. There is only a single point in each figure for FLDA, as it does not incorporate any fairness correction. Both Fair-FDA and Fair-FLDA$_c$ demonstrate comparable trade-offs between classification error and unfairness, since they are both derived from the Bayes optimal fair classifier.

**Results under $\beta = 2$.** We evaluate the methods under the Gaussian model with $\beta = 2$, while keeping other model parameters consistent with those in Appendix A.1. The patterns of disparity control are similar to those observed under $\beta = 1.5$. As illustrated in Figures 9-11, the classification errors are generally higher compared to the case of $\beta = 1.5$, due to the lower signal-to-noise ratio. Nonetheless, the excess risk decreases more rapidly with increasing $n$ for larger values of $\beta$, aligning well with our theoretical results.

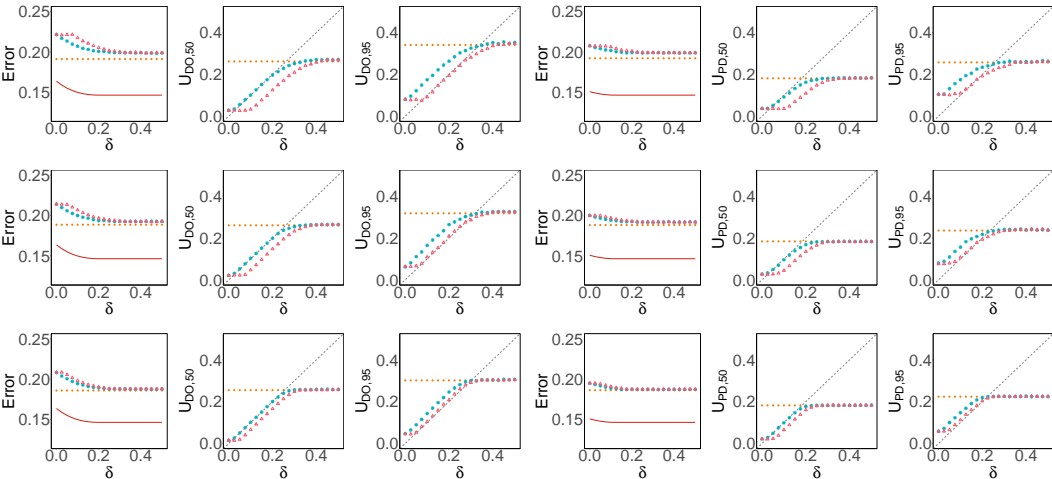

Figure 3: Disparity DO results under the Gaussian model, $\beta = 1.5$. Top: $n = 1000$; middle: $n = 2000$; bottom: $n = 5000$.

Figure 4: Disparity PD results under the Gaussian model, $\beta = 1.5$. Top: $n = 1000$; middle: $n = 2000$; bottom: $n = 5000$.

## A.3 Simulation results under non-Gaussian models

Although the proposed fairness-aware classifier is established based on the explicit form of the Radon–Nikodym derivative under Gaussian assumptions, it remains applicable in more general scenarios. However, when the Gaussian assumption is violated, the proposed classifier may no longer be Bayes optimal. To assess its performance beyond the Gaussian setting, we generate non-Gaussian stochastic processes by sampling $\zeta_{a,k} \sim \lambda_{a,k}^{1/2} \mathrm{Unif}(-\sqrt{3}, \sqrt{3})$, while keeping all other model parameters identical to those in Section A.1.

The results are presented in Figures 12-14. Despite the lack of Bayes optimality guarantees in this setting, the Fair-FLDA and Fair-FLDA$_c$ continue to exhibit effective disparity control and satisfactory classification accuracy. This robustness highlights the practical utility of our approach in more general, non-Gaussian scenarios.

## A.4 Results under perfect classification

Functional data exhibit a unique property known as perfect classification, where the classification error can vanish, a phenomenon that does not typically arise in multivariate data. To evaluate the

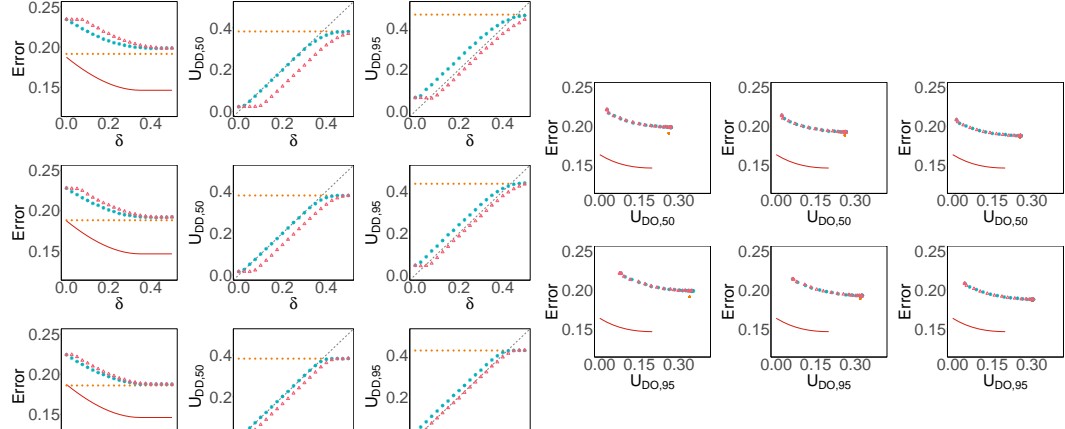

Figure 5: Disparity DD results under the Gaussian model, $\beta = 1.5$. Top: $n = 1000$; middle: $n = 2000$; bottom: $n = 5000$.

Figure 6: Error-unfairness trade-off for DO under the Gaussian model, $\beta = 1.5$. Left: $n = 1000$; middle: $n = 2000$; right: $n = 5000$.

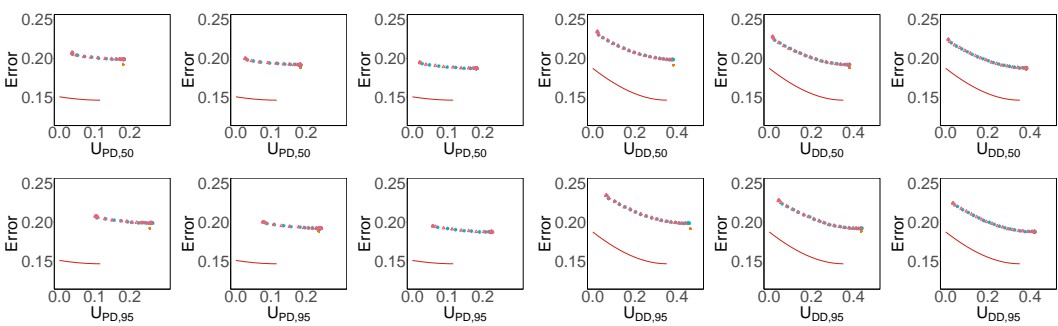

Figure 7: Error-unfairness trade-off for PD under the Gaussian model, $\beta = 1.5$. Left: $n = 1000$; middle: $n = 2000$; right: $n = 5000$.

Figure 8: Error-unfairness trade-off for DD under the Gaussian model, $\beta = 1.5$. Left: $n = 1000$; middle: $n = 2000$; right: $n = 5000$.

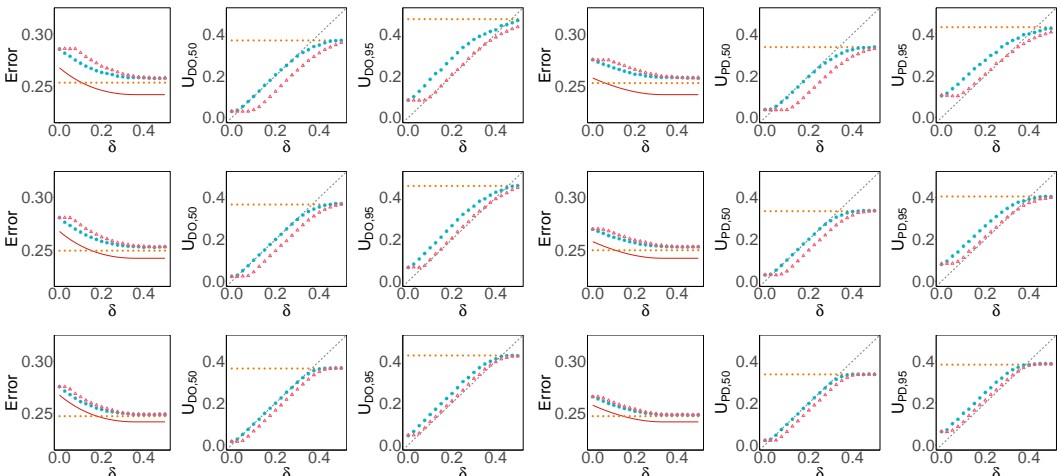

Figure 9: Disparity DO results under the Gaussian model, $\beta = 2$. Top: $n = 1000$; middle: $n = 2000$; bottom: $n = 5000$

Figure 10: Disparity PD results under the Gaussian model, $\beta = 2$. Top: $n = 1000$; middle: $n = 2000$; bottom: $n = 5000$.

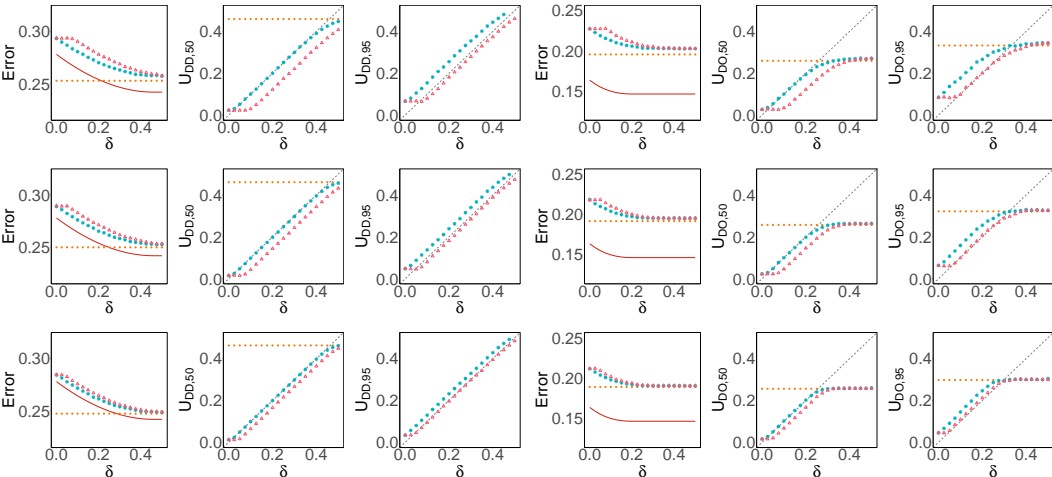

Figure 11: Disparity DD results under the Gaussian model, $\beta = 2$. Top: $n = 1000$; middle: $n = 2000$; bottom: $n = 5000$.

Figure 12: Disparity DO results under the non-Gaussian model, $\beta = 1.5$. Top: $n = 1000$; middle: $n = 2000$; bottom: $n = 5000$.

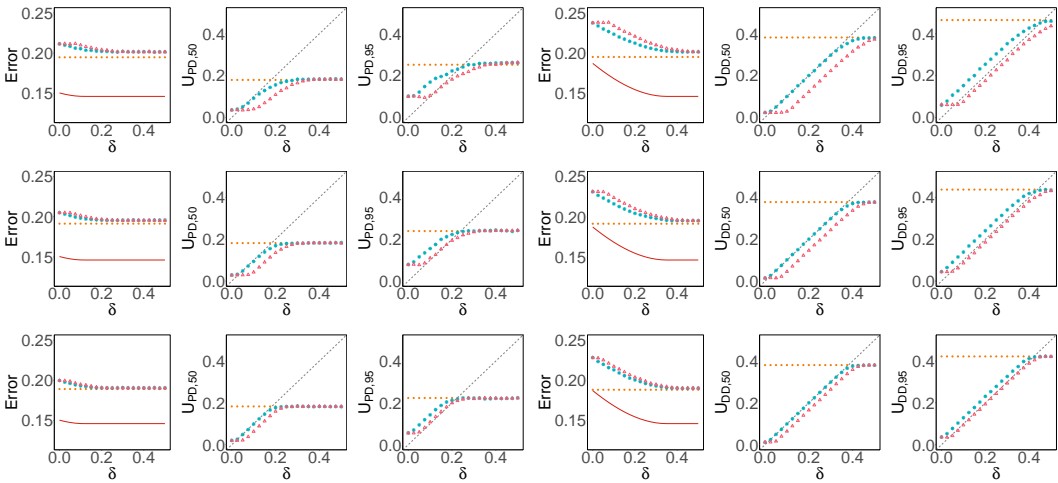

Figure 13: Disparity PD results under the non-Gaussian model, $\beta = 1.5$. Top: $n = 1000$; middle: $n = 2000$; bottom: $n = 5000$.

Figure 14: Disparity DD results under the non-Gaussian model, $\beta = 1.5$. Top: $n = 1000$; middle: $n = 2000$; bottom: $n = 5000$.

performance of our algorithm in such scenarios, we consider the Gaussian model in Appendix A.1 with $\beta = 0.5$, under which the signal-to-noise ratio is sufficiently high to mimic the perfect classification regime. For class probabilities, we examine two settings: (I) $\mathbb{P}(Y = 1|A = 0) = 0.4, \mathbb{P}(Y = 1|A = 1) = 0.7$; and (II) $\mathbb{P}(Y = 1|A = 0) = \mathbb{P}(Y = 1|A = 1) = 0.5$, while keeping all other model parameters in Appendix A.1 unchanged.

As discussed in Remark 2, under setting (I), the classical unconstrained Bayes classifier is automatically fair with respect to DO and PD. In setting (II), it is automatically fair with respect to all the three disparity measures DO, PD and DD. To visualise this difference, we plot DD as a function of $\tau$ in Figure 15.

The results are presented in Figures 16-19. Under the disparity measures DO and PD in setting (I), the classification errors of the fairness-aware classifiers are nearly zero, and the median disparity levels converge to zero across all values of $\delta$ as $n$ increases. This confirms that our approach naturally reduces to the classical FLDA classifier in such automatically fair cases. In contrast, under DD in setting (I), the fact that $|DD(\tau)| \equiv 0.3$ indicates that it is infeasible to achieve lower disparity levels

in this setting. By comparison, in setting (II), the unconstrained Bayes classifier is automatically fair under DD, and the empirical results exhibit a similar pattern to those observed for DO and PD in setting (I).

Overall, in perfect classification cases, our proposed algorithm continues to perform comparably to the oracle fairness-aware Bayes optimal classifier, further highlighting its effectiveness and adaptability.

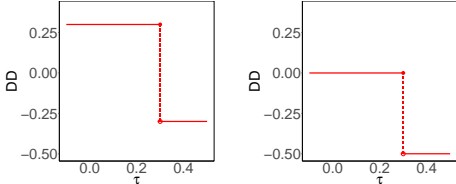

Figure 15: Oracle disparity DD versus $\tau$. Left: (I); right: (II).

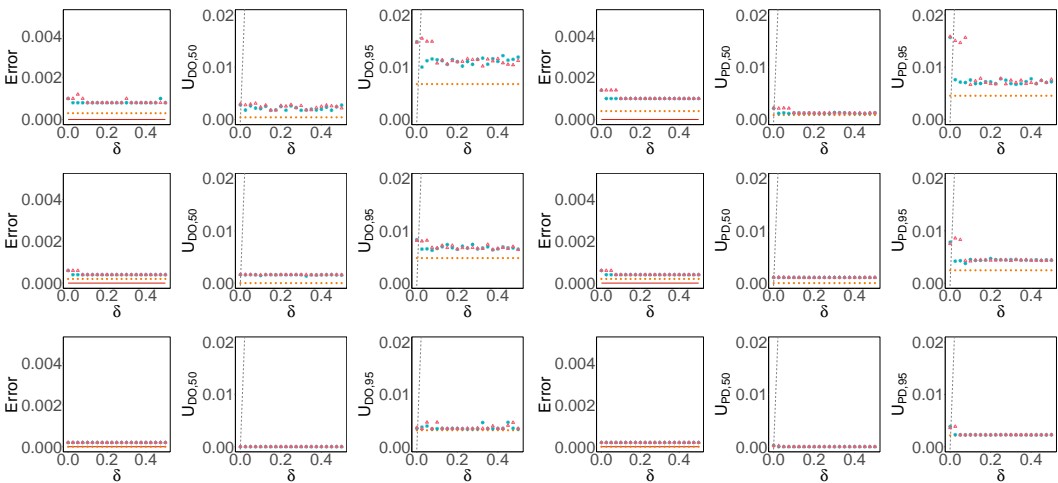

Figure 16: Disparity DO results under (I). Top: Figure 17: Disparity PD results under (I). Top: $n = 1000$; middle: $n = 2000$; bottom: $n = n = 1000$; middle: $n = 2000$; bottom: $n = 5000$.                  5000.

### A.5 Additional results for real data

In practice, we recommend tuning the calibration parameter $\kappa$ in Fair-FLDA$_c$ to achieve more reliable probabilistic disparity control. Specifically, we select the smallest value of $\kappa$ such that the empirical $1 - \rho$ quantile of the disparity remains below the pre-specified threshold $\delta$. To estimate this empirical quantile, we resort to random splitting. The data are randomly divided into two subsets, with one used to estimate the fairness-aware classifier, and the other to evaluate the resulting disparity. We repeat the process multiple times, e.g, 100 times, and the empirical $1 - \rho$ quantile is then computed from the empirical distribution of observed disparities.

The results obtained using the tuned calibration levels are reported in Figure 20. As shown, the tuned Fair-FLDA$_c$ consistently maintains disparity below $\delta$ with probability at least $1 - \rho$, except for a slight violation under one small $\delta$ under DO. This demonstrates the overall effectiveness of the proposed tuning strategy.

### A.6 Numerical comparisons with multivariate baselines

To the best of our knowledge, our work is the first to derive the Bayes optimal fair classifier and to establish explicit convergence rates in the context of functional data. There are no existing fair functional classifiers available for direct comparison.

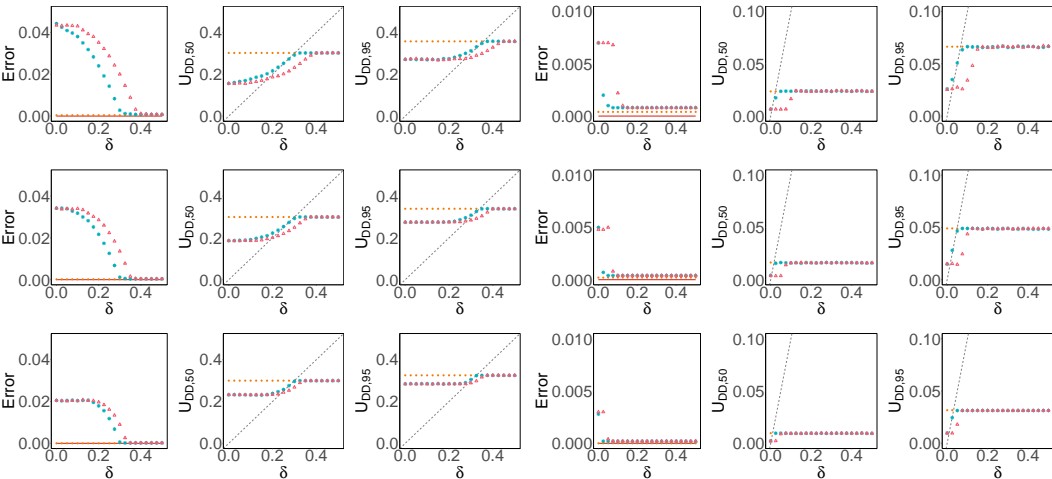

Figure 18: Disparity DD results under (I). Top: $n = 1000$; middle: $n = 2000$; bottom: $n = 5000$.

Figure 19: Disparity DD results under (II). Top: $n = 1000$; middle: $n = 2000$; bottom: $n = 5000$.

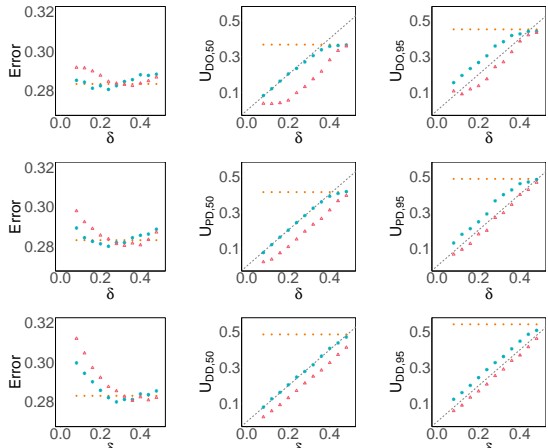

Figure 20: Results under NHANES with tuned calibration parameters over 100 Monte Carlo runs.

For comparison, we have incorporated additional baselines following a "dimension reduction + standard fair classification" strategy. Specifically, we first apply functional principal component analysis to extract features, and then employ fair classification methods designed for multivariate data. These include three post-processing methods FPIR (Zeng et al., 2024a), PPF (Chen et al., 2024) and PPOT (Xian et al., 2023), and one pre-processing approach FUDS (Zeng et al., 2024a), with default parameters as in the open source code of Zeng et al. (2024a).

Results in Tables 1 and 2 show that our proposed fair classifier, Fair-FLDA, consistently achieves the lowest classification errors while effectively controlling disparity under the pre-specified levels. In contrast, the other four baseline methods exhibit higher classification errors. In particular, PPF shows poor disparity control when $\delta = 0$, and FUDS fails to adequately control disparity on the NHANES dataset. These extensive numerical results highlight the superiority and practical necessity of our method for fair functional classification.

## A.7 Results on effects of eigenspace estimation and data splitting

**Eigenspace estimation.** Eigenspace estimation plays a fundamental role in FDA. We justify the performance of eigenspace estimation from both theoretical and numerical perspectives. In our paper, the theoretical guarantees for eigenfunction and eigenvalue estimations have been established

Table 1: Median classification error and DD over 500 runs under Gaussian $\beta = 1.5, n = 1000$.

| $\delta$ | 0.00 | | 0.05 | | 0.10 | | 0.15 | | 0.20 | |
|---|---|---|---|---|---|---|---|---|---|---|
| | Err | $U_{DD,50}$ | Err | $U_{DD,50}$ | Err | $U_{DD,50}$ | Err | $U_{DD,50}$ | Err | $U_{DD,50}$ |
| Fair-FLDA | 0.234 | 0.022 | 0.227 | 0.048 | 0.219 | 0.098 | 0.213 | 0.149 | 0.208 | 0.197 |
| FPIR | 0.276 | 0.024 | 0.269 | 0.039 | 0.264 | 0.080 | 0.258 | 0.122 | 0.253 | 0.168 |
| PPF | 0.240 | 0.323 | 0.269 | 0.039 | 0.264 | 0.080 | 0.258 | 0.121 | 0.253 | 0.167 |
| PPOT | 0.275 | 0.023 | 0.269 | 0.040 | 0.264 | 0.080 | 0.258 | 0.122 | 0.253 | 0.168 |
| FUDS | 0.276 | 0.030 | 0.269 | 0.040 | 0.263 | 0.087 | 0.256 | 0.142 | 0.251 | 0.195 |

Table 2: Median classification error and DD over 500 runs under NHANES.

| $\delta$ | 0.00 | | 0.05 | | 0.10 | | 0.15 | | 0.20 | |
|---|---|---|---|---|---|---|---|---|---|---|
| | Err | $U_{DD,50}$ | Err | $U_{DD,50}$ | Err | $U_{DD,50}$ | Err | $U_{DD,50}$ | Err | $U_{DD,50}$ |
| Fair-FLDA | 0.314 | 0.021 | 0.305 | 0.048 | 0.297 | 0.099 | 0.289 | 0.149 | 0.285 | 0.200 |
| FPIR | 0.385 | 0.016 | 0.377 | 0.050 | 0.369 | 0.101 | 0.360 | 0.151 | 0.354 | 0.200 |
| PPF | 0.343 | 0.722 | 0.378 | 0.047 | 0.369 | 0.097 | 0.361 | 0.144 | 0.355 | 0.193 |
| PPOT | 0.385 | 0.016 | 0.377 | 0.050 | 0.369 | 0.101 | 0.360 | 0.150 | 0.353 | 0.200 |
| FUDS | 0.403 | 0.157 | 0.371 | 0.189 | 0.358 | 0.254 | 0.351 | 0.317 | 0.346 | 0.349 |

in Lemmas 35 and 36, respectively. These results are optimal, matching the minimax rate established in Wahl (2022) up to poly-logarithmic factors.

To support our theory, we provide further simulation results in Table 3, where Fair-FLDA refers to the proposed classifier in Section 4; Truth uses true eigenfunctions and eigenvalues; Fourier replaces estimated eigenfunctions with Fourier basis and eigenvalues with covariance projection scores. Overall, disparity control is comparable across methods, with all meeting their fairness criteria. Comparing the results of Fair-FLDA with Truth, we see that the misclassification errors of the proposed classifiers are even smaller than those obtained without eigenspace estimation. This improvement is attributed to the data-adaptive nature of the estimated eigenfunctions, which captures more variance than the fixed true basis. Substituting the estimated eigenfunctions with the pre-specified Fourier basis leads to a noticeable increase in misclassification errors.

**Data splitting.** The calibration data are primarily introduced for technical convenience to bring independence among samples in our theoretical studies. In practice, our Algorithm 1 can be implemented by executing both steps **S1** and **S2** on the whole dataset. In all numerical experiments in Section 4, we mimic the effect of sample splitting via a cross-fitting approach detailed in Appendix

Table 3: Median classification error and DO over 500 runs under Gaussian $\beta = 1.5, n = 1000$.

| $\delta$ | Fair-FLDA | | Truth | | Fourier | |
|---|---|---|---|---|---|---|
| | Error | $U_{DO,50}$ | Error | $U_{DO,50}$ | Error | $U_{DO,50}$ |
| 0.00 | 0.221 | 0.029 | 0.235 | 0.028 | 0.274 | 0.027 |
| 0.05 | 0.213 | 0.049 | 0.228 | 0.055 | 0.266 | 0.053 |
| 0.10 | 0.207 | 0.099 | 0.223 | 0.101 | 0.261 | 0.101 |
| 0.15 | 0.203 | 0.150 | 0.220 | 0.140 | 0.256 | 0.151 |
| 0.20 | 0.200 | 0.191 | 0.218 | 0.172 | 0.253 | 0.200 |

Table 4: Effects of data splitting for Fair-FLDA in the real and simulated (Gaussian with $n = 1000, \beta = 1.5$) datasets. Results are reported as the median over 500 iterations. NoSplit: the results of Fair-FLDA applied without data splitting.

| $\delta$ | Fair-FLDA | | NoSplit | |
|---|---|---|---|---|
| | Error | $U_{DO,50}$ | Error | $U_{DO,50}$ |
| **Simulated data** | | | | |
| 0.00 | 0.221 | 0.029 | 0.211 | 0.029 |
| 0.05 | 0.213 | 0.049 | 0.203 | 0.059 |
| 0.10 | 0.207 | 0.099 | 0.198 | 0.110 |
| 0.15 | 0.203 | 0.150 | 0.195 | 0.164 |
| 0.20 | 0.200 | 0.191 | 0.193 | 0.208 |
| **Real data** | | | | |
| 0.00 | 0.291 | 0.034 | 0.284 | 0.039 |
| 0.05 | 0.286 | 0.054 | 0.281 | 0.056 |
| 0.10 | 0.284 | 0.099 | 0.278 | 0.101 |
| 0.15 | 0.282 | 0.148 | 0.277 | 0.152 |
| 0.20 | 0.281 | 0.196 | 0.276 | 0.205 |

A.1, where two classifiers are trained by alternating the roles of data used for model estimation and threshold calibration, and then averaged. To further illustrate the effect of sample splitting, we include additional numerical results on both simulated and real datasets in Table 3 below. Although the reduction in sample size from data splitting slightly increases the misclassification error, the unfairness measure under NoSplit is usually higher than the threshold $\delta$ due to the dependence of the data used in training and calibration.

## B  Extensions of our method

Our framework in Section 2.2 can be naturally extended to settings where the sensitive feature is unavailable at testing, as well as to multi-class classification problems.

**Extension to missing sensitive attributes.**  The extension to missing sensitive attributes during testing stage consists of three steps.

Step 1. A key ingredient in our framework, when sensitive features are available, is that both the misclassification error $R$ and disparity measure $D$ are linear in classifiers $f : \mathcal{X} \times \mathcal{A} \to [0,1]$. To extend the framework to settings where sensitive attributes are unavailable at testing, it is necessary to show that $R$ and $D$ remain linear to classifiers $f : \mathcal{X} \to [0,1]$, which is solely defined on $\mathcal{X}$. Following a similar idea as the proof of Proposition 1, it can be verified that DO, PD and DD are still linear and of the form $D(f) = \int_{\mathcal{X}} f(x) w_{X,D}(x) dP_{X|Y=0}(x)$, where $w_{X,D} : \mathcal{X} \to \mathbb{R}$ is a weight function.

Step 2. By the generalised Neyman–Pearson lemma and a similar argument to the one used in the proof of Theorem 2, we can derive an explicit formula for $\delta$-fair Bayes optimal classifier of the form $f_D^\star(x) = \mathbf{1}[\pi_1 \frac{dP_{X|Y=1}}{dP_{X|Y=0}}(x) - \pi_0 \geq \tau^\star w_{x,D}(x)]$.

Step 3. Construct a plug-in classifier. Further to the estimation in Algorithm 1, an additional non-parametric estimator can be used to estimate the probability of A given X and Y.

When sensitive information is not available in training time, the fair classification is a challenging problem because, without direct access to sensitive information, it is difficult to learn and correct for

the potential biases. Relatively few works have studied this issue. Existing approaches (Lahoti et al., 2020; Zhao et al., 2022; Veldanda et al., 2024) generally rely on the assumption that standard features $X$ are sufficiently informative, allowing bias mitigation through indirect inference for protected attributes. In Kallus et al. (2022), it is further shown that various disparities remain unidentifiable when $X$ lacks sufficient information.

In our work, we focus on developing a principled framework for achieving fairness in functional data classification when sensitive features are available, either during both training and testing or at least in the training stage. Extending our framework to settings where sensitive attributes are completely unavailable is an important direction for future research.

**Extension to multi-class classification.** Consider predictive disparity as an example. For a multi-class sensitive attribute $a \in \mathcal{A} = [1, \ldots, |\mathcal{A}|]$, motivated by the proof of Theorem 2 in our paper and Theorem 4.8 in Zeng et al. (2024a), we conjecture that the generalised Neyman-Pearson lemma leads to $f_{PD}^{\star}(x, a) = \mathbf{1}[\frac{dP_{a,1}}{dP_{a,0}}(x) \geq \frac{\pi_{a,0}+\tau_a^{\star}}{\pi_{a,1}}]$, where the threshold $\tau_a^{\star}$ is selected in a way similar to Equation (3).

## C   Proofs for Bayes optimal fairness-aware classifier

### C.1   Proof of Proposition 1

*Proof of Proposition 1.*

- For DO, we are to show that $s_{DO,a} = 2a - 1$ and $b_{DO,a} = 0$. It can be seen from the following that

$$DO(f) = \mathbb{P}\{\widehat{Y}_f(X, 1) = 1 | A = 1, Y = 1\} - \mathbb{P}\{\widehat{Y}_f(X, 0) = 1 | A = 0, Y = 1\}$$

$$= \int_{\mathcal{X}} f(x, 1) \frac{dP_{1,1}}{dP_{1,0}}(x) \, dP_{1,0}(x) - \int_{\mathcal{X}} f(x, 0) \frac{dP_{0,1}}{dP_{0,0}}(x) \, dP_{0,0}(x).$$

- For PD, we are to show that $s_{PD,a} = 0$ and $b_{PD,a} = 2a - 1$. It can be seen from the following that

$$PD(f) = \mathbb{P}\{\widehat{Y}_f(X, 1) = 1 | A = 1, Y = 0\} - \mathbb{P}\{\widehat{Y}_f(X, 0) = 1 | A = 0, Y = 0\}$$

$$= \int_{\mathcal{X}} f(x, 1) \, dP_{1,0}(x) - \int_{\mathcal{X}} f(x, 0) \, dP_{0,0}(x).$$

- For DD, we are to show that $s_{DD,a} = (2a - 1)\pi_{a,1}/\pi_a$ and $b_{DD,a} = (2a - 1)\pi_{a,0}/\pi_a$. It can be seen from the following that

$$DD(f) = \mathbb{P}\{\widehat{Y}_f(X, 1) = 1 | A = 1\} - \mathbb{P}\{\widehat{Y}_f(X, 0) = 1 | A = 0\}$$

$$= \int_{\mathcal{X}} f(x, 1) \frac{\pi_{1,1}}{\pi_1} \frac{dP_{1,1}}{dP_{1,0}}(x) \, dP_{1,0}(x) + \int_{\mathcal{X}} f(x, 1) \frac{\pi_{1,0}}{\pi_1} \, dP_{1,0}(x)$$

$$- \int_{\mathcal{X}} f(x, 0) \frac{\pi_{0,1}}{\pi_0} \frac{dP_{0,1}}{dP_{0,0}}(x) \, dP_{0,0}(x) - \int_{\mathcal{X}} f(x, 0) \frac{\pi_{0,0}}{\pi_0} \, dP_{0,0}(x)$$

$$= \int_{\mathcal{X}} f(x, 1) \left( \frac{\pi_{1,1}}{\pi_1} \frac{dP_{1,1}}{dP_{1,0}}(x) + \frac{\pi_{1,0}}{\pi_1} \right) dP_{1,0}(x)$$

$$- \int_{\mathcal{X}} f(x, 0) \left( \frac{\pi_{0,1}}{\pi_0} \frac{dP_{0,1}}{dP_{0,0}}(x) + \frac{\pi_{0,0}}{\pi_0} \right) dP_{0,0}(x).$$

$\square$

## C.2 Proof of Theorem 2

*Proof of Theorem 2.* If $|D(0)| \leq \delta$, the unconstrained Bayes optimal classifier satisfies the fairness constraint. Therefore, we have $\tau_{D,\delta}^\star = 0$ and the $\delta$-fair Bayes optimal classifier is given by $f_{D,\delta}^\star = g_{D,0}$.

If $D(0) > \delta$, by Proposition 7, we have $D(\tau_{D,\delta}^\star) = \delta$. Moreover, $\tau_{D,\delta}^\star > 0$. By Lemma 10,

$$g_{D,\tau_{D,\delta}^\star} = \underset{f \in \mathcal{F}}{\arg\min} \left\{ R(f) : |D(f)| \leq \frac{\tau_{D,\delta}^\star D(\tau_{D,\delta}^\star)}{|\tau_{D,\delta}^\star|} \right\} = \underset{f \in \mathcal{F}}{\arg\min} \left\{ R(f) : |D(f)| \leq \delta \right\}.$$

Analogously, we can establish the claim when $D(0) < -\delta$. This completes the proof. $\qquad\square$

## C.3 Proof of Proposition 4

*Proof of Proposition 4.* Let

$$\Lambda_a = \langle X - \mu_{a,0}, \mu_{a,1} - \mu_{a,0} \rangle_{K_a} = \sum_{j=1}^{\infty} \frac{(\zeta_{a,j} - \theta_{a,0,j})(\theta_{a,1,j} - \theta_{a,0,j})}{\lambda_{a,j}}.$$

Then, by standard properties of Gaussina processes, we have that

$$\Lambda_a | \{A = a, Y = 0\} \sim N\left(0, \sum_{j=1}^{\infty} \frac{(\theta_{a,1,j} - \theta_{a,0,j})^2}{\lambda_{a,j}}\right),$$

$$\Lambda_a | \{A = a, Y = 1\} \sim N\left(\sum_{j=1}^{\infty} \frac{(\theta_{a,1,j} - \theta_{a,0,j})^2}{\lambda_{a,j}}, \sum_{j=1}^{\infty} \frac{(\theta_{a,1,j} - \theta_{a,0,j})^2}{\lambda_{a,j}}\right).$$

The proposition then follows by a similar argument as the one used in the proof of Theorem 2 in Berrendero et al. (2018) and the format of $f_{D,\delta}^\star$ in (4).

$\qquad\square$

## C.4 Auxiliary results

**Proposition 7.** *Recall that $D(\tau) = D(g_{D,\tau})$, where $g_{D,\tau}$ is defined as*

$$g_{\mathrm{D},\tau}(x,a) = \mathbb{1}\left\{(\pi_{a,1} - \tau s_{\mathrm{D},a}) \frac{\mathrm{d}P_{a,1}}{\mathrm{d}P_{a,0}}(x) \geq \pi_{a,0} + \tau b_{\mathrm{D},a}\right\}.$$

*Then, under the assumptions in Theorem 2, the following properties hold.*

(i) *The disparity $D(\tau)$ is continuous and non-increasing.*

(ii) *The misclassification $R(g_{D,\tau})$ is non-increasing on $(-\infty, 0)$ and non-decreasing on $(0, +\infty)$.*

*Proof of Proposition 7.*

(i) Note that by Definition 4,

$$D(\tau) = \sum_{a \in \{0,1\}} \int_{\mathcal{X}} g_{D,\tau}(x,a) \left\{ s_{D,a} \frac{\mathrm{d}P_{a,1}}{\mathrm{d}P_{a,0}}(x) + b_{D,a} \right\} \mathrm{d}P_{a,0}(x).$$

Since $\mathrm{d}P_{a,1}/\mathrm{d}P_{a,0}(x)$ is a continuous random variable given $A = a \in \{0,1\}$ and $Y = y \in \{0,1\}$, we have that the function $\tau \mapsto \mathbb{P}_{X|A=a,Y=y}\big((\pi_{a,1} - \tau s_{D,a})\mathrm{d}P_{a,1}/\mathrm{d}P_{a,0}(x) > \pi_{a,0} + \tau b_{D,a}\big)$ is continuous for $a \in \{0,1\}$ and $y \in \{0,1\}$. Thus, the function $\tau \mapsto D(\tau)$ is continuous.

Define

$$\mathcal{E}_{a,+} = \left\{ x \in \mathcal{X} : s_{D,a} \frac{\mathrm{d}P_{a,1}}{\mathrm{d}P_{a,0}}(x) + b_{D,a} > 0 \right\},$$

and
$$\mathcal{E}_{a,-} = \left\{ x \in \mathcal{X} : s_{D,a} \frac{\mathrm{d}P_{a,1}}{\mathrm{d}P_{a,0}}(x) + b_{D,a} < 0 \right\}.$$

Let $\tau_1 < \tau_2$. For $a \in \{0,1\}$ and $x \in \mathcal{X}$,

$$g_{D,\tau_1}(x,a) - g_{D,\tau_2}(x,a) = \begin{cases} \mathbb{1}\left\{ \tau_1 < \dfrac{\pi_{a,1}\frac{\mathrm{d}P_{a,1}}{\mathrm{d}P_{a,0}}(x) - \pi_{a,0}}{s_{D,a}\frac{\mathrm{d}P_{a,1}}{\mathrm{d}P_{a,0}}(x) + b_{D,a}} \leq \tau_2 \right\}, & x \in \mathcal{E}_{a,+}; \\[2em] -\mathbb{1}\left\{ \tau_1 \leq \dfrac{\pi_{a,1}\frac{\mathrm{d}P_{a,1}}{\mathrm{d}P_{a,0}}(x) - \pi_{a,0}}{s_{D,a}\frac{\mathrm{d}P_{a,1}}{\mathrm{d}P_{a,0}}(x) + b_{D,a}} < \tau_2 \right\}, & x \in \mathcal{E}_{a,-}; \\[2em] 0, & \text{otherwise.} \end{cases}$$

We then have

$$D(\tau_1) - D(\tau_2)$$

$$= \sum_{a \in \{0,1\}} \int_{\mathcal{X}} \{ g_{D,\tau_1}(x,a) - g_{D,\tau_2}(x,a) \} \left\{ s_{D,a} \frac{\mathrm{d}P_{a,1}}{\mathrm{d}P_{a,0}}(x) + b_{D,a} \right\} \mathrm{d}P_{a,0}(x)$$

$$= \sum_{a \in \{0,1\}} \int_{x \in \mathcal{E}_{a,+}} \mathbb{1}\left\{ \tau_1 < \frac{\pi_{a,1}\frac{\mathrm{d}P_{a,1}}{\mathrm{d}P_{a,0}}(x) - \pi_{a,0}}{s_{D,a}\frac{\mathrm{d}P_{a,1}}{\mathrm{d}P_{a,0}}(x) + b_{D,a}} \leq \tau_2 \right\}$$

$$\cdot \left\{ s_{D,a} \frac{\mathrm{d}P_{a,1}}{\mathrm{d}P_{a,0}}(x) + b_{D,a} \right\} \mathrm{d}P_{a,0}(x)$$

$$- \sum_{a \in \{0,1\}} \int_{x \in \mathcal{E}_{a,-}} \mathbb{1}\left\{ \tau_1 \leq \frac{\pi_{a,1}\frac{\mathrm{d}P_{a,1}}{\mathrm{d}P_{a,0}}(x) - \pi_{a,0}}{s_{D,a}\frac{\mathrm{d}P_{a,1}}{\mathrm{d}P_{a,0}}(x) + b_{D,a}} < \tau_2 \right\}$$

$$\left\{ s_{D,a} \frac{\mathrm{d}P_{a,1}}{\mathrm{d}P_{a,0}}(x) + b_{D,a} \right\} \mathrm{d}P_{a,0}(x)$$

$$\geq 0.$$

Consequently, the function $\tau \mapsto D(\tau)$ is non-increasing.

(ii) We first consider $\tau_1 < \tau_2 < 0$. If $x \in \mathcal{E}_{a,+}$,

$$\mathbb{1}\left\{ \tau_1 < \frac{\pi_{a,1}\frac{\mathrm{d}P_{a,1}}{\mathrm{d}P_{a,0}}(x) - \pi_{a,0}}{s_{D,a}\frac{\mathrm{d}P_{a,1}}{\mathrm{d}P_{a,0}}(x) + b_{D,a}} \leq \tau_2 \right\}\left\{ \pi_{a,0} - \pi_{a,1}\frac{\mathrm{d}P_{a,1}}{\mathrm{d}P_{a,0}}(x) \right\}$$

$$\geq -\tau_2 \left\{ s_{D,a}\frac{\mathrm{d}P_{a,1}}{\mathrm{d}P_{a,0}}(x) + b_{D,a} \right\}\mathbb{1}\left\{ \tau_1 < \frac{\pi_{a,1}\frac{\mathrm{d}P_{a,1}}{\mathrm{d}P_{a,0}}(x) - \pi_{a,0}}{s_{D,a}\frac{\mathrm{d}P_{a,1}}{\mathrm{d}P_{a,0}}(x) + b_{D,a}} \leq \tau_2 \right\} \geq 0.$$

If $x \in \mathcal{E}_{a,-}$,

$$\mathbb{1}\left\{ \tau_1 \leq \frac{\pi_{a,1}\frac{\mathrm{d}P_{a,1}}{\mathrm{d}P_{a,0}}(x) - \pi_{a,0}}{s_{D,a}\frac{\mathrm{d}P_{a,1}}{\mathrm{d}P_{a,0}}(x) + b_{D,a}} < \tau_2 \right\}\left\{ \pi_{a,0} - \pi_{a,1}\frac{\mathrm{d}P_{a,1}}{\mathrm{d}P_{a,0}}(x) \right\}$$

$$\leq -\tau_2 \left\{ s_{D,a}\frac{\mathrm{d}P_{a,1}}{\mathrm{d}P_{a,0}}(x) + b_{D,a} \right\}\mathbb{1}\left\{ \tau_1 \leq \frac{\pi_{a,1}\frac{\mathrm{d}P_{a,1}}{\mathrm{d}P_{a,0}}(x) - \pi_{a,0}}{s_{D,a}\frac{\mathrm{d}P_{a,1}}{\mathrm{d}P_{a,0}}(x) + b_{D,a}} < \tau_2 \right\} \leq 0.$$

Then, by Lemma 8, it holds that

$$R(g_{D,\tau_1}) - R(g_{D,\tau_2})$$

$$= \sum_{a\in\{0,1\}} \int_{\mathcal{X}} \{g_{D,\tau_1}(x,a) - g_{D,\tau_2}(x,a)\}\left\{\pi_{a,0} - \pi_{a,1}\frac{\mathrm{d}P_{a,1}}{\mathrm{d}P_{a,0}}(x)\right\}\mathrm{d}P_{a,0}(x)$$

$$= \sum_{a\in\{0,1\}} \int_{x\in\mathcal{E}_{a,+}} \mathbb{1}\left\{\tau_1 < \frac{\pi_{a,1}\frac{\mathrm{d}P_{a,1}}{\mathrm{d}P_{a,0}}(x) - \pi_{a,0}}{s_{D,a}\frac{\mathrm{d}P_{a,1}}{\mathrm{d}P_{a,0}}(x) + b_{D,a}} \le \tau_2\right\}$$

$$\cdot \left\{\pi_{a,0} - \pi_{a,1}\frac{\mathrm{d}P_{a,1}}{\mathrm{d}P_{a,0}}(x)\right\}\mathrm{d}P_{a,0}(x)$$

$$- \sum_{a\in\{0,1\}} \int_{x\in\mathcal{E}_{a,-}} \mathbb{1}\left\{\tau_1 \le \frac{\pi_{a,1}\frac{\mathrm{d}P_{a,1}}{\mathrm{d}P_{a,0}}(x) - \pi_{a,0}}{s_{D,a}\frac{\mathrm{d}P_{a,1}}{\mathrm{d}P_{a,0}}(x) + b_{D,a}} < \tau_2\right\}$$

$$\cdot \left\{\pi_{a,0} - \pi_{a,1}\frac{\mathrm{d}P_{a,1}}{\mathrm{d}P_{a,0}}(x)\right\}\mathrm{d}P_{a,0}(x)$$

$$\ge 0.$$

Therefore, $\tau \mapsto R(g_{D,\tau})$ is non-increasing on $(-\infty, 0)$.

Consider $0 \le \tau_1 < \tau_2$. If $x \in \mathcal{E}_{a,+}$,

$$\mathbb{1}\left\{\tau_1 < \frac{\pi_{a,1}\frac{\mathrm{d}P_{a,1}}{\mathrm{d}P_{a,0}}(x) - \pi_{a,0}}{s_{D,a}\frac{\mathrm{d}P_{a,1}}{\mathrm{d}P_{a,0}}(x) + b_{D,a}} \le \tau_2\right\}\left\{\pi_{a,0} - \pi_{a,1}\frac{\mathrm{d}P_{a,1}}{\mathrm{d}P_{a,0}}(x)\right\}$$

$$\le -\tau_1\left\{s_{D,a}\frac{\mathrm{d}P_{a,1}}{\mathrm{d}P_{a,0}}(x) + b_{D,a}\right\}\mathbb{1}\left\{\tau_1 < \frac{\pi_{a,1}\frac{\mathrm{d}P_{a,1}}{\mathrm{d}P_{a,0}}(x) - \pi_{a,0}}{s_{D,a}\frac{\mathrm{d}P_{a,1}}{\mathrm{d}P_{a,0}}(x) + b_{D,a}} \le \tau_2\right\} \le 0.$$

If $x \in \mathcal{E}_{a,-}$,

$$\mathbb{1}\left\{\tau_1 \le \frac{\pi_{a,1}\frac{\mathrm{d}P_{a,1}}{\mathrm{d}P_{a,0}}(x) - \pi_{a,0}}{s_{D,a}\frac{\mathrm{d}P_{a,1}}{\mathrm{d}P_{a,0}}(x) + b_{D,a}} < \tau_2\right\}\left\{\pi_{a,0} - \pi_{a,1}\frac{\mathrm{d}P_{a,1}}{\mathrm{d}P_{a,0}}(x)\right\}$$

$$\ge -\tau_1\left\{s_{D,a}\frac{\mathrm{d}P_{a,1}}{\mathrm{d}P_{a,0}}(x) + b_{D,a}\right\}\mathbb{1}\left\{\tau_1 \le \frac{\pi_{a,1}\frac{\mathrm{d}P_{a,1}}{\mathrm{d}P_{a,0}}(x) - \pi_{a,0}}{s_{D,a}\frac{\mathrm{d}P_{a,1}}{\mathrm{d}P_{a,0}}(x) + b_{D,a}} < \tau_2\right\} \ge 0.$$

Then, we have

$$R(g_{D,\tau_1}) - R(g_{D,\tau_2})$$

$$= \sum_{a\in\{0,1\}} \int_{\mathcal{X}} \{g_{D,\tau_1}(x,a) - g_{D,\tau_2}(x,a)\}\left\{\pi_{a,0} - \pi_{a,1}\frac{\mathrm{d}P_{a,1}}{\mathrm{d}P_{a,0}}(x)\right\}\mathrm{d}P_{a,0}(x)$$

$$= \sum_{a\in\{0,1\}} \int_{x\in\mathcal{E}_{a,+}} \mathbb{1}\left\{\tau_1 < \frac{\pi_{a,1}\frac{\mathrm{d}P_{a,1}}{\mathrm{d}P_{a,0}}(x) - \pi_{a,0}}{s_{D,a}\frac{\mathrm{d}P_{a,1}}{\mathrm{d}P_{a,0}}(x) + b_{D,a}} \le \tau_2\right\}$$

$$\cdot \left\{\pi_{a,0} - \pi_{a,1}\frac{\mathrm{d}P_{a,1}}{\mathrm{d}P_{a,0}}(x)\right\}\mathrm{d}P_{a,0}(x)$$

$$- \sum_{a\in\{0,1\}} \int_{x\in\mathcal{E}_{a,-}} \mathbb{1}\left\{\tau_1 \le \frac{\pi_{a,1}\frac{\mathrm{d}P_{a,1}}{\mathrm{d}P_{a,0}}(x) - \pi_{a,0}}{s_{D,a}\frac{\mathrm{d}P_{a,1}}{\mathrm{d}P_{a,0}}(x) + b_{D,a}} < \tau_2\right\}$$

$$\cdot \left\{\pi_{a,0} - \pi_{a,1}\frac{\mathrm{d}P_{a,1}}{\mathrm{d}P_{a,0}}(x)\right\}\mathrm{d}P_{a,0}(x)$$

$$\leq 0.$$

Therefore, $\tau \mapsto R(g_{D,\tau})$ is non-decreasing on $[0, +\infty)$.

$\square$

**Lemma 8.** *For any classifier $f : \mathcal{X} \times \mathcal{A} \to [0, 1]$, we have*

$$R(f) = \sum_{a \in \{0,1\}} \int_{\mathcal{X}} f(x, a) \left\{ \pi_{a,0} - \pi_{a,1} \frac{\mathrm{d}P_{a,1}}{\mathrm{d}P_{a,0}}(x) \right\} \mathrm{d}P_{a,0}(x) + \mathbb{P}(Y = 1).$$

*Proof.* By definition, we have that

$$R(f) = \sum_{a,y \in \{0,1\}} \mathbb{P}\{\widehat{Y}_f(X, a) \neq y | A = a, Y = y\} P(A = a, Y = y)$$

$$= \sum_{a \in \{0,1\}} \Big[ \mathbb{E}\{1 - f(X, a)| A = a, Y = 1\} P(A = a, Y = 1)$$

$$+ \mathbb{E}\{f(X, a)| A = a, Y = 0\} P(A = a, Y = 0) \Big]$$

$$= \sum_{a \in \{0,1\}} \left\{ \int_{\mathcal{X}} f(x, a) \mathrm{d}P_{a,0}(x) \pi_{a,0} - \int_{\mathcal{X}} f(x, a) \mathrm{d}P_{a,1}(x) \pi_{a,1} \right\}$$

$$+ \sum_{a \in \{0,1\}} \mathbb{P}(A = a, Y = 1)$$

$$= \sum_{a \in \{0,1\}} \int_{\mathcal{X}} f(x, a) \left\{ \pi_{a,0} - \pi_{a,1} \frac{\mathrm{d}P_{a,1}}{\mathrm{d}P_{a,0}}(x) \right\} \mathrm{d}P_{a,0}(x) + \mathbb{P}(Y = 1).$$

$\square$

**Lemma 9.** *For the bilinear disparity measures DO, PD and DD, it holds that*

$$\pi_{a,1} - \tau_{D,\delta}^\star s_{D,a} > 0, \quad and \quad \pi_{a,0} + \tau_{D,\delta}^\star b_{D,a} > 0,$$

*with $s_{D,a}$ and $b_{D,a}$ defined in Definition 4.*

*Proof.*

1. **DO:** In this case, it suffices to show that $-\pi_{0,1} < \tau_{DO,\delta}^\star < \pi_{1,1}$. If $\tau = \pi_{1,1}$, then $g_{DO,\pi_{1,1}}(x, 1) = 0$ for all $x \in \mathcal{X}$, and,

$$DO(\pi_{1,1}) = DO(g_{DO,\pi_{1,1}}) = -\mathbb{P}\left\{ (\pi_{0,1} + \pi_{1,1}) \frac{\mathrm{d}P_{0,1}}{\mathrm{d}P_{0,0}}(x) > \pi_{0,0} \Big| A = 0, Y = 1 \right\} \leq 0.$$

   If $\tau = -\pi_{0,1}$, then $g_{DO,-\pi_{0,1}}(x, 0) = 0$ for all $x \in \mathcal{X}$. Then

$$DO(g_{DO,-\pi_{0,1}}) = \mathbb{P}\left\{ (\pi_{1,1} + \pi_{0,1}) \frac{\mathrm{d}P_{1,1}}{\mathrm{d}P_{1,0}}(x) > \pi_{1,0} \Big| A = 1, Y = 1 \right\} \geq 0.$$

   Note that if $|DO(0)| \leq \delta$, then $\tau_{DO,\delta}^\star = 0$. By Proposition 7(i), if $DO(0) > \delta$, then $0 < \tau_{DO,\delta}^\star < \pi_{1,1}$. If $DO(0) < -\delta$, then $-\pi_{0,1} < \tau_{DO,\delta}^\star < 0$. Therefore, we conclude that $-\pi_{0,1} < \tau_{DO,\delta}^\star < \pi_{1,1}$.

2. **PD** In this case, it suffices to show that $-\pi_{1,0} < \tau_{PD,\delta}^\star < \pi_{0,0}$. Note that if $\tau = -\pi_{1,0}$, we have $g_{PD,-\pi_{1,0}}(x, 1) = 1$ for all $x \in \mathcal{X}$. Then,

$$PD(g_{PD,-\pi_{1,0}}) = 1 - \mathbb{P}\left\{ \pi_{0,1} \frac{\mathrm{d}P_{0,1}}{\mathrm{d}P_{0,0}}(x) > \pi_{0,0} + \pi_{1,0} \Big| A = 0, Y = 0 \right\} \geq 0.$$

If $\tau = \pi_{0,0}$, we have $g_{PD,\pi_{0,0}}(x,0) = 1$ for all $x \in \mathcal{X}$. Then,

$$PD(g_{PD,\pi_{0,0}}) = \mathbb{P}\Big\{\pi_{1,1}\frac{\mathrm{d}P_{1,1}}{\mathrm{d}P_{1,0}}(x) > \pi_{1,0} + \pi_{0,0}\Big|A=1, Y=0\Big\} - 1 \leq 0.$$

Note that if $|PD(0)| \leq \delta$, then $\tau_{PD,\delta}^\star = 0$. By Proposition 7 (i), if $PD(0) > \delta$, then $0 < \tau_{PD,\delta}^\star < \pi_{0,0}$. If $PD(0) < -\delta$, then $-\pi_{1,0} < \tau_{PD,\delta}^\star < 0$. Therefore, we conclude that $-\pi_{1,0} < \tau_{PD,\delta}^\star < \pi_{0,0}$.

3. **DD:** In this case, it suffices to show that $|\tau_{DD,\delta}^\star| < \min\{\pi_0, \pi_1\}$. Note that if $\tau = \pi_1$ then $g_{DD,\pi_1}(x,1) = 0$ for all $x \in \mathcal{X}$. Also,

$$DD(\pi_1) = -\mathbb{P}\Big\{\Big(\pi_{0,1} + \frac{\pi_1\pi_{0,1}}{\pi_0}\Big)\frac{\mathrm{d}P_{0,1}}{\mathrm{d}P_{0,0}}(x) > \pi_{0,0} - \frac{\pi_1\pi_{0,0}}{\pi_0}\Big|A=0\Big\}$$

$$\begin{cases} = -1, & \pi_1 \geq \pi_0, \\ \leq 0, & \pi_1 < \pi_0. \end{cases}$$

If $\tau = \pi_0$, then $g_{DD,\pi_0}(x,0) = 1$ for all $x \in \mathcal{X}$. Then

$$DD(\pi_0) = \mathbb{P}\Big\{\Big(\pi_{1,1} - \frac{\pi_0\pi_{1,1}}{\pi_1}\Big)\frac{\mathrm{d}P_{1,1}}{\mathrm{d}P_{1,0}}(x) > \pi_{1,0} + \frac{\pi_0\pi_{1,0}}{\pi_1}\Big|A=1\Big\} - 1 \leq 0.$$

Moreover, if $\tau = -\pi_0$, then $g_{DD,-\pi_0}(x,0) = 0$ for all $x \in \mathcal{X}$. And,

$$DD(-\pi_0) = \mathbb{P}\Big\{\Big(\pi_{1,1} + \frac{\pi_0\pi_{1,1}}{\pi_1}\Big)\frac{\mathrm{d}P_{1,1}}{\mathrm{d}P_{1,0}}(x) > \pi_{1,0} - \frac{\pi_0\pi_{1,0}}{\pi_1}\Big|A=1\Big\}$$

$$\begin{cases} \geq 0, & \pi_1 \geq \pi_0, \\ = 1, & \pi_1 < \pi_0. \end{cases}$$

If $\tau = -\pi_1$, then $g_{DD,-\pi_1}(x,1) = 1$ for all $x \in \mathcal{X}$. Then,

$$DD(-\pi_1) = 1 - \mathbb{P}\Big\{\Big(\pi_{0,1} - \frac{\pi_1\pi_{0,1}}{\pi_0}\Big)\frac{\mathrm{d}P_{0,1}}{\mathrm{d}P_{0,0}}(x) > \pi_{0,0} + \frac{\pi_1\pi_{0,0}}{\pi_0}\Big|A=0\Big\} \geq 0.$$

If $|DD(0)| \leq \delta$, then $\tau_{DD,\delta}^\star = 0$. By Proposition 7 (i), if $DD(0) > \delta$, then $0 < \tau_{DD,\delta}^\star < \min\{\pi_0,\pi_1\}$. If $DD(0) < -\delta$, then $\max\{-\pi_0,-\pi_1\} < \tau_{DD,\delta}^\star < 0$. Therefore, we have $|\tau_{DD,\delta}^\star| < \min\{\pi_0,\pi_1\}$.

$\square$

**Lemma 10.** *Recall that $g_{D,\tau}$ is defined in Theorem 2 and $D(\tau) = D(g_{D,\tau})$. For any fixed $\tau \in \mathbb{R}$,*

$$g_{D,\tau} = \arg\min_{f \in \mathcal{F}}\Big\{R(f) : \frac{\tau D(f)}{|\tau|} \leq \frac{\tau D(\tau)}{|\tau|}\Big\}.$$

*Moreover, for all classifiers $f' \in \arg\min_{f \in \mathcal{F}}\big\{R(f) : \tau D(f)/|\tau| \leq \tau D(\tau)/|\tau|\big\}$, $f' = g_{D,\tau}$ almost surely with respect to $P_{X,A}$. In addition, if $\tau \in [\min(0, \tau_{D,0}^\star), \max(0, \tau_{D,0}^\star)]$,*

$$g_{D,\tau} = \arg\min_{f \in \mathcal{F}}\Big\{R(f) : |D(f)| \leq \frac{\tau D(\tau)}{|\tau|}\Big\}.$$

*Proof.* If $\tau = 0$, then the result follows because $g_{D,0}$ is the unconstrained Bayes optimal classifier.

If $\tau \neq 0$, take $\phi_0(x,a) = \pi_{a,1}\mathrm{d}P_{a,1}/\mathrm{d}P_{a,0}(x) - \pi_{a,0}$ and $\phi_1(x,a) = s_{D,a}\mathrm{d}P_{a,1}/\mathrm{d}P_{a,0}(x) + b_{D,a}$ in Lemma 43.

Write

$$g_{D,\tau}(x,a) = \mathbb{1}\Big\{\phi_0(x,a) > |\tau|\frac{\tau\phi_1(x,a)}{|\tau|}\Big\}.$$

Define

$$Acc(f) = 1 - R(f) = \sum_{a \in \{0,1\}} \int_{\mathcal{X}} f(x,a) \left\{ \pi_{a,1} \frac{\mathrm{d}P_{a,1}}{\mathrm{d}P_{a,0}}(x) - \pi_{a,0} \right\} \mathrm{d}P_{a,0}(x) + \mathbb{P}(Y=0),$$

$$\tilde{D}_\tau(f) = \sum_{a \in \{0,1\}} \int_{\mathcal{X}} f(x,a) \frac{\tau}{|\tau|} \phi_1(x,a) dP_{a,0}(x).$$

Let

$$\mathcal{F}_{\tau,=} = \left\{ f \in \mathcal{F} : \tilde{D}_\tau(f) = \frac{\tau D(\tau)}{|\tau|} \right\}; \quad \mathcal{F}_{\tau,|\cdot|,\leq} = \left\{ f \in \mathcal{F} : |\tilde{D}_\tau(f)| \leq \frac{\tau D(\tau)}{|\tau|} \right\};$$

$$\mathcal{F}_{\tau,\leq} = \left\{ f \in \mathcal{F} : \tilde{D}_\tau(f) \leq \frac{\tau D(\tau)}{|\tau|} \right\}.$$

Since $|\tau| \geq 0$, by Lemma 43,

$$g_{D,\tau} \in \arg\max_{f \in \mathcal{F}_{\tau,\leq}} Acc(f).$$

Moreover, since $\mathrm{d}P_{a,1}/\mathrm{d}P_{a,0}(x)$ is a continuous random variable given $A = a \in \{0,1\}$ and $Y = y \in \{0,1\}$, we have $\mathbb{P}_{X|A=a,Y=y}\big(\pi_{a,1}\mathrm{d}P_{a,1}/\mathrm{d}P_{a,0}(x) - \pi_{a,0} = \tau\{s_{D,a}\mathrm{d}P_{a,1}/\mathrm{d}P_{a,0}(x) + b_{D,a}\}\big) = 0$. Thus, for all $f' \in \arg\max_{f \in \mathcal{F}_{\tau,\leq}} Acc(f)$, $f' = g_{D,\tau}$ almost surely with respect to $P_{X,A}$.

By Lemma 43, we have $g_{D,\tau} \in \arg\max_{f \in \mathcal{F}_{\tau,=}} Acc(f)$. By result (i) of Proposition 7, if $\tau^\star_{D,0} \geq 0$, then we have $D(\tau) \geq 0$ for $\tau \in [0, \tau^\star_{D,0}]$. If $\tau^\star_{D,0} \leq 0$, then $D(\tau) \leq 0$ for $\tau \in [\tau^\star_{D,0}, 0]$. Therefore, when $\tau \in [\min(0, \tau^\star_{D,0}), \max(0, \tau^\star_{D,0})]$, we have $\tau D(\tau) \geq 0$. Consequently, $g_{D,\tau} \in \mathcal{F}_{\tau,=} \subseteq \mathcal{F}_{\tau,|\cdot|,\leq} \subseteq \mathcal{F}_{\tau,\leq}$, we have

$$\max_{f \in \mathcal{F}_{\tau,\leq}} Acc(f) = Acc(g_{D,\tau}) = \max_{f \in \mathcal{F}_{\tau,=}} Acc(f) \leq \max_{f \in \mathcal{F}_{\tau,|\cdot|,\leq}} Acc(f) \leq \max_{f \in \mathcal{F}_{\tau,\leq}} Acc(f).$$

Thus, we conclude that

$$g_{D,\tau} = \arg\max_{f \in \mathcal{F}_{\tau,|\cdot|,\leq}} Acc(f) = \arg\min_{f \in \mathcal{F}} \left\{ R(f) : |D(f)| \leq \frac{\tau D(\tau)}{|\tau|} \right\}.$$

$\square$

## D Proof of Theorem 3

*Proof of Theorem 3.* Conditioning on the training data $\widetilde{\mathcal{D}}$, by the Dvoretzky–Kiefer–Wolfowitz inequality (Dvoretzky et al., 1956; Massart, 1990), we have, for any $a \in \{0,1\}$, that

$$\mathbb{P}\left[ \sup_\tau \Big| \int_{\mathcal{X}} \widehat{g}_{D,\tau}(x,a) \Big\{ s_{D,a} \frac{\mathrm{d}\widehat{P}_{a,1}}{\mathrm{d}\widehat{P}_{a,0}}(x) + b_{D,a} \Big\} \, \mathrm{d}\widehat{P}_{a,0}(x) \right.$$

$$\left. - \int_{\mathcal{X}} \widehat{g}_{D,\tau} \Big\{ s_{D,a} \frac{\mathrm{d}P_{a,1}}{\mathrm{d}P_{a,0}}(x) + b_{D,a} \Big\} \, \mathrm{d}P_{a,0}(x) \Big| \geq \epsilon \right] \lesssim \exp\{-(n_{a,1} \wedge n_{a,0})\epsilon^2\},$$

where $\widehat{g}_{D,\tau}$ is given in Algorithm 1. Thus, by taking $\epsilon \asymp \sqrt{\log(1/\eta)/(n_{a,1} \wedge n_{a,0})}$ and applying a union bound argument over $a \in \{0,1\}$ and the event in Lemma 32, the theorem holds by taking another expectation with respect to $\widetilde{\mathcal{D}}$. $\square$

## E Proofs for excess risk control

To simplify the notation, we write $\widehat{\tau} = \widehat{\tau}_{D,\delta}$ and $\tau^\star = \tau^\star_{D,\delta}$ in this section. For $a \in \{0,1\}$, denote

$$T_a(X) = \sum_{j=1}^\infty \left\{ \frac{(\zeta_{a,j} - \theta_{a,0,j})(\theta_{a,1,j} - \theta_{a,0,j})}{\lambda_{a,j}} - \frac{(\theta_{a,1,j} - \theta_{a,0,j})^2}{2\lambda_{a,j}} \right\} - \log\left\{ \frac{\pi_{a,0}}{\pi_{a,1} - \tau^\star(2a-1)} \right\}, \tag{6}$$

and

$$\widehat{T}_a(X) = \sum_{j=1}^{\infty} \left\{ \frac{(\widehat{\zeta}_{a,j} - \widehat{\theta}_{a,0,j})(\widehat{\theta}_{a,1,j} - \widehat{\theta}_{a,0,j})}{\widehat{\lambda}_{a,j}} - \frac{(\widehat{\theta}_{a,1,j} - \widehat{\theta}_{a,0,j})^2}{2\widehat{\lambda}_{a,j}} \right\} - \log\left\{ \frac{\widehat{\pi}_{a,0}}{\widehat{\pi}_{a,1} - \widehat{\tau}(2a-1)} \right\}.$$

We further let

$$H_a(X) = \sum_{j=1}^{\infty} \left\{ \frac{(\zeta_{a,j} - \theta_{a,0,j})(\theta_{a,1,j} - \theta_{a,0,j})}{\lambda_{a,j}} - \frac{(\theta_{a,1,j} - \theta_{a,0,j})^2}{2\lambda_{a,j}} \right\}, \tag{7}$$

and

$$\widehat{H}_a(X) = \sum_{j=1}^{J} \left\{ \frac{(\widehat{\zeta}_{a,j} - \widehat{\theta}_{a,0,j})(\widehat{\theta}_{a,1,j} - \widehat{\theta}_{a,0,j})}{\widehat{\lambda}_{a,j}} - \frac{(\widehat{\theta}_{a,1,j} - \widehat{\theta}_{a,0,j})^2}{2\widehat{\lambda}_{a,j}} \right\}. \tag{8}$$

With the above notation, we can rewrite $T_a(X)$ and $\widehat{T}_a(X)$ as

$$T_a(X) = H_a(X) - \log\left\{ \frac{\pi_{a,0}}{\pi_{a,1} - \tau^\star(2a-1)} \right\},$$

and

$$\widehat{T}_a(X) = \widehat{H}_a(X) - \log\left\{ \frac{\widehat{\pi}_{a,0}}{\widehat{\pi}_{a,1} - \widehat{\tau}(2a-1)} \right\}.$$

### E.1 Proof of Theorem 5

*Proof of Theorem 5.* Theorem 5 is a general version of Corollary 6. Most of the proof follows from a similar argument to the one used in the proof of Corollary 6. We only include the difference here.

**Upper bound on $|\widehat{\tau} - \tau^\star|$.** Consider the following event, $\mathcal{E}_D = \{\sup_{\tau \in \mathbb{R}} |\widehat{D}(\tau) - D(\tau)| \leq \epsilon_D\}$. Then, condition on $\mathcal{E}_D$ happening, by the argument in the proof of Lemma 14, we have that with probability at least $1 - \eta$ that

$$C_D|\widehat{\tau} - \tau^\star|^{\frac{1}{\gamma}} \lesssim \epsilon_D \mathbb{1}\{\tau^\star \neq 0\}.$$

Thus, it holds that $|\widehat{\tau} - \tau^\star| \lesssim \epsilon_D^\gamma \mathbb{1}\{\tau_D^\star \neq 0\}$.

**Upper bound on $d_E(\widehat{f}_{D,\delta}, f_{D,\delta}^\star)$.** The proof follows from a similar argument leading to (11) and it suffices to verify $\mathbb{P}\{\mathcal{E}_\tau\} \wedge \mathbb{P}(\mathcal{E}_{T_0} \cap \mathcal{E}_{T_1}) \geq 1 - \eta$, with $\mathcal{E}_\tau, \mathcal{E}_{T_0}$ and $\mathcal{E}_{T_1}$ defined in the proof of Corollary 6. To control $\mathcal{E}_\tau$, since by assumption, it holds that $\pi_{a,1} - \tau^\star s_{D,a} \geq c$ and $\pi_{a,0} + \tau^\star b_{D,a} \geq c$, we have

$$\widehat{\pi}_{a,1} - \widehat{\tau} s_{D,a} \geq \pi_{a,1} - \epsilon_\pi - \tau^\star s_{D,a} - |s_{D,a}|\epsilon_\tau \geq \frac{c}{2},$$

where the first inequality follows from the fact that $\tau^\star$ and $\widehat{\tau}$ share the same sign, and the last inequality follows from the fact that $|s_{D,a}| \asymp 1$ and $\epsilon_\pi, \epsilon_\tau \ll 1$. Similarly, we can verify that $\widehat{\pi}_{a,0} + \widehat{\tau} b_{D,a} > c/2$. To control $\mathcal{E}_{T_a}$ for $a \in \{0, 1\}$, by a similar argument as the one in Lemma 12, pick $\epsilon_{T_a} \lesssim \epsilon_\eta + \epsilon_\pi + \epsilon_\tau$, then by a union bound argument, we have that $\mathbb{P}\{\mathcal{E}_\tau\} \wedge \mathbb{P}\{\mathcal{E}_{T_0} \cap \mathcal{E}_{T_1}\} \geq 1 - \eta$. (11) thus leads to $d_E(\widehat{f}_{D,\delta}, f_{D,\delta}^\star) \lesssim (\epsilon_\eta + \epsilon_\pi + \epsilon_\tau)^2$.

**Upper bound on $|R(\widehat{f}_{D,\delta}) - R(f_{D,\delta}^\star)|$.** This follows directly from the fact that

$$|R(f) - R(f_{D,\delta}^\star)| = d_E(f, f_{D,\delta}^\star) + |\tau^\star\{D(f_{D,\delta}^\star) - D(f)\}|$$

$$\lesssim d_E(f, f_{D,\delta}^\star) + |\tau^\star|\sqrt{\frac{\log(1/\eta)}{n}}.$$

$\square$

## E.2 Proof of Corollary 6

**Corollary 11** (Excess risk control under disparity of opportunity). *Suppose that the training and calibration data $\mathcal{D} \cup \widetilde{\mathcal{D}}$ are generated under Assumptions 1 and 2. For any $\delta \geq 0$, the classifier $\widehat{f}_{\mathrm{DO},\delta}$ output by Algorithm 1 satisfies the following properties.*

1. *For any truncation level $J \in \mathbb{N}_+$ in **S1** in Algorithm 1 such that*

$$J \gtrsim \log^2(J) \quad and \quad J^{2\alpha+2}\log^2(J)\log(\widetilde{n}/\widetilde{\eta}) \lesssim \widetilde{n}, \tag{9}$$

*and any arbitrarily small constants $\eta \in (0, n^{-1/2} \wedge \widetilde{n}^{(\alpha-2\beta+1)/(2\beta-\alpha)})$, denote*

$$\epsilon_\eta = \begin{cases} \sqrt{\frac{J^{\alpha-2\beta+4}\log(\widetilde{n}/\eta)\log(1/\eta)}{\widetilde{n}}} + \sqrt{J^{\alpha-2\beta+1}\log(1/\eta)}, & \frac{\alpha+1}{2} < \beta \leq \frac{\alpha+2}{2}, \\ \sqrt{\frac{J^2\log(\widetilde{n}/\eta)\log(1/\eta)}{\widetilde{n}}} + \sqrt{J^{\alpha-2\beta+1}\log(1/\eta)}, & \frac{\alpha+2}{2} < \beta \leq \frac{\alpha+3}{2}, \\ \sqrt{\frac{J\log(\widetilde{n}/\eta)\log(1/\eta)}{\widetilde{n}}} + \sqrt{J^{\alpha-2\beta+1}\log(1/\eta)}, & \beta > \frac{\alpha+3}{2}. \end{cases}$$

*Then, it holds with probability at least $1 - \eta$ that*

$$d_E(\widehat{f}_{\mathrm{DO},\delta}, f_{\mathrm{DO},\delta}^\star) \lesssim \epsilon_\eta^2 + \frac{\log(1/\eta)\mathbb{1}\{\tau_{D,\delta}^\star \neq 0\}}{n},$$

*if we additionally assume that $\mathrm{DO}(0) \notin (\delta - \epsilon_\eta - \sqrt{\log(1/\eta)/n}, \delta] \cup [-\delta, -\delta + \epsilon_\eta + \sqrt{\log(1/\eta)/n})$.*

2. *If we further assume that $n \asymp \widetilde{n} \asymp N$ up to poly-logrithmic factors and select the truncation level $J$ in Algorithm 1 as*

$$J \asymp N^{\frac{1}{2\alpha+2}} \cdot \mathbb{1}\left\{\frac{\alpha+1}{2} < \beta < \frac{3\alpha+2}{2}\right\} + N^{\frac{1}{2\beta-\alpha}} \cdot \mathbb{1}\left\{\beta \geq \frac{3\alpha+2}{2}\right\},$$

*then it holds that*

$$d_E(\widehat{f}_{\mathrm{DO},\delta}, f_{\mathrm{DO},\delta}^\star) = \begin{cases} O_{\mathrm{p}}\left(N^{\frac{\alpha-2\beta+1}{2\alpha+2}}\right), & \frac{\alpha+1}{2} < \beta < \frac{3\alpha+2}{2}, \\ O_{\mathrm{p}}\left(N^{\frac{\alpha-2\beta+1}{2\beta-\alpha}}\right), & \beta \geq \frac{3\alpha+2}{2}, \end{cases}$$

*and*

$$|R(\widehat{f}_{\mathrm{DO},\delta}) - R(f_{\mathrm{DO},\delta}^\star)| \leq d_E(\widehat{f}_{\mathrm{DO},\delta}, f_{\mathrm{DO},\delta}^\star) + |\tau_{\mathrm{DO},\delta}^\star|O_{\mathrm{p}}\left(N^{-\frac{1}{2}}\right).$$

*Proof of Corollary 11.* For any classifier $f : \mathcal{X} \times \{0,1\} \to [0,1]$, by Proposition 1, the fairness-aware excess risk under DO is defined as

$$d_E^{DO}(f, f_{D,\delta}^\star) = \sum_{a \in \{0,1\}} \int_{\mathcal{X}} \{f(x,a) - f_{D,\delta}^\star(x,a)\}$$

$$\cdot \left[\pi_{a,0} + \{\tau^\star(2a-1) - \pi_{a,1}\}\frac{\mathrm{d}P_{a,1}}{\mathrm{d}P_{a,0}}(x)\right]\mathrm{d}P_{a,0}(x).$$

With the notation in (6), we have that

$$\frac{\mathrm{d}P_{a,1}}{\mathrm{d}P_{a,0}} = \frac{\pi_{a,0}}{\pi_{a,1} - \tau^\star(2a-1)}\exp\{T_a(X)\}.$$

In addition, consider the following event $\mathcal{E}_{\widehat{\tau}} = \{\widehat{\tau} \in (-\widehat{\pi}_{0,1}, \widehat{\pi}_{1,1})\}$. When $\mathcal{E}_{\widehat{\tau}}$ holds, we can then control the fairness-aware excess risk $d_E^{DO}(\widehat{f}, f^\star)$ by

$$d_E^{DO}(\widehat{f}, f^\star)$$

$$= \int_{\widehat{T}_1(x) \geq 0} \pi_{1,0} + (\tau^\star - \pi_{1,1})\frac{\mathrm{d}P_{1,1}}{\mathrm{d}P_{1,0}}(x)\,\mathrm{d}P_{1,0}(x)$$

$$-\int_{T_1(x)\geq 0}\pi_{1,0}+\big(\tau^\star-\pi_{1,1}\big)\frac{\mathrm{d}P_{1,1}}{\mathrm{d}P_{1,0}}(x)\,\mathrm{d}P_{1,0}(x)$$

$$+\int_{\widehat{T}_0(x)\geq 0}\pi_{0,0}+\big(-\tau^\star-\pi_{0,1}\big)\frac{\mathrm{d}P_{0,1}}{\mathrm{d}P_{0,0}}(x)\,\mathrm{d}P_{0,0}(x)$$

$$-\int_{T_0(x)\geq 0}\pi_{0,0}+\big(-\tau^\star-\pi_{0,1}\big)\frac{\mathrm{d}P_{0,1}}{\mathrm{d}P_{0,0}}(x)\,\mathrm{d}P_{0,0}(x)$$

$$\leq\int_{\widehat{T}_1(x)\geq 0,T_1(x)<0}\pi_{1,0}+\big(\tau^\star-\pi_{1,1}\big)\frac{\mathrm{d}P_{1,1}}{\mathrm{d}P_{1,0}}(x)\,\mathrm{d}P_{1,0}(x)$$

$$+\int_{\widehat{T}_0(x)\geq 0,T_0(X)<0}\pi_{0,0}+\big(-\tau^\star-\pi_{0,1}\big)\frac{\mathrm{d}P_{0,1}}{\mathrm{d}P_{0,0}}(x)\,\mathrm{d}P_{0,0}(x)$$

$$=\pi_{1,0}\cdot\int_{T_1(x)-\widehat{T}_1(x)\leq T_1(x)<0}1-e^{T_1(x)}\,\mathrm{d}P_{1,0}(x)$$

$$+\pi_{0,0}\cdot\int_{T_0(x)-\widehat{T}_0(x)\leq T_0(x)<0}1-e^{T_0(x)}\,\mathrm{d}P_{0,0}(x)$$

$$=\pi_{1,0}\cdot\mathbb{E}_{P_{1,0}}\Big[\{1-e^{T_1(X)}\}\mathbb{1}\Big\{T_1(X)-\widehat{T}_1(X)\leq T_1(X)<0\Big\}\Big]$$

$$+\pi_{0,0}\cdot\mathbb{E}_{P_{0,0}}\Big[\{1-e^{T_0(X)}\}\mathbb{1}\Big\{T_0(X)-\widehat{T}_0(X)\leq T_0(X)<0\Big\}\Big]$$

$$=\pi_{1,0}\cdot\mathbb{E}_{P_{1,0}}\Big[\{1-e^{T_1(X)}\}\mathbb{1}\Big\{T_1(X)-\widehat{T}_1(X)\leq T_1(X)<0\Big\}\mathbb{1}\Big\{|T_1(X)-\widehat{T}_1(X)|\leq\epsilon_{T_1}\Big\}\Big]$$

$$+\pi_{1,0}\cdot\mathbb{E}_{P_{1,0}}\Big[\{1-e^{T_1(X)}\}\mathbb{1}\Big\{T_1(X)-\widehat{T}_1(X)\leq T_1(X)<0\Big\}\mathbb{1}\Big\{|T_1(X)-\widehat{T}_1(X)|>\epsilon_{T_1}\Big\}\Big]$$

$$+\pi_{0,0}\cdot\mathbb{E}_{P_{0,0}}\Big[\{1-e^{T_0(X)}\}\mathbb{1}\Big\{T_0(X)-\widehat{T}_0(X)\leq T_0(X)<0\Big\}\mathbb{1}\Big\{|T_0(X)-\widehat{T}_0(X)|\leq\epsilon_{T_0}\Big\}\Big]$$

$$+\pi_{0,0}\cdot\mathbb{E}_{P_{0,0}}\Big[\{1-e^{T_0(X)}\}\mathbb{1}\Big\{T_0(X)-\widehat{T}_0(X)\leq T_0(X)<0\Big\}\mathbb{1}\Big\{|T_0(X)-\widehat{T}_0(X)|>\epsilon_{T_0}\Big\}\Big].$$

For any $a\in\{0,1\}$, write $X_a\sim\mathcal{GP}(\mu_{a,0},K_a)$. We further denote

$$\mathcal{E}_{T_a}=\Big\{|\widehat{T}_a(X_a)-T_a(X_a)|\leq\epsilon_T\Big\}. \tag{10}$$

Consequently, we can further upper bound the fairness-aware excess risk by

$$d_E^{DO}(\widehat{f},f^\star)\leq\pi_{1,0}\cdot\epsilon_T\cdot\mathbb{E}_{P_{1,0}}\Big[\mathbb{1}\big\{-\epsilon_T\leq T_1(X)<0\big\}\Big]$$

$$+\pi_{0,0}\cdot\epsilon_T\cdot\mathbb{E}_{P_{0,0}}\Big[\mathbb{1}\big\{-\epsilon_T\leq T_0(X)<0\big\}\Big]$$

$$+\pi_{0,0}\mathbb{P}(\mathcal{E}_{T_0}^c)+\pi_{1,0}\mathbb{P}(\mathcal{E}_{T_1}^c)$$

$$\leq\epsilon_T^2\cdot\Big(\sup_{-\epsilon_T\leq t<0}f_{0,T_1}(t)\Big)+\epsilon_T^2\cdot\Big(\sup_{-\epsilon_T\leq t<0}f_{0,T_0}(t)\Big)+\mathbb{P}(\mathcal{E}_{T_0}^c)+\mathbb{P}(\mathcal{E}_{T_1}^c)$$

$$\lesssim\epsilon_T^2\cdot\frac{1}{\|\mu_{1,1}-\mu_{1,0}\|_{K_1}}\exp\Big\{-\frac{\|\mu_{1,1}-\mu_{1,0}\|_{K_1}^4\vee 1}{\|\mu_{1,1}-\mu_{1,0}\|_{K_1}^2}\Big\}$$

$$+\epsilon_T^2\cdot\frac{1}{\|\mu_{0,1}-\mu_{0,0}\|_{K_0}}\exp\Big\{-\frac{\|\mu_{0,1}-\mu_{0,0}\|_{K_0}^4\vee 1}{\|\mu_{0,1}-\mu_{0,0}\|_{K_0}^2}\Big\}+\mathbb{P}(\mathcal{E}_{T_0}^c)+\mathbb{P}(\mathcal{E}_{T_1}^c)$$

$$\lesssim \epsilon_T^2 + \mathbb{P}(\mathcal{E}_{T_0}^c \cup \mathcal{E}_{T_1}^c). \tag{11}$$

where for $a \in \{0,1\}$, $f_{0,T_a}$ denotes the density for $T_a$ given $Y = 0$, the first inequality holds as $1 - \exp\{T_a(x)\} \leq -T_a(X) \leq \widehat{T}_a(X) - T_a(X) \leq \epsilon_{T_a}$ and the last inequality follows from Lemma 17 and the fact that

$$T_a(X_a)|Y = 0 \sim N\Big( -\frac{\|\mu_{a,1} - \mu_{a,0}\|_{K_a}^2}{2} - \log\Big\{\frac{\pi_{a,0}}{\pi_{a,1} - \tau^\star(2a-1)}\Big\}, \ \|\mu_{a,1} - \mu_{a,0}\|_{K_a}^2\Big).$$

In the rest of the proof, it suffices to control the probability for $\mathcal{E}_\tau, \mathcal{E}_{T_1}, \mathcal{E}_{T_0}$ happening. By Lemmas 14, 17, 31 and a union bound argument, it holds with probability at least $1 - \eta/4$ that

$$\widehat{\pi}_{1,1} - \widehat{\tau} \geq \pi_{1,1} - \tau^\star - \Big\{\sqrt{\frac{\log(1/\eta)}{\widetilde{n}}} + \epsilon_H + \sqrt{\frac{\log(1/\eta)}{n}}\Big\} \geq c - \frac{c}{2} = \frac{c}{2}.$$

Analogously, we can also show that with probability at least $1 - \eta/4$ that, when $\widehat{\tau} < 0$,

$$\widehat{\pi}_{0,1} + \widehat{\tau} \geq \pi_{0,1} + \tau^\star - \Big\{\sqrt{\frac{\log(1/\eta)}{\widetilde{n}}} + \epsilon_H + \sqrt{\frac{\log(1/\eta)}{n}}\Big\} \geq \frac{c}{2}.$$

Moreover, taking $\epsilon_T = \epsilon_H + \sqrt{\log(1/\eta)/n} \cdot \mathbb{1}\{|DO(0)| > \delta - \epsilon_H - \sqrt{\log(1/\eta)/n}\}$ in (10), by Lemma 12 and an additional union bound argument, it holds that $\mathbb{P}\{\mathcal{E}_\tau\} \wedge \mathbb{P}(\mathcal{E}_{T_0} \cap \mathcal{E}_{T_1}) \geq 1 - \eta$. Thus, conditioning on $\mathcal{E}_\tau$ happening, we have with probability at least $1 - \eta$ that

$$d_E^{DO}(\widehat{f}, f^\star) \lesssim \Big\{\epsilon_H + \sqrt{\frac{\log(1/\eta)}{n}} \cdot \mathbb{1}\Big\{|DO(0)| > \delta - \epsilon_H - \sqrt{\frac{\log(1/\eta)}{n}}\Big\}\Big\}^2 + \eta$$

$$\lesssim \Big\{\epsilon_H + \sqrt{\frac{\log(1/\eta)}{n}} \cdot \mathbb{1}\Big\{|DO(0)| > \delta - \epsilon_H - \sqrt{\frac{\log(1/\eta)}{n}}\Big\}\Big\}^2,$$

whenever $\eta \in (0, n^{-1/2} \wedge \widetilde{n}^{(\alpha-2\beta+1)/(2\beta-\alpha)})$. $\qquad\square$

## E.3  Control of $|\widehat{T}_a(X) - T_a(X)|$

**Lemma 12.** *Suppose the training and calibration data $\mathcal{D} \cup \widetilde{\mathcal{D}}$ are generated under Assumptions 1 and 2. Then for any $a \in \{0,1\}$, $J \in \mathbb{N}_+$ such that*

$$J \gtrsim \log^2(J) \quad and \quad J^{2\alpha+2}\log^2(J)\log(\widetilde{n}/\widetilde{\eta}) \lesssim \widetilde{n},$$

*and any constant $\eta \in (0, 1/2)$, it holds that*

$$\mathbb{P}_{a,0}\Big[|\widehat{T}_a(X) - T_a(X)| \lesssim \epsilon_H + \sqrt{\frac{\log(1/\eta)}{n}} \cdot \mathbb{1}\Big\{|DO(0)| > \delta - \epsilon_H - \sqrt{\frac{\log(1/\eta)}{n}}\Big\}\Big] \geq 1 - \eta,$$

*where $\epsilon_H$ is defined in (12).*

*Proof.* It follows from the triangle inequality that

$$|\widehat{T}_a(X) - T_a(X)| \leq |\widehat{H}_a(X) - H_a(X)| + |\log(\widehat{\pi}_{a,0}) - \log(\pi_{a,0})|$$
$$+ |\log\{\widehat{\pi}_{a,1} - \widehat{\tau}(2a-1)\} - \log\{\pi_{a,1} - \tau^\star(2a-1)\}|.$$

For $X \sim \mathcal{GP}(\mu_{a,0}, K_a)$, consider the following events,

$$\mathcal{E}_\pi = \Big\{|\widehat{\pi}_{a,y} - \pi_{a,y}| \lesssim \sqrt{\frac{\log(1/\eta)}{\widetilde{n}}}, \quad a, y \in \{0,1\}\Big\},$$

$$\mathcal{E}_H = \Big\{|\widehat{H}_a(X) - H_a(X)| \lesssim \epsilon_H, \quad a \in \{0,1\}\Big\},$$

and

$$\mathcal{E}_\tau = \Big\{|\widehat{\tau} - \tau^\star| \lesssim \Big\{\epsilon_H + \sqrt{\frac{\log(1/\eta)}{n}}\Big\} \cdot \mathbb{1}\Big\{|DO(0)| > \delta - \epsilon_H - \sqrt{\frac{\log(1/\eta)}{n}}\Big\}\Big\}.$$

By Lemmas 13, 14, 31 and a union bound argument, we have that for any $a \in \{0,1\}$, $\mathbb{P}_{a,0}(\mathcal{E}_\pi \cap \mathcal{E}_H \cap \mathcal{E}_\tau) \geq 1 - \eta$. The rest of the proof is constructed conditioning on $\mathcal{E}_\pi \cap \mathcal{E}_H \cap \mathcal{E}_\tau$ happening. Therefore, we have that

$$|\widehat{T}_a(X) - T_a(X)| \lesssim |\widehat{H}_a(X) - H_a(X)| + |\widehat{\pi}_{a,0} - \pi_{a,0}| + |\widehat{\pi}_{a,1} - \pi_{a,1}| + |\widehat{\tau} - \tau^\star|$$

$$\lesssim \epsilon_H + \sqrt{\frac{\log(1/\eta)}{n}} \cdot \mathbb{1}\Big\{|DO(0)| > \delta - \epsilon_H - \sqrt{\frac{\log(1/\eta)}{n}}\Big\}.$$

$\square$

## E.4 Control of $|\widehat{H}_a(X) - H_a(X)|$

**Lemma 13.** *Under the same condition of Lemma 12, for any $a, y \in \{0,1\}$ and small constant $\eta \in (0, 1/2)$, it holds that*

$$\mathbb{P}_{a,y}\Big\{|\widehat{H}_a(X) - H_a(X)| \lesssim \epsilon_H\Big\} \geq 1 - \eta,$$

*where*

$$
\epsilon_H = \begin{cases}
\sqrt{\frac{J^{\alpha-2\beta+4}\log(\widetilde{n}/\eta)\log(1/\eta)}{\widetilde{n}}} + \sqrt{J^{\alpha-2\beta+1}\log(1/\eta)} & \text{when } \frac{\alpha+1}{2} < \beta \leq \frac{\alpha+2}{2}, \\
\sqrt{\frac{J^2\log(\widetilde{n}/\eta)\log(1/\eta)}{\widetilde{n}}} + \sqrt{J^{\alpha-2\beta+1}\log(1/\eta)}, & \text{when } \frac{\alpha+2}{2} < \beta \leq \frac{\alpha+3}{2}, \quad (12) \\
\sqrt{\frac{J\log(\widetilde{n}/\eta)\log(1/\eta)}{\widetilde{n}}} + \sqrt{J^{\alpha-2\beta+1}\log(1/\eta)}, & \text{when } \beta > \frac{\alpha+3}{2}.
\end{cases}
$$

*Proof.* Note that

$$|\widehat{H}_a(X) - H_a(X)|$$

$$\leq \Big| \sum_{j=1}^J \frac{(\zeta_{a,j} - \theta_{a,0,j})(\theta_{a,1,j} - \theta_{a,0,j})}{\lambda_{a,j}} - \sum_{j=1}^J \frac{(\widehat{\zeta}_j - \widehat{\theta}_{a,0,j})(\widehat{\theta}_{a,1,j} - \widehat{\theta}_{a,0,j})}{\widehat{\lambda}_{a,j}} \Big|$$

$$+ \Big| \sum_{j=J+1}^\infty \frac{(\zeta_{a,j} - \theta_{a,0,j})(\theta_{a,1,j} - \theta_{a,0,j})}{\lambda_{a,j}} \Big|$$

$$+ \Big| \sum_{j=1}^J \frac{(\theta_{a,1,j} - \theta_{a,0,j})^2}{2\lambda_{a,j}} - \frac{(\widehat{\theta}_{a,1,j} - \widehat{\theta}_{a,0,j})^2}{2\widehat{\lambda}_{a,j}} \Big|$$

$$+ \Big| \sum_{j=J+1}^\infty \frac{(\theta_{a,1,j} - \theta_{a,0,j})^2}{2\lambda_{a,j}} \Big|.$$

In the case when $X \sim \mathcal{GP}(\mu_{a,0}, K_a)$, $a \in \{0,1\}$, the lemma thus follows by applying a union bound argument to the results in Lemmas 19, 20 and 30, together with the fact that under Assumptions 2a and 2c, we have

$$\Big| \sum_{j=J+1}^\infty \frac{(\theta_{a,1,j} - \theta_{a,0,j})^2}{2\lambda_{a,j}} \Big| \lesssim J^{\alpha-2\beta+1}.$$

The case when $X \sim \mathcal{GP}(\mu_{a,1}, K_a)$, $a \in \{0,1\}$ can be justified similarly, hence is omitted here. $\square$

## E.5 Control of $|\widehat{\tau} - \tau^\star|$

**Lemma 14.** *Under the same condition as Lemma 12, for any small constant $\eta \in (0, 1/2)$, it holds with probability at least $1 - \eta$ that*

$$|\widehat{\tau} - \tau^\star| \lesssim \Big\{\epsilon_H + \sqrt{\frac{\log(1/\eta)}{n}}\Big\} \cdot \mathbb{1}\Big\{|DO(0)| > \delta - \epsilon_H - \sqrt{\frac{\log(1/\eta)}{n}}\Big\},$$

*where $\epsilon_H$ is explicitly defined in (12).*

*Proof.* Denote $\mathcal{E}_{DO} = \{\sup_{\tau \in \mathbb{R}} |\widehat{DO}(\tau) - DO(\tau)| \leq \epsilon_{DO}\}$. Taking

$$\epsilon_{DO} \asymp \epsilon_H + \sqrt{\frac{\log(1/\eta)}{n}},$$

it holds from Lemma 15 that $\mathbb{P}(\mathcal{E}_{DO}^c) \leq \eta$. The rest of the proof is divided into four cases.

**Case 1: When $|DO(0)| \leq \delta - \epsilon_{DO}$.** In this case, we have that $\tau^\star = 0$. Moreover, under the event $\mathcal{E}_{DO}$, $|\widehat{DO}(0)| \leq \delta$, then $\widehat{\tau} = 0$. Consequently, $|\widehat{\tau} - \tau^\star| = 0$.

**Case 2 : When $\tau^\star = 0$ and $\delta - \epsilon_{DO} < |DO(0)| \leq \delta$.** In this case, we have that

$$\mathbb{P}(\widehat{\tau} - \tau^\star > \varepsilon) = \mathbb{P}(\widehat{\tau} > \varepsilon)$$

$$\leq \mathbb{P}(\widehat{\tau} > \varepsilon, \mathcal{E}_{DO}) + \mathbb{P}(\mathcal{E}_{DO}^c)$$

$$\leq \mathbb{P}(\widehat{\tau} > \varepsilon, \widehat{DO}(0) > \delta, \mathcal{E}_{DO}) + \mathbb{P}(\mathcal{E}_{DO}^c)$$

$$\leq \mathbb{P}(\widehat{DO}(\varepsilon) > \delta, \mathcal{E}_{DO}) + \mathbb{P}(\mathcal{E}_{DO}^c)$$

$$\leq \mathbb{P}(\widehat{DO}(\varepsilon) - DO(\varepsilon) > \delta - DO(0) + c_1\varepsilon, \mathcal{E}_{DO}) + \mathbb{P}(\mathcal{E}_{DO}^c)$$

$$\leq \mathbb{P}(\mathcal{E}_{DO}^c),$$

where the second inequality follows as $\widehat{\tau} > 0$, the third inequality follows from Lemma 18, the fourth inequality follows as $DO(\epsilon) \leq DO(0) - c_1\varepsilon$ by Lemma 16 and the last inequality follows by taking $\varepsilon > \epsilon_{DO}/c_1$. Analogously,

$$\mathbb{P}(\widehat{\tau} < \tau^\star - \varepsilon) = \mathbb{P}(\widehat{\tau} < -\varepsilon)$$

$$\leq \mathbb{P}(\widehat{\tau} < -\varepsilon, \mathcal{E}_{DO}) + \mathbb{P}(\mathcal{E}_{DO}^c)$$

$$= \mathbb{P}(\widehat{\tau} < -\varepsilon, \widehat{DO}(0) < -\delta, \mathcal{E}_{DO}) + \mathbb{P}(\mathcal{E}_{DO}^c)$$

$$\leq \mathbb{P}(\widehat{DO}(-\varepsilon) < -\delta, \mathcal{E}_{DO}) + \mathbb{P}(\mathcal{E}_{DO}^c)$$

$$\leq \mathbb{P}(\widehat{DO}(-\varepsilon) - DO(-\varepsilon) < -\delta - DO(0) - c_1\varepsilon, \mathcal{E}_{DO}) + \mathbb{P}(\mathcal{E}_{DO}^c)$$

$$\leq \mathbb{P}(\mathcal{E}_{DO}^c),$$

where the second inequality follows from Lemma 18, the third inequality follows from the fact that $DO(-\varepsilon) \geq DO(0) + c_1\varepsilon$ by Lemma 16 and the last inequality follows by taking $\varepsilon > \epsilon_{DO}/c_1$.

**Case 3: When $DO(\tau^\star) = \delta$ and $\tau^\star > 0$.** In this case, by Proposition 7, it holds that $DO(0) > \delta$. Hence,

$$\mathbb{P}(\widehat{\tau} > \tau^\star + \varepsilon) \leq \mathbb{P}(\widehat{\tau} > \tau^\star + \varepsilon, \mathcal{E}_{DO}) + \mathbb{P}(\mathcal{E}_{DO}^c)$$

$$= \mathbb{P}(\widehat{\tau} > \tau^\star + \varepsilon, \widehat{DO}(0) > \delta, \mathcal{E}_{DO}) + \mathbb{P}(\mathcal{E}_{DO}^c)$$

$$\leq \mathbb{P}(\widehat{DO}(\tau^\star + \varepsilon) > \delta, \mathcal{E}_{DO}) + \mathbb{P}(\mathcal{E}_{DO}^c)$$

$$= \mathbb{P}(\widehat{DO}(\tau^\star + \varepsilon) - DO(\tau^\star + \varepsilon) > DO(\tau^\star) - DO(\tau^\star + \varepsilon), \mathcal{E}_{DO}) + \mathbb{P}(\mathcal{E}_{DO}^c)$$

$$\leq \mathbb{P}(\epsilon_{DO} > c_1\varepsilon) + \mathbb{P}(\mathcal{E}_{DO}^c) = \mathbb{P}(\mathcal{E}_{DO}^c),$$

where the first equality follows as $\mathbb{P}(\widehat{\tau} > \tau^\star + \varepsilon, \widehat{DO}(0) \leq \delta) = 0$, the second inequality follows from Lemma 18, the third inequality follows from Lemma 16 and the last equality follows by taking $\varepsilon > \epsilon_{DO}/c_1$. Similarly, by taking $\varepsilon > \epsilon_{DO}/c_1$, it also holds that

$$\mathbb{P}(\widehat{\tau} < \tau^\star - \varepsilon)$$

$$\leq \mathbb{P}(\widehat{\tau} < \tau^\star - \varepsilon, \mathcal{E}_{DO}) + \mathbb{P}(\mathcal{E}_{DO}^c)$$

$$\leq \mathbb{P}(\widehat{DO}(\tau^\star - \varepsilon) \leq \delta, \mathcal{E}_{DO}) + \mathbb{P}(\mathcal{E}_{DO}^c)$$

$$= \mathbb{P}\big(DO(\tau^\star - \varepsilon) - \widehat{DO}(\tau^\star - \varepsilon) \geq DO(\tau^\star - \varepsilon) - DO(\tau^\star), \mathcal{E}_{DO}\big) + \mathbb{P}(\mathcal{E}_{DO}^c)$$

$$\leq \mathbb{P}(\epsilon_{DO} > c_1 \varepsilon) + \mathbb{P}(\mathcal{E}_{DO}^c)$$

$$\leq \mathbb{P}(\mathcal{E}_{DO}^c).$$

**Case 4: When $DO(\tau^\star) = -\delta$ and $\tau^\star < 0$.** In this case, it holds that $DO(0) < -\delta$. The rest of the proof follows similarly to the proof for **Case 3**, we include them for completeness. For $\varepsilon > \epsilon_{DO}/c_1$,

$$\mathbb{P}(\widehat{\tau} > \tau^\star + \varepsilon)$$

$$\leq \mathbb{P}(\widehat{\tau} > \tau^\star + \varepsilon, \mathcal{E}_{DO}) + \mathbb{P}(\mathcal{E}_{DO}^c)$$

$$\leq \mathbb{P}(\widehat{DO}(\tau^\star + \varepsilon) \geq -\delta, \mathcal{E}_{DO}) + \mathbb{P}(\mathcal{E}_{DO}^c)$$

$$= \mathbb{P}(\widehat{DO}(\tau^\star + \varepsilon) - DO(\tau^\star + \varepsilon) \geq DO(\tau^\star) - DO(\tau^\star + \varepsilon), \mathcal{E}_{DO}) + \mathbb{P}(\mathcal{E}_{DO}^c)$$

$$\leq \mathbb{P}(\mathcal{E}_{DO}^c).$$

Since $\mathbb{P}(\widehat{\tau} < \tau^\star - \varepsilon, \widehat{DO}(0) \geq -\delta) = 0$, for $\varepsilon > \epsilon_{DO}/c_1$,

$$\mathbb{P}(\widehat{\tau} < \tau^\star - \varepsilon) \leq \mathbb{P}(\widehat{\tau} < \tau^\star - \varepsilon, \mathcal{E}_{DO}) + \mathbb{P}(\mathcal{E}_{DO}^c)$$

$$= \mathbb{P}(\widehat{\tau} < \tau^\star - \varepsilon, \widehat{DO}(0) < -\delta, \mathcal{E}_{DO}) + \mathbb{P}(\mathcal{E}_{DO}^c)$$

$$\leq \mathbb{P}(\widehat{DO}(\tau^\star - \varepsilon) < -\delta, \mathcal{E}_{DO}) + \mathbb{P}(\mathcal{E}_{DO}^c)$$

$$= \mathbb{P}(\widehat{DO}(\tau^\star - \varepsilon) - DO(\tau^\star - \varepsilon) < -\delta - DO(\tau^\star - \varepsilon), \mathcal{E}_{DO}) + \mathbb{P}(\mathcal{E}_{DO}^c)$$

$$\leq \mathbb{P}(\mathcal{E}_{DO}^c).$$

The lemma thus follows by combining results from all cases together.

$\square$

### E.6 Control of $|\widehat{DO}(\tau) - DO(\tau)|$

**Lemma 15.** *Under the same condition of Lemma 12, for any small constant $\eta \in (0, 1/2)$, it holds with probability at least $1 - \eta$ that*

$$\sup_{\tau \in \mathbb{R}} |\widehat{DO}(\tau) - DO(\tau)| \lesssim \epsilon_H + \sqrt{\frac{\log(1/\eta)}{n}},$$

*where $\epsilon_H$ is explicitly defined in* (12).

*Proof.* Note that, by triangle inequality,

$$\sup_{\tau \in \mathbb{R}} |\widehat{DO}(\tau) - DO(\tau)|$$

$$\leq \sup_{\tau \in \mathbb{R}} |\widehat{DO}(\tau) - \mathbb{E}\{\widehat{DO}(\tau)|\widetilde{\mathcal{D}}\}| + \sup_{\tau \in \mathbb{R}} |\mathbb{E}\{\widehat{DO}(\tau)|\widetilde{\mathcal{D}}\} - DO(\tau)|$$

$$= (I) + (II). \tag{13}$$

**Step 1: Upper bound on $(I)$.** To control $(I)$, by the Dvoretzky–Kiefer–Wolfowitz inequality (Dvoretzky et al., 1956; Massart, 1990), we have that

$$\mathbb{P}\left[\sup_{\tau \in \mathbb{R}} \left| \frac{1}{n_{1,1}} \sum_{i=1}^{n_{1,1}} \mathbb{1}\big\{(\widehat{\pi}_{1,1} - \tau)\widehat{\eta}_1(X_{1,1,i}) > \widehat{\pi}_{1,0}\big\} - \mathbb{P}\big\{(\widehat{\pi}_{1,1} - \tau)\widehat{\eta}_1(X_{1,1,i}) > \widehat{\pi}_{1,0}|\widetilde{\mathcal{D}}\big\} \right| \geq \epsilon \Big| \widetilde{\mathcal{D}} \right]$$

$$\leq 2\exp\{-2n_{1,1}\epsilon^2\},$$

$$\mathbb{P}\left[\sup_{\tau\in\mathbb{R}}\left|\frac{1}{n_{0,1}}\sum_{i=1}^{n_{0,1}}\mathbb{1}\left\{(\widehat{\pi}_{0,1}+\tau)\widehat{\eta}_0(X_{0,1,i})>\widehat{\pi}_{0,0}\right\}-\mathbb{P}\left\{(\widehat{\pi}_{0,1}+\tau)\widehat{\eta}_0(X_{0,1,i})>\widehat{\pi}_{0,0}|\widetilde{\mathcal{D}}\right\}\right|\geq\epsilon\Big|\widetilde{\mathcal{D}}\right]$$

$$\leq 2\exp\{-2n_{0,1}\epsilon^2\}.$$

Thus, by a union bound argument, it holds with probability at least $1-\eta/2$ that

$$\sup_{\tau\in\mathbb{R}}|\widehat{DO}(\tau)-\mathbb{E}\{\widehat{DO}(\tau)|\widetilde{\mathcal{D}}\}|\lesssim\sqrt{\frac{\log(1/\eta)}{n_{1,1}}}+\sqrt{\frac{\log(1/\eta)}{n_{0,1}}}\lesssim\sqrt{\frac{\log(1/\eta)}{n}}, \qquad (14)$$

where the last inequality follows by further conditioning on the event in Lemma 32 and a union bound argument.

**Step 2: Upper bound on $(II)$.** Note that

$$\mathbb{E}\{\widehat{DO}(\tau)|\widetilde{\mathcal{D}}\}$$

$$= \mathbb{P}_{1,1}\left\{(\widehat{\pi}_{1,1}-\tau)\widehat{\eta}_1(X)>\widehat{\pi}_{1,0}|\widetilde{\mathcal{D}}\right\}-\mathbb{P}_{0,1}\left\{(\widehat{\pi}_{0,1}+\tau)\widehat{\eta}_0(X)>\widehat{\pi}_{0,0}|\widetilde{\mathcal{D}}\right\}$$

$$= \begin{cases} \mathbb{P}_{1,1}\left[\log\{\widehat{\eta}_1(X)\}>\log(\frac{\widehat{\pi}_{1,0}}{\widehat{\pi}_{1,1}-\tau})\big|\widetilde{\mathcal{D}}\right]-\mathbb{P}_{0,1}\left[\log\{\widehat{\eta}_0(X)\}>\log(\frac{\widehat{\pi}_{0,0}}{\widehat{\pi}_{0,1}+\tau})\big|\widetilde{\mathcal{D}}\right], \\ \qquad\qquad\qquad\qquad\qquad\qquad\qquad\qquad\qquad\qquad \tau\in(-\widehat{\pi}_{0,1},\widehat{\pi}_{1,1}), \\ \qquad\qquad \mathbb{P}_{1,1}\left[\log\{\widehat{\eta}_1(X)\}>\log(\frac{\widehat{\pi}_{1,0}}{\widehat{\pi}_{1,1}-\tau})\big|\widetilde{\mathcal{D}}\right], \\ \qquad\qquad\qquad\qquad\qquad\qquad\qquad\qquad\qquad\qquad \tau\leq-\widehat{\pi}_{0,1}, \\ \qquad\qquad -\mathbb{P}_{0,1}\left[\log\{\widehat{\eta}_0(X)\}>\log(\frac{\widehat{\pi}_{0,0}}{\widehat{\pi}_{0,1}+\tau})\big|\widetilde{\mathcal{D}}\right], \\ \qquad\qquad\qquad\qquad\qquad\qquad\qquad\qquad\qquad\qquad \tau\geq\widehat{\pi}_{1,1}. \end{cases}$$

We denote $\Pi=(-\min(\pi_{0,1},\widehat{\pi}_{0,1}),\min(\pi_{1,1},\widehat{\pi}_{1,1}))$. In the rest of the proof, to control $(II)$, we will consider various cases.

**Step 2-Case 1:** $\tau\in\Pi$**.** Note that for $\tau\in\Pi$,

$$\mathbb{E}\{\widehat{DO}(\tau)|\widetilde{\mathcal{D}}\}$$

$$= \mathbb{P}\left\{(\widehat{\pi}_{1,1}-\tau)\widehat{\eta}_1(X_{1,1,i})>\widehat{\pi}_{1,0}|\widetilde{\mathcal{D}}\right\}-\mathbb{P}\left\{(\widehat{\pi}_{0,1}+\tau)\widehat{\eta}_0(X_{0,1,i})>\widehat{\pi}_{0,0}|\widetilde{\mathcal{D}}\right\}$$

$$= \mathbb{P}_{1,1}\left\{(\widehat{\pi}_{1,1}-\tau)\widehat{\eta}_1(X)>\widehat{\pi}_{1,0}|\widetilde{\mathcal{D}}\right\}-\mathbb{P}_{0,1}\left\{(\widehat{\pi}_{0,1}+\tau)\widehat{\eta}_0(X)>\widehat{\pi}_{0,0}|\widetilde{\mathcal{D}}\right\}$$

$$= \int_{\widehat{H}_1(x)-\log\{\widehat{\pi}_{1,0}/(\widehat{\pi}_{1,1}-\tau)\}>0}\mathrm{d}P_{1,1}(x)-\int_{\widehat{H}_0(x)-\log\{\widehat{\pi}_{0,0}/(\widehat{\pi}_{0,1}+\tau)\}>0}\mathrm{d}P_{0,1}(x),$$

with $\widehat{H}_a$ defined in (8). Similarly, we can rewrite $DO(\tau)$ as

$$DO(\tau)=\int_{H_1(x)-\log\{\pi_{1,0}/(\pi_{1,1}-\tau)\}>0}\mathrm{d}P_{1,1}(x)-\int_{H_0(x)-\log\{\pi_{0,0}/(\pi_{0,1}+\tau)\}>0}\mathrm{d}P_{0,1}(x),$$

with $H_a$ defined in (7). For $a\in\{0,1\}$ and $X\sim\mathcal{GP}(\mu_{a,1},K_a)$, consider the following event,

$$\mathcal{E}_{H_a}=\Big\{\sup_{\tau\in\Pi}\Big|\widehat{H}_a(X)-\log\Big(\frac{\widehat{\pi}_{a,0}}{\widehat{\pi}_{a,1}+(1-2a)\tau}\Big)-H_a(X)+\log\Big(\frac{\pi_{a,0}}{\pi_{a,1}+(1-2a)\tau}\Big)\Big|$$

$$\leq\epsilon_H,\ a\in\{0,1\}\Big\}.$$

Note that

$$\sup_{\tau\in\Pi}\Big|\widehat{H}_a(X)-\log\Big(\frac{\widehat{\pi}_{a,0}}{\widehat{\pi}_{a,1}+(1-2a)\tau}\Big)-H_a(X)+\log\Big(\frac{\pi_{a,0}}{\pi_{a,1}+(1-2a)\tau}\Big)\Big|$$

$$= \sup_{\tau\in\Pi}\Big\{|\widehat{H}_a(X)-H_a(X)|+|\log(\widehat{\pi}_{a,0})-\log(\pi_{a,0})|$$

$$+ |\log(\widehat{\pi}_{a,1} + (1-2a)\tau) - \log(\pi_{a,1} + (1-2a)\tau)|\Big\}$$

$$\lesssim |\widehat{H}_a(X) - H_a(X)| + |\widehat{\pi}_{a,0} - \pi_{a,0}| + |\widehat{\pi}_{a,1} - \pi_{a,1}|.$$

Thus, picking $\epsilon_H$ as the one in Equation (12), by Lemmas 13, 31 and a union bound argument, we have that $\mathbb{P}_{0,1}(\mathcal{E}_{H_0}) + \mathbb{P}_{1,1}(\mathcal{E}_{H_1}) \geq 1 - \eta/2$.

For any $\tau \in \Pi$, it holds that

$$\mathbb{E}\{\widehat{DO}(\tau)|\widetilde{\mathcal{D}}\} - DO(\tau)$$

$$= \int_{\widehat{H}_1(x) - \log\{\widehat{\pi}_{1,0}/(\widehat{\pi}_{1,1} - \tau)\} > 0} dP_{1,1}(x) - \int_{H_1(x) - \log\{\pi_{1,0}/(\pi_{1,1} - \tau)\} > 0} dP_{1,1}(x)$$

$$+ \int_{H_0(x) - \log\{\pi_{0,0}/(\pi_{0,1} + \tau)\} > 0} dP_{0,1}(x) - \int_{\widehat{H}_0(x) - \log\{\widehat{\pi}_{0,0}/(\widehat{\pi}_{0,1} + \tau)\} > 0} dP_{0,1}(x)$$

$$\leq \int_{0 > H_1(x) - \log\{\pi_{1,0}/(\pi_{1,1} - \tau)\} > H_1(x) - \log\{\pi_{1,0}/(\pi_{1,1} - \tau)\} - \widehat{H}_1(x) + \log\{\widehat{\pi}_{1,0}/(\widehat{\pi}_{1,1} - \tau)\}} dP_{1,1}(x)$$

$$+ \int_{0 < H_0(x) - \log\{\pi_{0,0}/(\pi_{0,1} + \tau)\} < H_0(x) - \log\{\pi_{0,0}/(\pi_{0,1} + \tau)\} - \widehat{H}_0(x) + \log\{\widehat{\pi}_{0,0}/(\widehat{\pi}_{0,1} + \tau)} dP_{0,1}(x)$$

$$\leq \epsilon_H \left[ \frac{1}{\|\mu_{1,1} - \mu_{1,0}\|_{K_1}} \exp\left\{ - \frac{\|\mu_{1,1} - \mu_{1,0}\|_{K_1}^4 \vee 1}{\|\mu_{1,1} - \mu_{1,0}\|_{K_1}^2} \right\} \right.$$

$$\left. + \frac{1}{\|\mu_{0,1} - \mu_{0,0}\|_{K_0}} \exp\left\{ - \frac{\|\mu_{0,1} - \mu_{0,0}\|_{K_0}^4 \vee 1}{\|\mu_{0,1} - \mu_{0,0}\|_{K_0}^2} \right\} \right] \tag{15}$$

$$\asymp \epsilon_H, \tag{16}$$

where the last inequality follows from a similar argument as the one leading to (11). Using a similar argument, we can also achieve a same (up to constant order) upper bound for $DO(\tau) - \mathbb{E}\{\widehat{DO}(\tau)|\widetilde{\mathcal{D}}\}$ as the one in (16).

**Step 2-Case 2: When** $-\pi_{0,1} < \tau \leq -\widehat{\pi}_{0,1}$. Note that in this case, by standard calculation, we have that

$$DO(\tau) = \int_{H_1(x) - \log\{\pi_{1,0}/(\pi_{1,1} - \tau)\} > 0} dP_{1,1}(x) - \int_{H_0(x) - \log\{\pi_{0,0}/(\pi_{0,1} + \tau)\} > 0} dP_{0,1}(x),$$

and

$$\mathbb{E}\{\widehat{DO}(\tau)|\widetilde{\mathcal{D}}\} = \int_{\widehat{H}_1(x) - \log\{\widehat{\pi}_{1,0}/(\widehat{\pi}_{1,1} - \tau)\} > 0} dP_{1,1}(x).$$

Consequently,

$$\sup_{-\pi_{0,1} < \tau \leq -\widehat{\pi}_{0,1}} \left| \mathbb{E}\{\widehat{DO}(\tau)|\widetilde{\mathcal{D}}\} - DO(\tau) \right|$$

$$\leq \sup_{-\pi_{0,1} < \tau \leq -\widehat{\pi}_{0,1}} \left| \int_{\widehat{H}_1(x) - \log\{\widehat{\pi}_{1,0}/(\widehat{\pi}_{1,1} - \tau)\} > 0} dP_{1,1}(x) - \int_{H_1(x) - \log\{\pi_{1,0}/(\pi_{1,1} - \tau)\} > 0} dP_{1,1}(x) \right|$$

$$+ \sup_{-\pi_{0,1} < \tau \leq -\widehat{\pi}_{0,1}} \left| \int_{H_0(x) - \log\{\pi_{0,0}/(\pi_{0,1} + \tau)\} > 0} dP_{0,1}(x) \right|$$

$$= (A) + (B).$$

Note that $(A)$ can be controlled using a similar argument as the one used in **Step 1** and we only present the upper bound on $(B)$ in the sequel. Consider the event $\mathcal{E}_\pi = \{|\pi_{0,1} - \widehat{\pi}_{0,1}| \lesssim$

$\sqrt{\log(1/\eta)/\widetilde{n}}\}$. By Lemma 31, it holds that $\mathbb{P}(\mathcal{E}_\pi) \geq 1 - \eta/2$. The rest of the proof is constructed conditioning on the event $\mathcal{E}_\pi$ happening. Therefore, we have that

$$(B) = \sup_{-\pi_{0,1} < \tau \leq -\widehat{\pi}_{0,1}} \mathbb{P}_{0,1}\Big\{H_0(X) > \log\Big(\frac{\pi_{0,0}}{\pi_{0,1}+\tau}\Big)\Big\}$$

$$\leq \mathbb{P}_{0,1}\Big\{H_0(X) > \log\Big(\frac{\pi_{0,0}}{\pi_{0,1}-\widehat{\pi}_{0,1}}\Big)\Big\}$$

$$\leq \mathbb{P}_{0,1}\Big\{H_0(X) > \log\Big(\sqrt{\frac{C_p^2\widetilde{n}}{\log(1/\eta)}}\Big)\Big\}$$

$$= \mathbb{P}_{0,1}\Big\{H_0(X) - \mathbb{E}_{P_{0,1}}\{H_0(X)\} > \log\Big(\sqrt{\frac{C_p^2\widetilde{n}}{\log(1/\eta)}}\Big) - \frac{\|\mu_{0,1}-\mu_{0,0}\|_{K_0}^2}{2}\Big\}$$

$$\lesssim \exp\Big[-\frac{1}{\|\mu_{0,1}-\mu_{0,0}\|_{K_0}^2}\log^2\Big(\sqrt{\frac{\widetilde{n}}{\log(1/\eta)}}\Big)\Big] \leq (17) \lesssim \epsilon_H, \qquad (17)$$

where the third inequality follows from the standard property of Gaussian random variables.

The proof for the other cases follows a similar argument as the one used in **Step2-Case 2**, hence it is omitted here. The lemma thus follows by substituting the results in (14) and (16) (or (17)) into (13) and a union bound argument. $\qquad\square$

### E.7   Behaviour of $DO$ around $\tau^\star$

**Lemma 16.** *Recall $DO$ given in Definition 3. Under Assumption 2, for any $\epsilon$ in a small neighbourhood of $0$, there exists absolute constants $c_1, c_2 > 0$ such that*

$$c_1\epsilon \leq DO(\tau^\star) - DO(\tau^\star + \epsilon) \leq c_2\epsilon, \quad c_1\epsilon \leq DO(\tau^\star - \epsilon) - DO(\tau^\star) \leq c_2\epsilon.$$

*Proof.* Note that for any $\tau \in \mathbb{R}$,

$$DO(\tau)$$

$$= \mathbb{P}\big\{\pi_{1,1}\eta_1(X) - \pi_{1,0} > \tau\eta_1(X)|A=1, Y=1\big\}$$

$$\quad - \mathbb{P}\big\{\pi_{0,1}\eta_0(X) - \pi_{0,0} > -\tau\eta_0(X)|A=0, Y=1\big\}$$

$$= \begin{cases} \mathbb{P}_{1,1}\big[\log\{\eta_1(X)\} > \log(\frac{\pi_{1,0}}{\pi_{1,1}-\tau})\big] - \mathbb{P}_{0,1}\big[\log\{\eta_0(X)\} > \log(\frac{\pi_{0,0}}{\pi_{0,1}+\tau})\big], \\ \qquad\qquad\qquad\qquad\qquad\qquad\qquad\qquad -\pi_{0,1} < \tau < \pi_{1,1}, \\ \mathbb{P}_{1,1}\big[\log\{\eta_1(X)\} > \log(\frac{\pi_{1,0}}{\pi_{1,1}-\tau})\big], \qquad\qquad\qquad \tau \leq -\pi_{0,1}, \\ -\mathbb{P}_{0,1}\big[\log\{\eta_0(X)\} > \log(\frac{\pi_{0,0}}{\pi_{0,1}+\tau})\big], \qquad\qquad\qquad \tau \geq \pi_{1,1}. \end{cases}$$

$$= \begin{cases} \Phi\Big[\frac{\|\mu_{1,1}-\mu_{1,0}\|_{K_1}}{2} - \frac{\log\{\pi_{1,0}/(\pi_{1,1}-\tau)\}}{\|\mu_{1,1}-\mu_{1,0}\|_{K_1}}\Big] - \Phi\Big[\frac{\|\mu_{0,1}-\mu_{0,0}\|_{K_0}}{2} - \frac{\log\{\pi_{0,0}/(\pi_{0,1}+\tau)\}}{\|\mu_{0,1}-\mu_{0,0}\|_{K_0}}\Big], \\ \qquad\qquad\qquad\qquad\qquad\qquad\qquad\qquad -\pi_{0,1} < \tau < \pi_{1,1}, \\ \Phi\Big[\frac{\|\mu_{1,1}-\mu_{1,0}\|_{K_1}}{2} - \frac{\log\{\pi_{1,0}/(\pi_{1,1}-\tau)\}}{\|\mu_{1,1}-\mu_{1,0}\|_{K_1}}\Big], \\ \qquad\qquad\qquad\qquad\qquad\qquad\qquad\qquad \tau \leq -\pi_{0,1}, \\ -\Phi\Big[\frac{\|\mu_{0,1}-\mu_{0,0}\|_{K_0}}{2} - \frac{\log\{\pi_{0,0}/(\pi_{0,1}+\tau)\}}{\|\mu_{0,1}-\mu_{0,0}\|_{K_0}}\Big], \\ \qquad\qquad\qquad\qquad\qquad\qquad\qquad\qquad \tau \geq \pi_{1,1}. \end{cases}$$

By standard calculation, the derivative of $DO$ with respect to $\tau$ is

$$DO'(\tau) = -\frac{1}{\|\mu_{1,1}-\mu_{1,0}\|_{K_1}(\pi_{1,1}-\tau)} \cdot \phi\Big[\frac{\|\mu_{1,1}-\mu_{1,0}\|_{K_1}}{2} - \frac{\log\{\pi_{1,0}/(\pi_{1,1}-\tau)\}}{\|\mu_{1,1}-\mu_{1,0}\|_{K_1}}\Big]$$

$$- \frac{1}{\|\mu_{0,1} - \mu_{0,0}\|_{K_0}(\pi_{0,1} + \tau)} \cdot \phi \left[ \frac{\|\mu_{0,1} - \mu_{0,0}\|_{K_0}}{2} - \frac{\log\left\{\pi_{0,0}/(\pi_{0,1} + \tau)\right\}}{\|\mu_{0,1} - \mu_{0,0}\|_{K_0}} \right], \quad (18)$$

where $\phi$ is the standard normal pdf. For any $\epsilon$ in a small right neighborhood of 0, by Lemma 17, it holds that $-\pi_{0,1} + c < \tau^\star - \epsilon, \tau^\star, \tau^\star + \epsilon < \pi_{1,1} - c$, where $c > 0$ is a small constant. Thus, by (18), we have that $-c_2 \leq DO'(\tau^\star - \epsilon), DO'(\tau^\star), DO'(\tau^\star + \epsilon) \leq -c_1$ for some universal constants $c_1, c_2 > 0$. Hence, by the mean value theorem, we have that

$$c_2\epsilon \geq DO(\tau^\star) - DO(\tau^\star + \epsilon) \geq c_1\epsilon, \quad c_2\epsilon \geq DO(\tau^\star - \epsilon) - DO(\tau^\star) \geq c_1\epsilon.$$

$\square$

### E.8 Auxiliary results

In this subsection, without loss of generality, we assume that $X \sim \mathcal{GP}(\mu_{a,0}, K)$ for $a \in \{0, 1\}$. The case when $X \sim \mathcal{GP}(\mu_{a,1}, K)$ can be justified similarly.

**Lemma 17.** *Recall the definition of $\tau^\star$ in (3) with $D$ chosen as DO. Under Assumptions 1 and 2, there exists a small absolute constant $0 < c < \min\{\pi_{0,1}, \pi_{1,1}\}$ such that*

$$-\pi_{0,1} + c \leq \tau^\star \leq \pi_{1,1} - c.$$

*Proof of Lemma 17.* By the definition of $DO$, we have that

$DO(\tau)$

$= \mathbb{P}\{\pi_{1,1}\eta_1(X) - \pi_{1,0} > \tau\eta_1(X)|A = 1, Y = 1\}$

$\quad - \mathbb{P}\{\pi_{0,1}\eta_0(X) - \pi_{0,0} > -\tau\eta_0(X)|A = 0, Y = 1\}$

$$= \begin{cases} \mathbb{P}_{1,1}\left[\log\{\eta_1(X)\} > \log(\frac{\pi_{1,0}}{\pi_{1,1}-\tau})\right] - \mathbb{P}_{0,1}\left[\log\{\eta_0(X)\} > \log(\frac{\pi_{0,0}}{\pi_{0,1}+\tau})\right], \\ \hfill -\pi_{0,1} < \tau < \pi_{1,1}, \\ \mathbb{P}_{1,1}\left[\log\{\eta_1(X)\} > \log(\frac{\pi_{1,0}}{\pi_{1,1}-\tau})\right], \hfill \tau \leq -\pi_{0,1}, \\ -\mathbb{P}_{0,1}\left[\log\{\eta_0(X)\} > \log(\frac{\pi_{0,0}}{\pi_{0,1}+\tau})\right], \hfill \tau \geq \pi_{1,1}. \end{cases}$$

$$= \begin{cases} \Phi\left[\frac{\|\mu_{1,1}-\mu_{1,0}\|_{K_1}}{2} - \frac{\log\left\{\pi_{1,0}/(\pi_{1,1}-\tau)\right\}}{\|\mu_{1,1}-\mu_{1,0}\|_{K_1}}\right] - \Phi\left[\frac{\|\mu_{0,1}-\mu_{0,0}\|_{K_0}}{2} - \frac{\log\left\{\pi_{0,0}/(\pi_{0,1}+\tau)\right\}}{\|\mu_{0,1}-\mu_{0,0}\|_{K_0}}\right], \\ \hfill -\pi_{0,1} < \tau < \pi_{1,1}, \\ \Phi\left[\frac{\|\mu_{1,1}-\mu_{1,0}\|_{K_1}}{2} - \frac{\log\left\{\pi_{1,0}/(\pi_{1,1}-\tau)\right\}}{\|\mu_{1,1}-\mu_{1,0}\|_{K_1}}\right], \\ \hfill \tau \leq -\pi_{0,1}, \\ -\Phi\left[\frac{\|\mu_{0,1}-\mu_{0,0}\|_{K_0}}{2} - \frac{\log\left\{\pi_{0,0}/(\pi_{0,1}+\tau)\right\}}{\|\mu_{0,1}-\mu_{0,0}\|_{K_0}}\right], \\ \hfill \tau \geq \pi_{1,1}. \end{cases}$$

In the case when $|DO(0)| \leq \delta$, the lemma holds trivially as $\tau^\star = 0$, and it is automatically bounded away from the boundary by a small constant. We will divide the following proof by conditioning on different assumptions.

Let $\tau_0$ denote the value such that $DO(\tau_0) = 0$. Then in the case when $DO(0) > \delta$, it holds that $\tau^\star > 0$ and $DO(\tau^\star) = \delta$. By Proposition 7 and the fact that $DO(\pi_{1,1}) < 0$, it must be the case that $0 < \tau^\star < \tau_0 < \pi_{1,1}$. Consequently, we write $\tau_0 = \pi_{1,1} - \epsilon$ where $0 < \epsilon_1 < \pi_{1,1}$, and in the rest of the proof it suffices to prove that $\epsilon_1 \asymp 1$. By the fact that $DO(\tau_0) = 0$ and $\Phi$ is a strictly increasing function, it holds that

$$\frac{\|\mu_{1,1} - \mu_{1,0}\|_{K_1}}{2} - \frac{\log\left\{\pi_{1,0}/(\pi_{1,1} - \tau_0)\right\}}{\|\mu_{1,1} - \mu_{1,0}\|_{K_1}} = \frac{\|\mu_{0,1} - \mu_{0,0}\|_{K_0}}{2} - \frac{\log\left\{\pi_{0,0}/(\pi_{0,1} + \tau_0)\right\}}{\|\mu_{0,1} - \mu_{0,0}\|_{K_0}},$$

hence implies

$$\frac{\|\mu_{1,1} - \mu_{1,0}\|_{K_1}}{2} - \frac{\log\left(\pi_{1,0}/\epsilon_1\right)}{\|\mu_{1,1} - \mu_{1,0}\|_{K_1}} = \frac{\|\mu_{0,1} - \mu_{0,0}\|_{K_0}}{2} - \frac{\log\left\{\pi_{0,0}/(\pi_{0,1} + \pi_{1,1} - \epsilon_1)\right\}}{\|\mu_{0,1} - \mu_{0,0}\|_{K_0}}. \quad (19)$$

Suppose that $\epsilon_1$ is a function of $n$ and $\widetilde{n}$ such that $\epsilon_1 \prec 1$. However, in this case $\log\left(\pi_{1,0}/\epsilon\right) \gg \log\left\{\pi_{0,0}/(\pi_{0,1} + \pi_{1,1} - \epsilon)\right\} \asymp 1$, hence (19) never holds when $\|\mu_{0,1} - \mu_{0,0}\|_{K_0} \asymp \|\mu_{1,1} - \mu_{1,0}\|_{K_1} \asymp 1$. Thus, we achieve a contradiction, and it must be the case that $\epsilon_1 = c_1$ where $c_1 > 0$ is an absolute constant. Therefore, we have that $\tau^\star > 0$ and $\pi_{1,1} - \tau^\star > \pi_{1,1} - \tau_0 = \epsilon_1 = c_1$.

Similarly, when $DO(0) < -\delta$, we have $\tau^\star < 0$ and $DO(\tau^\star) = -\delta$. With the same notation as above, by Proposition 7, it must be the case that $-\pi_{0,1} < -\pi_{0,1} + \epsilon_2 = \tau_0 < \tau^\star < 0$, where $0 < \epsilon_2 < -\pi_{0,1}$. Using a similar argument as above, it holds that

$$\frac{\|\mu_{1,1} - \mu_{1,0}\|_{K_1}}{2} - \frac{\log\left\{\pi_{1,0}/(\pi_{1,1} + \pi_{0,1} - \epsilon_2)\right\}}{\|\mu_{1,1} - \mu_{1,0}\|_{K_1}} = \frac{\|\mu_{0,1} - \mu_{0,0}\|_{K_0}}{2} - \frac{\log\left(\pi_{0,0}/\epsilon_2\right)}{\|\mu_{0,1} - \mu_{0,0}\|_{K_0}}. \quad (20)$$

Suppose that $\epsilon_2$ is a function of $n$ and $\widetilde{n}$ such that $\epsilon_2 \prec 1$. However, in this case, we have that $\log\left(\pi_{0,0}/\epsilon_2\right) \gg \log\left\{\pi_{1,0}/(\pi_{1,1} + \pi_{0,1} - \epsilon_2)\right\}$. Thus (20) never holds and we reach a contradiction. Hence, we conclude that $\epsilon_2 = c_2$ and $\tau^\star + \pi_{0,1} \geq \tau_0 + \pi_{0,1} = \epsilon_2 = c_2$. The lemma thus follows by combining the results in the three cases above. $\qquad\square$

**Lemma 18.** *Recall the empirical estimator for Disparity of Opportunity*

$$\widehat{DO}(\tau) = \frac{1}{n_{1,1}}\sum_{i=1}^{n_{1,1}} \mathbb{1}\left\{(\widehat{\pi}_{1,1} - \tau)\widehat{\eta}_1(X_{1,1,i}) > \widehat{\pi}_{1,0}\right\} - \frac{1}{n_{0,1}}\sum_{i=1}^{n_{0,1}} \mathbb{1}\left\{(\widehat{\pi}_{0,1} + \tau)\widehat{\eta}_0(X_{0,1,i}) > \widehat{\pi}_{0,0}\right\}.$$

*It holds that $\widehat{DO}$ is a non-increasing function. Moreover, $\widehat{DO}(\widehat{\pi}_{1,1}) \leq 0$ and $\widehat{DO}(-\widehat{\pi}_{0,1}) \geq 0$.*

*Proof of Lemma 18.* Recall that $s_{DO,a} = 2a - 1$ and $b_{DO,a} = 0$. Then $\{\tau \in \mathbb{R} : \widehat{\pi}_{a,0} + \tau b_{D,a} \geq 0, \widehat{\pi}_{a,1} - \tau s_{D,a} \geq 0, \forall a \in \{0,1\}\} = \{\tau \in \mathbb{R} : -\widehat{\pi}_{0,1} \leq \tau \leq \widehat{\pi}_{1,1}\}$. The estimated disparity function is

$$\widehat{DO}(\tau) = \frac{1}{n_{1,1}}\sum_{i=1}^{n_{1,1}} \mathbb{1}\left\{(\widehat{\pi}_{1,1} - \tau)\widehat{\eta}_1(X_{1,1,i}) > \widehat{\pi}_{1,0}\right\} - \frac{1}{n_{0,1}}\sum_{i=1}^{n_{0,1}} \mathbb{1}\left\{(\widehat{\pi}_{0,1} + \tau)\widehat{\eta}_0(X_{0,1,i}) > \widehat{\pi}_{0,0}\right\}$$

$$= \begin{cases} \frac{1}{n_{1,1}}\sum_{i=1}^{n_{1,1}} \mathbb{1}\left\{\widehat{\eta}_1(X_{1,1,i}) > \frac{\widehat{\pi}_{1,0}}{\widehat{\pi}_{1,1} - \tau}\right\} - \frac{1}{n_{0,1}}\sum_{i=1}^{n_{0,1}} \mathbb{1}\left\{\widehat{\eta}_0(X_{0,1,i}) > \frac{\widehat{\pi}_{0,0}}{\widehat{\pi}_{0,1} + \tau}\right\}, \\ \qquad\qquad\qquad\qquad\qquad\qquad\qquad\qquad \tau \in (-\widehat{\pi}_{0,1}, \widehat{\pi}_{1,1}), \\ \frac{1}{n_{1,1}}\sum_{i=1}^{n_{1,1}} \mathbb{1}\left\{\widehat{\eta}_1(X_{1,1,i}) > \frac{\widehat{\pi}_{1,0}}{\widehat{\pi}_{1,1} - \tau}\right\}, \\ \qquad\qquad\qquad\qquad\qquad\qquad\qquad\qquad \tau \in (-\infty, -\widehat{\pi}_{0,1}], \\ -\frac{1}{n_{0,1}}\sum_{i=1}^{n_{0,1}} \mathbb{1}\left\{\widehat{\eta}_0(X_{0,1,i}) > \frac{\widehat{\pi}_{0,0}}{\widehat{\pi}_{0,1} + \tau}\right\}, \\ \qquad\qquad\qquad\qquad\qquad\qquad\qquad\qquad \tau \in [\widehat{\pi}_{1,1}, +\infty). \end{cases}$$

For $-\widehat{\pi}_{0,1} \leq \tau_1 < \tau_2 \leq \widehat{\pi}_{1,1}$,

$$\widehat{DO}(\tau_1) - \widehat{DO}(\tau_2)$$

$$= \frac{1}{n_{1,1}}\sum_{i=1}^{n_{1,1}} \mathbb{1}\left\{\frac{\widehat{\pi}_{1,0}}{\widehat{\pi}_{1,1} - \tau_1} < \widehat{\eta}_1(X_{1,1,i}) \leq \frac{\widehat{\pi}_{1,0}}{\widehat{\pi}_{1,1} - \tau_2}\right\}$$

$$+ \frac{1}{n_{0,1}}\sum_{i=1}^{n_{0,1}} \mathbb{1}\left\{\frac{\widehat{\pi}_{0,0}}{\widehat{\pi}_{0,1} + \tau_2} < \widehat{\eta}_0(X_{0,1,i}) \leq \frac{\widehat{\pi}_{0,0}}{\widehat{\pi}_{0,1} + \tau_1}\right\}$$

$$\geq 0.$$

For $\tau_1 < \tau_2 < -\widehat{\pi}_{0,1}$,

$$\widehat{DO}(\tau_1) - \widehat{DO}(\tau_2)$$

$$= \frac{1}{n_{1,1}} \sum_{i=1}^{n_{1,1}} \mathbb{1}\left\{ \frac{\widehat{\pi}_{1,0}}{\widehat{\pi}_{1,1} - \tau_1} < \widehat{\eta}_1(X_{1,1,i}) \leq \frac{\widehat{\pi}_{1,0}}{\widehat{\pi}_{1,1} - \tau_2} \right\} \geq 0.$$

For $\widehat{\pi}_{1,1} < \tau_1 < \tau_2$,

$$\widehat{DO}(\tau_1) - \widehat{DO}(\tau_2)$$

$$= \frac{1}{n_{0,1}} \sum_{i=1}^{n_{0,1}} \mathbb{1}\left\{ \frac{\widehat{\pi}_{0,0}}{\widehat{\pi}_{0,1} + \tau_2} < \widehat{\eta}_0(X_{0,1,i}) \leq \frac{\widehat{\pi}_{0,0}}{\widehat{\pi}_{0,1} + \tau_1} \right\} \geq 0.$$

Hence, $\widehat{DO}(\tau)$ is a non-increasing function. Therefore, if $|\widehat{DO}(0)| \leq \delta$, then $\widehat{\tau} = 0$. If $\widehat{DO}(0) > \delta$, then $\widehat{\tau} \geq 0$. If $\widehat{DO}(0) < -\delta$, $\widehat{\tau} \leq 0$. Moreover, a straight calculation leads to

$$\widehat{DO}(\widehat{\pi}_{1,1}) \leq 0 \quad \text{and} \quad \widehat{DO}(-\widehat{\pi}_{0,1}) \geq 0.$$

$\square$

**Lemma 19.** *Conditioning on the training data $\widetilde{\mathcal{D}}$, under the same condition as the one in Lemma 12, for any small constant $\eta \in (0, 1/2)$, it holds with probability at least $1 - \eta$ that*

$$\left| \sum_{j=1}^{J} \frac{(\widehat{\zeta}_{a,j} - \widehat{\theta}_{a,0,j})(\widehat{\theta}_{a,1,j} - \widehat{\theta}_{a,0,j})}{\widehat{\lambda}_{a,j}} - \sum_{j=1}^{J} \frac{(\zeta_{a,j} - \theta_{a,0,j})(\theta_{a,1,j} - \theta_{a,0,j})}{\lambda_{a,j}} \right|$$

$$\lesssim \begin{cases} \sqrt{\dfrac{J^{\alpha-2\beta+4} \log(\widetilde{n}/\eta) \log(1/\eta)}{\widetilde{n}}} & \text{when } \frac{\alpha+1}{2} < \beta \leq \frac{\alpha+2}{2}, \\[2ex] \sqrt{\dfrac{J^2 \log(\widetilde{n}/\eta) \log(1/\eta)}{\widetilde{n}}}, & \text{when } \frac{\alpha+2}{2} < \beta \leq \frac{\alpha+3}{2}, \\[2ex] \sqrt{\dfrac{J \log(\widetilde{n}/\eta) \log(1/\eta)}{\widetilde{n}}}, & \text{when } \beta > \frac{\alpha+3}{2}. \end{cases}$$

*Proof.* Note that

$$\sum_{j=1}^{J} \frac{(\widehat{\zeta}_{a,j} - \widehat{\theta}_{a,0,j})(\widehat{\theta}_{a,1,j} - \widehat{\theta}_{a,0,j})}{\widehat{\lambda}_{a,j}} - \sum_{j=1}^{\infty} \frac{(\zeta_{a,j} - \theta_{a,0,j})(\theta_{a,1,j} - \theta_{a,0,j})}{\lambda_{a,j}}$$

$$= \sum_{j=1}^{J} \frac{(\widehat{\zeta}_{a,j} - \widehat{\theta}_{a,0,j})(\widehat{\theta}_{a,1,j} - \widehat{\theta}_{a,0,j})}{\widehat{\lambda}_{a,j}} - \sum_{j=1}^{J} \frac{(\zeta_{a,j} - \theta_{a,0,j})(\widehat{\theta}_{a,1,j} - \widehat{\theta}_{a,0,j})}{\widehat{\lambda}_{a,j}}$$

$$+ \sum_{j=1}^{J} \frac{(\zeta_{a,j} - \theta_{a,0,j})(\widehat{\theta}_{a,1,j} - \widehat{\theta}_{a,0,j})}{\widehat{\lambda}_{a,j}} - \sum_{j=1}^{J} \frac{(\zeta_{a,j} - \theta_{a,0,j})(\theta_{a,1,j} - \theta_{a,0,j})}{\lambda_{a,j}}$$

$$= \sum_{j=1}^{J} \frac{(\widehat{\zeta}_{a,j} - \widehat{\theta}_{a,0,j} - \zeta_{a,j} + \theta_{a,0,j})(\widehat{\theta}_{a,1,j} - \widehat{\theta}_{a,0,j})}{\widehat{\lambda}_{a,j}}$$

$$+ \sum_{j=1}^{J} (\zeta_{a,j} - \theta_{a,0,j})\left( \frac{\widehat{\theta}_{a,1,j} - \widehat{\theta}_{a,0,j}}{\widehat{\lambda}_{a,j}} - \frac{\theta_{a,1,j} - \theta_{a,0,j}}{\lambda_{a,j}} \right)$$

$$= (I) + (II). \tag{21}$$

When $X \sim \mathcal{GP}(\mu_{a,0}, K)$, by standard properties of Gaussian process, it holds that $(\zeta_{a,1}, \ldots, \zeta_{a,j})^\top \sim N(0, \Lambda)$ where $\Lambda = \text{diag}(\lambda_1, \ldots, \lambda_{a,j})$ and for any $j, k \in [J]$ such that $j \neq k$, $\text{var}(\zeta_{a,j}) = \lambda_{a,j}$,

$$\text{var}\left( \widehat{\zeta}_{a,j} - \widehat{\theta}_{a,0,j} - \zeta_{a,j} + \theta_{a,0,j} \right) = \int \int K_a(s,t)\{\widehat{\phi}_{a,j}(s) - \phi_{a,j}(s)\}\{\widehat{\phi}_{a,j}(t) - \phi_{a,j}(t)\} \, \mathrm{d}s \, \mathrm{d}t,$$

and

$$\mathrm{cov}\big(\widehat{\zeta}_{a,j} - \widehat{\theta}_{a,0,j} - \zeta_{a,j} + \theta_{a,0,j}, \widehat{\zeta}_{a,k} - \widehat{\theta}_{a,0,k} - \zeta_k + \theta_{a,0,k}\big)$$

$$= \int\int K_a(s,t)\big\{\widehat{\phi}_{a,j}(s) - \phi_{a,j}(s)\big\}\big\{\widehat{\phi}_{a,k}(t) - \phi_{a,k}(t)\big\}\,\mathrm{d}s\,\mathrm{d}t.$$

Therefore, it holds that

$$\sum_{j=1}^{J} \frac{(\widehat{\zeta}_{a,j} - \widehat{\theta}_{a,0,j} - \zeta_{a,j} + \theta_{a,0,j})(\widehat{\theta}_{a,1,j} - \widehat{\theta}_{a,0,j})}{\widehat{\lambda}_{a,j}} \sim N\big(Q_1, Q_2\big),$$

and

$$\sum_{j=1}^{J} (\zeta_{a,j} - \theta_{a,0,j})\Big(\frac{\widehat{\theta}_{a,1,j} - \widehat{\theta}_{a,0,j}}{\widehat{\lambda}_{a,j}} - \frac{\theta_{a,1,j} - \theta_{a,0,j}}{\lambda_{a,j}}\Big) \sim N\big(0, Q_3\big),$$

where

$$Q_1 = \sum_{j=1}^{J} \frac{(\widehat{\theta}_{a,1,j} - \widehat{\theta}_{a,0,j})}{\widehat{\lambda}_{a,j}} \int \big\{\mu_{a,0}(t) - \widehat{\mu}_{a,0}(t)\big\}\widehat{\phi}_{a,j}(t)\,\mathrm{d}t,$$

$$Q_2 = \sum_{j=1}^{J} \frac{(\widehat{\theta}_{a,1,j} - \widehat{\theta}_{a,0,j})^2}{\widehat{\lambda}_{a,j}^2} \int\int K_a(s,t)\big\{\widehat{\phi}_{a,j}(s) - \phi_{a,j}(s)\big\}\big\{\widehat{\phi}_{a,j}(t) - \phi_{a,j}(t)\big\}\,\mathrm{d}s\,\mathrm{d}t$$

$$+ 2 \sum_{1 \le j < k \le J} \frac{\widehat{\theta}_{a,1,j} - \widehat{\theta}_{a,0,j}}{\widehat{\lambda}_{a,j}} \cdot \frac{\widehat{\theta}_{a,1,k} - \widehat{\theta}_{a,0,k}}{\widehat{\lambda}_{a,k}}$$

$$\cdot \int\int K_a(s,t)\big\{\widehat{\phi}_{a,j}(s) - \phi_{a,j}(s)\big\}\big\{\widehat{\phi}_{a,k}(t) - \phi_{a,k}(t)\big\}\,\mathrm{d}s\,\mathrm{d}t,$$

and

$$Q_3 = \sum_{j=1}^{J} \lambda_{a,j}\Big(\frac{\widehat{\theta}_{a,1,j} - \widehat{\theta}_{a,0,j}}{\widehat{\lambda}_{a,j}} - \frac{\theta_{a,1,j} - \theta_{a,0,j}}{\lambda_{a,j}}\Big)^2.$$

Consequently, by standard Gaussian tail properties (e.g. Proposition 2.1.2 in Vershynin, 2018), it holds with probability at least $1 - \eta$ that

$$|(I)| \lesssim |Q_1| + \sqrt{Q_2 \log(1/\eta)}, \quad \text{and} \quad |(II)| \lesssim \sqrt{Q_3 \log(1/\eta)}. \tag{22}$$

Substituting (22) into (21), the lemma thus follows by applying a union bound argument to the results in Lemmas 21, 23, 24 and 27. □

**Lemma 20.** *Under Assumptions 1 and 2, if we assume that $X \sim \mathcal{GP}(\mu_{a,0}, K)$ for $a \in \{0,1\}$, then for any small constant $\eta \in (0, 1/2)$, it holds with probability at least $1 - \eta$ that*

$$\Big|\sum_{j=J+1}^{\infty} \frac{(\zeta_{a,j} - \theta_{a,0,j})(\theta_{a,1,j} - \theta_{a,0,j})}{\lambda_{a,j}}\Big| \lesssim \sqrt{J^{\alpha - 2\beta + 1}\log(1/\eta)}.$$

*Proof.* When $X \sim \mathcal{GP}(\mu_{a,0}, K)$, it holds that

$$\sum_{j=J+1}^{\infty} \frac{(\zeta_{a,j} - \theta_{a,0,j})(\theta_{a,1,j} - \theta_{a,0,j})}{\lambda_{a,j}} \sim N\Big(0, \sum_{j=J+1}^{\infty} \frac{(\theta_{a,1,j} - \theta_{a,0,j})^2}{\lambda_{a,j}}\Big).$$

Consequently, by standard Gaussian tail properties (e.g. Proposition 2.1.2 in Vershynin, 2018), we have with probability at least $1 - \eta$ that

$$\Big|\sum_{j=J+1}^{\infty} \frac{(\zeta_{a,j} - \theta_{a,0,j})(\theta_{a,1,j} - \theta_{a,0,j})}{\lambda_{a,j}}\Big| \lesssim \sqrt{\log(1/\eta)\sum_{j=J+1}^{\infty}\frac{(\theta_{a,1,j} - \theta_{a,0,j})^2}{\lambda_{a,j}}}$$

$$\lesssim \sqrt{J^{\alpha - 2\beta + 1}\log(1/\eta)},$$

where the last inequality follows from Assumptions 2a and 2c. The lemma thus follows. □

**Lemma 21.** *Under the same condition of Lemma 12, for any small constant $\eta \in (0, 1/2)$, it holds with probability at least $1 - \eta$ that*

$$Q_1 = \Big| \sum_{j=1}^{J} \frac{(\widehat{\theta}_{a,1,j} - \widehat{\theta}_{a,0,j})}{\widehat{\lambda}_{a,j}} \int \{\mu_{a,0}(t) - \widehat{\mu}_{a,0}(t)\} \widehat{\phi}_{a,j}(t) \, \mathrm{d}t \Big|$$

$$\lesssim \begin{cases} \sqrt{\frac{J \log(\widetilde{n}/\eta)}{\widetilde{n}}}, & \text{when } \frac{\alpha+1}{2} < \beta \le \frac{\alpha+2}{2}, \\ \sqrt{\frac{\log(\widetilde{n}/\eta)}{\widetilde{n}}}, & \text{when } \beta > \frac{\alpha+2}{2}. \end{cases}$$

*Proof.* By Lemmas 30 and 22, the event in (44) and a union bound argument, we have with probability at least $1 - \eta/2$ that

$$\Big| \sum_{j=1}^{J} \frac{(\widehat{\theta}_{a,1,j} - \widehat{\theta}_{a,0,j})}{\widehat{\lambda}_{a,j}} \int \{\mu_{a,0}(t) - \widehat{\mu}_{a,0}(t)\} \widehat{\phi}_{a,j}(t) \, \mathrm{d}t \Big|$$

$$\le \sum_{j=1}^{J} \Big| \frac{\widehat{\theta}_{a,1,j} - \widehat{\theta}_{a,0,j}}{\widehat{\lambda}_{a,j}} \Big| \cdot \Big| \int \{\mu_{a,0}(t) - \widehat{\mu}_{a,0}(t)\} \widehat{\phi}_{a,j}(t) \, \mathrm{d}t \Big|$$

$$\lesssim \sum_{j=1}^{J} \Big| \frac{\widehat{\theta}_{a,1,j} - \widehat{\theta}_{a,0,j}}{\sqrt{\lambda_{a,j}}} \Big| \frac{1}{\sqrt{\lambda_{a,j}}} \sqrt{\frac{j^{-\alpha} \log(\widetilde{n}/\eta)}{\widetilde{n}}}$$

$$\lesssim \sum_{j=1}^{J} \frac{|\widehat{\theta}_{a,1,j} - \widehat{\theta}_{a,0,j}|}{\sqrt{\lambda_{a,j}}} \sqrt{\frac{\log(\widetilde{n}/\eta)}{\widetilde{n}}}. \tag{23}$$

To further control (23), we will consider three different cases.

**Case 1: When** $(\alpha + 1)/2 < \beta \le (\alpha + 2)/2$**.** In this scenario, by a union bound argument, we have with probability at least $1 - \eta$ that

$$(23) \lesssim \sqrt{\sum_{j=1}^{J} \frac{(\widehat{\theta}_{a,1,j} - \widehat{\theta}_{a,0,j})^2}{\widehat{\lambda}_{a,j}}} \sqrt{\sum_{j=1}^{J} \frac{\log(\widetilde{n}/\eta)}{\widetilde{n}}}$$

$$\lesssim \sqrt{\frac{J \log(\widetilde{n}/\eta)}{\widetilde{n}}} \Big\{ 1 \vee \Big( \frac{J^2 \log^2(J) \log(\widetilde{n}/\eta)}{\widetilde{n}} \Big)^{\frac{1}{4}} \Big\}$$

$$\lesssim \sqrt{\frac{J \log(\widetilde{n}/\eta)}{\widetilde{n}}},$$

where the first inequality follows from Cauchy–Schwarz inequality, the second inequality follows from Lemma 30 and the last inequality follows from the fact that

$$\sqrt{\frac{J^2 \log^2(J) \log(\widetilde{n}/\eta)}{\widetilde{n}}} \lesssim 1.$$

**Case 2: When** $(\alpha + 2)/2 < \beta \le (3\alpha + 2)/2$**.** In this case, by Lemma 39, we have with probability at least $1 - \eta/2$ that, for any $j \in [J]$, $|\widehat{\theta}_{a,1,j} - \widehat{\theta}_{a,0,j}| \lesssim j^{-\beta}$. Therefore, we have that

$$(23) \lesssim \sqrt{\frac{\log(\widetilde{n}/\eta)}{\widetilde{n}}} \sum_{j=1}^{J} j^{\frac{\alpha}{2} - \beta} \lesssim \sqrt{\frac{\log(\widetilde{n}/\eta)}{\widetilde{n}}},$$

where the last inequality follows as $\alpha/2 - \beta < -1$.

**Case 3: When** $\beta > (3\alpha + 2)/2$**.** In this case, by Lemma 39, we have that with probability at least $1 - \eta/2$ that, for any $j \in [J]$,

$$|\widehat{\theta}_{a,1,j} - \widehat{\theta}_{a,0,j}| \lesssim j^{-\beta} + \sqrt{\frac{j^{-\alpha} \log(\widetilde{n}/\eta)}{\widetilde{n}}}.$$

Therefore, we have that

$$(23) \lesssim \sqrt{\frac{\log(\widetilde{n}/\eta)}{\widetilde{n}}} \Big\{ \sum_{j=1}^{J} j^{\frac{\alpha}{2}-\beta} + \sum_{j=1}^{J} \sqrt{\frac{\log(\widetilde{n}/\eta)}{\widetilde{n}}} \Big\}$$

$$\lesssim \sqrt{\frac{\log(\widetilde{n}/\eta)}{\widetilde{n}}} + \frac{J \log(\widetilde{n}/\eta)}{\widetilde{n}}$$

$$\lesssim \sqrt{\frac{\log(\widetilde{n}/\eta)}{\widetilde{n}}},$$

where the second inequality follows as $\alpha/2 - \beta < -1$ and the last inequality follows as $J^2 \log(\widetilde{n}/\eta) \lesssim \widetilde{n}$.

The lemma thus follows by combining results in three cases together. $\qquad\square$

**Lemma 22.** *Under Assumptions 1 and 2a, for any small constant $\eta \in (0, 1/2)$, it holds with probability at least $1 - \eta$ that, for any $a, y \in \{0, 1\}$ and $j \in [J]$ such that $J^{2\alpha+2} \log(\widetilde{n}/\eta) \lesssim \widetilde{n}$,*

$$\Big| \int \{\mu_{a,y}(t) - \widehat{\mu}_{a,y}(t)\}\widehat{\phi}_{a,j}(t) \, \mathrm{d}t \Big| \lesssim \sqrt{\frac{j^{-\alpha} \log(\widetilde{n}/\eta)}{\widetilde{n}}}.$$

*Proof.* Consider the following events,

$$\mathcal{E}_1 = \Big\{ \|\widehat{\mu}_{a,y}(t) - \mu_{a,y}(t)\|_{L^2} \lesssim \sqrt{\frac{\log(1/\eta)}{\widetilde{n}}}, \text{ for } a, y \in \{0, 1\} \Big\},$$

$$\mathcal{E}_2 = \Big\{ \|\widehat{\phi}_{a,j} - \phi_{a,j}\|_{L^2} \lesssim \sqrt{\frac{j^2 \log(\widetilde{n}/\eta)}{\widetilde{n}}}, \quad \text{for } a \in \{0, 1\}, j \in [J] \Big\},$$

and

$$\mathcal{E}_3 = \Big\{ \Big| \frac{1}{\widetilde{n}_{a,y}} \sum_{i=1}^{\widetilde{n}_{a,y}} \int \{\widetilde{X}_{a,y}^i(t) - \mu_{a,y}(t)\}\phi_{a,j}(t) \, \mathrm{d}t \Big| \lesssim \sqrt{\frac{j^{-\alpha} \log(\widetilde{n}/\eta)}{\widetilde{n}}}, \text{ for } a \in \{0, 1\}, j \in [J] \Big\}.$$

By Lemmas 32, 33, 35 and 40, it holds from a union-bound argument that $\mathbb{P}(\mathcal{E}_1 \cap \mathcal{E}_2 \cap \mathcal{E}_3) \geq 1 - \eta$. The rest of the proof is constructed conditioning on the events happening. Note that

$$\Big| \int \{\mu_{a,y}(t) - \widehat{\mu}_{a,y}(t)\}\widehat{\phi}_{a,j}(t) \, \mathrm{d}t \Big|$$

$$= \Big| \int \{\widehat{\mu}_{a,y}(t) - \mu_{a,y}(t)\}\{\widehat{\phi}_{a,j}(t) - \phi_{a,j}(t)\} \, \mathrm{d}t \Big| + \Big| \int \{\widehat{\mu}_{a,y}(t) - \mu_{a,y}(t)\}\phi_{a,j}(t) \, \mathrm{d}t \Big|$$

$$= (I) + (II), \tag{24}$$

and in the rest of the proof, we will control $(I)$ and $(II)$ individually.

To control $(I)$, it holds from Cauchy–Schwarz inequality that

$$|(I)| \leq \sqrt{\int \{\mu_{a,y}(t) - \widehat{\mu}_{a,y}(t)\}^2 \, \mathrm{d}t \int \{\widehat{\phi}_{a,j}(t) - \phi_{a,j}(t)\}^2 \, \mathrm{d}t}$$

$$\lesssim \sqrt{\frac{\log(1/\eta)}{\widetilde{n}} \frac{j^2 \log(\widetilde{n}/\eta)}{\widetilde{n}}} = \sqrt{\frac{j^2 \log(\widetilde{n}/\eta) \log(1/\eta)}{\widetilde{n}^2}}, \tag{25}$$

where the last inequality follows from $\mathcal{E}_1$ and $\mathcal{E}_2$.

To control $(II)$, it holds that

$$|(II)| = \Big| \frac{1}{\widetilde{n}_{a,y}} \sum_{i=1}^{\widetilde{n}_{a,y}} \int \{\widetilde{X}_{a,y}^i(t) - \mu_{a,y}(t)\}\phi_{a,j}(t) \, \mathrm{d}t \Big|$$

$$\lesssim \sqrt{\frac{j^{-\alpha}\log(\widetilde{n}/\eta)}{\widetilde{n}}}, \tag{26}$$

where the inequality holds from $\mathcal{E}_3$. Substituting the results in (25) and (26) into (24), we have that for any $j \in [J]$,

$$\left| \int \{\mu_{a,y}(t) - \widehat{\mu}_{a,y}(t)\} \widehat{\phi}_{a,j}(t) \, \mathrm{d}t \right| \lesssim \sqrt{\frac{j^2 \log(\widetilde{n}/\eta)\log(1/\eta)}{\widetilde{n}^2}} + \sqrt{\frac{j^{-\alpha}\log(\widetilde{n}/\eta)}{\widetilde{n}}}$$

$$\lesssim \sqrt{\frac{j^{-\alpha}\log(\widetilde{n}/\eta)}{\widetilde{n}}},$$

whenever $J^{\alpha+2}\log(1/\eta) \lesssim \widetilde{n}$. Thus, the lemma follows. $\qquad\square$

**Lemma 23.** *Under the same condition of Lemma 12, for any small constant $\eta \in (0, 1/2)$, it holds with probability at least $1 - \eta$ that*

$$\left| \sum_{j=1}^{J} \frac{(\widehat{\theta}_{a,1,j} - \widehat{\theta}_{a,0,j})^2}{\widehat{\lambda}_{a,j}^2} \int \int K_a(s,t)\{\widehat{\phi}_{a,j}(s) - \phi_{a,j}(s)\}\{\widehat{\phi}_{a,j}(t) - \phi_{a,j}(t)\} \, \mathrm{d}s \, \mathrm{d}t \right|$$

$$\lesssim \begin{cases} \frac{J^{\alpha-2\beta+3}\log(\widetilde{n}/\eta)}{\widetilde{n}}, & \text{when } \frac{\alpha+1}{2} < \beta \leq \frac{\alpha+2}{2}, \\ \frac{J\log(\widetilde{n}/\eta)}{\widetilde{n}}, & \text{when } \frac{\alpha+2}{2} < \beta \leq \frac{\alpha+3}{2}, \\ \frac{\log(\widetilde{n}/\eta)}{\widetilde{n}}, & \text{when } \beta > \frac{\alpha+3}{2}. \end{cases}$$

*Proof.* Consider the following event

$$\mathcal{E}_1 = \left\{ \lambda_{a,j}/2 \leq \widehat{\lambda}_{a,j} \leq 3\lambda_{a,j}/2, \quad \text{for } j \in [J], a \in \{0, 1\} \right\}. \tag{27}$$

By a similar argument as the one used to control (44), we have that $\mathbb{P}(\mathcal{E}_1) \geq 1 - \eta$. The rest of the proof is constructed conditioning on $\mathcal{E}_1$. Note that, by triangle inequality, we have that

$$\left| \sum_{j=1}^{J} \frac{(\widehat{\theta}_{a,1,j} - \widehat{\theta}_{a,0,j})^2}{\widehat{\lambda}_{a,j}^2} \int \int K_a(s,t)\{\widehat{\phi}_{a,j}(s) - \phi_{a,j}(s)\}\{\widehat{\phi}_{a,j}(t) - \phi_{a,j}(t)\} \, \mathrm{d}s \, \mathrm{d}t \right|$$

$$\leq \sum_{j=1}^{J} \frac{(\widehat{\theta}_{a,1,j} - \widehat{\theta}_{a,0,j})^2}{\widehat{\lambda}_{a,j}^2} \left| \int \int K_a(s,t)\{\widehat{\phi}_{a,j}(s) - \phi_{a,j}(s)\}\{\widehat{\phi}_{a,j}(t) - \phi_{a,j}(t)\} \, \mathrm{d}s \, \mathrm{d}t \right|. \tag{28}$$

We divide the following proof into three cases depending on the value of $\beta$. For any $\alpha > 1$, by Lemma 25, we have with probability at least $1 - \eta$ that

$$\left| \int \int K_a(s,t)\{\widehat{\phi}_{a,j}(s) - \phi_{a,j}(s)\}\{\widehat{\phi}_{a,j}(t) - \phi_{a,j}(t)\} \, \mathrm{d}s \, \mathrm{d}t \right| \lesssim \frac{j^{2-\alpha}\log(\widetilde{n}/\eta)}{\widetilde{n}}.$$

**Case 1: When $(\alpha+1)/2 < \beta \leq (\alpha+2)/2$.** In this scenario, by Lemma 39, we have with probability at least $1 - \eta/2$ that, for any $j \in [J]$, $|\widehat{\theta}_{a,1,j} - \widehat{\theta}_{a,0,j}| \lesssim j^{-\beta}$. Therefore, we have that

$$(28) \lesssim \frac{\log(\widetilde{n}/\eta)}{\widetilde{n}} \sum_{j=1}^{J} j^{-2\beta+2\alpha+2-\alpha} = \frac{\log(\widetilde{n}/\eta)}{\widetilde{n}} \sum_{j=1}^{J} j^{-2\beta+\alpha+2} \leq \frac{J^{\alpha-2\beta+3}\log(\widetilde{n}/\eta)}{\widetilde{n}},$$

where the last inequality follows from the fact that $\alpha - 2\beta + 2 \geq 0$.

**Case 2: When $(\alpha+2)/2 < \beta \leq (\alpha+3)/2$.** In this scenario, by Lemma 39, we still have with probability at least $1 - \eta/2$ that, for any $j \in [J]$, $|\widehat{\theta}_{a,1,j} - \widehat{\theta}_{a,0,j}| \lesssim j^{-\beta}$. Therefore, we have that

$$(28) \lesssim \frac{\log(\widetilde{n}/\eta)}{\widetilde{n}} \sum_{j=1}^{J} j^{-2\beta+\alpha+2} \leq \frac{J\log(\widetilde{n}/\eta)}{\widetilde{n}},$$

where the last inequality follows from the fact that $-1 \leq -2\beta + \alpha + 2 < 0$.

**Case 3: When** $\beta > (\alpha + 3)/2$**.** In this case, by Lemma 39, we have that with probability at least $1 - \eta/2$ that, for any $j \in [J]$,

$$|\widehat{\theta}_{a,1,j} - \widehat{\theta}_{a,0,j}| \lesssim j^{-\beta} + \sqrt{\frac{j^{-\alpha}\log(\widetilde{n}/\eta)}{\widetilde{n}}}.$$

Therefore, we have that

$$(28) \lesssim \frac{\log(\widetilde{n}/\eta)}{\widetilde{n}} \sum_{j=1}^{J}\{j^{-2\beta} \vee \frac{j^{-\alpha}\log(\widetilde{n}/\eta)}{\widetilde{n}}\}j^{2+\alpha} \leq \frac{\log(\widetilde{n}/\eta)}{\widetilde{n}}\Big\{\sum_{j=1}^{J} j^{\alpha-2\beta+2} + \sum_{j=1}^{J} \frac{j^2}{\widetilde{n}}\Big\}$$

$$\leq \frac{\log(\widetilde{n}/\eta)}{\widetilde{n}}\Big\{1 \vee \frac{J^3\log(\widetilde{n})}{\widetilde{n}}\Big\} \leq \frac{\log(\widetilde{n}/\eta)}{\widetilde{n}},$$

where the third inequality follows from the fact that $\alpha - 2\beta + 2 < -1$ in this case and the last inequality follows from the assumption on $J$.

The lemma thus follows by combining the results for all six cases above. $\qquad\square$

**Lemma 24.** *Under the same condition of Lemma 12, for any small constant $\eta \in (0, 1/2)$, it holds with probability at least $1 - \eta$ that*

$$\Big| \sum_{1 \leq j < k \leq J} \frac{\widehat{\theta}_{a,1,j} - \widehat{\theta}_{a,0,j}}{\widehat{\lambda}_{a,j}} \cdot \frac{\widehat{\theta}_{a,1,k} - \widehat{\theta}_{a,0,k}}{\widehat{\lambda}_{a,k}}$$

$$\cdot \int\int K_a(s,t)\{\widehat{\phi}_{a,j}(s) - \phi_{a,j}(s)\}\{\widehat{\phi}_{a,k}(t) - \phi_{a,k}(t)\} \, \mathrm{d}s \, \mathrm{d}t \Big|$$

$$\lesssim \begin{cases} \frac{J^{\alpha-2\beta+4}\log(\widetilde{n}/\eta)}{\widetilde{n}}, & when \ \frac{\alpha+1}{2} < \beta \leq \frac{\alpha+2}{2}, \\ \frac{J^2\log(\widetilde{n}/\eta)}{\widetilde{n}}, & when \ \frac{\alpha+2}{2} < \beta \leq \frac{\alpha+3}{2}, \\ \frac{J\log(\widetilde{n}/\eta)}{\widetilde{n}}, & when \ \frac{\alpha+3}{2} < \beta \leq \frac{\alpha+4}{2}, \\ \frac{\log(\widetilde{n}/\eta)}{\widetilde{n}}, & when \ \beta > \frac{\alpha+4}{2}. \end{cases}$$

*Proof.* Note that by the triangle inequality, we have that

$$\Big| \sum_{1 \leq j < k \leq J} \frac{\widehat{\theta}_{a,1,j} - \widehat{\theta}_{a,0,j}}{\widehat{\lambda}_{a,j}} \cdot \frac{\widehat{\theta}_{a,1,k} - \widehat{\theta}_{a,0,k}}{\widehat{\lambda}_{a,k}}$$

$$\cdot \int\int K_a(s,t)\{\widehat{\phi}_{a,j}(s) - \phi_{a,j}(s)\}\{\widehat{\phi}_{a,k}(t) - \phi_{a,k}(t)\} \, \mathrm{d}s \, \mathrm{d}t \Big|$$

$$\leq \sum_{1 \leq j < k \leq J} \frac{|\widehat{\theta}_{a,1,j} - \widehat{\theta}_{a,0,j}|}{\widehat{\lambda}_{a,j}} \cdot \frac{|\widehat{\theta}_{a,1,k} - \widehat{\theta}_{a,0,k}|}{\widehat{\lambda}_{a,k}}$$

$$\cdot \Big| \int\int K_a(s,t)\{\widehat{\phi}_{a,j}(s) - \phi_{a,j}(s)\}\{\widehat{\phi}_{a,k}(t) - \phi_{a,k}(t)\} \, \mathrm{d}s \, \mathrm{d}t \Big|$$

$$= \sum_{j=1}^{J-1}\sum_{k=j+1}^{J} \frac{|\widehat{\theta}_{a,1,j} - \widehat{\theta}_{a,0,j}|}{\widehat{\lambda}_{a,j}} \cdot \frac{|\widehat{\theta}_{a,1,k} - \widehat{\theta}_{a,0,k}|}{\widehat{\lambda}_{a,k}}$$

$$\cdot \Big| \int\int K_a(s,t)\{\widehat{\phi}_{a,j}(s) - \phi_{a,j}(s)\}\{\widehat{\phi}_{a,k}(t) - \phi_{a,k}(t)\} \, \mathrm{d}s \, \mathrm{d}t \Big|$$

$$(29)$$

The rest of the proof then follows a similar argument as the one used in the proof of Lemma 23. The proof below is constructed conditioning on the event in (27) and we will divide the proof into various cases. For any $\alpha > 1$, by Lemma 26, we have with probability at least $1 - \eta$ that

$$\left| \int \int K_a(s,t) \{\widehat{\phi}_{a,j}(s) - \phi_{a,j}(s)\} \{\widehat{\phi}_{a,k}(t) - \phi_{a,k}(t)\} \, \mathrm{d}s \, \mathrm{d}t \right| \lesssim \sqrt{\frac{j^{2-\alpha} k^{2-\alpha} \log^2(\widetilde{n}/\eta)}{\widetilde{n}^2}}.$$

**Case 1: When $(\alpha+1)/2 < \beta \le (\alpha+2)/2$.** In this scenario, by Lemma 39, we have with probability at least $1 - \eta/2$ that, for any $j \in [J]$, $|\widehat{\theta}_{a,1,j} - \widehat{\theta}_{a,0,j}| \lesssim j^{-\beta}$. Therefore, we have that

$$(29) \lesssim \frac{\log(\widetilde{n}/\eta)}{\widetilde{n}} \sum_{j=1}^{J-1} j^{1-\frac{\alpha}{2}} \frac{|\widehat{\theta}_{a,1,j} - \widehat{\theta}_{a,0,j}|}{\widehat{\lambda}_{a,j}} \sum_{k=j+1}^{J} k^{1-\frac{\alpha}{2}} \frac{|\widehat{\theta}_{a,1,k} - \widehat{\theta}_{a,0,k}|}{\widehat{\lambda}_{a,k}}$$

$$\lesssim \frac{\log(\widetilde{n}/\eta)}{\widetilde{n}} \sum_{j=1}^{J-1} j^{\frac{\alpha}{2}-\beta+1} \sum_{k=j+1}^{J} k^{\frac{\alpha}{2}-\beta+1} \le \frac{\log(\widetilde{n}/\eta)}{\widetilde{n}} \sum_{j=1}^{J-1} j^{\frac{\alpha}{2}-\beta+1} \cdot J^{\frac{\alpha}{2}-\beta+2}$$

$$\le \frac{J^{\alpha-2\beta+4} \log(\widetilde{n}/\eta)}{\widetilde{n}},$$

where the third inequality follows as $\alpha/2 - \beta + 1 > 0$.

**Case 2: When $(\alpha + 2)/2 < \beta \le (\alpha + 3)/2$.** In this scenario, by Lemma 39, we still have with probability at least $1 - \eta/2$ that, for any $j \in [J]$, $|\widehat{\theta}_{a,1,j} - \widehat{\theta}_{a,0,j}| \lesssim j^{-\beta}$. Therefore, we have that

$$(29) \lesssim \frac{\log(\widetilde{n}/\eta)}{\widetilde{n}} \sum_{j=1}^{J-1} j^{\frac{\alpha}{2}-\beta+1} \sum_{k=j+1}^{J} k^{\frac{\alpha}{2}-\beta+1} \le \frac{J \log(\widetilde{n}/\eta)}{\widetilde{n}} \sum_{j=1}^{J-1} j^{\alpha-2\beta+2} \le \frac{J^2 \log(\widetilde{n}/\eta)}{\widetilde{n}},$$

where the second inequality follows as $\sum_{k=j+1}^{J} k^{\alpha/2-\beta+1} \le J j^{\alpha/2-\beta+1}$ as $-1 < \alpha/2-\beta+1 < 0$ and the last inequality follows as $-1 \le \alpha - 2\beta + 2 < 0$.

**Case 3: When $(\alpha + 3)/2 < \beta \le (\alpha + 4)/2$.** In this scenario, by Lemma 39, we still have with probability at least $1 - \eta/2$ that, for any $j \in [J]$, $|\widehat{\theta}_{a,1,j} - \widehat{\theta}_{a,0,j}| \lesssim j^{-\beta}$. Therefore, we have that

$$(29) \lesssim \frac{\log(\widetilde{n}/\eta)}{\widetilde{n}} \sum_{j=1}^{J-1} j^{\frac{\alpha}{2}-\beta+1} \sum_{k=j+1}^{J} k^{\frac{\alpha}{2}-\beta+1} \le \frac{J \log(\widetilde{n}/\eta)}{\widetilde{n}} \sum_{j=1}^{J-1} j^{\alpha-2\beta+2} \le \frac{J \log(\widetilde{n}/\eta)}{\widetilde{n}},$$

where the second inequality follows as $-1 \le \alpha/2 - \beta + 1 < 0$ and the last inequality follows as $\alpha - 2\beta + 2 < -1$.

**Case 4: When $\beta > (\alpha + 4)/2$.** In this case, we have that with probability at least $1 - \eta/2$ that, for any $j \in [J]$,

$$|\widehat{\theta}_{a,1,j} - \widehat{\theta}_{a,0,j}| \lesssim j^{-\beta} + \sqrt{\frac{j^{-\alpha} \log(\widetilde{n}/\eta)}{\widetilde{n}}}.$$

Therefore, we have that

$$(29) \lesssim \frac{\log(\widetilde{n}/\eta)}{\widetilde{n}} \sum_{j=1}^{J-1} \left\{ j^{\frac{\alpha}{2}-\beta+1} \vee \sqrt{\frac{j^2 \log(\widetilde{n}/\eta)}{\widetilde{n}}} \right\} \sum_{k=j+1}^{J} \left\{ k^{\frac{\alpha}{2}-\beta+1} \vee \sqrt{\frac{k^2 \log(\widetilde{n}/\eta)}{\widetilde{n}}} \right\}$$

$$\le \frac{\log(\widetilde{n}/\eta)}{\widetilde{n}} \sum_{j=1}^{J-1} \left\{ j^{\frac{\alpha}{2}-\beta+1} \vee \sqrt{\frac{j^2 \log(\widetilde{n}/\eta)}{\widetilde{n}}} \right\} \left\{ 1 \vee \sqrt{\frac{J^4 \log(\widetilde{n}/\eta)}{\widetilde{n}}} \right\}$$

$$\le \frac{\log(\widetilde{n}/\eta)}{\widetilde{n}} \left\{ \sum_{j=1}^{J-1} j^{\frac{\alpha}{2}-\beta+1} \vee \sum_{j=1}^{J-1} \sqrt{\frac{j^2 \log(\widetilde{n}/\eta)}{\widetilde{n}}} \right\} \le \frac{\log(\widetilde{n}/\eta)}{\widetilde{n}},$$

where the second inequality follows as $\alpha/2 - \beta + 1 < -1$ and the third inequality follows from the assumption of $J$.

The lemma thus follows by combining results for all cases together. $\square$

**Lemma 25.** *Under Assumptions 1 and 2a, for $a \in \{0,1\}$ and any small constant $\eta \in (0, 1/2)$, it holds with probability at least $1 - \eta$ that, for any $j \in [J]$ such that $J^{2\alpha+2} \log(1/\eta) \lesssim \widetilde{n}$,*

$$\left| \int \int K_a(s,t) \{\widehat{\phi}_{a,j}(s) - \phi_{a,j}(s)\}\{\widehat{\phi}_{a,j}(t) - \phi_{a,j}(t)\} \, \mathrm{d}s \, \mathrm{d}t \right| \lesssim \frac{j^{2-\alpha} \log(\widetilde{n}/\eta)}{\widetilde{n}}.$$

*Proof.* By Lemma 46, we have that

$$\left| \int \int K_a(s,t) \{\widehat{\phi}_{a,j}(t) - \phi_{a,j}(t)\}\{\widehat{\phi}_{a,j}(s) - \phi_{a,j}(s)\} \, \mathrm{d}s \, \mathrm{d}t \right|$$

$$\leq \left| \int \{\widehat{\phi}_{a,j}(m) - \phi_{a,j}(m)\}\phi_{a,j}(m) \, \mathrm{d}m \cdot \int \int K_a(s,t)\phi_{a,j}(t)\{\widehat{\phi}_{a,j}(s) - \phi_{a,j}(s)\} \, \mathrm{d}s \, \mathrm{d}t \right|$$

$$+ \left| \sum_{l:l \neq j} (\widehat{\lambda}_{a,j} - \lambda_{a,l})^{-1} \int \int K_a(s,t)\phi_{a,l}(t)\{\widehat{\phi}_{a,j}(s) - \phi_{a,j}(s)\} \, \mathrm{d}s \, \mathrm{d}t \right.$$

$$\left. \cdot \int \int \{\widehat{K}_a(m,\ell) - K_a(m,\ell)\}\widehat{\phi}_{a,j}(m)\phi_{a,l}(\ell) \, \mathrm{d}m \, \mathrm{d}\ell \right|, \tag{30}$$

and for any $l \in \mathbb{N}+$, it holds that

$$\int \int K_a(s,t)\phi_{a,l}(t)\{\widehat{\phi}_{a,j}(s) - \phi_{a,j}(s)\} \, \mathrm{d}s \, \mathrm{d}t$$

$$= \int \int K_a(s,t)\phi_{a,l}(t)\phi_{a,j}(s) \, \mathrm{d}s \, \mathrm{d}t \int \{\widehat{\phi}_{a,j}(m) - \phi_{a,j}(m)\}\phi_{a,j}(m) \, \mathrm{d}m$$

$$+ \sum_{r:r \neq j} (\widehat{\lambda}_{a,j} - \lambda_{a,r})^{-1} \int \int K_a(s,t)\phi_{a,l}(t)\phi_{a,r}(s) \, \mathrm{d}s \, \mathrm{d}t$$

$$\cdot \int \int \{\widehat{K}_a(m,\ell) - K_a(m,\ell)\}\widehat{\phi}_{a,j}(m)\phi_{a,r}(\ell) \, \mathrm{d}m \, \mathrm{d}\ell$$

$$= \begin{cases} (\widehat{\lambda}_{a,j} - \lambda_{a,l})^{-1}\lambda_{a,l} \int \int \{\widehat{K}_a(m,\ell) - K_a(m,\ell)\}\widehat{\phi}_{a,j}(m)\phi_{a,l}(\ell) \, \mathrm{d}m \, \mathrm{d}\ell, & \text{when } l \neq j, \\ \lambda_{a,j} \int \{\widehat{\phi}_{a,j}(m) - \phi_{a,j}(m)\}\phi_{a,j}(m) \, \mathrm{d}m & \text{when } l = j. \end{cases} \tag{31}$$

Substituting the results in (31) into (30), we have that

$$\left| \int \int K_a(s,t) \{\widehat{\phi}_{a,j}(t) - \phi_{a,j}(t)\}\{\widehat{\phi}_{a,j}(s) - \phi_{a,j}(s)\} \, \mathrm{d}s \, \mathrm{d}t \right|$$

$$\leq \left| \lambda_{a,j} \left\{ \int \{\widehat{\phi}_{a,j}(m) - \phi_{a,j}(m)\}\phi_{a,j}(m) \, \mathrm{d}m \right\}^2 \right|$$

$$+ \left| \sum_{l:l \neq j} \lambda_{a,j}(\widehat{\lambda}_{a,j} - \lambda_{a,l})^{-2} \left\{ \int \int \{\widehat{K}_a(m,\ell) - K_a(m,\ell)\}\widehat{\phi}_{a,j}(m)\phi_{a,l}(\ell) \, \mathrm{d}m \, \mathrm{d}\ell \right\}^2 \right|$$

$$= (I) + (II). \tag{32}$$

Consider the following events,

$$\mathcal{E}_1 = \{(\widehat{\lambda}_{a,j} - \lambda_{a,k})^{-2} \leq 2(\lambda_{a,j} - \lambda_{a,k})^{-2}, \quad \text{for } k \in \mathbb{N}\backslash\{j\}, \ j \in [J], \ a \in \{0,1\}\},$$

$$\mathcal{E}_2 = \left\{ \|\widehat{K}_a - K_a\|_{L^2} \lesssim \sqrt{\frac{\log(1/\eta)}{\widetilde{n}}}, \ a \in \{0,1\} \right\},$$

$$\mathcal{E}_3 = \left\{ \|\widehat{\phi}_{a,j} - \phi_{a,j}\|_{L^2} \lesssim \sqrt{\frac{j^2 \log(\widetilde{n}/\eta)}{\widetilde{n}}}, \quad \text{for } j \in [J], \ a \in \{0,1\} \right\},$$

and

$$\mathcal{E}_4 = \left\{ \left| \int \int \{\widehat{K}_a(m,\ell) - K_a(m,\ell)\} \phi_{a,j}(m) \phi_{a,k}(\ell) \, dm \, d\ell \right| \lesssim \sqrt{\frac{j^{-\alpha} k^{-\alpha} \log(\widetilde{n}/\eta)}{\widetilde{n}}}, \right.$$

$$\left. k, j \in [J], \ a \in \{0,1\} \right\}.$$

By Lemmas 32, 34, 35, 37 and a similar argument as the one leads to (51), it holds from a union-bound argument that $\mathbb{P}(\mathcal{E}_1 \cap \mathcal{E}_2 \cap \mathcal{E}_3 \cap \mathcal{E}_4) \geq 1 - \eta$. The rest of the proof is constructed conditioning on these events happening. To control $(I)$, we have that

$$(I) \leq j^{-\alpha} \int \{\widehat{\phi}_{a,j}(m) - \phi_{a,j}(m)\}^2 \, dm \cdot \int \phi_{a,j}^2(m) \, dm \lesssim \frac{j^{2-\alpha} \log(\widetilde{n}/\eta)}{\widetilde{n}}, \tag{33}$$

where the first inequality follows from Cauchy–Schwarz inequality. To control $(II)$, note that by the triangle inequality, we have that

$$(II) \leq \left| \sum_{l: l \neq j} \lambda_{a,l} (\widehat{\lambda}_{a,j} - \lambda_{a,l})^{-2} \left\{ \int \int \{\widehat{K}_a(m,\ell) - K_a(m,\ell)\} \phi_{a,j}(m) \phi_{a,l}(\ell) \, dm \, d\ell \right\}^2 \right|$$

$$+ \left| \sum_{l: l \neq j} \lambda_{a,l} (\widehat{\lambda}_{a,j} - \lambda_{a,l})^{-2} \right.$$

$$\left. \cdot \left\{ \int \int \{\widehat{K}_a(m,\ell) - K_a(m,\ell)\} \{\widehat{\phi}_{a,j}(m) - \phi_{a,j}(m)\} \phi_{a,l}(\ell) \, dm \, d\ell \right\}^2 \right|$$

$$= (II)_1 + (II)_2.$$

To control $(II)_1$ in a separate large probability event, by a similar argument as **Steps 2 and 3** in the proof of Lemma 41, we have that

$$(II)_1 \lesssim \sum_{l: l \neq j} \lambda_{a,l} (\widehat{\lambda}_{a,j} - \lambda_{a,l})^{-2} \cdot \frac{j^{-\alpha} l^{-\alpha} \log(\widetilde{n}/\eta)}{\widetilde{n}}$$

$$\leq \frac{j^{-\alpha} \log(\widetilde{n}/\eta)}{\widetilde{n}} \sum_{l: l \neq j} (\lambda_{a,j} - \lambda_{a,l})^{-2} l^{-2\alpha} \lesssim \frac{j^{2-\alpha} \log(\widetilde{n}/\eta)}{\widetilde{n}}, \tag{34}$$

where the last inequality follows from Lemma 44. Similarly, to control $(II)_2$ in a large probability event, we have that

$$(II)_2 \lesssim \sum_{l: l \neq j} \lambda_{a,l} (\lambda_{a,j} - \lambda_{a,l})^{-2} \|\widehat{K}_a - K_a\|_{L^2}^2 \|\widehat{\phi}_{a,j} - \phi_{a,j}\|_{L^2}^2 \|\phi_{a,l}\|_{L^2}^2$$

$$\lesssim \frac{\log(\widetilde{n}/\eta) \log(1/\eta)}{\widetilde{n}^2} \sum_{l: l \neq j} (\lambda_{a,j} - \lambda_{a,l})^{-2} j^2 l^{-\alpha}$$

$$\lesssim \frac{j^{4+\alpha} \log(\widetilde{n}/\eta) \log(1/\eta)}{\widetilde{n}^2} \lesssim \frac{j^{2-\alpha} \log(\widetilde{n}/\eta)}{\widetilde{n}}, \tag{35}$$

where the first inequality follows from Cauchy–Schwarz inequality, the third inequality follows from Lemma 44 and the last inequality follows from the assumption that $J^{2\alpha+2} \lesssim \widetilde{n}$. The lemma thus follows by substituting the results in (33), (34) and (35) into (32). □

**Lemma 26.** *Under Assumptions 1 and 2a, for any $a \in \{0,1\}$ and small constant $\eta \in (0, 1/2)$, it holds with probability at least $1 - \eta$ that, for any $j, k \in [J]$ such that $1 \leq j < k \leq J$, $J^{2\alpha+2} \log(1/\eta) \lesssim \widetilde{n}$,*

$$\left| \int \int K_a(s,t) \{\widehat{\phi}_{a,j}(s) - \phi_{a,j}(s)\} \{\widehat{\phi}_{a,k}(t) - \phi_{a,k}(t)\} \, ds \, dt \right| \lesssim \sqrt{\frac{j^{2-\alpha} k^{2-\alpha} \log^2(\widetilde{n}/\eta)}{\widetilde{n}^2}}.$$

*Proof.* By Lemma 46, we have that

$$
\left| \int \int K_a(s,t)\{\widehat{\phi}_{a,j}(t) - \phi_{a,j}(t)\}\{\widehat{\phi}_{a,k}(s) - \phi_{a,k}(s)\} \, \mathrm{d}s \, \mathrm{d}t \right|
$$

$$
= \left| \int \{\widehat{\phi}_{a,j}(m) - \phi_{a,j}(m)\}\phi_{a,j}(m) \, \mathrm{d}m \cdot \int \int K_a(s,t)\phi_{a,j}(t)\{\widehat{\phi}_{a,k}(s) - \phi_{a,k}(s)\} \, \mathrm{d}s \, \mathrm{d}t \right|
$$

$$
+ \left| \sum_{l:l\neq j} (\widehat{\lambda}_{a,j} - \lambda_{a,l})^{-1} \int \int K_a(s,t)\phi_{a,l}(t)\{\widehat{\phi}_{a,k}(s) - \phi_{a,k}(s)\} \, \mathrm{d}s \, \mathrm{d}t \right.
$$

$$
\left. \cdot \int \int \{\widehat{K}_a(m,\ell) - K_a(m,\ell)\}\widehat{\phi}_{a,j}(m)\phi_{a,l}(\ell) \, \mathrm{d}m \, \mathrm{d}\ell \right|,
\tag{36}
$$

and for any $l \in \mathbb{N}+$,

$$
\int \int K_a(s,t)\phi_{a,l}(t)\{\widehat{\phi}_{a,k}(s) - \phi_{a,k}(s)\} \, \mathrm{d}s \, \mathrm{d}t
$$

$$
= \int \int K_a(s,t)\phi_{a,l}(t)\phi_{a,k}(s) \, \mathrm{d}s \, \mathrm{d}t \int \{\widehat{\phi}_{a,k}(m) - \phi_{a,k}(m)\}\phi_{a,k}(m) \, \mathrm{d}m
$$

$$
+ \sum_{r:r\neq k} (\widehat{\lambda}_{a,k} - \lambda_{a,r})^{-1} \int \int K_a(s,t)\phi_{a,l}(t)\phi_{a,r}(s) \, \mathrm{d}s \, \mathrm{d}t
$$

$$
\cdot \int \int \{\widehat{K}_a(m,\ell) - K_a(m,\ell)\}\widehat{\phi}_{a,k}(m)\phi_{a,r}(\ell) \, \mathrm{d}m \, \mathrm{d}\ell. \tag{37}
$$

Substituting the results in (37) into (36) and using the property of eigenfunctions, we have that

$$
\left| \int \int K_a(s,t)\{\widehat{\phi}_{a,j}(t) - \phi_{a,j}(t)\}\{\widehat{\phi}_{a,k}(s) - \phi_{a,k}(s)\} \, \mathrm{d}s \, \mathrm{d}t \right|
$$

$$
= \left| \int \{\widehat{\phi}_{a,j}(m) - \phi_{a,j}(m)\}\phi_{a,j}(m) \, \mathrm{d}m \cdot (\widehat{\lambda}_{a,k} - \lambda_{a,j})^{-1}\lambda_{a,j} \right.
$$

$$
\left. \cdot \int \int \{\widehat{K}_a(m,\ell) - K_a(m,\ell)\}\widehat{\phi}_{a,k}(m)\phi_{a,j}(\ell) \, \mathrm{d}m \, \mathrm{d}\ell \right|
$$

$$
+ \left| (\widehat{\lambda}_{a,j} - \lambda_{a,k})^{-1}\lambda_{a,k} \int \{\widehat{\phi}_{a,k}(m) - \phi_{a,k}(m)\}\phi_{a,k}(m) \, \mathrm{d}m \right.
$$

$$
\left. \cdot \int \int \{\widehat{K}_a(m,\ell) - K_a(m,\ell)\}\widehat{\phi}_{a,j}(m)\phi_{a,k}(\ell) \, \mathrm{d}m \, \mathrm{d}\ell \right|
$$

$$
+ \left| \sum_{l:l\neq j,k} (\widehat{\lambda}_{a,j} - \lambda_{a,l})^{-1}(\widehat{\lambda}_{a,k} - \lambda_{a,l})^{-1}\lambda_{a,l} \right.
$$

$$
\cdot \int \int \{\widehat{K}_a(m,\ell) - K_a(m,\ell)\}\widehat{\phi}_{a,k}(m)\phi_{a,l}(\ell) \, \mathrm{d}m \, \mathrm{d}\ell
$$

$$
\left. \cdot \int \int \{\widehat{K}_a(m,\ell) - K_a(m,\ell)\}\widehat{\phi}_{a,j}(m)\phi_{a,l}(\ell) \, \mathrm{d}m \, \mathrm{d}\ell \right|
$$

$$
= (I) + (II) + (III). \tag{38}
$$

Consider the following events,

$$
\mathcal{E}_1 = \left\{ |\widehat{\lambda}_{a,j} - \lambda_{a,k}|^{-1} \leq \sqrt{2}|\lambda_{a,j} - \lambda_{a,k}|^{-1}, \quad \text{for } k \in \mathbb{N}\backslash\{j\}, \ j \in [J], \ a \in \{0,1\} \right\},
$$

$$
\mathcal{E}_2 = \left\{ \|\widehat{K}_a - K_a\|_{L^2} \lesssim \sqrt{\frac{\log(1/\eta)}{\widetilde{n}}}, \text{ for } a \in \{0,1\} \right\},
$$

$$\mathcal{E}_3 = \left\{ \|\widehat{\phi}_{a,j} - \phi_{a,j}\|_{L^2} \lesssim \sqrt{\frac{j^2 \log(\widetilde{n}/\eta)}{\widetilde{n}}}, \quad \text{for } j \in [J], \ a \in \{0,1\} \right\},$$

and

$$\mathcal{E}_4 = \left\{ \left| \int \int \{\widehat{K}_a(m,\ell) - K_a(m,\ell)\} \phi_{a,j}(m) \phi_{a,k}(\ell) \, \mathrm{d}m \, \mathrm{d}\ell \right| \right.$$
$$\left. \lesssim \sqrt{\frac{j^{-\alpha} k^{-\alpha} \log(\widetilde{n}/\eta)}{\widetilde{n}}}, \ k,j \in [J], \ a \in \{0,1\} \right\}.$$

By Lemmas 32, 34, 35, 37 and a similar argument as the one leads to (51), it holds from a union-bound argument that $\mathbb{P}(\mathcal{E}_1 \cap \mathcal{E}_2 \cap \mathcal{E}_3 \cap \mathcal{E}_4) \geq 1 - \eta$. The rest of the proof is constructed conditioning on these events happening, and we will control the three terms in (38) above separately in three large probability events below.

**Step 1: Upper bound on** $(I)$**.** Note that by triangle inequality, we have that

$$(I) \leq \left| \int \{\widehat{\phi}_{a,j}(m) - \phi_{a,j}(m)\} \phi_{a,j}(m) \, \mathrm{d}m \cdot (\widehat{\lambda}_{a,k} - \lambda_{a,j})^{-1} \lambda_{a,j} \right.$$
$$\left. \cdot \int \int \{\widehat{K}_a(m,\ell) - K_a(m,\ell)\} \{\widehat{\phi}_{a,k}(m) - \phi_{a,k}(m)\} \phi_{a,j}(\ell) \, \mathrm{d}m \, \mathrm{d}\ell \right|$$
$$+ \left| \int \{\widehat{\phi}_{a,j}(m) - \phi_{a,j}(m)\} \phi_{a,j}(m) \, \mathrm{d}m \cdot (\widehat{\lambda}_{a,k} - \lambda_{a,j})^{-1} \lambda_{a,j} \right.$$
$$\left. \cdot \int \int \{\widehat{K}_a(m,\ell) - K_a(m,\ell)\} \phi_{a,k}(m) \phi_{a,j}(\ell) \, \mathrm{d}m \, \mathrm{d}\ell \right|$$
$$= (I)_1 + (I)_2.$$

Note that the control of $(I)_1$ is similar to the argument in (35) and will be dominated by the upper bound we give on $(I)_2$. We omit the proof here. To control $(I)_2$, we have that

$$(I)_2 \leq \lambda_{a,j} |\lambda_{a,k} - \lambda_{a,j}|^{-1} \left| \int \{\widehat{\phi}_{a,j}(m) - \phi_{a,j}(m)\} \phi_{a,j}(m) \, \mathrm{d}m \right|$$
$$\cdot \left| \int \int \{\widehat{K}_a(m,\ell) - K_a(m,\ell)\} \phi_{a,k}(m) \phi_{a,j}(\ell) \, \mathrm{d}m \, \mathrm{d}\ell \right|$$
$$\leq j^{-\alpha} |\lambda_{a,k} - \lambda_{a,j}|^{-1} \|\widehat{\phi}_{a,j} - \phi_{a,j}\|_{L^2} \|\phi_{a,j}\|_{L^2} \sqrt{\frac{j^{-\alpha} k^{-\alpha} \log(\widetilde{n}/\eta)}{\widetilde{n}}}$$
$$\leq |\lambda_{a,k} - \lambda_{a,j}|^{-1} \sqrt{\frac{j^{2-3\alpha} k^{-\alpha} \log^2(\widetilde{n}/\eta)}{\widetilde{n}^2}},$$

where the second inequality follows from Cauchy–Schwarz inequality. Therefore, we have that

$$(I) \lesssim |\lambda_{a,k} - \lambda_{a,j}|^{-1} \sqrt{\frac{j^{2-3\alpha} k^{-\alpha} \log^2(\widetilde{n}/\eta)}{\widetilde{n}^2}}. \tag{39}$$

**Step 2: Upper bound on** $(II)$**.** The argument to control $(II)$ is similar to the one used in **Step 1**. We omit the proof here and we have that

$$(II) \lesssim |\lambda_{a,k} - \lambda_{a,j}|^{-1} \sqrt{\frac{k^{2-3\alpha} j^{-\alpha} \log^2(\widetilde{n}/\eta)}{\widetilde{n}^2}}. \tag{40}$$

**Step 3: Upper bound on** $(III)$**.** To control $(III)$, by Cauchy–Schwarz inequality, we have that

$$(III)$$

$$\leq \sqrt{\sum_{l:l\neq j,k} \lambda_{a,l}(\widehat{\lambda}_{a,j} - \lambda_{a,l})^{-2}\Big\{\int\int\{\widehat{K}_a(m,\ell) - K_a(m,\ell)\}\widehat{\phi}_{a,j}(m)\phi_{a,l}(\ell)\,\mathrm{d}m\,\mathrm{d}\ell\Big\}^2}$$

$$\cdot \sqrt{\sum_{l:l\neq j,k} \lambda_{a,l}(\widehat{\lambda}_{a,k} - \lambda_{a,l})^{-2}\Big\{\int\int\{\widehat{K}_a(m,\ell) - K_a(m,\ell)\}\widehat{\phi}_{a,k}(m)\phi_{a,l}(\ell)\,\mathrm{d}m\,\mathrm{d}\ell\Big\}^2}$$

$$= \sqrt{(III)_1} \cdot \sqrt{(III)_2}.$$

To control $(III)_1$, note that

$$(III)_1$$

$$\lesssim \sum_{l:l\neq j,k} \lambda_{a,l}(\lambda_{a,j} - \lambda_{a,l})^{-2}$$

$$\cdot \Big\{\int\int\{\widehat{K}_a(m,\ell) - K_a(m,\ell)\}\{\widehat{\phi}_{a,j}(m) - \phi_{a,j}(m)\}\phi_{a,l}(\ell)\,\mathrm{d}m\,\mathrm{d}\ell\Big\}^2$$

$$+ \sum_{l:l\neq j,k} \lambda_{a,l}(\lambda_{a,j} - \lambda_{a,l})^{-2}\Big\{\int\int\{\widehat{K}_a(m,\ell) - K_a(m,\ell)\}\phi_{a,j}(m)\phi_{a,l}(\ell)\,\mathrm{d}m\,\mathrm{d}\ell\Big\}^2$$

$$= (A) + (B).$$

Similarly, the upper bound we provide on $(A)$ will be masked off by the upper bound we provide on $(B)$, and we focus on the upper bound on $(B)$ below. We have, from a similar argument as the one used in **Steps 2 and 3** in the proof of Lemma 41, that

$$(B) \leq \sum_{l:l\neq j,k} \lambda_{a,l}(\lambda_{a,j} - \lambda_{a,l})^{-2} \cdot \frac{j^{-\alpha}l^{-\alpha}\log(\widetilde{n}/\eta)}{\widetilde{n}}$$

$$= \frac{j^{-\alpha}\log(\widetilde{n}/\eta)}{\widetilde{n}} \sum_{l:l\neq j,k}(\lambda_{a,j} - \lambda_{a,l})^{-2}l^{-2\alpha} \lesssim \frac{j^{2-\alpha}\log(\widetilde{n}/\eta)}{\widetilde{n}},$$

where the last inequality follows from Lemma 44. Similarly, we can show that

$$(III)_2 \lesssim \frac{k^{2-\alpha}\log(\widetilde{n}/\eta)}{\widetilde{n}}.$$

Therefore, we have that

$$(III) \lesssim \sqrt{\frac{j^{2-\alpha}\log(\widetilde{n}/\eta)}{\widetilde{n}} \cdot \frac{k^{2-\alpha}\log(\widetilde{n}/\eta)}{\widetilde{n}}} \lesssim \sqrt{\frac{j^{2-\alpha}k^{2-\alpha}\log^2(\widetilde{n}/\eta)}{\widetilde{n}^2}}. \tag{41}$$

**Step 4: Combine results together.** Substituting results in (39), (40) and (41) into (38) and applying a union bound argument, we have that for any $1 \leq j < k \leq J$

$$(41) \lesssim |\lambda_{a,k} - \lambda_{a,j}|^{-1}\sqrt{\frac{j^{2-3\alpha}k^{-\alpha}\log^2(\widetilde{n}/\eta)}{\widetilde{n}^2}}$$

$$+ |\lambda_{a,k} - \lambda_{a,j}|^{-1}\sqrt{\frac{k^{2-3\alpha}j^{-\alpha}\log^2(\widetilde{n}/\eta)}{\widetilde{n}^2}} + \sqrt{\frac{j^{2-\alpha}k^{2-\alpha}\log^2(\widetilde{n}/\eta)}{\widetilde{n}^2}}$$

$$\lesssim |\lambda_{a,k} - \lambda_{a,j}|^{-1}\sqrt{\frac{j^{2-3\alpha}k^{-\alpha}\log^2(\widetilde{n}/\eta)}{\widetilde{n}^2}} + \sqrt{\frac{j^{2-\alpha}k^{2-\alpha}\log^2(\widetilde{n}/\eta)}{\widetilde{n}^2}}$$

$$\lesssim \begin{cases} \sqrt{\frac{j^{4-\alpha}k^{-\alpha}\log^2(\widetilde{n}/\eta)}{\widetilde{n}^2}} + \sqrt{\frac{j^{2-\alpha}k^{2-\alpha}\log^2(\widetilde{n}/\eta)}{\widetilde{n}^2}} & \text{when } j < k \leq \{2j \wedge J\} \\[2ex] \sqrt{\frac{j^{2-\alpha}k^{-\alpha}\log^2(\widetilde{n}/\eta)}{\widetilde{n}^2}} + \sqrt{\frac{j^{2-\alpha}k^{2-\alpha}\log^2(\widetilde{n}/\eta)}{\widetilde{n}^2}} & \text{when } k > \{2j \wedge J\} \end{cases}$$

$$\lesssim \sqrt{\frac{j^{2-\alpha}k^{2-\alpha}\log^2(\widetilde{n}/\eta)}{\widetilde{n}^2}},$$

where the second and the last inequality follows from the fact that $j < k$, the third inequality follows from the fact that

$$|\lambda_{a,j} - \lambda_{a,k}| \gtrsim \begin{cases} |j-k|j^{-\alpha-1}, & \text{if } j/2 \leq k \leq 2j, \\ j^{-\alpha}, & \text{if } k > 2j. \end{cases}$$

The lemma thus follows.

$\square$

**Lemma 27.** *Under the same condition of Lemma 12, for any small constant $\eta \in (0,1/2)$, it holds with probability at least $1 - \eta$ that*

$$Q_3 = \sum_{j=1}^{J} \lambda_{a,j} \Big( \frac{\widehat{\theta}_{a,1,j} - \widehat{\theta}_{a,0,j}}{\widehat{\lambda}_{a,j}} - \frac{\theta_{a,1,j} - \theta_{a,0,j}}{\lambda_{a,j}} \Big)^2$$

$$\lesssim \begin{cases} \frac{J^{\alpha-2\beta+3}\log^2(J)\log(\widetilde{n}/\eta)}{\widetilde{n}}, & \text{when } \frac{\alpha+1}{2} < \beta \leq \frac{\alpha+2}{2}, \\ \frac{J\log(\widetilde{n}/\eta)}{\widetilde{n}}, & \text{when } \beta > \frac{\alpha+2}{2}. \end{cases}$$

*Proof.* To control $Q_3$, we have that

$$\sum_{j=1}^{J} \lambda_{a,j} \Big( \frac{\widehat{\theta}_{a,1,j} - \widehat{\theta}_{a,0,j}}{\widehat{\lambda}_{a,j}} - \frac{\theta_{a,1,j} - \theta_{a,0,j}}{\lambda_{a,j}} \Big)^2$$

$$= \sum_{j=1}^{J} \lambda_{a,j} \Big( \frac{\widehat{\theta}_{a,1,j} - \widehat{\theta}_{a,0,j}}{\widehat{\lambda}_{a,j}} - \frac{\theta_{a,1,j} - \theta_{a,0,j}}{\widehat{\lambda}_{a,j}} + \frac{\theta_{a,1,j} - \theta_{a,0,j}}{\widehat{\lambda}_{a,j}} - \frac{\theta_{a,1,j} - \theta_{a,0,j}}{\lambda_{a,j}} \Big)^2$$

$$\lesssim \sum_{j=1}^{J} \frac{\lambda_{a,j}}{\widehat{\lambda}_{a,j}^2} (\widehat{\theta}_{a,1,j} - \widehat{\theta}_{a,0,j} - \theta_{a,1,j} + \theta_{a,0,j})^2 + \sum_{j=1}^{J} (\theta_{a,1,j} - \theta_{a,0,j})^2 \lambda_{a,j} \Big| \frac{1}{\widehat{\lambda}_{a,j}} - \frac{1}{\lambda_{a,j}} \Big|^2$$

$$\lesssim \sum_{j=1}^{J} \frac{\lambda_{a,j}}{\widehat{\lambda}_{a,j}^2} (\widehat{\theta}_{a,1,j} - \widehat{\theta}_{a,0,j} - \theta_{a,1,j} + \theta_{a,0,j})^2 + \sum_{j=1}^{J} \frac{(\theta_{a,1,j} - \theta_{a,0,j})^2}{\lambda_{a,j}} \cdot \frac{|\widehat{\lambda}_{a,j} - \lambda_{a,j}|^2}{\widehat{\lambda}_{a,j}^2}.$$

The lemma thus follows by applying a union bound argument to results in Lemmas 28 and 29. $\square$

**Lemma 28.** *Under the same condition of Lemma 12, for any small constant $\eta \in (0,1/2)$, it holds with probability at least $1 - \eta$ that*

$$\sum_{j=1}^{J} \frac{\lambda_{a,j}}{\widehat{\lambda}_{a,j}^2} (\widehat{\theta}_{a,1,j} - \widehat{\theta}_{a,0,j} - \theta_{a,1,j} + \theta_{a,0,j})^2$$

$$\lesssim \begin{cases} \frac{J^{\alpha-2\beta+3}\log^2(J)\log(\widetilde{n}/\eta)}{\widetilde{n}}, & \text{when } \frac{\alpha+1}{2} < \beta \leq \frac{\alpha+2}{2}, \\ \frac{J\log(\widetilde{n}/\eta)}{\widetilde{n}}, & \text{when } \beta > \frac{\alpha+2}{2}. \end{cases}$$

*Proof.* Consider the following events

$$\mathcal{E}_1 = \Big\{ \lambda_{a,j}/2 \leq \widehat{\lambda}_{a,j} \leq 3\lambda_{a,j}/2 \text{ and } \Big| \frac{1}{\widehat{\lambda}_{a,j}} - \frac{1}{\lambda_{a,j}} \Big| \lesssim \frac{1}{\lambda_{a,j}^2} \sqrt{\frac{j^{-2\alpha}\log(\widetilde{n}/\eta)}{\widetilde{n}}},$$

$$j \in [J], a \in \{0,1\} \Big\},$$

$$\mathcal{E}_2 = \Big\{ |\widehat{\theta}_{a,1,j} - \theta_{a,1,j} + \widehat{\theta}_{a,0,j} - \theta_{a,0,j}| \lesssim \sqrt{\frac{j^{2-2\beta}\log^2(j)\log(\widetilde{n}/\eta)}{\widetilde{n}}},$$

$$j \in [J], a \in \{0, 1\}, \frac{\alpha + 1}{2} < \beta \leq \frac{\alpha + 2}{2} \Big\},$$

and

$$\mathcal{E}_3 = \left\{ |\widehat{\theta}_{a,1,j} - \theta_{a,1,j} + \widehat{\theta}_{a,0,j} - \theta_{a,0,j}| \lesssim \sqrt{\frac{j^{-\alpha} \log(\widetilde{n}/\eta)}{\widetilde{n}}}, \; j \in [J], a \in \{0, 1\}, \beta > \frac{\alpha + 2}{2} \right\}.$$

By Lemma 39, a similar argument used to control (44) and a union bound argument, it holds that $\mathbb{P}(\mathcal{E}_1 \cap \mathcal{E}_2 \cap \mathcal{E}_3) \geq 1 - \eta$. The rest of the proof is constructed conditioning on $\mathcal{E}_1 \cap \mathcal{E}_2 \cap \mathcal{E}_3$ happening and we have that

$$\sum_{j=1}^{J} \frac{\lambda_{a,j}}{\widehat{\lambda}_{a,j}^2} (\widehat{\theta}_{a,1,j} - \widehat{\theta}_{a,0,j} - \theta_{a,1,j} + \theta_{a,0,j})^2 \leq 4 \sum_{j=1}^{J} \frac{1}{\lambda_{a,j}} (\widehat{\theta}_{a,1,j} - \widehat{\theta}_{a,0,j} - \theta_{a,1,j} + \theta_{a,0,j})^2. \quad (42)$$

In the rest of the proof, we will present upper bounds on (42) in several cases depending on the relationship between $\alpha$ and $\beta$.

**Case 1: When $(\alpha + 1)/2 < \beta \leq (\alpha + 2)/2$.** In this case we have that

$$(42) \lesssim \sum_{j=1}^{J} j^{\alpha} \cdot \frac{j^{2-2\beta} \log^2(j) \log(\widetilde{n}/\eta)}{\widetilde{n}} \leq \frac{\log^2(J) \log(\widetilde{n}/\eta)}{\widetilde{n}} \sum_{j=1}^{J} j^{\alpha - 2\beta + 2}$$

$$\leq \frac{J^{\alpha - 2\beta + 3} \log^2(J) \log(\widetilde{n}/\eta)}{\widetilde{n}},$$

where the last inequality follows as $\alpha - 2\beta + 2 \geq 0$.

**Case 2: When $\beta > (\alpha + 2)/2$.** In this case we have that

$$(42) \lesssim \sum_{j=1}^{J} j^{\alpha} \cdot \frac{j^{-\alpha} \log(\widetilde{n}/\eta)}{\widetilde{n}} \lesssim \frac{J \log(\widetilde{n}/\eta)}{\widetilde{n}}.$$

$\square$

**Lemma 29.** *Under Assumptions 1 and 2, for any small constant $\eta \in (0, 1/2)$ and $J \in \mathbb{N}_+$ such that $J^{2\alpha + 2} \log^2(J) \log(\widetilde{n}/\eta) \lesssim \widetilde{n}$, it holds with probability at least $1 - \eta$ that*

$$\sum_{j=1}^{J} (\theta_{a,1,j} - \theta_{a,0,j})^2 \lambda_{a,j} \left| \frac{1}{\widehat{\lambda}_{a,j}} - \frac{1}{\lambda_{a,j}} \right|^2 \lesssim \frac{\log(\widetilde{n}/\eta)}{\widetilde{n}}.$$

*Proof.* Consider the following event,

$$\mathcal{E}_1 = \left\{ \lambda_{a,j}/2 \leq \widehat{\lambda}_{a,j} \leq 3\lambda_{a,j}/2 \text{ and } \left| \frac{1}{\widehat{\lambda}_{a,j}} - \frac{1}{\lambda_{a,j}} \right| \lesssim \frac{1}{\lambda_{a,j}^2} \sqrt{\frac{j^{-2\alpha} \log(\widetilde{n}/\eta)}{\widetilde{n}}}, \right.$$

$$\left. \text{for } j \in [J], a \in \{0, 1\} \right\}.$$

By a similar argument used to control (44), we have that $\mathbb{P}(\mathcal{E}_1) \geq 1 - \eta$. The rest of the proof is constructed conditioning on $\mathcal{E}_1$ happening. We thus have that

$$\sum_{j=1}^{J} (\theta_{a,1,j} - \theta_{a,0,j})^2 \lambda_{a,j} \left| \frac{1}{\widehat{\lambda}_{a,j}} - \frac{1}{\lambda_{a,j}} \right|^2 \lesssim \sum_{j=1}^{J} (\theta_{a,1,j} - \theta_{a,0,j})^2 \lambda_{a,j} \cdot \frac{1}{\lambda_{a,j}^4} \frac{j^{-2\alpha} \log(\widetilde{n}/\eta)}{\widetilde{n}}$$

$$\lesssim \frac{\log(\widetilde{n}/\eta)}{\widetilde{n}} \sum_{j=1}^{J} \frac{(\theta_{a,1,j} - \theta_{a,0,j})^2}{\lambda_{a,j}} \asymp \frac{\log(\widetilde{n}/\eta)}{\widetilde{n}}.$$

The lemma thus follows. $\square$

**Lemma 30.** *Under the same condition of Lemma 12, for any small constant $\eta \in (0, 1/2)$, it holds with probability at least $1 - \eta$ that*

$$\Big| \sum_{j=1}^{J} \frac{(\widehat{\theta}_{a,1,j} - \widehat{\theta}_{a,0,j})^2}{\widehat{\lambda}_{a,j}} - \sum_{j=1}^{J} \frac{(\theta_{a,1,j} - \theta_{a,0,j})^2}{\lambda_{a,j}} \Big|$$

$$\lesssim \begin{cases} \sqrt{\dfrac{J^2 \log^2(J) \log(\widetilde{n}/\eta)}{\widetilde{n}}}, & \text{when } \frac{\alpha+1}{2} < \beta \le \frac{\alpha+2}{2}, \\ \sqrt{\dfrac{\log(\widetilde{n}/\eta)}{\widetilde{n}}}, & \text{when } \beta > \frac{\alpha+2}{2}. \end{cases}$$

*Proof.* By triangle inequality, we have that

$$\Big| \sum_{j=1}^{J} \frac{(\widehat{\theta}_{a,1,j} - \widehat{\theta}_{a,0,j})^2}{\widehat{\lambda}_{a,j}} - \sum_{j=1}^{J} \frac{(\theta_{a,1,j} - \theta_{a,0,j})^2}{\lambda_{a,j}} \Big|$$

$$\lesssim \Big| \sum_{j=1}^{J} \frac{(\widehat{\theta}_{a,1,j} - \widehat{\theta}_{a,0,j})^2 - (\theta_{a,1,j} - \theta_{a,0,j})^2}{\widehat{\lambda}_{a,j}} \Big|$$

$$+ \Big| \sum_{j=1}^{J} \frac{(\theta_{a,1,j} - \theta_{a,0,j})^2}{\widehat{\lambda}_{a,j}} - \sum_{j=1}^{J} \frac{(\theta_{a,1,j} - \theta_{a,0,j})^2}{\lambda_{a,j}} \Big|$$

$$= \Big| \sum_{j=1}^{J} \frac{(\widehat{\theta}_{a,1,j} - \widehat{\theta}_{a,0,j} + \theta_{a,1,j} - \theta_{a,0,j})(\widehat{\theta}_{a,1,j} - \widehat{\theta}_{a,0,j} - \theta_{a,1,j} + \theta_{a,0,j})}{\widehat{\lambda}_{a,j}} \Big|$$

$$+ \Big| \sum_{j=1}^{J} \frac{(\theta_{a,1,j} - \theta_{a,0,j})^2}{\widehat{\lambda}_{a,j}} - \sum_{j=1}^{J} \frac{(\theta_{a,1,j} - \theta_{a,0,j})^2}{\lambda_{a,j}} \Big|$$

$$\lesssim \sum_{j=1}^{J} \frac{(|\widehat{\theta}_{a,1,j} - \widehat{\theta}_{a,0,j}| + |\theta_{a,1,j} - \theta_{a,0,j}|) \cdot |\widehat{\theta}_{a,1,j} - \widehat{\theta}_{a,0,j} - \theta_{a,1,j} + \theta_{a,0,j}|}{\widehat{\lambda}_{a,j}}$$

$$+ \Big| \sum_{j=1}^{J} \frac{(\theta_{a,1,j} - \theta_{a,0,j})^2}{\widehat{\lambda}_{a,j}} - \sum_{j=1}^{J} \frac{(\theta_{a,1,j} - \theta_{a,0,j})^2}{\lambda_{a,j}} \Big|. \tag{43}$$

Consider the events

$$\mathcal{E}_1 = \Big\{ \lambda_{a,j}/2 \le \widehat{\lambda}_{a,j} \le 3\lambda_{a,j}/2 \text{ and } \Big| \frac{1}{\widehat{\lambda}_{a,j}} - \frac{1}{\lambda_{a,j}} \Big| \lesssim \frac{1}{\lambda_{a,j}^2} \sqrt{\frac{j^{-2\alpha} \log(\widetilde{n}/\eta)}{\widetilde{n}}},$$

$$\text{for } j \in [J], a \in \{0,1\} \Big\}, \tag{44}$$

$$\mathcal{E}_2 = \Big\{ |\widehat{\theta}_{a,1,j} - \widehat{\theta}_{a,0,j}| \le j^{-\beta}, \quad \text{for } j \in [J], a \in \{0,1\}, \frac{\alpha+1}{2} < \beta \le \frac{3\alpha+2}{2} \Big\},$$

and

$$\mathcal{E}_3 = \Big\{ |\widehat{\theta}_{a,1,j} - \widehat{\theta}_{a,0,j}| \lesssim \sqrt{\frac{j^{-\alpha} \log(\widetilde{n}/\eta)}{\widetilde{n}}}, \quad \text{for } j \in [J], a \in \{0,1\}, \beta > \frac{3\alpha+2}{2} \Big\}.$$

Note that by (58), it holds with probability at least $1 - \eta/6$ that

$$\frac{\lambda_{a,j}}{2} \lesssim \lambda_{a,j} - \sqrt{\frac{j^{-2\alpha} \log(\widetilde{n}/\eta)}{\widetilde{n}}} \le \widehat{\lambda}_{a,j} \lesssim \lambda_{a,j} + \sqrt{\frac{j^{-2\alpha} \log(\widetilde{n}/\eta)}{\widetilde{n}}} \le \frac{3\lambda_{a,j}}{2},$$

where the first and the fourth inequality follow from the fact that

$$\sqrt{\frac{j^{-2\alpha} \log(\widetilde{n}/\eta)}{\widetilde{n}}} \lesssim j^{-\alpha} \le \frac{\lambda_{a,j}}{2}.$$

By Lemma 36, we also have with probability at least $1 - \eta/6$ that for all $j \in [J]$,

$$\left|\frac{1}{\widehat{\lambda}_{a,j}} - \frac{1}{\lambda_{a,j}}\right| = \frac{|\lambda_{a,j} - \widehat{\lambda}_{a,j}|}{\lambda_{a,j}\widehat{\lambda}_{a,j}} \lesssim \frac{1}{\lambda_{a,j}^2}\sqrt{\frac{j^{-2\alpha}\log(\widetilde{n}/\eta)}{\widetilde{n}}} \lesssim \frac{1}{\lambda_{a,j}}\sqrt{\frac{\log(\widetilde{n}/\eta)}{\widetilde{n}}}.$$

Additionally, by Lemma 39, we have with probability at least $1 - \eta/3$ that for all $j \in [J]$,

$$|\widehat{\theta}_{a,1,j} - \widehat{\theta}_{a,0,j}| \lesssim \begin{cases} |\theta_{a,1,j} - \theta_{a,0,j}| + \sqrt{\frac{j^{2-2\beta}\log^2(j)\log(\widetilde{n}/\eta)}{\widetilde{n}}}, & \text{when } \frac{\alpha+1}{2} < \beta \le \frac{\alpha+2}{2}, \\ |\theta_{a,1,j} - \theta_{a,0,j}| + \sqrt{\frac{j^{-\alpha}\log(\widetilde{n}/\eta)}{\widetilde{n}}}, & \text{when } \beta > \frac{\alpha+2}{2}, \end{cases}$$

$$\lesssim \begin{cases} j^{-\beta}, & \text{when } \frac{\alpha+1}{2} < \beta \le \frac{3\alpha+2}{2}, \\ j^{-\beta} \vee \sqrt{\frac{j^{-\alpha}\log(\widetilde{n}/\eta)}{\widetilde{n}}}, & \text{when } \beta > \frac{3\alpha+2}{2}, \end{cases}$$

where the second inequality follows from the fact that

- When $\frac{\alpha+1}{2} < \beta \le \frac{\alpha+2}{2}$, it holds that

$$\sqrt{\frac{j^{2-2\beta}\log^2(j)\log(\widetilde{n}/\eta)}{\widetilde{n}}} \asymp j^{-\beta}\sqrt{\frac{j^2\log^2(j)\log(\widetilde{n}/\eta)}{\widetilde{n}}} \lesssim j^{-\beta}. \tag{45}$$

- When $\frac{\alpha+2}{2} < \beta \le \frac{3\alpha+2}{2}$, it holds that

$$\sqrt{\frac{j^{-\alpha}\log(\widetilde{n}/\eta)}{\widetilde{n}}} \asymp j^{-\beta}\sqrt{\frac{j^{2\beta-\alpha}\log(\widetilde{n}/\eta)}{\widetilde{n}}} \lesssim j^{-\beta}\sqrt{\frac{j^{3\alpha+2-\alpha}\log(\widetilde{n}/\eta)}{\widetilde{n}}} \lesssim j^{-\beta}.$$

- When $\beta > \frac{3\alpha+2}{2}$, different term dominates depending on the value of $j \in [J]$.

Therefore, by a union bound argument, we have that $\mathbb{P}(\mathcal{E}_k) \ge 1 - \eta/3$ for any $k \in \{1, 2, 3\}$. In addition, consider the following disjoint events,

$$\mathcal{E}_4 = \left\{|\widehat{\theta}_{a,1,j} - \theta_{a,1,j} + \widehat{\theta}_{a,0,j} - \theta_{a,0,j}| \lesssim \sqrt{\frac{j^{2-2\beta}\log^2(j)\log(\widetilde{n}/\eta)}{\widetilde{n}}},\right.$$

$$\left. j \in [J], a \in \{0,1\}, \frac{\alpha+1}{2} < \beta \le \frac{\alpha+2}{2}\right\},$$

$$\mathcal{E}_5 = \left\{|\widehat{\theta}_{a,1,j} - \theta_{a,1,j} + \widehat{\theta}_{a,0,j} - \theta_{a,0,j}| \lesssim \sqrt{\frac{j^{-\alpha}\log(\widetilde{n}/\eta)}{\widetilde{n}}},\ j \in [J], a \in \{0,1\}, \beta > \frac{\alpha+2}{2}\right\},$$

and by Lemma 39, we have that for each $k \in \{4, 5\}$, $\mathbb{P}(\mathcal{E}_k) \ge 1 - \eta/3$. In the rest of the proof, we will consider 3 different cases conditioning on various events happening based on the range of $\beta$.

**Case 1: When $(\alpha + 1)/2 < \beta \le (\alpha + 2)/2$.** In this case, the proof is constructed conditioning on $\mathcal{E}_1 \cap \mathcal{E}_2 \cap \mathcal{E}_4$ happening and by a union bound argument we have that $\mathbb{P}(\mathcal{E}_1 \cap \mathcal{E}_2 \cap \mathcal{E}_4) \ge 1 - \eta$. Consequently, it holds that

$$(43) \lesssim \sum_{j=1}^{J}\frac{j^{-\beta}}{\lambda_{a,j}}\sqrt{\frac{j^{2-2\beta}\log^2(j)\log(\widetilde{n}/\eta)}{\widetilde{n}}} + \sum_{j=1}^{J}\left|\frac{1}{\widehat{\lambda}_{a,j}} - \frac{1}{\lambda_{a,j}}\right|(\theta_{a,1,j} - \theta_{a,0,j})^2$$

$$\lesssim \sqrt{\frac{\log^2(J)\log(\widetilde{n}/\eta)}{\widetilde{n}}}\sum_{j=1}^{J}j^{\alpha-2\beta+1} + \sqrt{\frac{\log(\widetilde{n}/\eta)}{\widetilde{n}}}\sum_{j=1}^{J}\frac{(\theta_{a,1,j} - \theta_{a,0,j})^2}{\lambda_{a,j}}$$

$$\lesssim \sqrt{\frac{J^2\log^2(J)\log(\widetilde{n}/\eta)}{\widetilde{n}}},$$

where the last inequality follows from the fact that $-1 \leq \alpha - 2\beta + 1 < 0$.

**Case 2: When** $(\alpha + 2)/2 < \beta \leq (3\alpha + 2)/2$. In this case, the proof is constructed conditioning on $\mathcal{E}_1 \cap \mathcal{E}_2 \cap \mathcal{E}_5$ happening and by a union bound argument we have that $\mathbb{P}(\mathcal{E}_1 \cap \mathcal{E}_2 \cap \mathcal{E}_5) \geq 1 - \eta$. Consequently, it holds that

$$
(43) \lesssim \sum_{j=1}^{J} \frac{j^{-\beta}}{\lambda_{a,j}} \sqrt{\frac{j^{-\alpha}\log(\widetilde{n}/\eta)}{\widetilde{n}}} + \sqrt{\frac{\log(\widetilde{n}/\eta)}{\widetilde{n}}} \sum_{j=1}^{J} \frac{(\theta_{a,1,j} - \theta_{a,0,j})^2}{\lambda_{a,j}}
$$

$$
= \sqrt{\frac{\log(\widetilde{n}/\eta)}{\widetilde{n}}} \sum_{j=1}^{J} j^{\frac{\alpha}{2}-\beta} + \sqrt{\frac{\log(\widetilde{n}/\eta)}{\widetilde{n}}}
$$

$$
\lesssim \sqrt{\frac{\log(\widetilde{n}/\eta)}{\widetilde{n}}},
$$

where the last inequality follows from the fact that $\alpha/2 - \beta < -1$.

**Case 3: When** $\beta > (3\alpha + 2)/2$. In this case, the proof is constructed conditioning on $\mathcal{E}_1 \cap \mathcal{E}_3 \cap \mathcal{E}_5$ happening and by a union bound argument we have that $\mathbb{P}(\mathcal{E}_1 \cap \mathcal{E}_3 \cap \mathcal{E}_5) \geq 1 - \eta$. Consequently, it holds that

$$
(43) \lesssim \sum_{j=1}^{J} \frac{1}{\lambda_{a,j}} \left\{ j^{-\beta} \vee \sqrt{\frac{j^{-\alpha}\log(\widetilde{n}/\eta)}{\widetilde{n}}} \right\} \sqrt{\frac{j^{-\alpha}\log(\widetilde{n}/\eta)}{\widetilde{n}}} + \sqrt{\frac{\log(\widetilde{n}/\eta)}{\widetilde{n}}}
$$

$$
= \sum_{j=1}^{J} \frac{1}{\lambda_{a,j}} \left\{ \sqrt{\frac{j^{-2\beta-\alpha}\log(\widetilde{n}/\eta)}{\widetilde{n}}} \vee \frac{j^{-\alpha}\log(\widetilde{n}/\eta)}{\widetilde{n}} \right\} + \sqrt{\frac{\log(\widetilde{n}/\eta)}{\widetilde{n}}}
$$

$$
\lesssim \sum_{j=1}^{J} j^{\frac{\alpha}{2}-\beta} \sqrt{\frac{\log(\widetilde{n}/\eta)}{\widetilde{n}}} + \sum_{j=1}^{J} \frac{\log(\widetilde{n}/\eta)}{\widetilde{n}} + \sqrt{\frac{\log(\widetilde{n}/\eta)}{\widetilde{n}}}
$$

$$
\lesssim \frac{J\log(\widetilde{n}/\eta)}{\widetilde{n}} + \sqrt{\frac{\log(\widetilde{n}/\eta)}{\widetilde{n}}} \lesssim \sqrt{\frac{\log(\widetilde{n}/\eta)}{\widetilde{n}}},
$$

where the third inequality follows from the fact that $\alpha/2 - \beta < -1$ and the last inequality follows whenever $J^{2\alpha+2}\log^2(J)\log(\widetilde{n}/\eta) \lesssim \widetilde{n}$. The lemma thus follows by combining results from three cases. $\qquad \square$

# F   Proofs for class probability estimation

In this section, we present auxiliary lemmas related to class probabilities. Results below holds for any $a, y \in \{0, 1\}$.

**Lemma 31.** *Under Assumption 1, for any small $\epsilon > 0$ and $a, y \in \{0, 1\}$, it holds that*

$$
\mathbb{P}\left(|\widehat{\pi}_{a,y} - \pi_{a,y}| \geq \epsilon\right) \lesssim \exp\left(-\widetilde{n}\epsilon^2\right).
$$

*Proof.* Consider the sequence of bounded random variables $\{\mathbb{1}\{Y_j = y, A_j = a\}\}_{j=1}^{\widetilde{n}}$, then it holds that

$$
\widehat{\pi}_{a,y} = \frac{\widetilde{n}_{a,y}}{\widetilde{n}} = \frac{1}{\widetilde{n}} \sum_{i=1}^{\widetilde{n}} \mathbb{1}\{Y_j = y, A_j = a\}.
$$

Therefore, the lemma follows by applying Hoeffdings inequality for general bounded random variables (e.g. Theorem 2.2.6 in Vershynin, 2018) $\qquad \square$

Note that in the lemma below, the constant $1/5$ is arbitrary.

**Lemma 32.** *With $0 < C_p, C_p' < 1/5$ being the absolute constants in Assumption 1, consider the following events,*

$$\mathcal{E}_1 = \left\{ \left( \frac{C_p}{2} \wedge \frac{(1 - C_p')}{2} \right) \widetilde{n} \leq \widetilde{n}_{a,y} \leq \widetilde{n}, \ \text{for all} \ a, y \in \{0, 1\} \right\},$$

*and*

$$\mathcal{E}_2 = \left\{ \left( \frac{C_p}{2} \wedge \frac{(1 - C_p')}{2} \right) n \leq n_{a,y} \leq n, \ \text{for all} \ a, y \in \{0, 1\} \right\}.$$

*We have that $\mathbb{P}(\mathcal{E}_1 \cap \mathcal{E}_2) \geq 1 - \eta$ for some small $\eta \in (0, 1/2)$ whenever $\{\widetilde{n} \wedge n\} \gtrsim \log(1/\eta)$.*

*Proof.* For $y \in \{0, 1\}$ and $a \in \{0, 1\}$, consider the sequence of bounded random variables $\{\mathbb{1}\{Y_i = y, A_i = a\}\}_{i=1}^{\widetilde{n}}$. By Hoeffdings inequality for general bounded random variables (e.g. Theorem 2.2.6 in Vershynin, 2018), we have that for any $\epsilon_1, \epsilon_2 > 0$,

$$\mathbb{P}\left\{ \left| \sum_{i=1}^{\widetilde{n}} \mathbb{1}\{Y_i = 1, A_i = 1\} - \pi_{1,1}\widetilde{n} \right| \geq \epsilon_1 \right\} \leq \exp\left( -\frac{\epsilon_1^2}{\widetilde{n}} \right),$$

and

$$\mathbb{P}\left\{ \left| \sum_{i=1}^{\widetilde{n}} \mathbb{1}\{Y_i = 0, A_i = 1\} - (1 - \pi_{1,1})\widetilde{n} \geq \epsilon_2 \right| \right\} \leq \exp\left( -\frac{\epsilon_2^2}{\widetilde{n}} \right).$$

Therefore, we have that with probability at least $1 - \eta/4$ that

$$n_{1,1} = \sum_{i=1}^{\widetilde{n}} \mathbb{1}\{Y_i = 1, A = 1\} \geq \pi_{1,1}\widetilde{n} - \sqrt{\widetilde{n} \log(1/\eta)} \geq C_p\widetilde{n} - \frac{C_p}{2}\widetilde{n} \geq \frac{C_p\widetilde{n}}{2},$$

whenever $\widetilde{n} \geq 4\log(1/\eta)/C_p^2$. Similarly, we also have with probability at least $1 - \eta/4$ that

$$n_{1,0} = \sum_{i=1}^{\widetilde{n}} \mathbb{1}\{Y_i = 0, A = 1\} \geq (1 - \pi_0 - \pi_{1,1})\widetilde{n} - \sqrt{\widetilde{n} \log(1/\eta)}$$

$$\geq (1 - 3C_p')\widetilde{n} - \frac{(1 - 5C_p')}{2}\widetilde{n} \geq \frac{(1 - C_p')\widetilde{n}}{2},$$

whenever $\widetilde{n} \geq 4\log(1/\eta)/(1 - 5C_p')^2$. The other cases can be justified similarly. The lemma thus follows from a union bound argument. □

# G Proofs for functional data estimation

In this section, we present auxiliary lemmas related to mean, covariance, eigenvalue, eigenfunction and score estimation for the training data $\widetilde{\mathcal{D}}$. Denote $\widetilde{n}_a = \widetilde{n}_{a,0} + \widetilde{n}_{a,1}$ and the group-wise mean and covariance function by

$$\widehat{\mu}_{a,y}(t) = \frac{1}{\widetilde{n}_{a,y}} \sum_{i=1}^{\widetilde{n}_{a,y}} \widetilde{X}_{a,y}^i(t),$$

and

$$\widehat{K}_a(s, t) = \sum_{y \in \{0, 1\}} \frac{\widetilde{n}_{a,y}}{\widetilde{n}_{a,0} + \widetilde{n}_{a,1}} \frac{1}{\widetilde{n}_{a,y} - 1} \sum_{i=1}^{\widetilde{n}_{a,y}} \{\widetilde{X}_{a,y}^i(s) - \widehat{\mu}_{a,y}(s)\} \{\widetilde{X}_{a,y}^i(t) - \widehat{\mu}_{a,y}(t)\}.$$

We further let $\{\widehat{\lambda}_{a,j}\}_{j \geq 1}$ and $\{\widehat{\phi}_{a,j}\}_{j \geq 1}$ denote the eigenvalues and eigenfunctions of $\widehat{K}_a$ obtained by spectral expansion. The lemmas in the rest of the section holds for all $a \in \{0, 1\}$ and $y \in \{0, 1\}$.

## G.1 Mean and covariance function

**Lemma 33** (Lemma 1 in Zapata et al., 2022). *Assume Assumptions 1 and 2a hold. For any small $\epsilon_1, \epsilon_2 > 0$, it holds that*

$$\mathbb{P}\Big(\|\widehat{\mu}_{a,y} - \mu_{a,y}\|_{L^2} \geq \epsilon_1\Big) \lesssim \exp(-\widetilde{n}_{a,y}\epsilon_1^2) \quad and \quad \mathbb{P}\Big(\|\widehat{K}_{a,y} - K_a\|_{L^2} \geq \epsilon_2\Big) \lesssim \exp(-\widetilde{n}_{a,y}\epsilon_2^2),$$

*where $\widehat{K}_{a,y} = 1/(\widetilde{n}_{a,y} - 1)\sum_{i=1}^{\widetilde{n}_{a,y}}\{\widetilde{X}_{a,y}^i(s) - \widehat{\mu}_{a,y}(s)\}\{\widetilde{X}_{a,y}^i(t) - \widehat{\mu}_{a,y}(t)\}$.*

*Proof.* By standard properties of Gaussian processes and Assumption 2a, Assumption 2 in Zapata et al. (2022) is automatically satisfied. Hence, the lemma follows.

$\square$

**Lemma 34.** *Assume Assumptions 1 and 2a hold. It holds for any small $0 < \epsilon \lesssim 1$ that*

$$\mathbb{P}\Big(\|\widehat{K}_a - K_a\|_{L^2} \geq \epsilon\Big) \lesssim \exp(-\widetilde{n}_a\epsilon^2).$$

*Proof.* Note that

$$\widehat{K}_a(s,t) - K_a(s,t)$$

$$\asymp \frac{1}{\widetilde{n}_a}\sum_{y\in\{0,1\}}\sum_{i=1}^{n_{a,y}}\Big[\big\{\widetilde{X}_{a,y}^i(s) - \mu_{a,y}(s)\big\} - \big\{\widehat{\mu}_{a,y}(s) - \mu_{a,y}(s)\big\}\Big]$$

$$\cdot\Big[\big\{\widetilde{X}_{a,y}^i(t) - \mu_{a,y}(t)\big\} - \big\{\widehat{\mu}_{a,y}(t) - \mu_{a,y}(t)\big\}\Big] - \mathbb{E}\Big\{\big\{\widetilde{X}_{a,y}^i(s) - \mu_{a,y}(s)\big\}\big\{\widetilde{X}_{a,y}^i(t) - \mu_{a,y}(t)\big\}\Big\}$$

$$\asymp \frac{1}{\widetilde{n}_a}\sum_{y\in\{0,1\}}\sum_{i=1}^{n_{a,y}}\Big[\big\{\widetilde{X}_{a,y}^i(s) - \mu_{a,y}(s)\big\}\big\{\widetilde{X}_{a,y}^i(t) - \mu_{a,y}(t)\big\}$$

$$- \mathbb{E}\Big\{\big\{\widetilde{X}_{a,y}^i(s) - \mu_{a,y}(s)\big\}\big\{\widetilde{X}_{a,y}^i(t) - \mu_{a,y}(t)\big\}\Big\}\Big]$$

$$- \frac{1}{\widetilde{n}_a}\sum_{y\in\{0,1\}}\widetilde{n}_{a,y}\big\{\widehat{\mu}_{a,y}(s) - \mu_{a,y}(s)\big\}\big\{\widehat{\mu}_{a,y}(t) - \mu_{a,y}(t)\big\}. \tag{46}$$

Therefore, triangle inequality implies that

$$\|\widehat{K}_a(s,t) - K_a(s,t)\|_{L^2}$$

$$\lesssim \Big\|\frac{1}{\widetilde{n}_a}\sum_{y\in\{0,1\}}\sum_{i=1}^{n_{a,y}}\Big[\big\{\widetilde{X}_{a,y}^i(s) - \mu_{a,y}(s)\big\}\big\{\widetilde{X}_{a,y}^i(t) - \mu_{a,y}(t)\big\}$$

$$- \mathbb{E}\Big\{\big\{\widetilde{X}_{a,y}^i(s) - \mu_{a,y}(s)\big\}\big\{\widetilde{X}_{a,y}^i(t) - \mu_{a,y}(t)\big\}\Big\}\Big]\Big\|_{L^2}$$

$$+ \frac{1}{\widetilde{n}_a}\sum_{y\in\{0,1\}}\widetilde{n}_{a,y}\sqrt{\int\int\big\{\widehat{\mu}_{a,y}(s) - \mu_{a,y}(s)\big\}^2\big\{\widehat{\mu}_{a,y}(t) - \mu_{a,y}(t)\big\}^2\,\mathrm{d}s\,\mathrm{d}t}$$

$$= \Big\|\frac{1}{\widetilde{n}_a}\sum_{y\in\{0,1\}}\sum_{i=1}^{n_{a,y}}\Big[\big\{\widetilde{X}_{a,y}^i(s) - \mu_{a,y}(s)\big\}\big\{\widetilde{X}_{a,y}^i(t) - \mu_{a,y}(t)\big\}$$

$$- \mathbb{E}\Big\{\big\{\widetilde{X}_{a,y}^i(s) - \mu_{a,y}(s)\big\}\big\{\widetilde{X}_{a,y}^i(t) - \mu_{a,y}(t)\big\}\Big\}\Big]\Big\|_{L^2}$$

$$+ \frac{1}{\widetilde{n}_a}\sum_{y\in\{0,1\}}\widetilde{n}_{a,y}\|\widehat{\mu}_{a,y} - \mu_{a,y}\|_{L^2}^2$$

$$= (I) + (II). \tag{47}$$

To control $(I)$, note we have for any $\ell, k \in \mathbb{N}_+$ that

$$\int \int \Big[ \big\{ \widetilde{X}^i_{a,y}(s) - \mu_{a,y}(s) \big\} \big\{ \widetilde{X}^i_{a,y}(t) - \mu_{a,y}(t) \big\}$$

$$- \mathbb{E}\Big\{ \big\{ \widetilde{X}^i_{a,y}(s) - \mu_{a,y}(s) \big\} \big\{ \widetilde{X}^i_{a,y}(t) - \mu_{a,y}(t) \big\} \Big\} \Big] \phi_{a,\ell}(s) \phi_{a,k}(t) \, \mathrm{d}s \, \mathrm{d}t$$

$$= \frac{1}{\widetilde{n}_a} \sum_{y \in \{0,1\}} \sum_{i=1}^{n_{a,y}} \big( \xi^i_{a,y,\ell} \xi^i_{a,y,k} - \lambda_{a,\ell} \mathbb{1}\{\ell = k\} \big),$$

where for $\ell \in \mathbb{N}_+, \xi^i_{a,y,\ell} = \int \{\widetilde{X}^i_{a,y}(s) - \mu_{a,y}(s)\} \phi_{a,\ell}(s) \, \mathrm{d}s$. Therefore, by Lemma 45 and a similar argument as the one in Lemma 1 in Zapata et al. (2022) or Lemma 6 in Qiao et al. (2019), we have that for some small $0 < \epsilon \lesssim 1$,

$$\mathbb{P}\Big( (I) \geq \epsilon \Big) \lesssim \exp(-\widetilde{n}_a \epsilon^2). \tag{48}$$

To control $(II)$, by Lemma 33, it holds that for $y \in \{0, 1\}$,

$$\mathbb{P}\Big( \|\widehat{\mu}_{a,y} - \mu_{a,y}\|_{L^2} \geq \sqrt{\frac{\widetilde{n}_a \epsilon^2}{\widetilde{n}_{a,y}}} \Big) \lesssim \exp(-\widetilde{n}_a \epsilon^2). \tag{49}$$

Applying a union bound argument and substituting (48) and (49) into (47), we have that with probability at least $1 - \exp(-\widetilde{n}\epsilon^2)$,

$$\|\widehat{K}_a - K_a\|_{L^2} \lesssim \epsilon + \frac{1}{\widetilde{n}_a} \sum_{y \in \{0,1\}} \frac{\widetilde{n}_{a,y}\widetilde{n}_a \epsilon^2}{\widetilde{n}_{a,y}} = \epsilon + \frac{2\widetilde{n}_a \epsilon^2}{\widetilde{n}_a} \lesssim \epsilon + \epsilon^2 \lesssim \epsilon,$$

where the last inequality follows from the fact that $\epsilon \lesssim 1$. Thus, the Lemma follows. $\qquad \square$

## G.2 Eigenfunction

**Lemma 35.** *Assume Assumptions 1 and 2a hold. For any $j \in \mathbb{N}_+$ such that $j \leq J_{\widetilde{n}}$ where $J_{\widetilde{n}} > 0$ is a function of $\widetilde{n}$ and any small $0 < \epsilon \lesssim j J_{\widetilde{n}}^{-(\alpha+1)}$, we have that*

$$\mathbb{P}\Big( \|\widehat{\phi}_{a,j} - \phi_{a,j}\|_{L^2} \geq \epsilon \Big) \lesssim \exp\Big( -\frac{\widetilde{n}_a \epsilon^2}{j^2} \Big).$$

*Proof.* Note that by Lemma 46 and (5.16) in Hall and Horowitz (2007), it holds that

$$\|\widehat{\phi}_{a,j} - \phi_{a,j}\|_{L^2}$$

$$\leq \sqrt{2} \Big\| \sum_{k:k \neq j} \phi_{a,k}(\cdot)(\widehat{\lambda}_{a,j} - \lambda_{a,k})^{-1} \int \int \{\widehat{K}_a(s,t) - K_a(s,t)\} \widehat{\phi}_{a,j}(s) \phi_{a,k}(t) \, \mathrm{d}s \, \mathrm{d}t \Big\|_{L^2}$$

$$\leq \sqrt{2} \Big\| \sum_{k:k \neq j} \phi_{a,k}(\cdot)(\widehat{\lambda}_{a,j} - \lambda_{a,k})^{-1} \int \int \{\widehat{K}_a(s,t) - K_a(s,t)\} \phi_{a,j}(s) \phi_{a,k}(t) \, \mathrm{d}s \, \mathrm{d}t \Big\|_{L^2}$$

$$+ \sqrt{2} \Big\| \sum_{k:k \neq j} \phi_{a,k}(\cdot)(\widehat{\lambda}_{a,j} - \lambda_{a,k})^{-1}$$

$$\cdot \int \int \{\widehat{K}_a(s,t) - K_a(s,t)\}\{\widehat{\phi}_{a,j}(s) - \phi_{a,j}(s)\} \phi_{a,k}(t) \, \mathrm{d}s \, \mathrm{d}t \Big\|_{L^2}, \tag{50}$$

where the second inequality follows from the triangle inequality. Next, consider the event $\mathcal{E}_1 = \{\|\widehat{K}_a - K_a\|_{L^2} \leq \epsilon/j\}$ and we have that $\mathbb{P}(\mathcal{E}_1) \gtrsim 1 - \exp(-\widetilde{n}\epsilon^2/j^2)$ by Lemma 34. The rest of the proof is constructed conditioning on $\mathcal{E}_1$. Construct another event

$$\mathcal{E}_2 = \Big\{ (\widehat{\lambda}_{a,j} - \lambda_{a,k})^{-2} \leq 2(\lambda_{a,j} - \lambda_{a,k})^{-2} \lesssim J_{\widetilde{n}}^{2(\alpha+1)}, \quad \text{for } k \in \mathbb{N}\setminus\{j\}, \ j \in [J_{\widetilde{n}}] \Big\}. \tag{51}$$

We want to show $\mathcal{E}_2$ holds. By Weyl's inequality, it holds that $|\widehat{\lambda}_{a,j} - \lambda_{a,j}| \leq \|\widehat{K}_a - K_a\|_{L^2} \leq \epsilon/j$ for any $j \in \mathbb{N}_+$. This then implies that

$$|\widehat{\lambda}_{a,j} - \lambda_{a,k}| \geq |\lambda_{a,k} - \lambda_{a,j}| - |\widehat{\lambda}_{a,j} - \lambda_{a,j}| \geq |\lambda_{a,k} - \lambda_{a,j}| - \epsilon/j$$

$$\geq |\lambda_{a,k} - \lambda_{a,j}| - (1 - 2^{-1/2})|\lambda_{a,k} - \lambda_{a,j}| \geq 2^{-1/2}|\lambda_{a,k} - \lambda_{a,j}|, \qquad (52)$$

where the first inequality follows from triangle inequality and the third inequality follows from Assumption 2a,

$$|\lambda_{a,j} - \lambda_{a,k}| \gtrsim \begin{cases} k^{-\alpha}, & \text{if } k < j/2, \\ |j-k|j^{-\alpha-1}, & \text{if } j/2 \leq k \leq 2j, \\ j^{-\alpha}, & \text{if } k > 2j, \end{cases}$$

$$\gtrsim J_{\widetilde{n}}^{-(\alpha+1)},$$

and the fact that for $0 < \epsilon \leq (1 - 2^{-1/2})jJ_{\widetilde{n}}^{-(\alpha+1)}$, we have that

$$\frac{\epsilon}{j} \leq (1 - 2^{-1/2})J_{\widetilde{n}}^{-(\alpha+1)} \lesssim (1 - 2^{-1/2})|\lambda_{a,k} - \lambda_{a,j}|.$$

Thus, from (52), we have that conditioning on $\mathcal{E}_1$, $\mathcal{E}_2$ holds with probability 1. The rest of the proof is constructed conditioning on both $\mathcal{E}_1$ and $\mathcal{E}_2$. To control (50), we can further upper bound it by

(50)

$$= \sqrt{2}\Big[ \sum_{k:k\neq j} (\widehat{\lambda}_{a,j} - \lambda_{a,k})^{-2} \Big\{ \int \int \{\widehat{K}_a(s,t) - K_a(s,t)\}\phi_{a,j}(s)\phi_{a,k}(t)\, \mathrm{d}s\, \mathrm{d}t \Big\}^2 \Big]^{\frac{1}{2}}$$

$$+ \sqrt{2}\Big[ \sum_{k:k\neq j} (\widehat{\lambda}_{a,j} - \lambda_{a,k})^{-2} \Big\{ \int \int \{\widehat{K}_a(s,t) - K_a(s,t)\}$$

$$\cdot \{\widehat{\phi}_{a,j}(s) - \phi_{a,j}(s)\}\phi_{a,k}(t)\, \mathrm{d}s\, \mathrm{d}t \Big\}^2 \Big]^{\frac{1}{2}}$$

$$\leq 2\Big[ \sum_{k:k\neq j} (\lambda_{a,j} - \lambda_{a,k})^{-2} \Big\{ \int \int \{\widehat{K}_a(s,t) - K_a(s,t)\}\phi_{a,j}(s)\phi_{a,k}(t)\, \mathrm{d}s\, \mathrm{d}t \Big\}^2 \Big]^{\frac{1}{2}}$$

$$+ 2\Big[ \sum_{k:k\neq j} (\lambda_{a,j} - \lambda_{a,k})^{-2} \Big\{ \int \int \{\widehat{K}_a(s,t) - K_a(s,t)\}$$

$$\cdot \{\widehat{\phi}_{a,j}(s) - \phi_{a,j}(s)\}\phi_{a,k}(t)\, \mathrm{d}s\, \mathrm{d}t \Big\}^2 \Big]^{\frac{1}{2}}$$

$$= 2\Big\| \sum_{k:k\neq j} \phi_{a,k}(\cdot)(\lambda_{a,j} - \lambda_{a,k})^{-1} \int \int \{\widehat{K}_a(s,t) - K_a(s,t)\}\phi_{a,j}(s)\phi_{a,k}(t)\, \mathrm{d}s\, \mathrm{d}t \Big\|_{L^2}$$

$$+ 2\Big\| \sum_{k:k\neq j} \phi_{a,k}(\cdot)(\lambda_{a,j} - \lambda_{a,k})^{-1} \int \int \{\widehat{K}_a(s,t) - K_a(s,t)\}$$

$$\cdot \{\widehat{\phi}_{a,j}(s) - \phi_{a,j}(s)\}\phi_{a,k}(t)\, \mathrm{d}s\, \mathrm{d}t \Big\|_{L^2}$$

$$= 2(I) + 2(II),$$

where the last inequality follows from $\mathcal{E}_2$ and the orthogonality of eigenfunctions. In the rest of the proof, we construct large probability events to control upper bounds on $(I)$ and $(II)$.

**Step 1: upper bound on** $(I)$**.** Using the result in (46), we can further write $(I)$ as

$$(I) \lesssim \Big\| \frac{1}{\widetilde{n}_a} \sum_{y\in\{0,1\}} \sum_{i=1}^{\widetilde{n}_{a,y}} \sum_{k:k\neq j} \phi_{a,k}(\cdot)(\lambda_{a,j} - \lambda_{a,k})^{-1}$$

$$\cdot \int \int \Big[ \{\widetilde{X}^i_{a,y}(s) - \mu_{a,y}(s)\}\{\widetilde{X}^i_{a,y}(t) - \mu_{a,y}(t)\}$$

$$- \mathbb{E}\Big\{ \{\widetilde{X}^i_{a,y}(s) - \mu_{a,y}(s)\}\{\widetilde{X}^i_{a,y}(t) - \mu_{a,y}(t)\} \Big\} \Big] \phi_{a,j}(s)\phi_{a,k}(t) \, \mathrm{d}s \, \mathrm{d}t \Big\|_{L^2}$$

$$+ \Big\| \frac{1}{\widetilde{n}_a} \sum_{y\in\{0,1\}} \sum_{k:k\neq j} \phi_{a,k}(\cdot)(\lambda_{a,j} - \lambda_{a,k})^{-1} \widetilde{n}_{a,y}$$

$$\cdot \int \int \{\widehat{\mu}_{a,y}(s) - \mu_{a,y}(s)\}\{\widehat{\mu}_{a,y}(t) - \mu_{a,y}(t)\}\phi_{a,j}(s)\phi_{a,k}(t) \, \mathrm{d}s \, \mathrm{d}t \Big\|_{L^2}$$

$$= \Big\| \frac{1}{\widetilde{n}_a} \sum_{y\in\{0,1\}} \sum_{i=1}^{\widetilde{n}_{a,y}} \sum_{k:k\neq j} \phi_{a,k}(\cdot)(\lambda_{a,j} - \lambda_{a,k})^{-1} \xi^i_{a,y,j} \xi^i_{a,y,k} \Big\|_{L^2}$$

$$+ \Big\| \frac{1}{\widetilde{n}_a} \sum_{y\in\{0,1\}} \sum_{k:k\neq j} \phi_{a,k}(\cdot)(\lambda_{a,j} - \lambda_{a,k})^{-1} \widetilde{n}_{a,y} \bar{\xi}_{a,y,j} \bar{\xi}_{a,y,k} \Big\|_{L^2}$$

$$= (I)_1 + (I)_2$$

where for $k \in \mathbb{N}_+$, $\xi^i_{a,y,k} = \int \{\widetilde{X}^i_{a,y}(s) - \mu_{a,y}(s)\}\phi_{a,k}(s) \, \mathrm{d}s \sim N(0, \lambda_{a,k})$, $\bar{\xi}_{a,y,k} = \widetilde{n}_{a,y}^{-1} \sum_{i=1}^{\widetilde{n}_{a,y}} \xi^i_{a,y,k}$.

**Step 1-1: upper bound on** $(I)_1$. Denote $\{\{z_{i,j}\}_{i=1}^{\widetilde{n}_a}\}_{j\in\mathbb{N}_+}$ and $\{\{z_{i,k}\}_{i=1}^{\widetilde{n}_a}\}_{k\in\mathbb{N}_+}$ two collections of independent standard Gaussian random variables. With the above notation, by Lemma 45, to control $(I)_1$, it suffices to control

$$\sum_{i=1}^{\widetilde{n}_a} \mathbb{E}\Big\{ \Big\| \sum_{k:k\neq j} \phi_{a,k}(\cdot)(\lambda_{a,j} - \lambda_{a,k})^{-1} \sqrt{\lambda_{a,j}\lambda_{a,k}} z_{i,j} z_{i,k} \Big\|_{L^2}^b \Big\}.$$

Note that for any $i \in [\widetilde{n}_a]$, we have that

$$\mathbb{E}\Big\{ \Big\| \sum_{k:k\neq j} \phi_{a,k}(\cdot)(\lambda_{a,j} - \lambda_{a,k})^{-1} \sqrt{\lambda_{a,j}\lambda_{a,k}} z_{i,j} z_{i,k} \Big\|_{L^2}^b \Big\}$$

$$= \mathbb{E}\Big[ \Big\{ \sum_{k:k\neq j} (\lambda_{a,j} - \lambda_{a,k})^{-2} \lambda_{a,j}\lambda_{a,k} z_{i,j}^2 z_{i,k}^2 \Big\}^{\frac{b}{2}} \Big]$$

$$\lesssim j^{-\frac{b\alpha}{2}} \mathbb{E}\Big[ \Big\{ \sum_{k:k\neq j} (\lambda_{a,j} - \lambda_{a,k})^{-2} k^{-\alpha} z_{i,j}^2 z_{i,k}^2 \Big\}^{\frac{b}{2}} \Big]$$

$$= j^{-\frac{b\alpha}{2}} \Big\{ \sum_{k:k\neq j} (\lambda_{a,j} - \lambda_{a,k})^{-2} k^{-\alpha} \Big\}^{\frac{b}{2}} \mathbb{E}\Big[ \Big\{ \frac{\sum_{k:k\neq j} (\lambda_{a,j} - \lambda_{a,k})^{-2} k^{-\alpha} z_{i,j}^2 z_{i,k}^2}{\sum_{k:k\neq j} (\lambda_{a,j} - \lambda_{a,k})^{-2} k^{-\alpha}} \Big\}^{\frac{b}{2}} \Big]$$

$$\leq j^{-\frac{b\alpha}{2}} \Big\{ \sum_{k:k\neq j} (\lambda_{a,j} - \lambda_{a,k})^{-2} k^{-\alpha} \Big\}^{\frac{b}{2}} \frac{\sum_{k:k\neq j} (\lambda_{a,j} - \lambda_{a,k})^{-2} k^{-\alpha} \mathbb{E}(z_{i,j}^b z_{i,k}^b)}{\sum_{k:k\neq j} (\lambda_{a,j} - \lambda_{a,k})^{-2} k^{-\alpha}}$$

$$\lesssim j^{-\frac{b\alpha}{2}} (1 + j^{\alpha+2})^{\frac{b}{2}-1} \sum_{k:k\neq j} (\lambda_{a,j} - \lambda_{a,k})^{-2} k^{-\alpha} \mathbb{E}(z_{i,j}^b z_{i,k}^b)$$

$$\lesssim j^{-\frac{b\alpha}{2}} (1 + j^{\alpha+2})^{\frac{b}{2}-1} \sum_{k:k\neq j} (\lambda_{a,j} - \lambda_{a,k})^{-2} k^{-\alpha} \sqrt{\mathbb{E}(z_{i,j}^{2b})\mathbb{E}(z_{i,k}^{2b})}$$

$$\lesssim j^{-\frac{b\alpha}{2}} (1 + j^{\alpha+2})^{\frac{b}{2}-1} 2^b b! (1 + j^{\alpha+2}) = 2^b b! j^2 j^{b-2},$$

where the first equality follows from the orthogonality of $\{\phi_{a,k}\}_{k\in\mathbb{N}_+}$, the first inequality follows from Assumption 2a, the second inequality follows from Jensen's inequality, the third inequality follows from Lemma 44, the fourth inequality follows from Cauchy–Schwartz inequality and the fifth inequality follows from Lemma 44 and the fact that

$$\mathbb{E}\big(z_{i,j}^{2b}\big) = \pi^{-1/2}2^b\Gamma\Big(\frac{2b+1}{2}\Big) \leq 2^b b!.$$

Therefore, pick $L_1 \asymp j^2$ and $L_2 \asymp j$, we have that

$$\sum_{i=1}^{\widetilde{n}_a}\mathbb{E}\Big\{\Big\|\sum_{k:k\neq j}\phi_{a,k}(\cdot)(\lambda_{a,j}-\lambda_{a,k})^{-1}\sqrt{\lambda_{a,j}\lambda_{a,k}}z_{i,j}z_{i,k}\Big\|_{L^2}^b\Big\}$$

$$\lesssim 4b!Nj^2(2j)^{b-2} \lesssim b!NL_1L_2^{b-2}.$$

Hence, it holds from Lemma 45 that

$$\mathbb{P}\Big\{(I)_1 \geq \epsilon\Big\} \leq 2\exp\Big(-\frac{\widetilde{n}_a\epsilon^2}{2j^2+2j\epsilon}\Big) \lesssim \exp\Big(-\frac{\widetilde{n}_a\epsilon^2}{j^2}\Big), \tag{53}$$

whenever $\epsilon \lesssim j$.

**Step 1-2: upper bound on $(I)_2$.** To control $(I)_2$, it holds that

$$(I)_2 \leq \sum_{y\in\{0,1\}}\frac{\widetilde{n}_{a,y}}{\widetilde{n}_a}\Big\|\sum_{k:k\neq j}\phi_{a,k}(\cdot)(\lambda_{a,j}-\lambda_{a,k})^{-1}\bar{\xi}_{a,y,j}\bar{\xi}_{a,y,k}\Big\|_{L^2}$$

$$= \sum_{y\in\{0,1\}}\frac{\widetilde{n}_{a,y}}{\widetilde{n}_a}\sqrt{\sum_{k:k\neq j}(\lambda_{a,j}-\lambda_{a,k})^{-2}\bar{\xi}_{a,y,j}^2\bar{\xi}_{a,y,k}^2}, \tag{54}$$

where the first inequality follows from triangle inequality and the last equality follows from the orthonormality of $\{\phi_{a,k}\}_{k\in\mathbb{N}_+}$. Also, since for any $k\in\mathbb{N}_+$ and $a,y\in\{0,1\}$, we have that $\xi_{a,y,k}^i \overset{\text{i.i.d.}}{\sim} N(0,\lambda_{a,k})$ by the independence property. Then, by standard property of independent Gaussian random variables, this implies that $\bar{\xi}_{a,y,k} \sim N(0,\lambda_{a,k}/\widetilde{n}_{a,y})$.

Moreover, by standard properties of sub-Gaussian variables and Lemma 47, we have that $\sum_{k:k\neq j}(\lambda_{a,j}-\lambda_{a,k})^{-2}\bar{\xi}_{a,y,j}^2\bar{\xi}_{a,y,k}^2$ follows a sub-Weibull distribution with parameter $1/2$. We next upper bound its sub-Weibull norm. Note that

$$\Big\|\sum_{k:k\neq j}(\lambda_{a,j}-\lambda_{a,k})^{-2}\bar{\xi}_{a,y,j}^2\bar{\xi}_{a,y,k}^2\Big\|_{\psi_{1/2}} \leq \sum_{k:k\neq j}(\lambda_{a,j}-\lambda_{a,k})^{-2}\big\|\bar{\xi}_{a,y,j}^2\bar{\xi}_{a,y,k}^2\big\|_{\psi_{1/2}}$$

$$\leq \sum_{k:k\neq j}(\lambda_{a,j}-\lambda_{a,k})^{-2}\big\|\bar{\xi}_{a,y,j}\big\|_{\psi_2}^2\big\|\bar{\xi}_{a,y,k}\big\|_{\psi_2}^2 \leq \sum_{k:k\neq j}(\lambda_{a,j}-\lambda_{a,k})^{-2}\frac{\lambda_{a,k}\lambda_{a,j}}{\widetilde{n}_{a,y}^2}$$

$$\leq \frac{j^{-\alpha}}{\widetilde{n}_{a,y}^2}\sum_{k:k\neq j}(\lambda_{a,j}-\lambda_{a,k})^{-2}k^{-\alpha} \lesssim \frac{j^2}{\widetilde{n}_{a,y}^2},$$

where the first inequality follows from triangle inequality, the second inequality follows from standard property of sub-Gaussian random variables (e.g. Lemma 2.7.7 in Vershynin, 2018), the third inequality follows from the fact that for any $k\in\mathbb{N}_0$, $\big\|\bar{\xi}_{a,y,k}\big\|_{\psi_2} \lesssim \sqrt{\lambda_{a,k}/\widetilde{n}_{a,y}}$, the fourth inequality follows from Assumption 2a and the last inequality follows from Lemma 44. Therefore, by Lemma 47.1, it holds for any small $\delta > 0$ that

$$\mathbb{P}\Big(\Big|\sum_{k:k\neq j}(\lambda_{a,j}-\lambda_{a,k})^{-2}\bar{\xi}_{a,y,j}^2\bar{\xi}_{a,y,k}^2\Big| \geq \delta\Big) \lesssim \exp\Big\{-\Big(\frac{\widetilde{n}_{a,y}^2\delta}{j^2}\Big)^{\frac{1}{2}}\Big\}.$$

Pick $\delta = \widetilde{n}_a^2\epsilon^4/(\widetilde{n}_{a,y}^2j^2)$, we have that

$$\mathbb{P}\Big(\Big|\sum_{k:k\neq j}(\lambda_{a,j}-\lambda_{a,k})^{-2}\bar{\xi}_{a,y,j}^2\bar{\xi}_{a,y,k}^2\Big| \geq \frac{\widetilde{n}_a^2\epsilon^4}{\widetilde{n}_{a,y}^2j^2}\Big) \lesssim \exp\Big(-\frac{\widetilde{n}_a\epsilon^2}{j^2}\Big). \tag{55}$$

Substituting the result in (55) into (54), we have that with probability at least $1 - \exp(-\widetilde{n}_a \epsilon^2 / j^2)$ that

$$(I)_2 \lesssim \sum_{y \in \{0,1\}} \frac{\widetilde{n}_{a,y}}{\widetilde{n}_a} \sqrt{\frac{\widetilde{n}_a^2 \epsilon^4}{\widetilde{n}_{a,y}^2 j^2}} \lesssim \frac{\epsilon^2}{j} \leq \epsilon, \tag{56}$$

where the last inequality follows from the fact that $j \geq 1$ and $\epsilon < 1$.

**Step 2: upper bound on** $(II)$. To control $(II)$, we have that

$$(II)^2 = \sum_{k:k \neq j} (\lambda_{a,j} - \lambda_{a,k})^{-2}$$

$$\cdot \left[ \int \int \{\widehat{K}_a(s,t) - K_a(s,t)\}\{\widehat{\phi}_{a,j}(s) - \phi_{a,j}(s)\}\phi_{a,k}(t) \, ds \, dt \right]^2$$

$$\lesssim J_{\widetilde{n}}^{2(\alpha+1)} \sum_{k=1}^{\infty} \left[ \int \left\{ \int \{\widehat{K}_a(s,t) - K_a(s,t)\}\{\widehat{\phi}_{a,j}(s) - \phi_{a,j}(s)\} \, ds \right\} \phi_{a,k}(t) \, dt \right]^2$$

$$= J_{\widetilde{n}}^{2(\alpha+1)} \int \left[ \int \{\widehat{K}_a(s,t) - K_a(s,t)\}\{\widehat{\phi}_{a,j}(s) - \phi_{a,j}(s)\} \, ds \right]^2 dt$$

$$\leq J_{\widetilde{n}}^{2(\alpha+1)} \left[ \int \{\widehat{\phi}_{a,j}(s) - \phi_{a,j}(s)\}^2 \, ds \right] \left[ \int \int \{\widehat{K}_a(s,t) - K_a(s,t)\}^2 \, ds \, dt \right]$$

$$= J_{\widetilde{n}}^{2(\alpha+1)} \|\widehat{\phi}_{a,j} - \phi_{a,j}\|_{L^2}^2 \|\widehat{K}_a - K_a\|_{L^2}^2$$

$$\leq J_{\widetilde{n}}^{2(\alpha+1)} \frac{\epsilon^2}{j^2} \|\widehat{\phi}_{a,j} - \phi_{a,j}\|_{L^2}^2 \leq (1 - 2^{-1/2})^2 \|\widehat{\phi}_{a,j} - \phi_{a,j}\|_{L^2}^2,$$

where the first inequality follows from $\mathcal{E}_2$ in (51), the first equality follows from Parseval's identity, the second inequality follows from Cauchy–Schwartz inequality, the third inequality follows from $\mathcal{E}_1$ and the last inequality follows for all $\epsilon$ such that $0 < \epsilon \lesssim j J_{\widetilde{n}}^{-(\alpha+1)}$. Therefore, we have that

$$\mathbb{P}\left\{ (II) \geq (1 - 2^{-1/2}) \|\widehat{\phi}_{a,j} - \phi_{a,j}\|_{L^2} \right\} \lesssim \exp\left( - \frac{\widetilde{n}_a \epsilon^2}{j^2} \right). \tag{57}$$

**Step 3: Combine results together.** Substituting the results in (53), (56) and (57) into (50) and applying a union bound argument, it holds with probability at least $1 - \exp(-\widetilde{n}_a \epsilon^2 / j^2)$ that

$$\|\widehat{\phi}_{a,j} - \phi_{a,j}\|_{L^2} \leq \epsilon + \epsilon + (1 - 2^{-1/2}) \|\widehat{\phi}_{a,j} - \phi_{a,j}\|_{L^2},$$

which implies that $\|\widehat{\phi}_{a,j} - \phi_{a,j}\|_{L^2} \lesssim \epsilon$ and the lemma thus follows.

$\square$

### G.3 Eigenvalue

**Lemma 36.** *Under Assumptions 1 and 2a, for any small constant $\eta \in (0, 1/2)$, it holds with probability at least $1 - \eta$ that, for any $j \in [J]$ such that $J^{2\alpha+2} \log(\widetilde{n}/\eta) \lesssim \widetilde{n}_a$,*

$$|\widehat{\lambda}_{a,j} - \lambda_{a,j}| \lesssim \sqrt{\frac{j^{-2\alpha} \log(\widetilde{n}_a/\eta)}{\widetilde{n}_a}}.$$

*Proof.* By Lemma 46 and the triangle inequality, for any $j \in [J]$, it holds that

$$\left(1 - \|\widehat{\phi}_{a,j} - \phi_{a,j}\|_{L^2}\right)|\widehat{\lambda}_{a,j} - \lambda_{a,j}|$$

$$\leq \left| \int \int \{\widehat{K}_a(s,t) - K_a(s,t)\}\phi_{a,j}(s)\phi_{a,j}(t) \, ds \, dt \right|$$

$$+ \|\widehat{\phi}_{a,j} - \phi_{a,j}\|_{L^2} \left\| \int \{\widehat{K}_a(s,t) - K_a(s,t)\}\phi_{a,j}(s) \, \mathrm{d}s \right\|_{L^2}$$

$$\leq \left| \int \int \{\widehat{K}_a(s,t) - K_a(s,t)\}\phi_{a,j}(s)\phi_{a,j}(t) \, \mathrm{d}s \, \mathrm{d}t \right| + \|\widehat{\phi}_{a,j} - \phi_{a,j}\|_{L^2} \left\| \widehat{K}_a - K_a \right\|_{L^2},$$

where the last inequality follows from the fact that

$$\sup_{j\in[J]} \left\| \int \{\widehat{K}_a(s,t) - K_a(s,t)\}\phi_{a,j}(s) \, \mathrm{d}s \right\|_{L^2} \leq \left\| \widehat{K}_a - K_a \right\|_{L^2}.$$

Therefore, by Lemmas 34, 35 and 37 and a union bound argument, we have that with probability at least $1 - 3\eta$,

$$|\widehat{\lambda}_{a,j} - \lambda_{a,j}| \lesssim \sqrt{\frac{j^{-2\alpha} \log(\widetilde{n}_a/\eta)}{\widetilde{n}_a}} + \sqrt{\frac{j^2 \log^2(\widetilde{n}_a/\eta)}{\widetilde{n}_a^2}} \lesssim \sqrt{\frac{j^{-2\alpha} \log(\widetilde{n}_a/\eta)}{\widetilde{n}_a}}, \quad \text{for all } j \in [J],$$

(58)

where the last inequality follows since $J^{2\alpha+2} \log(\widetilde{n}_a/\eta) \lesssim \widetilde{n}_a$. $\qquad\square$

## G.4 Projection of difference between covariance function and its estimator

**Lemma 37.** *Assume Assumptions 1 and 2a hold. Then for $\ell, k \in \mathbb{N}_+$ and $0 < \epsilon \lesssim \sqrt{\widetilde{n}_{a,y}(\ell k)^{-\alpha}/\widetilde{n}_a}$, it holds that*

$$\mathbb{P}\left\{ \left| \int \int \{\widehat{K}_a(s,t) - K_a(s,t)\}\phi_{a,\ell}(s)\phi_{a,k}(t) \, \mathrm{d}s \, \mathrm{d}t \right| \geq \epsilon \right\} \lesssim \exp\left( -\frac{\widetilde{n}_a\epsilon^2}{(\ell k)^{-\alpha}} \right).$$

*Proof.* Note that

$$\widehat{K}_a(s,t) - K_a(s,t) = \frac{1}{\widetilde{n}_a} \sum_{y\in\{0,1\}} \sum_{i=1}^{n_{a,y}} \left\{ \widetilde{X}_{a,y}^i(s) - \widehat{\mu}_{a,y}(s) \right\}\left\{ \widetilde{X}_{a,y}^i(t) - \widehat{\mu}_{a,y}(t) \right\}$$

$$- \mathbb{E}\left\{ \left\{ \widetilde{X}_{a,y}^i(s) - \mu_{a,y}(s) \right\}\left\{ \widetilde{X}_{a,y}^i(t) - \mu_{a,y}(t) \right\} \right\}$$

We have that for any $k, \ell \in \mathbb{N}_+$,

$$\int \int \{\widehat{K}_a(s,t) - K_a(s,t)\}\phi_{a,\ell}(s)\phi_{a,k}(t) \, \mathrm{d}s \, \mathrm{d}t$$

$$\asymp \frac{1}{\widetilde{n}_a} \sum_{y\in\{0,1\}} \sum_{i=1}^{n_{a,y}} (\xi_{a,y,\ell}^i - \bar{\xi}_{a,y,\ell})(\xi_{a,y,k}^i - \bar{\xi}_{a,y,k}) - \lambda_{a,\ell}\mathbb{1}\{\ell = k\}$$

$$\asymp \sum_{y\in\{0,1\}} \frac{\widetilde{n}_{a,y}}{\widetilde{n}_a}\left\{ \frac{1}{\widetilde{n}_{a,y}} \sum_{i=1}^{n_{a,y}} \left( \xi_{a,y,\ell}^i \xi_{a,y,k}^i - \lambda_{a,\ell}\mathbb{1}\{\ell = k\} \right) \right\}$$

$$- \sum_{y\in\{0,1\}} \frac{\widetilde{n}_{a,y}}{\widetilde{n}_a}\left\{ \frac{1}{\widetilde{n}_{a,y}} \sum_{i=1}^{\widetilde{n}_{a,y}} \xi_{a,y,\ell}^i \right\}\left\{ \frac{1}{\widetilde{n}_{a,y}} \sum_{j=1}^{\widetilde{n}_{a,y}} \xi_{a,y,k}^j \right\},$$

(59)

where for $\ell \in \mathbb{N}_+$, $\xi_{a,y,\ell}^i = \int\{\widetilde{X}_{a,y}^i(s) - \mu_{a,y}(s)\}\phi_{a,\ell}(s) \, \mathrm{d}s \sim N(0, \lambda_{a,\ell})$, $\bar{\xi}_{a,y,\ell} = \widetilde{n}_{a,y}^{-1} \sum_{i=1}^{\widetilde{n}_{a,y}} \xi_{a,y,\ell}^i$. By standard properties of sub-Gaussian random variables (e.g. Lemma 2.7.7 in Vershynin, 2018) and Assumption 2a, we have that $\|\xi_{a,y,\ell}^i \xi_{a,y,k}^i\|_{\psi_1} \leq \sqrt{\lambda_{a,\ell}\lambda_{a,k}} \asymp (\ell k)^{-\alpha/2}$ for any $y \in \{0,1\}$ and $i \in [\widetilde{n}_{a,y}]$. Using the standard property of the covariance operator, it holds that

$$\mathbb{E}(\xi_{a,y,\ell}^i \xi_{a,y,k}^i) = \mathbb{E}\left[ \int \int \{\widetilde{X}_{a,y}^i(s) - \mu_{a,y}(s)\}\{\widetilde{X}_{a,y}^i(t) - \mu_{a,y}(t)\phi_{a,\ell}(s)\phi_{a,\ell}(t) \, \mathrm{d}s \, \mathrm{d}t \right]$$

$$= \int \int K_a(s,t)\phi_{a,\ell}(t) \, \mathrm{d}s \, \mathrm{d}t = \lambda_{a,\ell}\mathbb{1}\{\ell = k\}.$$

Hence, for $y \in \{0,1\}$ it holds from Bernsteins inequality (e.g. Theorem 2.8.1 in Vershynin, 2018), we have that for any $\delta_{1,y} > 0$,

$$\mathbb{P}\Big\{\Big|\frac{1}{\widetilde{n}_{a,y}}\sum_{i=1}^{n_{a,y}}\big(\xi_{a,y,\ell}^i\xi_{a,y,k}^i - \lambda_{a,\ell}\mathbb{1}\{\ell = k\}\big)\Big| \geq \delta_{1,y}\Big\} \lesssim \exp\Big\{-\Big(\frac{\widetilde{n}_{a,y}\delta_{1,y}^2}{(\ell k)^{-\alpha}} \wedge \frac{\widetilde{n}_{a,y}\delta_{1,y}}{(\ell k)^{-\alpha/2}}\Big)\Big\}.$$

Moreover, it holds from General Hoeffding inequality (e.g. Theorem 2.6.2 in Vershynin, 2018) that for any $\delta_{2,y}, \delta_{3,y} > 0$,

$$\mathbb{P}\Big\{\Big|\frac{1}{\widetilde{n}_{a,y}}\sum_{i=1}^{\widetilde{n}_{a,y}}\xi_{a,y,\ell}^i\Big| \geq \delta_{2,y}\Big\} \lesssim \exp\Big(-\frac{\widetilde{n}_{a,y}\delta_{2,y}^2}{\ell^{-\alpha}}\Big),$$

and

$$\mathbb{P}\Big\{\Big|\frac{1}{\widetilde{n}_{a,y}}\sum_{i=1}^{\widetilde{n}_{a,y}}\xi_{a,y,k}^i\Big| \geq \delta_{3,y}\Big\} \lesssim \exp\Big(-\frac{\widetilde{n}_{a,y}\delta_{3,y}^2}{k^{-\alpha}}\Big).$$

Pick

$$\delta_{1,y} = \epsilon\sqrt{\frac{\widetilde{n}_a}{\widetilde{n}_{a,y}}}, \quad \delta_{2,y} = \epsilon\sqrt{\frac{\widetilde{n}_a}{\widetilde{n}_{a,y}k^{-\alpha}}}, \quad \text{and} \quad \delta_{3,y} = \epsilon\sqrt{\frac{\widetilde{n}_a}{\widetilde{n}_{a,y}\ell^{-\alpha}}},$$

and by a union-bound argument, we have that

$$\mathbb{P}\Big\{\Big|\int\int\big\{\widehat{K}_a(s,t) - K_a(s,t)\big\}\phi_{a,\ell}(s)\phi_{a,k}(t) \, \mathrm{d}s \, \mathrm{d}t\Big| \leq \sum_{y\in\{0,1\}}\frac{\widetilde{n}_{a,y}}{\widetilde{n}_a}\big(\delta_{1,y} + \delta_{2,y}\delta_{3,y}\big)\Big\}$$

$$\gtrsim 1 - \exp\Big(-\frac{\widetilde{n}_a\epsilon^2}{(\ell k)^{-\alpha}}\Big).$$

Note that

$$\sum_{a,y\in\{0,1\}}\frac{\widetilde{n}_{a,y}}{\widetilde{n}_a}\big(\delta_{1,y} + \delta_{2,y}\delta_{3,y}\big) = \sum_{a,y\in\{0,1\}}\frac{\widetilde{n}_{a,y}}{\widetilde{n}_a}\Big\{\epsilon\sqrt{\frac{\widetilde{n}_a}{\widetilde{n}_{a,y}}} + \epsilon^2\frac{\widetilde{n}_a}{\widetilde{n}_{a,y}}\sqrt{\frac{1}{(\ell k)^{-\alpha}}}\Big\}$$

$$\lesssim \epsilon + \epsilon^{3/2}\sqrt{\frac{\epsilon}{(\ell k)^{-\alpha}}} \lesssim \epsilon,$$

where the last inequality follows from the fact that $\epsilon \lesssim \sqrt{\widetilde{n}_{a,y}(\ell k)^{-\alpha}/\widetilde{n}_a}$, hence

$$\sqrt{\frac{\epsilon}{(\ell k)^{-\alpha}}} \lesssim \sqrt{\frac{\widetilde{n}_{a,y}}{\widetilde{n}_a}} \lesssim 1$$

and the lemma follows. $\qquad\square$

**Lemma 38.** *Assume Assumptions 1 and 2a hold. For $j \in \mathbb{N}_+$ and $0 < \epsilon \lesssim j^{-\alpha/2}$, we have that*

$$\mathbb{P}\Big\{\Big\|\int\big\{K_a(s,t) - \widehat{K}_a(s,t)\big\}\phi_{a,j}(t) \, \mathrm{d}t\Big\|_{L^2} \geq \epsilon\Big\} \lesssim \exp\Big(-\frac{\widetilde{n}_a\epsilon^2}{j^{-\alpha}}\Big).$$

*Proof.* The proof follows from a similar and even simpler argument as the one used in **Step 1-1** in the proof of Lemma 35. We only include the difference in the following. With the same notation as the one used in the proof of Lemma 35, it holds that

$$\mathbb{E}\Big\{\Big\|\{\widetilde{X}_{a,y}^i(\cdot) - \mu_{a,y}(\cdot)\}\xi_{a,y,j}^i - \lambda_{a,j}\phi_{a,j}(\cdot)\Big\|_{L^2}^b\Big\}$$

$$= \mathbb{E}\Big[\Big\{\sum_{\ell=1}^{\infty}\big(\xi_{a,y,\ell}^i\xi_{a,y,j}^i - \lambda_{a,j}\mathbb{1}\{\ell = j\}\big)^2\Big\}^{\frac{b}{2}}\Big]$$

$$= \mathbb{E}\Big[\Big\{\sum_{\ell=1}^{\infty} \lambda_{a,j}\lambda_{a,\ell}\big(Z_{i\ell}Z_{ij} - \mathbb{1}\{\ell = j\}\big)^2\Big\}^{\frac{b}{2}}\Big]$$

$$\leq j^{-\frac{\alpha b}{2}}\Big(\sum_{\ell=1}^{\infty} \lambda_{a,\ell}\Big)^{\frac{b}{2}-1} \sum_{\ell=1}^{\infty} \mathbb{E}\Big\{\big(Z_{i\ell}Z_{ij} - \mathbb{1}\{\ell = j\}\big)^b\Big\}$$

$$\lesssim b!j^{-\alpha}(j^{-\frac{\alpha}{2}})^{b-2}.$$

Therefore, by picking $L_1 = j^{-\alpha}$ and $L_2 = j^{-\frac{\alpha}{2}}$ in Lemma 45, we have that

$$\mathbb{P}\Big\{\Big\|\int \big\{K_a(s,t) - \widehat{K}_a(s,t)\big\}\phi_{a,j}(t)\,\mathrm{d}t\Big\|_{L^2} \geq \epsilon\Big\} \lesssim \exp\Big(-\frac{\widetilde{n}_a\epsilon^2}{j^{-\alpha} + j^{-\frac{\alpha}{2}}\epsilon}\Big).$$

The lemma thus holds by picking $\epsilon$ satisfying $j^{-\alpha} \gtrsim j^{-\frac{\alpha}{2}}\epsilon$. $\qquad\square$

### G.5 Projection score

**Lemma 39.** *Under Assumptions 1 and 2a, we have with probability at least $1 - \eta$ that, for any $j \in [J]$ such that $J^{2\alpha+2}\log^2(J)\log(\widetilde{n}/\eta) \lesssim \widetilde{n}$,*

$$|\widehat{\theta}_{a,1,j} - \widehat{\theta}_{a,0,j} - (\theta_{a,1,j} - \theta_{a,0,j})| \lesssim \begin{cases} \sqrt{\frac{j^{2-2\beta}\log^2(j)\log(\widetilde{n}/\eta)}{\widetilde{n}}}, & \text{when } \frac{\alpha+1}{2} < \beta \leq \frac{\alpha+2}{2}, \\ \sqrt{\frac{j^{-\alpha}\log(\widetilde{n}/\eta)}{\widetilde{n}}}, & \text{when } \beta > \frac{\alpha+2}{2}. \end{cases}$$

*Proof.* Applying a union bound argument to the arguments in Lemmas 40, 41 and 42, we have with probability at least $1 - \eta$ that

$$|\widehat{\theta}_{a,1,j} - \widehat{\theta}_{a,0,j} - (\theta_{a,1,j} - \theta_{a,0,j})|$$

$$\leq \Big|\int \big\{\widehat{\mu}_{a,1}(t) - \mu_{a,1}(t) - \widehat{\mu}_{a,0}(t) + \mu_{a,0}(t)\big\}\phi_{a,j}(t)\,\mathrm{d}t\Big|$$

$$+ \Big|\int \big\{\widehat{\mu}_{a,1}(t) - \widehat{\mu}_{a,0}(t)\big\}\big\{\widehat{\phi}_{a,j}(t) - \phi_{a,j}(t)\big\}\,\mathrm{d}t\Big|$$

$$\leq \sum_{y \in \{0,1\}} \Big|\frac{1}{\widetilde{n}_{a,y}}\sum_{i=1}^{\widetilde{n}_{a,y}} \int \big\{\widetilde{X}_{a,y}^i(t) - \mu_{a,y}(t)\big\}\phi_{a,j}(t)\,\mathrm{d}t\Big|$$

$$+ \sum_{y \in \{0,1\}} \Big|\frac{1}{\widetilde{n}_{a,y}}\sum_{i=1}^{\widetilde{n}_{a,y}} \int \big\{\widetilde{X}_{a,y}^i(t) - \mu_{a,y}(t)\big\}\big\{\widehat{\phi}_{a,j}(t) - \phi_{a,j}(t)\big\}\,\mathrm{d}t\Big|$$

$$+ \Big|\int \big\{\mu_{a,1}(t) - \mu_{a,0}(t)\big\}\big\{\widehat{\phi}_{a,j}(t) - \phi_{a,j}(t)\big\}\,\mathrm{d}t\Big|$$

$$\lesssim \sqrt{\frac{j^{2-2\beta}\log(\widetilde{n}/\eta)}{\widetilde{n}}} + \sqrt{\frac{j^2\log(\widetilde{n}/\eta)\log(1/\eta)}{\widetilde{n}^2}}\big\{1 \vee j^{1+\alpha-\beta}\log(j)\big\}$$

$$+ \sqrt{\frac{j^{-\alpha}\log(\widetilde{n}/\eta)}{\widetilde{n}}}\big\{1 \vee j^{\frac{\alpha}{2}-\beta+1}\log(j)\big\}.$$

whenever $J^2\log^2(J)\log(\widetilde{n}/\eta) \lesssim \widetilde{n}$. Thus, the result follows. $\qquad\square$

**Lemma 40.** *Under Assumptions 1 and 2a, for any small $\epsilon > 0$, we have that*

$$\mathbb{P}\Big\{\Big|\frac{1}{\widetilde{n}_{a,y}}\sum_{i=1}^{\widetilde{n}_{a,y}} \int \big\{\widetilde{X}_{a,y}^i(t) - \mu_{a,y}(t)\big\}\phi_{a,j}(t)\,\mathrm{d}t\Big| \geq \epsilon\Big\} \lesssim \exp\Big(-\frac{\widetilde{n}_{a,y}\epsilon^2}{j^{-\alpha}}\Big).$$

*Proof.* By the standard property of Gaussian processes, we have that for each $i \in [\widetilde{n}_{a,y}]$,

$$\int \{\widetilde{X}_{a,y}^i(t) - \mu_{a,y}(t)\}\phi_{a,j}(t)\mathrm{d}t \overset{\text{i.i.d.}}{\sim} N(0, \lambda_{a,j}).$$

Therefore, the lemma follows by Assumption 2a and General Hoeffding inequality (e.g. Theorem 2.6.2 in Vershynin, 2018). $\qquad\square$

**Lemma 41.** *Under Assumptions 1 and 2a, for any small constant $\eta \in (0, 1/2)$, it holds with probability at least $1 - \eta$ that*

$$\left|\frac{1}{\widetilde{n}_{a,y}} \sum_{i=1}^{\widetilde{n}_{a,y}} \int \{\widetilde{X}_{a,y}^i(t) - \mu_{a,y}(t)\}\{\widehat{\phi}_{a,j}(t) - \phi_{a,j}(t)\}\,\mathrm{d}t\right| \lesssim \sqrt{\frac{j^{2-\alpha}\log^2(j)\log^2(\widetilde{n}/\eta)}{\widetilde{n}^2}},$$

*for any $j \in [J]$, $a, y \in \{0, 1\}$, with $J^{2\alpha+2}\log(1/\eta) \lesssim \widetilde{n}$.*

*Proof.* Consider the following events:

$$\mathcal{E}_1 = \left\{|\widehat{\lambda}_{a,j} - \lambda_{a,k}|^{-1} \leq \sqrt{2}|\lambda_{a,j} - \lambda_{a,k}|^{-1}, \quad \text{for } a \in \{0,1\}, \ k \in \mathbb{N}\backslash\{j\}, \ j \in [J]\right\},$$

$$\mathcal{E}_2 = \left\{\|\widehat{K}_a - K_a\|_{L^2} \lesssim \sqrt{\frac{\log(1/\eta)}{\widetilde{n}}}, \quad a \in \{0,1\}\right\},$$

$$\mathcal{E}_3 = \left\{\|\widehat{\phi}_{a,j} - \phi_{a,j}\|_{L^2} \lesssim \sqrt{\frac{j^2\log(\widetilde{n}/\eta)}{\widetilde{n}}}, \quad \text{for } j \in [J], \ a \in \{0,1\}\right\},$$

$$\mathcal{E}_4 = \left\{\left|\frac{1}{\widetilde{n}_{a,y}} \sum_{i=1}^{\widetilde{n}_{a,y}} \int \{\widetilde{X}_{a,y}^i(t) - \mu_{a,y}(t)\}\phi_{a,j}(t)\,\mathrm{d}t\right| \lesssim \sqrt{\frac{j^{-\alpha}\log(\widetilde{n}/\eta)}{\widetilde{n}}},\right.$$

$$\left. \text{for } j \in [J], \ a, y \in \{0,1\}\right\},$$

and

$$\mathcal{E}_5 = \left\{\widetilde{n}_{a,y} \asymp \widetilde{n}, \ \text{ for any } \ a, y \in \{0,1\}\right\}.$$

Applying a union bound argument, it holds from Lemmas 32, 34, 35 and 40 and a similar argument as the one leads to (51) that $\mathbb{P}(\mathcal{E}_1 \cap \mathcal{E}_2 \cap \mathcal{E}_3 \cap \mathcal{E}_4 \cap \mathcal{E}_5) \geq 1 - \eta/4$, for some small $\eta \in (0, 1/2)$. The rest of the proof is constructed conditioning on $\mathcal{E}_1 \cap \mathcal{E}_2 \cap \mathcal{E}_3 \cap \mathcal{E}_4 \cap \mathcal{E}_5$ happening.

Note that by Lemma 46, we have that

$$\frac{1}{\widetilde{n}_{a,y}} \sum_{i=1}^{\widetilde{n}_{a,y}} \int \{\widetilde{X}_{a,y}^i(t) - \mu_{a,y}(t)\}\{\widehat{\phi}_{a,j}(t) - \phi_{a,j}(t)\}\,\mathrm{d}t$$

$$= \frac{1}{\widetilde{n}_{a,y}} \sum_{i=1}^{\widetilde{n}_{a,y}} \int \{\widetilde{X}_{a,y}^i(t) - \mu_{a,y}(t)\}\phi_{a,j}(t)\,\mathrm{d}t \int \{\widehat{\phi}_{a,j}(s) - \phi_{a,j}(s)\}\phi_{a,j}(s)\,\mathrm{d}s$$

$$+ \sum_{k:k\neq j} (\widehat{\lambda}_{a,j} - \lambda_{a,k})^{-1} \left\{\frac{1}{\widetilde{n}_{a,y}} \sum_{i=1}^{\widetilde{n}_{a,y}} \int \{\widetilde{X}_{a,y}^i(t) - \mu_{a,y}(t)\}\phi_{a,k}(t)\,\mathrm{d}t\right\}$$

$$\cdot \int \int \{\widehat{K}_a(s, \ell) - K_a(s, \ell)\}\widehat{\phi}_{a,j}(s)\phi_{a,k}(\ell)\,\mathrm{d}s\,\mathrm{d}\ell$$

$$= \frac{1}{\widetilde{n}_{a,y}} \sum_{i=1}^{\widetilde{n}_{a,y}} \int \{\widetilde{X}_{a,y}^i(t) - \mu_{a,y}(t)\}\phi_{a,j}(t)\,\mathrm{d}t \int \{\widehat{\phi}_{a,j}(s) - \phi_{a,j}(s)\}\phi_{a,j}(s)\,\mathrm{d}s$$

$$+ \sum_{k:k \neq j} (\widehat{\lambda}_{a,j} - \lambda_{a,k})^{-1} \Big\{ \frac{1}{\widetilde{n}_{a,y}} \sum_{i=1}^{\widetilde{n}_{a,y}} \int \{ \widetilde{X}_{a,y}^i(t) - \mu_{a,y}(t) \} \phi_{a,k}(t) \, \mathrm{d}t \Big\}$$

$$\cdot \int \int \{ \widehat{K}_a(s, \ell) - K_a(s, \ell) \} \{ \widehat{\phi}_{a,j}(s) - \phi_{a,j}(s) \} \phi_{a,k}(\ell) \, \mathrm{d}s \, \mathrm{d}\ell$$

$$+ \sum_{k:k \neq j} (\widehat{\lambda}_{a,j} - \lambda_{a,k})^{-1} \Big\{ \frac{1}{\widetilde{n}_{a,y}} \sum_{i=1}^{\widetilde{n}_{a,y}} \int \{ \widetilde{X}_{a,y}^i(t) - \mu_{a,y}(t) \} \phi_{a,k}(t) \, \mathrm{d}t \Big\}$$

$$\cdot \int \int \{ \widehat{K}_a(s, \ell) - K_a(s, \ell) \} \phi_{a,j}(s) \phi_{a,k}(\ell) \, \mathrm{d}s \, \mathrm{d}\ell$$

$$= (I) + (II) + (III). \tag{60}$$

In the rest of the proof, we construct seperate large probability events which give control to upper bounds on terms $(I)$, $(II)$ and $(III)$.

**Step 1 : upper bound on** $(I)$**.** To control $(I)$, we have that with probability at least $1 - \eta/4$ that, for any $j \in [J]$,

$$|(I)| \leq \Big| \frac{1}{\widetilde{n}_{a,y}} \sum_{i=1}^{\widetilde{n}_{a,y}} \int \{ \widetilde{X}_{a,y}^i(t) - \mu_{a,y}(t) \} \phi_{a,j}(t) \, \mathrm{d}t \Big| \cdot \Big| \int \{ \widehat{\phi}_{a,j}(s) - \phi_{a,j}(s) \} \phi_{a,j}(s) \, \mathrm{d}s \Big|$$

$$\leq \Big| \frac{1}{\widetilde{n}_{a,y}} \sum_{i=1}^{\widetilde{n}_{a,y}} \int \{ \widetilde{X}_{a,y}^i(t) - \mu_{a,y}(t) \} \phi_{a,j}(t) \, \mathrm{d}t \Big|$$

$$\cdot \sqrt{\int \{ \widehat{\phi}_{a,j}(s) - \phi_{a,j}(s) \}^2 \, \mathrm{d}s} \sqrt{\int \phi_{a,j}^2(s) \, \mathrm{d}s}$$

$$\lesssim \sqrt{\frac{j^{-\alpha} \log(\widetilde{n}/\eta)}{\widetilde{n}}} \sqrt{\frac{j^2 \log(\widetilde{n}/\eta)}{\widetilde{n}}} \asymp \sqrt{\frac{j^{2-\alpha} \log^2(\widetilde{n}/\eta)}{\widetilde{n}^2}}, \tag{61}$$

where the second inequality follows from Cauchy–Schwarz inequality and the last inequality follows from $\mathcal{E}_3$ and $\mathcal{E}_4$.

**Step 2: upper bound on** $(II)$**.** To control $(II)$, we have that

$$|(II)| \leq \| \widehat{\phi}_{a,j}(s) - \phi_{a,j}(s) \|_{L_2} \| \widehat{K}_a - K_a \|_{L_2} \sum_{k:k \neq j} |\widehat{\lambda}_{a,j} - \lambda_{a,k}|^{-1} \Big| \frac{1}{\widetilde{n}_{a,y}}$$

$$\cdot \sum_{i=1}^{\widetilde{n}_{a,y}} \int \{ \widetilde{X}_{a,y}^i(t) - \mu_{a,y}(t) \} \phi_{a,k}(t) \, \mathrm{d}t \Big|$$

$$\lesssim \| \widehat{\phi}_{a,j}(s) - \phi_{a,j}(s) \|_{L_2} \| \widehat{K}_a - K_a \|_{L_2}$$

$$\cdot \sum_{k:k \neq j} |\lambda_{a,j} - \lambda_{a,k}|^{-1} \Big| \frac{1}{\widetilde{n}_{a,y}} \sum_{i=1}^{\widetilde{n}_{a,y}} \int \{ \widetilde{X}_{a,y}^i(t) - \mu_{a,y}(t) \} \phi_{a,k}(t) \, \mathrm{d}t \Big|, \tag{62}$$

where the first inequality follows from Cauchy–Schwarz inequality and the second inequality follows from $\mathcal{E}_1$. Note that by the standard property of Gaussian process, we have that

$$\frac{1}{\widetilde{n}_{a,y}} \sum_{i=1}^{\widetilde{n}_{a,y}} \int \{ \widetilde{X}_{a,y}^i(t) - \mu_{a,y}(t) \} \phi_k(t) \mathrm{d}t \overset{\text{i.i.d.}}{\sim} N\Big( 0, \frac{\lambda_{a,k}}{\widetilde{n}_{a,y}} \Big).$$

Therefore, it holds from standard properties of sub-Gaussian norms that

$$\Big\| \sum_{k:k \neq j} |\lambda_{a,j} - \lambda_{a,k}|^{-1} \Big| \frac{1}{\widetilde{n}_{a,y}} \sum_{i=1}^{\widetilde{n}_{a,y}} \int \{ \widetilde{X}_{a,y}^i(t) - \mu_{a,y}(t) \} \phi_{a,k}(t) \, \mathrm{d}t \Big| \Big\|_{\psi_2}$$

$$\leq \sum_{k:k\neq j} |\lambda_{a,j} - \lambda_{a,k}|^{-1} \Big\| \frac{1}{\widetilde{n}_{a,y}} \sum_{i=1}^{\widetilde{n}_{a,y}} \int \big\{ \widetilde{X}_{a,y}^i(t) - \mu_{a,y}(t) \big\} \phi_{a,k}(t) \; \mathrm{d}t \Big\|_{\psi_2}$$

$$\leq \sum_{k:k\neq j} |\lambda_{a,j} - \lambda_{a,k}|^{-1} \sqrt{\frac{\lambda_{a,k}}{\widetilde{n}_{a,y}}} \lesssim \sqrt{\frac{1}{\widetilde{n}_{a,y}}} \sum_{k:k\neq j} |\lambda_{a,j} - \lambda_{a,k}|^{-1} k^{-\alpha/2} \lesssim \sqrt{\frac{j^{2+\alpha} \log^2(j)}{\widetilde{n}_{a,y}}},$$

where the last inequality follows from Lemma 44. Thus, by standard properties of sub-Gaussian random variables (e.g. Proposition 2.5.2 in Vershynin, 2018), we have that for any $\delta_1 > 0$,

$$\mathbb{P}\Big\{ \sum_{k:k\neq j} |\lambda_{a,j} - \lambda_{a,k}|^{-1} \Big| \frac{1}{\widetilde{n}_{a,y}} \sum_{i=1}^{\widetilde{n}_{a,y}} \int \big\{ \widetilde{X}_{a,y}^i(t) - \mu_{a,y}(t) \big\} \phi_{a,k}(t) \; \mathrm{d}t \Big| \geq \delta_1 \Big\}$$

$$\lesssim \exp\Big( -\frac{\delta_1^2 \widetilde{n}_{a,y}}{j^{2+\alpha} \log^2(j)} \Big).$$

Pick $\delta_1 = \sqrt{j^{2+\alpha} \log^2(j) \log(\widetilde{n}/\eta)/\widetilde{n}_{a,y}} \asymp \sqrt{j^{2+\alpha} \log^2(j) \log(\widetilde{n}/\eta)/\widetilde{n}}$, we then have that with probability at least $1 - \eta/4$ that

$$\sum_{k:k\neq j} |\lambda_{a,j} - \lambda_{a,k}|^{-1} \Big| \frac{1}{\widetilde{n}_{a,y}} \sum_{i=1}^{\widetilde{n}_{a,y}} \int \big\{ \widetilde{X}_{a,y}^i(t) - \mu_{a,y}(t) \big\} \phi_{a,k}(t) \; \mathrm{d}t \Big|$$

$$\lesssim \sqrt{\frac{j^{2+\alpha} \log^2(j) \log(\widetilde{n}/\eta)}{\widetilde{n}}}. \tag{63}$$

Substituting (63) into (62), it holds from a union-bound argument that with probability at least $1 - \eta/4$ that

$$|(II)| \lesssim \sqrt{\frac{j^2 \log(\widetilde{n}/\eta)}{\widetilde{n}}} \sqrt{\frac{\log(1/\eta)}{\widetilde{n}}} \sqrt{\frac{j^{2+\alpha} \log^2(j) \log(\widetilde{n}/\eta)}{\widetilde{n}}}$$

$$\asymp \sqrt{\frac{j^{4+\alpha} \log^2(j) \log^2(\widetilde{n}/\eta) \log(1/\eta)}{\widetilde{n}^3}}, \tag{64}$$

for any $j \in [J]$.

**Step 3: Upper bound on** $(III)$**.** To control $(III)$, firstly, note that under $\mathcal{E}_1$, it holds that

$$|(III)| \lesssim \sum_{k:k\neq j} |\widehat{\lambda}_{a,j} - \lambda_{a,k}|^{-1} \Big| \frac{1}{\widetilde{n}_{a,y}} \sum_{i=1}^{\widetilde{n}_{a,y}} \int \big\{ \widetilde{X}_{a,y}^i(t) - \mu_{a,y}(t) \big\} \phi_{a,k}(t) \; \mathrm{d}t \Big|$$

$$\cdot \Big| \int \int \big\{ \widehat{K}_a(s,\ell) - K_a(s,\ell) \big\} \phi_{a,j}(s) \phi_{a,k}(\ell) \; \mathrm{d}s \; \mathrm{d}\ell \Big|$$

$$\lesssim \sum_{k:k\neq j} |\lambda_{a,j} - \lambda_{a,k}|^{-1} \Big| \frac{1}{\widetilde{n}_{a,y}} \sum_{i=1}^{\widetilde{n}_{a,y}} \int \big\{ \widetilde{X}_{a,y}^i(t) - \mu_{a,y}(t) \big\} \phi_{a,k}(t) \; \mathrm{d}t \Big|$$

$$\cdot \Big| \int \int \big\{ \widehat{K}_a(s,\ell) - K_a(s,\ell) \big\} \phi_{a,j}(s) \phi_{a,k}(\ell) \; \mathrm{d}s \; \mathrm{d}\ell \Big|.$$

Next, we control its $\psi_1$-orcliz norm and we have that

$$\|(III)\|_{\psi_1} \leq \sum_{k:k\neq j} (\lambda_{a,j} - \lambda_{a,k})^{-1} \Big\| \frac{1}{\widetilde{n}_{a,y}} \sum_{i=1}^{\widetilde{n}_{a,y}} \int \big\{ \widetilde{X}_{a,y}^i(t) - \mu_{a,y}(t) \big\} \phi_{a,k}(t) \; \mathrm{d}t \Big\|_{\psi_2}$$

$$\cdot \left\| \int \int \{\widehat{K}_a(s,\ell) - K_a(s,\ell)\} \phi_{a,j}(s) \phi_{a,k}(\ell) \, ds \, d\ell \right\|_{\psi_2}$$

$$\lesssim \sum_{k:k \neq j} (\lambda_{a,j} - \lambda_{a,k})^{-1} \sqrt{\frac{k^{-\alpha}}{\widetilde{n}}} \sqrt{\frac{j^{-\alpha} k^{-\alpha}}{\widetilde{n}}} \lesssim \frac{j^{-\frac{\alpha}{2}}}{\widetilde{n}} \sum_{k:k \neq j} |\lambda_{a,j} - \lambda_{a,k}|^{-1} k^{-\alpha}$$

$$\lesssim \frac{j^{1-\frac{\alpha}{2}} \log(j)}{\widetilde{n}},$$

where the first inequality follows from the triangle inequality and Lemma 2.7.7 in Vershynin (2018), the second inequality follows from Lemmas 37 and 40 and the last inequality follows from Lemma 44. Consequently, by standard properties of sub-Exponential random variables (e.g. Proposition 2.7.1 in Vershynin, 2018), it holds for any $\delta_2 > 0$ that

$$\mathbb{P}\Big\{ |(III)| \geq \delta_2 \Big\} \lesssim \exp\Big\{ -\frac{\delta_2 \widetilde{n}}{j^{1-\frac{\alpha}{2}} \log(j)} \Big\}$$

By a union bound argument and by picking $\delta_2 = j^{1-\alpha/2} \log(j) \log(\widetilde{n}/\eta)/\widetilde{n}$, we have that with probability at least $1 - \eta/4$ that

$$|(III)| \lesssim \frac{j^{1-\frac{\alpha}{2}} \log(j) \log(\widetilde{n}/\eta)}{\widetilde{n}}, \tag{65}$$

for any $j \in [J]$.

**Step 4: Combine results.** Substituting the results in (61), (64) and (65) into (60) and applying a union bound argument, we have with probability at least $1 - \eta$ that

$$\frac{1}{\widetilde{n}_{a,y}} \sum_{i=1}^{\widetilde{n}_{a,y}} \int \{\widetilde{X}_{a,y}^i(t) - \mu_{a,y}(t)\} \{\widehat{\phi}_{a,j}(t) - \phi_{a,j}(t)\} \, dt$$

$$\lesssim \sqrt{\frac{j^{2-\alpha} \log^2(\widetilde{n}/\eta)}{\widetilde{n}^2}} + \sqrt{\frac{j^{4+\alpha} \log^2(j) \log^3(\widetilde{n}/\eta)}{\widetilde{n}^3}} + \frac{j^{1-\frac{\alpha}{2}} \log(j) \log(\widetilde{n}/\eta)}{\widetilde{n}}$$

$$\lesssim \sqrt{\frac{j^{2-\alpha} \log^2(j) \log^2(\widetilde{n}/\eta)}{\widetilde{n}^2}},$$

for any $j \in [J]$ such that $J^{2\alpha+2} \log(1/\eta) \lesssim \widetilde{n}$.

$$\square$$

**Lemma 42.** *Under Assumptions 1 and 2a, for any small constant $\eta \in (0, 1/2)$, it holds with probability at least $1 - \eta$ that*

$$\Big| \int \{\mu_{a,1}(t) - \mu_{a,0}(t)\} \{\widehat{\phi}_{a,j}(t) - \phi_{a,j}(t)\} \, dt \Big|$$

$$\lesssim \sqrt{\frac{j^{2-2\beta} \log(\widetilde{n}/\eta)}{\widetilde{n}}} + \sqrt{\frac{j^2 \log(\widetilde{n}/\eta) \log(1/\eta)}{\widetilde{n}^2}} \{1 + j^{1+\alpha-\beta} \log(j)\}$$

$$+ \sqrt{\frac{j^{-\alpha} \log(\widetilde{n}/\eta)}{\widetilde{n}}} \{1 + j^{\frac{\alpha}{2}-\beta+1} \log(j)\}$$

*for any $j \in [J]$ with $J^{2\alpha+2} \lesssim_{\log} \widetilde{n}$.*

*Proof.* The proof follows using a similar and simpler argument as the one used in the proof of Lemma 41. We only include the difference here.

By Lemma 46, we have that

$$\int \{\mu_{a,1}(t) - \mu_{a,0}(t)\} \{\widehat{\phi}_{a,j}(t) - \phi_{a,j}(t)\} \, dt$$

$$= \int \{\mu_{a,1}(t) - \mu_{a,0}(t)\}\phi_{a,j}(t) \, \mathrm{d}t \int \{\widehat{\phi}_{a,j}(s) - \phi_{a,j}(s)\}\phi_{a,j}(s) \, \mathrm{d}s$$

$$+ \sum_{k:k\neq j} (\widehat{\lambda}_{a,j} - \lambda_{a,k})^{-1} \int \{\mu_{a,1}(t) - \mu_{a,0}(t)\}\phi_{a,k}(t) \, \mathrm{d}t$$

$$\cdot \int \int \{\widehat{K}_a(s,\ell) - K_a(s,\ell)\}\widehat{\phi}_{a,j}(s)\phi_{a,k}(\ell) \, \mathrm{d}s \, \mathrm{d}\ell$$

$$= \int \{\mu_{a,1}(t) - \mu_{a,0}(t)\}\phi_{a,j}(t) \, \mathrm{d}t \int \{\widehat{\phi}_{a,j}(s) - \phi_{a,j}(s)\}\phi_{a,j}(s) \, \mathrm{d}s$$

$$+ \sum_{k:k\neq j} (\widehat{\lambda}_{a,j} - \lambda_{a,k})^{-1} \int \{\mu_{a,1}(t) - \mu_{a,0}(t)\}\phi_{a,k}(t) \, \mathrm{d}t$$

$$\cdot \int \int \{\widehat{K}_a(s,\ell) - K_a(s,\ell)\}\{\widehat{\phi}_{a,j}(s) - \phi_{a,j}(s)\}\phi_{a,k}(\ell) \, \mathrm{d}s \, \mathrm{d}\ell$$

$$+ \sum_{k:k\neq j} (\widehat{\lambda}_{a,j} - \lambda_{a,k})^{-1} \int \{\mu_{a,1}(t) - \mu_{a,0}(t)\}\phi_{a,k}(t) \, \mathrm{d}t$$

$$\cdot \int \int \{\widehat{K}_a(s,\ell) - K_a(s,\ell)\}\phi_{a,j}(s)\phi_{a,k}(\ell) \, \mathrm{d}s \, \mathrm{d}\ell$$

$$= (I) + (II) + (III). \tag{66}$$

**Step 1: Upper bound on** $(I)$ Using a similar argument as the one used in **Step 1** in the proof of Lemma 41, we have with probability at least $1 - \eta/3$ that

$$|(I)| \leq \left| \int \{\mu_{a,1}(t) - \mu_{a,0}(t)\}\phi_{a,j}(t) \, \mathrm{d}t \right| \cdot \left| \int \{\widehat{\phi}_{a,j}(s) - \phi_{a,j}(s)\}\phi_{a,j}(s) \, \mathrm{d}s \right|$$

$$\leq \left| \int \{\mu_{a,1}(t) - \mu_{a,0}(t)\}\phi_{a,j}(t) \, \mathrm{d}t \right| \sqrt{\int \{\widehat{\phi}_{a,j}(s) - \phi_{a,j}(s)\}^2 \, \mathrm{d}s} \sqrt{\int \phi_{a,j}^2(s) \, \mathrm{d}s}$$

$$\lesssim j^{-\beta} \sqrt{\frac{j^2 \log(\widetilde{n}/\eta)}{\widetilde{n}}} \asymp \sqrt{\frac{j^{2-2\beta} \log(\widetilde{n}/\eta)}{\widetilde{n}}}, \tag{67}$$

where the third inequality follows from Assumption 2c.

**Step 2: Upper bound on** $(II)$ To control $(II)$, then using a similar argument as the one used in **Step 2** in the proof of Lemma 41, we have with probability at least $1 - \eta/3$ that

$$|(II)| \lesssim \|\widehat{\phi}_{a,j}(s) - \phi_{a,j}(s)\|_{L_2} \|\widehat{K}_a - K_a\|_{L_2}$$

$$\cdot \sum_{k:k\neq j} |\lambda_{a,j} - \lambda_{a,k}|^{-1} \left| \int \{\mu_{a,1}(t) - \mu_{a,0}(t)\}\phi_{a,k}(t) \, \mathrm{d}t \right|$$

$$\lesssim \sqrt{\frac{j^2 \log(\widetilde{n}/\eta) \log(1/\eta)}{\widetilde{n}^2}} \sum_{k:k\neq j} |\lambda_{a,j} - \lambda_{a,k}|^{-1} k^{-\beta}$$

$$\lesssim \sqrt{\frac{j^2 \log(\widetilde{n}/\eta) \log(1/\eta)}{\widetilde{n}^2}} \{1 + j^{1+\alpha-\beta} \log(j)\}. \tag{68}$$

**Step 3: Upper bound on** $(III)$ To control $(III)$, then using a similar argument as the one used in **Step 3** in the proof of Lemma 41, we have with probability at least $1 - \eta/3$ that

$$|(III)| \lesssim \sum_{k:k\neq j} |\lambda_{a,j} - \lambda_{a,k}|^{-1} \left| \int \{\mu_{a,1}(t) - \mu_{a,0}(t)\}\phi_{a,k}(t) \, \mathrm{d}t \right|$$

$$\cdot \Big| \int \int \big\{ \widehat{K}_a(s, \ell) - K_a(s, \ell) \big\} \phi_{a,j}(s) \phi_{a,k}(\ell) \, \mathrm{d}s \, \mathrm{d}\ell \Big|$$

$$\lesssim \sqrt{\frac{j^{-\alpha} \log(\widetilde{n}/\eta)}{\widetilde{n}}} \big\{ 1 + j^{\frac{\alpha}{2} - \beta + 1} \log(j) \big\}. \tag{69}$$

**Step 4: Combine results.** Substituting the results in (67), (68) and (69) into (66) and applying a union bound argument, it holds with probability at least $1 - \eta$ that

$$\Big| \int \big\{ \mu_{a,1}(t) - \mu_{a,0}(t) \big\} \big\{ \widehat{\phi}_{a,j}(t) - \phi_{a,j}(t) \big\} \, \mathrm{d}t \Big|$$

$$\lesssim \sqrt{\frac{j^{2-2\beta} \log(\widetilde{n}/\eta)}{\widetilde{n}}} + \sqrt{\frac{j^2 \log(\widetilde{n}/\eta) \log(1/\eta)}{\widetilde{n}^2}} \big\{ 1 + j^{1+\alpha-\beta} \log(j) \big\}$$

$$+ \sqrt{\frac{j^{-\alpha} \log(\widetilde{n}/\eta)}{\widetilde{n}}} \big\{ 1 + j^{\frac{\alpha}{2} - \beta + 1} \log(j) \big\}.$$

$\square$

# H  Technical lemmas

For completeness, we provide all technical lemmas in this section.

**Lemma 43** (Generalized Neyman–Pearson lemma, e.g. Lemma 3.1 in Zeng et al., 2024a). *Let $\phi_0, \phi_1, \ldots, \phi_m$ be $m + 1$ real-valued functions defined on a Euclidean space $\mathcal{X}$. Assume they are $\nu$-integrable for a $\sigma$-finite measure $\nu$. Let $f^* \in \mathcal{F}$ be any function of the form*

$$f^*(x) = \begin{cases} 1, & \phi_0(x) > \sum_{i=1}^m c_i \phi_i(x), \\ \tau(x), & \phi_0(x) = \sum_{i=1}^m c_i \phi_i(x), \\ 0, & \phi_0(x) < \sum_{i=1}^m c_i \phi_i(x), \end{cases}$$

*where $0 \leq \tau(x) \leq 1$ for all $x \in \mathcal{X}$. For given constants $t_1, \ldots, t_m \in \mathbb{R}$, let $\mathcal{F}_{\leq}$ be the class of measurable functions $f : \mathcal{X} \to \mathbb{R}$ satisfying*

$$\int_{\mathcal{X}} f \phi_i \, d\nu \leq t_i, \quad i \in \{1, 2, \ldots, m\}, \tag{70}$$

*and let $\mathcal{F}_=$ be the set of functions in $\mathcal{F}_{\leq}$ satisfying (70) with all inequalities replaced by equalities.*

*(1) If $f^* \in \mathcal{F}_=$, then*

$$f^* \in \arg \max_{f \in \mathcal{F}_=} \int_{\mathcal{X}} f \phi_0 \, d\nu.$$

*Moreover, if $\nu(\{x : \phi_0(x) = \sum_{i=1}^m c_i \phi_i(x)\}) = 0$, for all $f' \in \arg \max_{f \in \mathcal{F}_=} \int_{\mathcal{X}} f \phi_0 \, d\nu$, $f' = f^*$ almost everywhere with respect to $\nu$.*

*(2) Moreover, if $c_i \geq 0$ for all $i = 1, \ldots, m$, then*

$$f^* \in \arg \max_{f \in \mathcal{F}_{\leq}} \int_{\mathcal{X}} f \phi_0 \, d\nu.$$

*Moreover, if $\nu(\{x : \phi_0(x) = \sum_{i=1}^m c_i \phi_i(x)\}) = 0$, for all $f' \in \arg \max_{f \in \mathcal{F}_{\leq}} \int_{\mathcal{X}} f \phi_0 \, d\nu$, we have $f'(x) = f^*(x)$ almost everywhere with respect to $\nu$.*

**Lemma 44** (Lemma 7 in Dou et al., 2012). *Under Assumption 2a, for each $r \geq 1$, there exists a constant $C_r > 0$ depending on $r$ such that*

$$\sum_{k \in \mathbb{N}} \mathbb{1}_{\{j \neq k\}} \frac{k^{-\gamma}}{|\lambda_{a,j} - \lambda_{a,k}|^r} \leq \begin{cases} C_r(1 + j^{r(1+\alpha)-\gamma}), & \text{if } r > 1, \\ C_1(1 + j^{1+\alpha-\gamma} \log j), & \text{if } r = 1, \end{cases}$$

*for all $a \in \{0, 1\}$ and $j \in \mathbb{N}_+$.*

**Lemma 45** (Theorem 2.5 in Bosq, 2000)**.** *Let $\{X_i\}_{i=1}^n$ be independent random variables in a separable Hilbert space with norm $\|\cdot\|$. If $\mathbb{E}[X_i] = 0$ for all $i \in [n]$ and*

$$\sum_{i=1}^n \mathbb{E}(\|X_i\|^b) \le \frac{b!}{2} n L_1 L_2^{b-2}, \quad for\ b = 2, 3, \dots,$$

*with $L_1, L_2 > 0$ being two constants, then for all $\epsilon > 0$, it holds that*

$$\mathbb{P}\Big(\Big\|\frac{1}{n}\sum_{i=1}^n X_i\Big\| \ge \epsilon\Big) \le 2\exp\Big(-\frac{n\epsilon^2}{2L_1 + 2L_2\epsilon}\Big).$$

**Lemma 46** (Lemma 5.1 in Hall and Horowitz, 2007)**.** *If we can write*

$$K(s,t) = \sum_{j=1}^\infty \lambda_j \phi_j(s)\phi_j(t) \quad and \quad \widehat{K}(s,t) = \sum_{j=1}^\infty \widehat{\lambda}_j \widehat{\phi}_j(s)\widehat{\phi}_j(t),$$

*then it holds for any $j \in \mathbb{N}_+$ that*

$$\Big|\widehat{\lambda}_j - \lambda_j - \int\int \big\{\widehat{K}(s,t) - K(s,t)\big\}\phi_j(s)\phi_j(t)\ \mathrm{d}s\ \mathrm{d}t\Big|$$

$$\le \|\widehat{\phi}_j - \phi_j\|_{L^2}\Big(|\widehat{\lambda}_j - \lambda_j| + \Big\|\int\big\{\widehat{K}(s,t) - K(s,t)\big\}\phi_j(s)\ \mathrm{d}s\Big\|_{L^2}\Big).$$

*Moreover, if $\inf_{k \ne j}|\widehat{\lambda}_j - \lambda_k| > 0$, it holds for $\ell \in [0, 1]$ that*

$$\widehat{\phi}_j(\ell) - \phi_j(\ell) = \phi_j(\ell)\int\big\{\widehat{\phi}_j(s) - \phi_j(s)\big\}\phi_j(s)\ \mathrm{d}s$$

$$+ \sum_{k:k \ne j}\phi_k(\ell)(\widehat{\lambda}_j - \lambda_k)^{-1}\int\int\big\{\widehat{K}(s,t) - K(s,t)\big\}\widehat{\phi}_j(s)\phi_k(t)\ \mathrm{d}s\ \mathrm{d}t.$$

**Lemma 47** (Wong et al., 2020, Sub-Weibull properties)**.** *Let $X$ be a random variable. Then the following statements are equivalent for every $\alpha > 0$. The constants $C_1, C_2, C_3 > 0$ differ at most by a constant depending only on $\alpha$.*

1. *The tail of $X$ satisfies*

$$\mathbb{P}\big\{|X| > t\big\} \le 2\exp\big\{-(t/C_1)^\alpha\big\}, for\ all\ \ t \ge 0.$$

2. *The moments of $X$ satisfy*

$$\|X\|_p := \big(\mathbb{E}\big[|X|^p\big]\big)^{1/p} \le C_2 p^{1/\alpha}, for\ all\ \ p \ge 1 \wedge \alpha.$$

3. *The moment generating function of $|X|^\alpha$ is finite at some point; namely*

$$\mathbb{E}\big[\exp(|X|/C_3)^\alpha\big] \le 2.$$

*We further call a random variable $X$ which satisfies any of the properties above a sub-Weibull random variable with parameter $\alpha$.*

