# OpenReview forum: "Fairness-aware Bayes Optimal Functional Classification"
_NeurIPS.cc/2025/Conference — NeurIPS 2025 poster_

### Official Review · Reviewer_V7nb · 2025-06-02

**Clarity:** 1
**Significance:** 2
**Originality:** 3
**Rating:** 4
**Confidence:** 2

**Summary:**

This paper introduces a novel framework for fairness-aware functional classification, a rapidly growing area within machine learning that addresses disparities across subpopulations. The authors propose a unified approach to classify functional data while controlling the disparity level below a pre-specified threshold, overcoming challenges like the absence of density ratios and intractable posterior probabilities in infinite-dimensional functional spaces. They also develop a post-processing algorithm called Fair Functional Linear Discriminant Analysis (Fair-FLDA), designed for homoscedastic Gaussian processes, which achieves fairness through group-wise thresholding. The paper provides theoretical guarantees for both fairness and excess risk control for Fair-FLDA, including excess risk control for standard FLDA as a byproduct. Some numerical experiments on synthetic and real datasets validate the practicality of their proposed algorithm.

**Questions:**

How does the fairness problem specifically manifest within functional classification, and what motivated this particular area of investigation?

What is the justification for assuming Gaussian processes for non-sensitive features in the proposed algorithm, and what literature supports this choice in the functional fairness setting?

Will the numerical experiments using less common datasets have a broad impact on the fairness community, and what are the authors' plans to address this?

**Ethical Concerns:**

["NO or VERY MINOR ethics concerns only"]

**Final Justification:**

I'm really not quite sure how much theoretical contribution there is (since theory takes up a very large proportion in the paper, it's necessary to consider whether theoretical innovation is really important in this article). Due to the author's detailed explanation and supplementary experiments, I tend to increase my score.

**Quality:**

3

**Strengths And Weaknesses:**

### Strengths

The theoretical analysis here is incredibly robust, spanning over 50 pages of proofs – I truly appreciate the authors' dedication to that! It's clear they've put a lot of effort into deriving numerous theoretical results, even tackling fairness guarantees. This feels like a significant theoretical step forward, as, to my knowledge, not many papers in this field offer theoretical guarantees for fairness-aware algorithms.

---

### Weaknesses

First off, I have to point out that this paper leans heavily on theory (with proofs taking up pages 18 to 70 in the appendix!). Compared to all that intricate theory, the fairness background feels relatively straightforward. This makes me wonder why "Social and economic aspects of machine learning" was chosen as the Primary Area instead of "learning theory," which seems like a much better fit for the paper's core.

Setting aside the scope for "Social and economic aspects of machine learning," I also found the paper's motivation a bit unclear and, at times, a little far-fetched.

* For example, the logical connection between the second and third paragraphs in the introduction is weak. Paragraph 2 dives into functional classification, but then paragraph 3 jumps straight to the fairness issues of existing algorithms without explicitly linking them to "functional classification." Why investigate fairness specifically within functional classification? That connection needs to be clearer.
* Another point: the proposed post-processing algorithm assumes non-sensitive features are Gaussian processes, but there's no explanation for this choice. What's the rationale behind it? Is there existing literature that supports using Gaussian processes in this specific context?
* Finally, the numerical experiments feel a bit limited, using datasets that aren't widely established in prior literature. I'm not entirely convinced that the setup in this paper will have a broad impact on the fairness community.

**I want to emphasize that, if I've misunderstood anything, please correct me – I'm happy to adjust my rating accordingly! Thanks!**

---

Overall, this strikes me as more of a pure theoretical paper (specifically, a learning theory paper, rather than one focused on "Social and economic aspects of machine learning"), analyzing scenarios somewhat related to fairness. It's a bit outside my usual area, and I'm not familiar with some of the theoretical tools, so I can't really judge the novelty of the theoretical contributions. However, given that "Social and economic aspects of machine learning" is the chosen primary area, I think these concerns need to be properly addressed for the paper to truly resonate with the fairness community.

---

> ### Author Rebuttal · Authors · 2025-07-30
>
> We thank the reviewer for acknowledging our theoretical contributions and recognising the importance of fairness in functional classification. Below, we address the weakness (W) point by point and further elaborate on the motivation behind our work. We respectfully hope the reviewer could consider raising the score in light of the additional motivation provided, comparisons to adapted methods, and clarifications of our model assumptions.
>
> **Practical motivation**
>
> In our paper, we focus on developing a novel and practically effective methodology for fairness-aware classification in functional data settings. This is an important and underexplored problem with significant real-world relevance. To the best of our knowledge, our proposed framework, Fair-FLDA, is the first to explicitly address fairness in the context of functional classification. Functional data pose unique challenges due to their infinite-dimensional nature, which makes it difficult or ineffective to directly apply fairness techniques developed for multivariate data.
>
> Our motivation is primarily practical: in many applications involving functional data, such as biometric signals, longitudinal health records, or time-varying sensor data, ensuring fair and accurate classification is crucial. As demonstrated in Section 4, applying standard classification algorithms to such data can lead to substantial fairness violations (see the second and third columns in the right panel of Figure 2). In this response, we also examine a simple ``dimension reduction + standard fair classification" strategy: applying dimension reduction followed by fair multivariate classification. Additional baseline results using this approach are included in Tables 1 and 2 at the end of this response, along with detailed comparisons between Fair-FLDA and three post-processing and one pre-processing fairness methods originally designed for multivariate data. Across all experiments, Fair-FLDA consistently achieves lower misclassification rates while effectively mitigating disparity, highlighting its practical merits. In the revised version, we will provide a clearer explanation of the methodological and practical motivations that guide our study.
>
> **Theoretical contribution**
>
> While the main thrust of our contribution lies in methodology and empirical performance, we include theoretical analysis to support and validate the proposed approach. This analysis serves to reinforce the practical methodology rather than being the central focus of the paper. We appreciate your recognition of this theoretical component. Notably, theoretical guarantees for functional classification remain largely open in general settings, particularly when the underlying eigenspace is unknown. Our results address this challenging scenario while incorporating fairness constraints.
>
> **Writing for the second and third paragraphs (W1)**
>
> We appreciate your suggestions on our writing in the introduction. In the revised version, we will rewrite these two paragraphs to improve the logical flow and strengthen their connection. Specifically, in the third paragraph, we will refer to the real data example in Section 4 to demonstrate that existing classifiers exhibit substantial unfairness, thereby highlighting the practical importance of addressing fairness in functional data classification. We will also emphasise the lack of existing work in the literature for fair functional data classification to further motivate our contribution.
>
> **Assumptions on Gaussianity (W2)**
>
> Thank you for your question regarding our model assumptions. The classification problem for Gaussian processes has been extensively studied in the literature (e.g., [1], [2], [3]). In functional data analysis, it is well known that linear methods can often achieve optimal performance, largely due to the infinite dimensionality of the functional space (e.g., [1]). In some scenarios, linear classifiers can even lead to perfect classification (see Remark 2). When the data is generated from a Gaussian process, this optimality can be further understood through its connection to the structure of reproducing kernel Hilbert spaces (RKHS), as discussed in [2] and [3].
>
> In our paper, the Gaussianity assumption offers us a theoretically tractable foundation for studying fairness in functional classification. Under this assumption, explicit expressions for the Radon–Nikodym derivative can be derived, which in turn facilitate the construction of the plug-in classifier, Fair-FLDA. To demonstrate the robustness of our method, we also include simulation results under non-Gaussian models in Appendix A.3, highlighting its practical performance beyond the Gaussian setting.
>
> As our work is, to the best of our knowledge, the first to study fairness in functional data classification, we consider the Gaussian process model a natural and principled starting point. It provides a clean and interpretable platform for exploring fairness constraints in infinite-dimensional spaces. In future work, we plan to extend our methodology to more general, non-Gaussian settings.
>
> **Additional empirical evaluation (W3)**
>
> Our primary goal in this work is to introduce the first principled framework for fair functional classification with theoretical guarantees, providing the fairness community with a trustworthy algorithm.  Our theoretical contributions are numerically justified through both synthetic and real datasets, along with additional experiments done in the rebuttal stage.
>
> Since no prior work specifically targets fair functional classification, there are no widely established benchmark datasets in this context. We therefore use the publicly available dataset NHANES, which is frequently used in the functional data literature [4,5], to demonstrate the practical significance of our fair functional classifier. It is important to note that our method is broadly applicable to a wide range of functional data classification problems where fairness is a concern. For example, the Siena Scalp EEG dataset in the PhysioNet database contains EEG recordings from male and female subjects and can serve as a natural application of our fair functional classifier, where the response variable can be defined as the occurrence of a seizure.
>
> Moreover, we have included more comparisons with other baseline methods to further demonstrate the superiority and practical relevance of our method for fair functional classification. We have incorporated additional baselines following a ``dimension reduction + standard fair classification" strategy. Specifically, we first apply functional principal component analysis (FPCA) to extract features, and then employ fair classification methods designed for multivariate data. These include three post-processing methods FPIR [6], PPF [7] and PPOT [8], and one pre-processing approach FUDS [6].
>
> Results in Tables 1 and 2 show that our proposed fair classifier, Fair-FLDA, consistently achieves the lowest classification errors while effectively controlling disparity under the pre-specified levels. In contrast, the other four baseline methods exhibit higher classification errors. In particular, PPF shows poor disparity control when $\delta=0$, and FUDS fails to adequately control disparity on the NHANES dataset. These extensive numerical results highlight the superiority and practical necessity of our method for fair functional classification.
>
> Overall, the proposed fair functional classifier, equipped with theoretical guarantees, fills an important gap in the literature on fair functional classification and contributes to advancing research in the fairness community. We hope these clarifications adequately address your concerns.
>
> Table 1: Median classification error and DD over 500 runs under Gaussian with mean difference $\beta=1.5$ and sample size $n=1000$.
>
> |$\delta$|0.00||0.05||0.10||0.15||0.20||
> |--|--|--|--|--|--|--|--|--|--|--|
> | |Err|$U_{DD,50}$|Err|$U_{DD,50}$|Err|$U_{DD,50}$|Err|$U_{DD,50}$|Err|$U_{DD,50}$|
> |Fair-FLDA|0.234|0.022|0.227|0.048|0.219|0.098|0.213|0.149|0.208|0.197|
> |FPIR|0.276|0.024|0.269|0.039|0.264|0.080|0.258|0.122|0.253|0.168|
> |PPF|0.240|0.323|0.269|0.039|0.264|0.080|0.258|0.121|0.253|0.167|
> |PPOT|0.275|0.023|0.269|0.040|0.264|0.080|0.258|0.122|0.253|0.168|
> |FUDS|0.276|0.030|0.269|0.040|0.263|0.087|0.256|0.142|0.251|0.195|
>
> Table 2: Median classification error and DD over 500 runs under NHANES.
>
> |$\delta$|0.00||0.05||0.10||0.15||0.20||
> |--|--|--|--|--|--|--|--|--|--|--|
> | |Err|$U_{DD,50}$|Err|$U_{DD,50}$|Err|$U_{DD,50}$|Err|$U_{DD,50}$|Err|$U_{DD,50}$|
> |Fair-FLDA|0.314|0.021|0.305|0.048|0.297|0.099|0.289|0.149|0.285|0.200|
> |FPIR|0.385|0.016|0.377|0.050|0.369|0.101|0.360|0.151|0.354|0.200|
> |PPF|0.343|0.722|0.378|0.047|0.369|0.097|0.361|0.144|0.355|0.193|
> |PPOT|0.385|0.016|0.377|0.050|0.369|0.101|0.360|0.150|0.353|0.200|
> |FUDS|0.403|0.157|0.371|0.189|0.358|0.254|0.351|0.317|0.346|0.349|
>
> [1] Delaigle and Hall. Achieving near-perfect classification for functional data. JRSSB, 2012.
>
> [2] Berrendero et al. On the use of reproducing kernel Hilbert spaces in functional classification. JASA, 2018.
>
> [3] Torrecilla et al. Optimal classification of Gaussian processes in homo-and heteroscedastic settings. Statistics and Computing, 2020.
>
> [4] Chang and McKeague, Empirical likelihood-based inference for functional means with application to wearable device data. JRSSB, 2022.
>
> [5] Lin et al., Causal inference on distribution functions. JRSSB, 2023.
>
> [6] Zeng et al., Bayes-optimal fair classification with linear disparity constraints via pre-, in-, and post-processing. arXiv:2402.02817.
>
> [7] Chen et al., Post-hoc bias scoring is optimal for fair classification. ICLR 2024.
>
> [8] Xian et al., Fair and optimal classification via post-processing. ICML 2023.

---

> > ### Comment · Reviewer_V7nb · 2025-08-01
> >
> > Thank you for your detailed response and explanation. Please add more information about the significance of your motivation in the revised version. I'm really not quite sure how much theoretical contribution there is (since theory takes up a very large proportion in the paper, it's necessary to consider whether theoretical innovation is really important in this article). Due to the author's detailed explanation and supplementary experiments, I tend to increase my score.

---

> > > ### Author Response · Authors · 2025-08-01
> > >
> > > Thank you very much.  We will endeavour to enhancing the motivation part in the revision, as you have suggested.

---

### Official Review · Reviewer_uzpK · 2025-07-03

**Clarity:** 3
**Significance:** 2
**Originality:** 2
**Rating:** 4
**Confidence:** 3

**Summary:**

In this work the authors consider Bayes-Optimal classification of functional data. While the notions of fairness and Bayes-optimal classifiers does not change in any meaningful way, standard analysis of these results does not readily extend to functional data. For this the authors systematically develop the necessary tools and propose an algorithm Fair Functional Linear Discriminant Analysis classifier (Fair-FLDA) which is a post-processing algorithm to enforce fairness constraints in functional data. This requires a held-out calibration dataset of i.i.d. samples. The authors validate briefly on synthetic and real datasets.

**Questions:**

See questions in the weaknesses sections.

**Ethical Concerns:**

["NO or VERY MINOR ethics concerns only"]

**Final Justification:**

The authors have convincingly updated their results to address my concerns and I have increased my score in line with my clarified understanding of the work.

**Limitations:**

The empirical validation of this approach is substantially below what we should expect.

**Paper Formatting Concerns:**

None.

**Quality:**

3

**Strengths And Weaknesses:**

Strengths:

The problem of fair classification is a substantial one and in the context of medical applications (neuroscience or genetics) and legal decisions (perhaps based on hand-writing recognition) unfairness or discrimination can cause significant harm. The authors do a good job of motivating the project, though I would remove the White House line given the recent 180 shift from the White House along the lines the authors highlight.

The problem is comprehensively explored and clearly enumerated. The method the authors develop to improve fairness classification also seems to be reasonable (I was unable to check all details of their proofs).

Weaknesses:


While the high-level motivation is clear and the transition between each of the derivations is clear, there are several key aspects that I am unclear on. Additionally, I find the experimental validation to be below standard. I enumerate my questions about some areas that lack clarity below:   Are there any works that establish unfairness of classifiers on functional data in particular? The problem of fairness is an important one, but practically the actual problem solved by this paper seems minimal and the experimental validation is lacking.   The authors to not label their axes, provide any legend, or properly outline their experimental set up. Additionally, there are many post- and pre- processing fairness method that would clearly work in this setting and the authors make no attempt (as far as I can see) to discuss or benchmark against them. Without this the evaluation of the paper is incomplete. Why are no pre-processing e.g., reweighing considered as baselines?   In real-world settings how much does the need for calibration data affect the results? As it seems most functional datasets are of limited size, and if you have to split into test, train, validation, and calibration datasets does this not substantially reduce the utility of the proposed approach? This also may need to be explored.   Perhaps this is too harsh of a phrasing, but the communication both in the figure and description of the experimental section is significantly inadequate to understand (1) the scope of fairness issues in the context, (2) the performance of the proposed method versus baselines, (3) the robustness of the proposal to various assumptions. I am happy to hear the counter arguments by the authors on these points and change my mind if they are compelling.

---

> ### Author Rebuttal · Authors · 2025-07-30
>
> We thank the reviewer for appreciating the strong motivation of our work, the importance of the problem and the comprehensive exploration we conducted. Below, we address the weaknesses and limitations raised by the reviewer. We respectfully hope the reviewer considers raising the score given additional comparisons with alternative methods, and more clarifications and interpretation regarding the advantages of our approach.
>
> **Fairness in functional data classification**
>
> We appreciate your recognition of the importance of fairness. Our data application demonstrates that unfairness can indeed arise in functional classification settings, highlighting the practical relevance of this problem.  This can be seen from Figures 4-6 in the appendix of our submission. Despite its significance, to the best of our knowledge, there is no previous work addressing unfairness in functional data classification. Our work bridges the gap by introducing a principled approach with theoretical guarantees.
>
> Regarding the experimental studies, we provide additional comparisons with multivariate methods adapted to functional data, please see the response to your following point. Moreover, we emphasise that our approach is broadly applicable to a wide range of functional data classification problems where fairness is a concern. For example, the Siena Scalp EEG dataset in the PhysioNet database contains EEG recordings from male and female subjects and can serve as a natural application of our fair functional classifier, where the response variable can be defined as the occurrence of a seizure.
>
> **Comparisons with methods adapted to functional data**
>
> Thanks for the comment on the numerical evaluation. Since no existing fair classifiers can be directly applied to functional data, we adopt a ``dimension reduction + standard fair classifiers" strategy for comparison. Specifically, we first apply functional principal component analysis (FPCA) to extract features, and then employ fair classification methods designed for multivariate data. These include three post-processing methods FPIR [1], PPF [2] and PPOT [3], and one pre-processing approach FUDS [1]. Since the open-source code provided by [1] only supports evaluation using the DD measure, we currently report results under DD with the default parameters. In the revised version, we will include the complete numerical results under the other measures PD and DO.
>
> Based on the numerical results, our proposed fair classifier, Fair-FLDA, consistently achieves the lowest classification errors while effectively controlling disparity under the pre-specified levels. In contrast, the other four baseline methods exhibit higher classification errors. In particular, PPF shows poor disparity control when $\delta=0$, and FUDS fails to adequately control disparity on the NHANES dataset. These extensive numerical results highlight the superiority and practical necessity of our method for fair functional classification.
>
> |   | $\delta=0.00$ |  | 0.05 |  | 0.10 |  | 0.15 | | 0.20 | |
> |----|---|---|---|---|---|---|---|---|---|---|
> |            | Error   | $U_{DD,50}$ | Error   | $U_{DD,50}$ | Error   | $U_{DD,50}$ | Error   | $U_{DD,50}$ | Error   | $U_{DD,50}$ |
> | Fair-FLDA  | 0.234   | 0.022     | 0.227   | 0.048     | 0.219   | 0.098     | 0.213   | 0.149     | 0.208   | 0.197     |
> | FPIR       | 0.276   | 0.024     | 0.269   | 0.039     | 0.264   | 0.080     | 0.258   | 0.122     | 0.253   | 0.168     |
> | PPF        | 0.240   | 0.323     | 0.269   | 0.039     | 0.264   | 0.080     | 0.258   | 0.121     | 0.253   | 0.167     |
> | PPOT       | 0.275   | 0.023     | 0.269   | 0.040     | 0.264   | 0.080     | 0.258   | 0.122     | 0.253   | 0.168     |
> | FUDS       | 0.276   | 0.030     | 0.269   | 0.040     | 0.263   | 0.087     | 0.256   | 0.142     | 0.251   | 0.195     |
>
> Table 1: The median classification error and median DD measure over 500 repetitions under the Gaussian setting with mean difference $\beta=1.5$ and sample size $n=1000$.
>
> |   | $\delta=0.00$ |  | 0.05 |  | 0.10 |  | 0.15 | | 0.20 | |
> |---|---|---|---|---|---|---|---|---|---|---|
> |            | Error   | $U_{DD,50}$ | Error   | $U_{DD,50}$ | Error   | $U_{DD,50}$ | Error   | $U_{DD,50}$ | Error   | $U_{DD,50}$ |
> | Fair-FLDA | 0.314        | 0.021            | 0.305        | 0.048            | 0.297        | 0.099            | 0.289        | 0.149            | 0.285        | 0.200            |
> | FPIR      | 0.385        | 0.016            | 0.377        | 0.050            | 0.369        | 0.101            | 0.360        | 0.151            | 0.354        | 0.200            |
> | PPF       | 0.343        | 0.722            | 0.378        | 0.047            | 0.369        | 0.097            | 0.361        | 0.144            | 0.355        | 0.193            |
> | PPOT      | 0.385        | 0.016            | 0.377        | 0.050            | 0.369        | 0.101            | 0.360        | 0.150            | 0.353        | 0.200            |
> | FUDS      | 0.403        | 0.157            | 0.371        | 0.189            | 0.358        | 0.254            | 0.351        | 0.317            | 0.346        | 0.349            |
>
> Table 2: The median classification error and median DD measure over 500 repetitions under NHANES data.
>
> **Presentation of numerical experiments**
>
> Thank you for your suggestions on our numerical presentations. In the revised version, we will rewrite Section 4 to update figures to include axis labels and legends, and clarify the experimental setup to provide more insights into our results.
>
> Due to the restrictions of the rebuttal format, we will talk through Figure 2 in the original submission here as an example for clarification.  In Figure 2 of the original paper, we compare the performance of Fair-FLDA in terms of both misclassification error and fairness control as the disparity threshold $\delta$ varies. Focusing on the left panel of Figure 2, the orange dots represent the FLDA classifier, which ignores fairness constraints and serves as the baseline in the comparison. The blue stars and pink triangles are the results for Fair-FLDA and Fair-FLDA$_c$, respectively. Focusing on the misclassification error plots in the first column, we observe that in the fair-impacted regime (i.e. $\delta$ is small), the error decreases as $\delta$ increases, and eventually approaches the baseline as the fairness constraint is relaxed. To evaluate fairness performance, the second and third columns report disparity measures under various values of $\delta$. Specifically, points below the grey dashed line indicate that the fairness constraint is satisfied. We show that FLDA consistently fails to meet these fairness requirements, while Fair-FLDA maintains the desired median disparity level, and Fair-FLDA$_c$ ensures that the disparity remains below $\delta$ with at least 95\% probability. We have also included results on model misspecification in Appendix A.3 to illustrate the robustness of Fair-FLDA.
>
> **Usage of calibration data**
>
> The calibration data are primarily introduced here for technical convenience to bring independence among samples in our theoretical studies. In practice, our Algorithm 1 can be implemented by executing both **S1** and **S2** on the whole dataset. In all numerical experiments in Section 4, we mimic the effect of sample splitting via a cross-fitting approach detailed in Appendix A.1, where two classifiers are trained by alternating the roles of data used for model estimation and threshold calibration, and then averaged. To further illustrate the effect of sample splitting, we include additional numerical results on both simulated and real datasets in Table 3 below. Although the reduction in sample size from data splitting slightly increases the misclassification error, the unfairness measure under NoSplit is usually higher than the threshold $\delta$ due to the dependence of the data used in training and calibration.
>
> |           |     Fair-FLDA      |              |   NoSplit     |              |
> |----|---|---|---|---|
> | $\delta$    | Error          | $U_{DO,50}$  | Error          | $U_{DO,50}$  |
> | **Simulated data:**  |               |               |               |              |
> | 0.00        | 0.221         | 0.029        | 0.211         | 0.029        |
> | 0.05        | 0.213         | 0.049        | 0.203         | 0.059        |
> | 0.10        | 0.207         | 0.099        | 0.198         | 0.110        |
> | 0.15        | 0.203         | 0.150        | 0.195         | 0.164        |
> | 0.20        | 0.200         | 0.191        | 0.193         | 0.208        |
> | **Real data:**            |        |              |      |              |
> | 0.00        | 0.291         | 0.034        | 0.284         | 0.039        |
> | 0.05        | 0.286         | 0.054        | 0.281         | 0.056        |
> | 0.10        | 0.284         | 0.099        | 0.278         | 0.101        |
> | 0.15        | 0.282         | 0.148        | 0.277         | 0.152        |
> | 0.20        | 0.281         | 0.196        | 0.276         | 0.205        |
>
> Table 3: Comparison of data splitting for Fair-FLDA on the real and simulated datasets. Results are reported as the median over 500 iterations. NoSplit: the results of Fair-FLDA applied without data splitting.
>
> [1] Zeng et al., Bayes-optimal fair classification with linear disparity constraints via pre-, in-, and post-processing. arXiv:2402.02817.
>
> [2] Chen et al., Post-hoc bias scoring is optimal for fair classification. ICLR 2024.
>
> [3] Xian et al., Fair and optimal classification via post-processing. ICML 2023.

---

> > ### Author Response · Authors · 2025-08-06
> >
> > Dear Reviewer uzpK,
> >
> > As the discussion phase is coming to a close, we would like to check if our responses have fully addressed your concerns. If there are any remaining issues that need clarification, please let us know, we would be happy to provide additional details.
> >
> > Thank you!

---

> > ### Comment · Reviewer_uzpK · 2025-08-06
> > **Thank you for careful consideration**
> >
> > I would like to thank the authors for their careful and comprehensive address of my concerns. It is clear that a tremendous amount of work was done in a short time, and I do find that work to be convincing in addressing almost all of my critiques. I have increased my score to reflect this.

---

> > > ### Author Response · Authors · 2025-08-06
> > >
> > > Thank you very much for your appreciation.

---

### Official Review · Reviewer_4ZfG · 2025-07-04

**Clarity:** 3
**Significance:** 3
**Originality:** 3
**Rating:** 5
**Confidence:** 3

**Summary:**

This paper introduces the first framework for fairness-aware classification in the context of functional data analysis (FDA). The authors provide a theoretical characterization of the Bayes-optimal fair classifier under disparity constraints (Theorem 2), propose a practical post-processing algorithm (Fair-FLDA), and establish high-probability guarantees for both fairness satisfaction (Theorem 3) and excess classification risk (Theorem 5). This work bridges the gap between algorithmic fairness and FDA, a domain where fairness considerations have not been previously explored. The mathematical development is rigorous, and the results generalize known finite-dimensional theories to the functional setting.

**Questions:**

Please see weaknesses.

**Ethical Concerns:**

["NO or VERY MINOR ethics concerns only"]

**Final Justification:**

The paper takes the first steps toward fairness-aware functional classification, and the theoretical results are sound. The authors have addressed the concerns through the rebuttal, and I keep the positive score.

**Quality:**

3

**Strengths And Weaknesses:**

Strengths:
1. The idea of fairness-aware functional classification is new and important for the FDA community, and the paper makes the first steps to address it in a principled way.
2. The theoretical results are sound – they appear to be derived correctly and with appropriate conditions, and they generalize known results.

Weaknesses:
1. While the paper is carefully crafted and mathematically rigorous, the core methodology—such as group-wise threshold adjustment grounded in the Neyman–Pearson principle and use of Radon–Nikodym derivatives—builds upon existing techniques in fairness-aware classification and functional data analysis. The novelty primarily lies in thoughtfully adapting and unifying these tools in the functional setting, rather than proposing fundamentally new statistical models or algorithmic paradigms.
2. Limited diversity in real-world functional datasets; generalizability to broader FDA applications is not demonstrated.

---

> ### Author Rebuttal · Authors · 2025-07-30
>
> We thank the reviewer for recognising the importance of fairness-aware functional classification and our first contribution in this area. Below, we address the weaknesses (W) raised by the reviewer.
>
> **1. Methodological and theoretical novelty (W1)**
>
> Thank you for your appreciation of our work. To the best of our knowledge, this paper is the first to study fairness with functional data.  While there is a rich body of literature on fairness-aware classification for multivariate data, most existing methods do not extend naturally to functional data due to the infinite dimensionality of the function space. Rather than the posterior probabilities $\mathbb{P}(Y=1|A=a, X=x)$ considered in most of the previous papers, our methodology fully respects the functional nature of data by constructing the classifier based on the Radon--Nikodym derivative $\mathrm{d}P_{a,1}(x)/\mathrm{d}P_{a,0}$, which is a more natural functional used for functional classification. Theoretically, we provide not only a fairness guarantee of the designed classifier Fair-FLDA in Theorem 3, but also an explicit convergence rate of excess risk in Theorem 5.
>
> To further demonstrate the advantages of our approach, we have included additional numerical results in which dimension reduction using functional principal component analysis (FPCA) is first performed, followed by the application of existing post- and pre-processing fairness methods. Specifically, we first apply FPCA to extract features, and then employ fair classification methods designed for multivariate data. These include three post-processing methods FPIR [1], PPF [2] and PPOT [3], and one pre-processing approach FUDS [1].
>
> Results in Tables 1 and 2 show that our proposed fair classifier, Fair-FLDA, consistently achieves the lowest classification errors while effectively controlling disparity under the pre-specified levels. In contrast, the other four baseline methods exhibit higher classification errors. In particular, PPF shows poor disparity control when $\delta=0$, and FUDS fails to adequately control disparity on the NHANES dataset. These extensive numerical results highlight the superiority and practical necessity of our method for fair functional classification.
>
> |   | $\delta=0.00$ |  | 0.05 |  | 0.10 |  | 0.15 | | 0.20 | |
> |----|---|---|---|---|---|---|---|---|---|---|
> |            | Error   | $U_{DD,50}$ | Error   | $U_{DD,50}$ | Error   | $U_{DD,50}$ | Error   | $U_{DD,50}$ | Error   | $U_{DD,50}$ |
> | Fair-FLDA  | 0.234   | 0.022     | 0.227   | 0.048     | 0.219   | 0.098     | 0.213   | 0.149     | 0.208   | 0.197     |
> | FPIR       | 0.276   | 0.024     | 0.269   | 0.039     | 0.264   | 0.080     | 0.258   | 0.122     | 0.253   | 0.168     |
> | PPF        | 0.240   | 0.323     | 0.269   | 0.039     | 0.264   | 0.080     | 0.258   | 0.121     | 0.253   | 0.167     |
> | PPOT       | 0.275   | 0.023     | 0.269   | 0.040     | 0.264   | 0.080     | 0.258   | 0.122     | 0.253   | 0.168     |
> | FUDS       | 0.276   | 0.030     | 0.269   | 0.040     | 0.263   | 0.087     | 0.256   | 0.142     | 0.251   | 0.195     |
>
> Table 1: The median classification error and median DD measure over 500 repetitions under the Gaussian setting with mean difference $\beta=1.5$ and sample size $n=1000$.
>
> |   | $\delta=0.00$ |  | 0.05 |  | 0.10 |  | 0.15 | | 0.20 | |
> |---|---|---|---|---|---|---|---|---|---|---|
> |            | Error   | $U_{DD,50}$ | Error   | $U_{DD,50}$ | Error   | $U_{DD,50}$ | Error   | $U_{DD,50}$ | Error   | $U_{DD,50}$ |
> | Fair-FLDA | 0.314        | 0.021            | 0.305        | 0.048            | 0.297        | 0.099            | 0.289        | 0.149            | 0.285        | 0.200            |
> | FPIR      | 0.385        | 0.016            | 0.377        | 0.050            | 0.369        | 0.101            | 0.360        | 0.151            | 0.354        | 0.200            |
> | PPF       | 0.343        | 0.722            | 0.378        | 0.047            | 0.369        | 0.097            | 0.361        | 0.144            | 0.355        | 0.193            |
> | PPOT      | 0.385        | 0.016            | 0.377        | 0.050            | 0.369        | 0.101            | 0.360        | 0.150            | 0.353        | 0.200            |
> | FUDS      | 0.403        | 0.157            | 0.371        | 0.189            | 0.358        | 0.254            | 0.351        | 0.317            | 0.346        | 0.349            |
>
> Table 2: The median classification error and median DD measure over 500 repetitions under NHANES data.
>
> **2. Diversity and generalizability (W2)**
>
> Thank you for the comment. In our work, we use the publicly available NHANES dataset as a representative data example to demonstrate the effectiveness of our method. It is worth noting that our approach is broadly applicable to a wide range of functional data classification problems where fairness is a concern. For example, the Siena Scalp EEG dataset in the PhysioNet database contains EEG recordings from male and female subjects and can serve as a natural application of our fair functional classifier, where the response variable can be defined as the occurrence of a seizure. Overall, our method offers a principled and broadly applicable tool for fair classification in functional data analysis. We will include a relevant discussion on this point in the revised version.
>
> [1] Zeng et al., Bayes-optimal fair classification with linear disparity constraints via pre-, in-, and post-processing. arXiv:2402.02817.
>
> [2] Chen et al., Post-hoc bias scoring is optimal for fair classification. ICLR 2024.
>
> [3] Xian et al., Fair and optimal classification via post-processing. ICML 2023.

---

> > ### Comment · Reviewer_4ZfG · 2025-08-05
> > **Response to author rebuttal**
> >
> > Thank you for your response. I will keep my score.

---

> > > ### Author Response · Authors · 2025-08-06
> > >
> > > Thank you very much.

---

### Official Review · Reviewer_tKM6 · 2025-07-09

**Clarity:** 3
**Significance:** 4
**Originality:** 2
**Rating:** 4
**Confidence:** 3

**Summary:**

This paper addresses fairness-aware binary classification for functional data by introducing a post-processing algorithm called Fair Functional Linear Discriminant Analysis (Fair-FLDA). The main contributions include theoretical guarantees on fairness and excess risk control under homoscedastic Gaussian processes, and a novel analysis of eigenspace estimation effects on classification performance. The work provides both theoretical foundations with finite samples and algorithmic implementation, validated through synthetic and real data experiment.

**Questions:**

1. Why choose this sophisticated approach over simpler alternatives (e.g., dimension reduction + standard fair classification)?
2. How does the method scale computationally and handle practical challenges (eigenspace estimation, noisy/missing sensitive attributes)?
3. Can the framework extend to multiple sensitive attributes or multi-class classification?
4. Can we compare the proposed method with standard fairness methods adapted to functional data?

**Ethical Concerns:**

["NO or VERY MINOR ethics concerns only"]

**Final Justification:**

I appreciate the authors for their detailed response to my concerns and questions. Few remaining questions includes 1) how such comparing methods for additional experiments selected (as some are not SOTA methods); 2) concern on sensitive information not available in training time. However, as most of my questions have been answered in rebuttal, I am leaning towards raising the final score.

**Limitations:**

1. Limited empirical evaluation (multiple datasets, baselines, adapted fairness methods) and lack of justification on this
2. Justification against simpler alternatives
3. Analysis of practical limitations and computational efficiency
4. Broader applicability (multiple attributes, missing data scenarios)

**Quality:**

3

**Strengths And Weaknesses:**

**Strengths:**
1. Novel theoretical contribution combining fairness and functional data analysis, with comprehensive guarantees under weak assumptions
2. Clear theoretical connections to both FDA and fairness literature and strong motivation of the topic

**Weaknesses:**

1. Limited empirical validation (single synthetic/real dataset) and missing comparisons with adapted existing fairness methods
2. Lack of justification against simpler alternatives (e.g., dimensionality reduction + standard fair classification)
3. Practical limitations: In many cases, sensitive attributes may not be avilalbe during both training and testing. Also, missing discussion on multi-class sensitive attribute (e.g., race, age category, etc.)
4. Concerns around eigenspace estimation reliability and computational efficiency

---

> ### Author Rebuttal · Authors · 2025-07-30
>
> We thank the reviewer for the appreciation of our theoretical contributions and the strong motivation for studying fair functional classification. Below we address the weaknesses (W), questions (Q) and limitations (L) raised by the reviewer. We respectfully hope the reviewer could consider raising the score given the additional comparisons to adapted methods, the clarified advantages of our method and extensions to scenarios with missing or multiple sensitive attributes.
>
> **1. Additional empirical evaluation (W1, Q4, L1)**
>
> **Comparisons with standard fairness methods adapted to functional data.** To the best of our knowledge, our work is the first to derive the Bayes optimal fair classifier and to establish explicit convergence rates in the context of functional data. There are no existing fair functional classifiers available for direct comparison.
>
> As suggested, we have incorporated additional baselines following a "dimension reduction + standard fair classification" strategy. Specifically, we first apply functional principal component analysis to extract features, and then employ fair classification methods designed for multivariate data. These include three post-processing methods FPIR [1], PPF [2] and PPOT [3], and one pre-processing approach FUDS [1], with default parameters as in the open source code of [1].
>
> Results in Tables 1 and 2 show that our proposed fair classifier, Fair-FLDA, consistently achieves the lowest classification errors while effectively controlling disparity under the pre-specified levels. In contrast, the other four baseline methods exhibit higher classification errors. In particular, PPF shows poor disparity control when $\delta=0$, and FUDS fails to adequately control disparity on the NHANES dataset. These extensive numerical results highlight the superiority and practical necessity of our method for fair functional classification.
>
> Table 1: Median classification error and DD over 500 runs under Gaussian with mean difference $\beta=1.5$ and sample size $n=1000$.
>
> |$\delta$|0.00||0.05||0.10||0.15||0.20||
> |--|--|--|--|--|--|--|--|--|--|--|
> | |Err|$U_{DD,50}$|Err|$U_{DD,50}$|Err|$U_{DD,50}$|Err|$U_{DD,50}$|Err|$U_{DD,50}$|
> |Fair-FLDA|0.234|0.022|0.227|0.048|0.219|0.098|0.213|0.149|0.208|0.197|
> |FPIR|0.276|0.024|0.269|0.039|0.264|0.080|0.258|0.122|0.253|0.168|
> |PPF|0.240|0.323|0.269|0.039|0.264|0.080|0.258|0.121|0.253|0.167|
> |PPOT|0.275|0.023|0.269|0.040|0.264|0.080|0.258|0.122|0.253|0.168|
> |FUDS|0.276|0.030|0.269|0.040|0.263|0.087|0.256|0.142|0.251|0.195|
>
> Table 2: Median classification error and DD over 500 runs under NHANES.
>
> |$\delta$|0.00||0.05||0.10||0.15||0.20||
> |--|--|--|--|--|--|--|--|--|--|--|
> | |Err|$U_{DD,50}$|Err|$U_{DD,50}$|Err|$U_{DD,50}$|Err|$U_{DD,50}$|Err|$U_{DD,50}$|
> |Fair-FLDA|0.314|0.021|0.305|0.048|0.297|0.099|0.289|0.149|0.285|0.200|
> |FPIR|0.385|0.016|0.377|0.050|0.369|0.101|0.360|0.151|0.354|0.200|
> |PPF|0.343|0.722|0.378|0.047|0.369|0.097|0.361|0.144|0.355|0.193|
> |PPOT|0.385|0.016|0.377|0.050|0.369|0.101|0.360|0.150|0.353|0.200|
> |FUDS|0.403|0.157|0.371|0.189|0.358|0.254|0.351|0.317|0.346|0.349|
>
> **Synthetic/real datasets.** In the original submission, we conducted extensive simulations under various model configurations to illustrate the effects of sample sizes, alignment of mean difference, model misspecification and perfect classification. These synthetic experiments provide insights into the practical implications of our theoretical convergence rates. Detailed results are deferred to the Supplementary Material due to space constraints.
>
> Since no prior work specifically targets fair functional classification, there are no widely established benchmark datasets in this context. We therefore use the NHANES dataset, a common dataset in functional data studies [4, 5], to demonstrate the practical significance of our fair functional classifier. It is important to note that our method is broadly applicable; for instance, it can be used on the Siena Scalp EEG dataset in the PhysioNet database to classify seizures while ensuring gender fairness.
>
> **2. Advantages of our method over simpler alternatives (W2, Q1, L2)**
>
> We clarify our advantages from both theoretical and numerical aspects. Theoretically, our method fully respects the functional nature of the data by deriving the explicit form of the Bayes optimal fair classifier in Eq(4), which naturally motivates its practical approximation via truncation in Algorithm 1. We establish that the excess risk converges to zero and provide an explicit convergence rate in Theorem 5. In contrast, simpler alternatives such as "dimensional reduction + standard fair classifiers" lack any theoretical guarantees. Numerically, through extensive comparisons with FPIR, PPF PPOT and FUDS on both synthetic and real data, our method consistently achieves the lowest classification error while effectively controlling disparity, significantly outperforming these simpler alternatives.
>
> **3. Extensions to scenarios with missing or multiple sensitive attributes (W3, Q3, L3, L4)**
>
> Our framework in Section 2.2 can be naturally extended to settings where the sensitive feature is unavailable at testing, as well as to multi-class classification problems. We provide a sketch of methods in this response, and we will incorporate this addition in the revision.
>
> **Missing sensitive attributes.** The extension consists of three steps.
>
> **Step 1.** A key ingredient in our framework, when sensitive features are available, is that both the misclassification error $R$ and disparity measure $D$ are linear in classifiers $f:\mathcal{X}\times \mathcal{A}\rightarrow [0,1]$. To extend the framework to settings where sensitive attributes are unavailable at testing, it is necessary to show that $R$ and $D$ remain linear to classifiers $f:\mathcal{X} \rightarrow [0,1]$, which is solely defined on $\mathcal{X}$. Following a similar idea as in the proof of Prop.1, it can be verified that DO, PD and DD are still linear and of the form $D(f) = \int_{\mathcal{X}}f(x)w_{X, D}(x)dP_{X|Y=0}(x)$, where $w_{X,D}:\mathcal{X} \rightarrow \mathbb{R}$ is a weight function.
>
> **Step 2.** By the generalised Neyman--Pearson lemma and a similar argument to the one used in the proof of Theorem 2, we can derive an explicit formula for $\delta$-fair Bayes optimal classifier of the form $f_{D}^\star(x) =\mathbf{1}[\pi_{1} \frac{dP_{X|Y=1}}{dP_{X|Y=0}}(x) -\pi_{0}  \geq \tau^\star w_{x, D}(x)].$
>
> **Step 3.** Construct a plug-in classifier. Further to the estimation in Algorithm 1, an additional non-parametric estimator can be used to estimate the probability of A given X and Y.
>
> **Multi-class classification.**  Consider predictive disparity as an example. For a multi-class sensitive attribute $a \in \mathcal{A} = [1, \ldots, |\mathcal{A}|]$, motivated by the proof of Theorem 2 in our paper and Theorem 4.8 in [1], we conjecture that the generalised Neyman-Pearson lemma leads to $f_{PD}^\star(x,a) = \mathbf{1} [\frac{dP_{a,1}}{dP_{a,0}}(x) \geq  \frac{\pi_{a,0}+\tau^\star_a}{\pi_{a,1}}]$, where the threshold $\tau^\star_a$ is selected in a way similar to Eq(3) in the paper.
>
> **4. Eigenspace estimation and computational efficiency (W4, Q2, L3)**
>
> Eigenspace estimation plays a fundamental role in FDA. We justify the performance of eigenspace estimation from both theoretical and numerical perspectives. In our paper, the theoretical guarantees for eigenfunction and eigenvalue estimation have been established in Lemmas 35 and 36, respectively. These results are optimal, matching the minimax rate established in [6] up to poly-logarithmic factors.
>
> To support our theory, we provide further simulation results in Table 3, where Fair-FLDA refers to the proposed classifier in Section 4; Truth uses true eigenfunctions and eigenvalues; Fourier replaces estimated eigenfunctions with Fourier basis and eigenvalues with covariance projection scores. Overall, disparity control is comparable across methods, with all meeting their fairness criteria. Comparing the results of Fair-FLDA with Truth, we see that the misclassification errors of the proposed classifiers are even smaller than those obtained without eigenspace estimation. This improvement is attributed to the data-adaptive nature of the estimated eigenfunctions, which captures more variance than the fixed true basis. Substituting the estimated eigenfunctions with the pre-specified Fourier basis leads to a noticeable increase in misclassification errors.
>
> Regarding computational complexity, the eigensystem is estimated using a discretisation-based approximation in implementation, with an overall complexity of $O(K^2J)$, where $K$ is the number of evaluated points and $J$ is the truncation level. Due to the inherent smoothness of functional data, choosing $J$ in 50–100 will provide a sufficiently accurate approximation.
>
>
>
> Table 3: Median classification error and DO over 500 runs under Gaussian with mean diff $\beta=1.5$ and sample size $n=1000$.
>
> |$\delta$|Fair-FLDA||Truth||Fourier||
> |--|--|--|--|--|--|--|
> | |Err|$U_{DO,50}$|Err|$U_{DO,50}$|Err|$U_{DO,50}$|
> |0.00|0.221|0.029|0.235|0.028|0.274|0.027|
> |0.05|0.213|0.049|0.228|0.055|0.266|0.053|
> |0.10|0.207|0.099|0.223|0.101|0.261|0.101|
> |0.15|0.203|0.150|0.220|0.140|0.256|0.151|
> |0.20|0.200|0.191|0.218|0.172|0.253|0.200|
>
> [1] Zeng et al., Bayes-optimal fair classification with linear disparity constraints via pre-, in-, and post-processing. arXiv:2402.02817.
>
> [2] Chen et al., Post-hoc bias scoring is optimal for fair classification. ICLR 2024.
>
> [3] Xian et al., Fair and optimal classification via post-processing. ICML 2023.
>
> [4] Chang and McKeague, Empirical likelihood-based inference for functional means with application to wearable device data. JRSSB, 2022.
>
> [5] Lin et al., Causal inference on distribution functions. JRSSB, 2023.
>
> [6] Wahl. Lower bounds for invariant statistical models with applications to principal component analysis. Ann. Inst. H. Poincaré Probab. Statist, 2022.

---

> > ### Comment · Reviewer_tKM6 · 2025-08-05
> > **Author response review**
> >
> > I appreciate the authors for their detailed response to my concerns and questions. Few remaining questions includes 1) how such comparing methods for additional experiments selected (as some are not SOTA methods); 2) concern on sensitive information not available in training time. However, as most of my questions have been answered in rebuttal, I am leaning towards raising the final score.

---

> > > ### Author Response · Authors · 2025-08-06
> > >
> > > We are encouraged that the rebuttal has addressed most of your questions. We respond to the remaining ones below.
> > >
> > > **Choices of additional methods**
> > >
> > > Our work focuses on deriving the $\delta$-fair Bayes optimal fair classifier for general linear disparity measures, such as DO, DD and PD, in the functional setting. As suggested, we compared with the multivariate alternatives by first applying dimension reduction. In multivariate settings, the most relevant work is Zeng et al. [1], who established a theoretical framework for the $\delta$-fair Bayes optimal classifier and proposed theoretically optimal algorithms for general linear disparity measures, including pre-processing method FUDS and the post-processing method FPIR.
> > >
> > > We selected FUDS and FPIR algorithms as baselines primarily because they are theoretically optimal under the considered fairness metrics, rather than relying on heuristics. For comprehensiveness, we also included two additional methods PPF [2] and PPOT [3] for comparison. We acknowledge that these methods may not represent all recent SOTA algorithms. We would be glad to include additional comparisons if the reviewer can suggest specific methods or references.
> > >
> > > **Sensitive information not available in training time**
> > >
> > > Thank you for raising this important question. When sensitive information is not available in training time, the fair classification is indeed a challenging problem because, without direct access to sensitive information, it is difficult to learn and correct for the potential biases. Relatively few works have studied this issue. Existing approaches [4, 5, 6] generally rely on the assumption that standard features $X$ are sufficiently informative, allowing bias mitigation through indirect inference for protected attributes. In [7], it is further shown that various disparities remain unidentifiable when $X$ lacks sufficient information.
> > >
> > > In our work, we focus on developing a principled framework for achieving fairness in functional data classification when sensitive features are available, either during both training and testing or at least in the training stage. Extending our framework to settings where sensitive attributes are completely unavailable is an important direction for future research, and we will add a more detailed discussion of this scenario and related literature in the revised manuscript.
> > >
> > >
> > >
> > > [1] Zeng et al., Bayes-optimal fair classification with linear disparity constraints via pre-, in-, and post-processing. arXiv:2402.02817.
> > >
> > > [2] Chen et al., Post-hoc bias scoring is optimal for fair classification. ICLR, 2024.
> > >
> > > [3] Xian et al., Fair and optimal classification via post-processing. ICML, 2023.
> > >
> > > [4] Lahoti et al., Fairness without Demographics through Adversarially Reweighted Learning. NeurIPS, 2020.
> > >
> > > [5] Zhao et al., Towards Fair Classifiers Without Sensitive Attributes: Exploring Biases in Related Features. International Conference on Web Search and Data Mining, 2022.
> > >
> > > [6] Veldanda et al., Hyper-parameter tuning for fair classification without sensitive attribute access. TMLR, 2024.
> > >
> > > [7] Kallus et al., Assessing algorithmic fairness with unobserved protected class using data combination. Management Science, 2022.

---

### Note · Authors · 2025-08-12

We sincerely thank the reviewers for their insightful and constructive comments, as well as their engagement during the rebuttal. Below, we briefly summarise the contributions and strengths of our work.

**Fair functional data classification.**
Functional data classification is increasingly prevalent in recent years, with diverse applications in fields such as neuroscience, genetics and others. However, as demonstrated by our numerical analysis on the publicly available NHANES dataset in Section 4, applying standard functional classification algorithms can lead to substantial unfairness. Despite the recognition of the importance of fairness in the literature, to the best of our knowledge, there is no previous work addressing unfairness in functional data classification. Our work bridges the gap by combining methodological innovation with a rigorous theoretical analysis of both fairness and excess risk controls, highlighting the practicality and reliability of our designed classifiers.

**Advantages over simpler alternatives.** The infinite-dimensional nature of functional data poses unique challenges, rendering fairness methods developed for multivariate data ineffective. Our framework fully respects the functional nature by focusing on a more natural functional, the Radon-Nikodym derivative, used in functional data classification. In additional numerical experiments carried out during rebuttal, we include further baselines by applying four different multivariate fairness methods to functional data after dimension reduction. In both simulated and real data analysis, our methods consistently achieve the lowest classification errors while effectively controlling disparity under the pre-specified levels. These numerical results highlight the superiority and practical necessity of our method for fair functional classification from a numerical perspective.

**Flexibility and generality.**
In addition to the setups considered in Section 2.1 of the paper, our unified framework for fair functional classification in Theorem 2 is sufficiently general to accommodate a broad array of extensions. In the rebuttal, we have demonstrated that our framework can be easily extended to settings where the sensitive feature is unavailable at testing, as well as to multi-class classification problems. These extensions further illustrate the flexibility of our framework and its potential to serve as a foundation for future developments in fairness-aware functional data analysis.

---

### Decision · Program_Chairs · 2025-09-17

**Decision:**

Accept (poster)

**Comment:**

The manuscript introduces the first framework for fairness-aware classification in the context of functional data analysis (FDA), providing a theoretical characterization of the Bayes-optimal fair classifier under disparity constraints, proposing a practical post-processing algorithm (Fair-FLDA), and establishing high-probability guarantees for both fairness satisfaction and excess classification risk. Reviewers acknowledged that this work bridges the gap between algorithmic fairness and FDA, a domain where fairness considerations have not previously been explored. Nonetheless, concerns were raised regarding the novelty, as the core methodology draws on existing techniques in fairness-aware classification and FDA, such as group-wise threshold adjustment grounded in the Neyman-Pearson principle and the use of Radon-Nikodym derivatives, and about the absence of simple baselines involving multivariate fairness methods applied to functional data after dimension reduction. Most of these concerns were addressed in the rebuttal and subsequent author–reviewer discussion, including through additional experiments in extended settings where the sensitive feature is unavailable at test time and for multi-class classification problems. In the end, all reviewers agreed that the manuscript is suitable for publication, though they emphasized the need for a careful revision of the motivation and a more thorough discussion of scenarios where sensitive information is not available during training.